# GraphOmni: A Comprehensive and Extensible Benchmark Framework for Large Language Models on Graph-theoretic Tasks

[1,7]**Hao Xu**[*], [2]**Xiangru Jian**[*,†,✉], [1]**Xinjian Zhao**[*], [2]**Wei Pang**[*], [2]**Chao Zhang**, [3]**Suyuchen Wang**,
[4]**Qixin Zhang**, [2]**Zhengyuan Dong**, [5]**Joao Monteiro**[‡], [3]**Bang Liu**, [6]**Qiuzhuang Sun**, [1]**Tianshu Yu**[✉]

[1] The Chinese University of Hong Kong, Shenzhen, [2]University of Waterloo
[3] Université de Montréal / Mila - Quebec AI Institute, [4] City University of Hong Kong
[5] Apple MLR, [6] Singapore Management University [7] The University of Sydney

✺ Project Page: https://gai-community.github.io/Graph-Omni/
⦿ https://github.com/GAI-Community/GraphOmni
🤗 https://huggingface.co/datasets/G-A-I/GraphOmni

## Abstract

This paper introduces GraphOmni, a comprehensive benchmark designed to evaluate the reasoning capabilities of LLMs on graph-theoretic tasks articulated in natural language. GraphOmni spans diverse graph types, serialization formats, and prompting schemes, substantially extending upon prior efforts in both scope and depth. Through systematic evaluation, we uncover critical interactions among these dimensions, revealing their decisive impact on model performance. Our experiments show that state-of-the-art closed-source models such as Claude-3.5 and o4-mini consistently lead overall, yet still leave considerable headroom, while open-source models display pronounced sensitivity to various design choices. Beyond the standard scope, larger graphs, real-world graphs, and additional NP-hard tasks are further discussed. We further analyze efficiency via output token usage, highlighting cost–accuracy trade-offs, and introduce a reinforcement learning-based optimizer that adaptively selects factor combinations, reducing evaluation cost by 75% while retaining strong accuracy. This flexible and extensible benchmark not only deepens understanding of LLM performance on structured graph reasoning but also establishes a robust foundation for advancing model design and evaluation.

## 1 Introduction

Large Language Models (LLMs) have emerged as a transformative force in natural language processing (NLP), demonstrating state-of-the-art performance in tasks such as open-ended text generation, summarization, and problem-solving (Radford et al., 2019; Brown et al., 2020; Raffel et al., 2020; Lewis et al., 2020). However, their application to structured reasoning on graph-based data remains relatively underexplored. Graphs, defined by their nodes and edges, encapsulate complex relationships that are crucial to many real-world applications, including social network analysis (Easley et al., 2010), recommendation systems (Wu et al., 2022; Jian & Wang, 2023), out-of-distribution detection (Fang et al., 2025a), and drug discovery (Gaudelet et al., 2021; Wang & Zhuang, 2025).

Traditional approaches to graph analysis primarily rely on Graph Neural Networks (GNNs) that are designed with specialized representations and training paradigms tailored for tasks such as node classification (Wu et al., 2020), link prediction (Zhang & Chen, 2018), and community detection (Su

---

[*]Equal contribution. ✉ Corresponding to: xiangru.jian@uwaterloo.ca, yutianshu@cuhk.edu.cn
[†]Project Lead
[‡]Work done before joining Apple

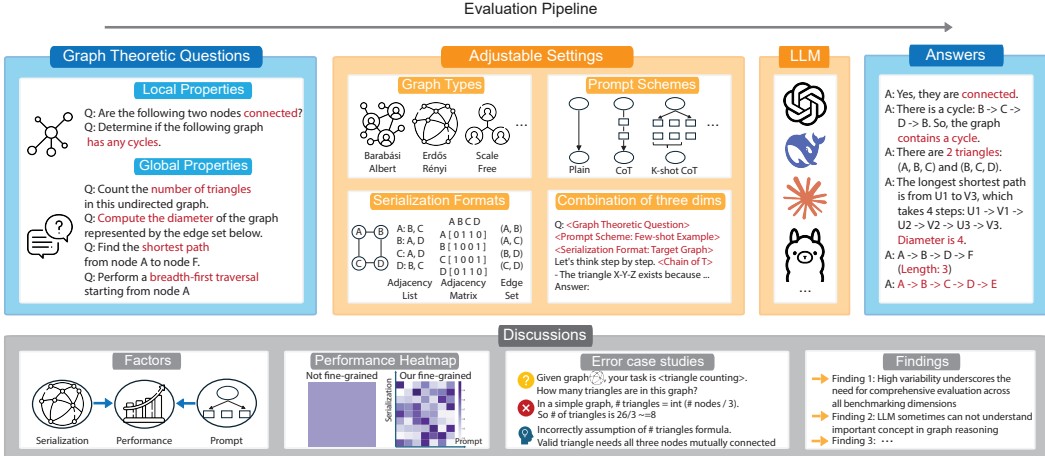

Figure 1: **GRAPHOMNI Evaluation Pipeline.** We convert graph-theoretic tasks into text-based questions about local and global properties. In the adjustable settings, we vary three dimensions, i.e., graph type, serialization format, and prompt scheme, and then generate every possible combination.

et al., 2022). In contrast, LLMs are trained on vast quantities of unstructured or semi-structured text and excel at reasoning about entities and relationships described linguistically, as evidenced by benchmarks like MMLU (Hendrycks et al., 2021a) and MATH (Hendrycks et al., 2021b). This discrepancy raises a pivotal question: **Can LLMs be effectively harnessed to understand and manipulate graph-theoretic concepts when graphs are articulated in natural language?**

To address this question, a multi-dimensional evaluation is required rather than tuning a single knob. Prior work has examined individual components in isolation, including prompting strategies (Wang et al., 2023; Fatemi et al., 2024), textual graph serialization (Xypolopoulos et al., 2024), or specific graph families (Zhang et al., 2024b), but this piecemeal view obscures how these choices interact. We therefore vary three interacting dimensions jointly. First, **graph type**: different graph types exhibit distinct structures, so we use synthetic generators (ER, BA, scale-free, bipartite) to produce them, which in turn affects how readily a text description can capture these structures. Second, **serialization format**: the same graph written as an adjacency list or matrix, an edge set, or a richer schema can help or hinder model reading. Third, **prompt scheme**: the way the question is posed (zero-shot, few-shot, instructive, algorithmic, chain-of-thought) can shift answers even with identical inputs. As summarized in Table 1, previous studies do not vary these dimensions together, so they cannot determine whether gains come from the model, the representation, or the instruction, nor explain why a setting that benefits one model may harm another. Consequently, we still lack a comprehensive and robust understanding of LLM capabilities in graph reasoning.

Table 1: **Comparison of existing graph-related benchmarks for LLM with our GRAPHOMNI.** We evaluate their inclusion of different types of graphs, serialization formats, and prompt schemes, noting a gap between recent works and ours. Additionally, GRAPHOMNI is the only work with a random baseline as well as a modularized and expandable framework design. More related works are included in Detailed Related Works in Appendix F.

| Benchmarks | Graph Sources | | | Serializations | | Prompt Schemes | | Evaluation Framework | |
|---|---|---|---|---|---|---|---|---|---|
| | # Samples | # Graph Types* | Node Size | Multiple Types | # Types | Multiple Types | # Types | Random Baseline | Modularized |
| LLM4DyG (Zhang et al., 2024b) | 900 (100 per task) | 4 | 5 to 20 | ✗ | 1 | ✓ | 4 | ✓ | ✗ |
| GraphInstruct (Luo et al., 2024b) | N/A | 3 | 5 to 35 | ✓ | 3 | ✗ | 1 | ✗ | ✓ |
| MAGMA (Taylor et al., 2024) | ∼ 400 | 1 | 5 to 50 | ✗ | 1 | ✗ | 1 | ✗ | ✗ |
| NLGraph (Wang et al., 2023) | 5,902 | 1 | 5 to 35 | ✗ | 1 | ✓ | 5 | ✗ | ✗ |
| GPC (Dai et al., 2024) | 350 | 1 | 5 to 35 | ✓ | 2 | ✗ | N/A | ✗ | ✗ |
| GraphWiz (Chen et al., 2024a) | 3,600 | 1 | 2 to 100 | ✗ | 1 | ✗ | 1 | ✗ | ✗ |
| GPT4Graph (Guo et al., 2024a) | N/A | 1 | 10 to 20 | ✓ | 4 | ✓ | 6 | ✗ | ✗ |
| GraphArena (Tang et al., 2025) | 10,000 | N/A | 5 to 30† | ✗ | 1 | ✗ | 1 | ✗ | ✗ |
| GraphQA (Fatemi et al., 2024) | 2,300 | 7 | 5 to 20 | ✗ (only via text) | 1 | ✓ | 6 | ✗ | ✓ |
| NLGift (Zhang et al., 2024a) | 37,000 | 2 | 3 to 25 | ✗ | 1 | ✗ | 1 | ✗ | ✗ |
| GraphWild (Zhang et al., 2025) | 49,224 | 5 | N/A | ✗ | 1 | ✗ | 1 | N/A | N/A |
| **GRAPHOMNI** | 241,726 | 7 | 5 to 30 | ✓ | 7 | ✓ | 9 | ✓ | ✓ |

* Note that # Graph Types is targeted for synthetic datasets and reflects the number of types of random graph generators.
† The range is for all non-trivial tasks, excluding nearest neighbor and shortest distance.

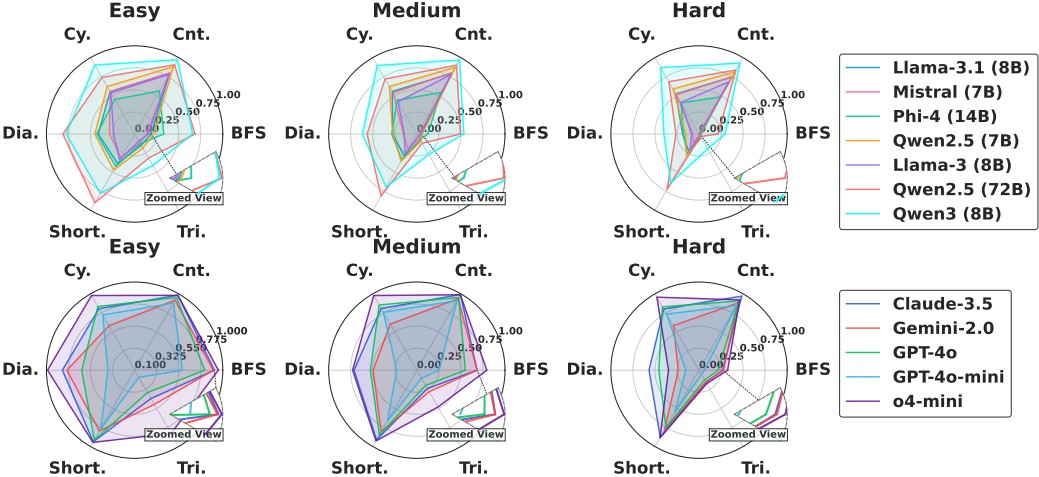

Figure 2: Radar charts comparing the performance of open-source (top row) and closed-source (bottom row) LLMs across six canonical graph reasoning tasks at three difficulty levels.

To address this gap, we propose GRAPHOMNI, a unified benchmark with an extensible framework, summarized in Figure 1. It represents the most comprehensive graph-theory-based evaluation framework developed to date, compared with all related works in Table 1. It spans various graph types, serialization formats, and prompt schemes, surpassing previous works in scope and granularity. Furthermore, our framework is designed as an extensible and flexible evaluation system. Researchers can easily incorporate new graph generators, serialization methods, and prompt strategies, thereby ensuring that the benchmark remains current with evolving methodologies in both LLM research and graph theory. A random baseline is then implemented to ensure a fair evaluation.

With the help of GRAPHOMNI we clearly demonstrate that no single serialization or prompt works best for all models and accuracy varies widely across graph types, serializations, and prompts, which validates the need for our multi-dimensional design and per-task configuration. Additionally, model performance requires further improvement overall: Claude-3.5 and o4-mini lead across tasks and difficulty levels, yet even they fall short of the near-perfect accuracy a non-specialist human evaluator could achieve on 5–30 node problems given sufficient time. To verify the robustness of the evaluation results, we extend the analysis to larger graphs, NP-hard tasks, and conduct a representativeness check on real-world graphs, all of which yield the same trends. Motivated by these results, we introduce a simple RL-inspired selector that chooses the optimal settings (prompt + serialization) for each task, thereby improving accuracy at a minimal extra cost. We summarize our contributions as:

∗ **Novel benchmark:** We introduce GRAPHOMNI, the most comprehensive benchmark to our knowledge for evaluating graph-theoretic reasoning in LLMs, covering a wide range of synthetic graph types, diverse serialization formats, and varied prompt schemes.

∗ **Comprehensive evaluation framework:** We design a flexible and extensible evaluation framework that allows for the seamless addition or removal of graph generators, serialization methods, and prompt schemes, ensuring adaptability to future research developments. We also include extended studies on larger graphs (30–50 nodes), real-world datasets, and NP-Hard tasks, which together confirm the robustness and transferability of our conclusions.

∗ **Insightful empirical observations:** State-of-the-art models still exhibit considerable room for improvement overall. Our experiments reveal substantial performance variance, with notable accuracy differences across different serialization and prompting configurations, emphasizing the need for comprehensive evaluation across all dimensions to provide fair and trustworthy understandings.

∗ **Practical methods inspired by observations:** Motivated by the above observations, we develop an RL-based adaptive mechanism that dynamically selects the optimal factors, achieving near-optimal performance with only a small exploration cost.

## 2 GRAPHOMNI

**Overview and Statistics.** GRAPHOMNI rigorously evaluates LLM performance on graph reasoning by examining the interplay between graph structure, textual representation, and prompt formulation. It comprises four key components: **Benchmark Tasks**, **Graph Types**, **Prompt Schemes**, and **Serialization Formats**. Figure 1 illustrates how these four components form our end-to-end evaluation pipeline. **Benchmark Tasks** cover canonical graph problems that test both local and global reasoning. **Graph Types** are defined by diverse synthetic datasets generated by different random graph generators, including stochastic, scale-free, and bipartite models. **Prompt Schemes** incorporate various query designs such as algorithmic, chain-of-thought, k-shot, instructive, and zero-shot approaches. **Serialization Formats** convert graph data into text using methods like adjacency lists, matrices, and the GMoL. Moreover, we have designed three difficulty modes for all graph-related tasks, determined by the number of nodes: Easy (5–10 nodes), Medium (10–20 nodes), and Hard (20–30 nodes). This unified and extensible framework distinguishes itself by integrating multiple dimensions of graph reasoning into a single evaluation platform, thereby providing comprehensive insights into LLM performance on complex, structured data. The basic statistics of GRAPHOMNI are presented in Table 2, while token statistics for different combinations are shown in Figure 3. In summary, our dataset contains a total of 241,726 queries. More detailed statistics are in Appendix B.

**Graph Tasks.** We consider **6 canonical tasks** that capture both local and global properties of graphs, thereby requiring diverse reasoning capabilities from LLMs. Connectivity involves determining whether a path exists between two designated nodes, testing the model's understanding of local linkages. Cycle detection requires verifying the presence of any cycle, which probes the model's ability to recognize recurring patterns in connectivity. Diameter calculation demands calculating the maximum distance between any two nodes, thereby challenging the model to grasp the global network structure. BFS order tests the ability to generate an ordered sequence of nodes as encountered in a breadth-first search, assessing sequential output and structured reasoning. Triangle counting requires precise numerical enumeration of 3-cycles, blending quantitative precision with structural insight. Shortest path tasks compel the model to identify the most efficient route between two nodes. Collectively, these tasks provide a robust measure of performance across both binary decisions and nuanced numerical analyses. For

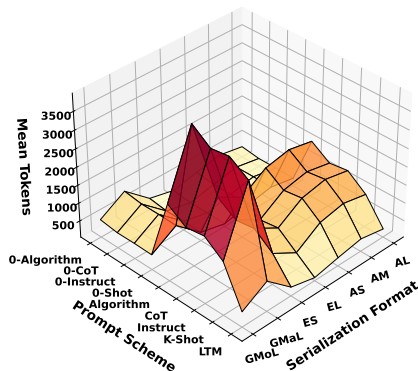

Figure 3: **Token usage for prompt-serialization combinations by GPT-4 tokenizer**. More detailed statistics are included in Figures 6a and 6b.

more details on the design of the graph task, please refer to Appendix A.3, where we further discuss the rationale behind the task selection and analyze the distinct capability demands of each task in Appendix A.3.1. We also include NP-hard tasks for extended discussion, elaborated in Appendix C.4.

**Graph Generators (Types of Graphs).** To mirror the diversity found in real-world networks, our benchmark incorporates a broad array of graph families of **7 types**, each presenting unique structural characteristics that challenge LLM reasoning. ER Graphs are generated by random sampling from the space of all graphs with $n$ vertices. Within this family, ERM employs a fixed edge count $m$, randomly chosen between 1 and $\frac{n(n-1)}{2}$, while ERP uses a probability-based approach with an edge probability drawn uniformly from $[0, 1]$. Extending these models to capture structured variations, Bipartite ER Graphs (denoted as BERM and BERP) impose bipartite constraints that yield additional topological diversity. To reflect the power-law distributions prevalent in real-world networks, we include Barabási–Albert Graphs(BAG), generated by initializing a complete graph of $m_0$ vertices (with $m_0$ randomly chosen up to $\frac{n}{3}$) and sequentially adding nodes that form $m = m_0 + 1$ connections via preferential attachment. Recognizing that many practical

Table 2: **Statistical summary of GRAPHOMNI over tasks at all difficulty levels**. More statistics are in Table 7.

| Difficulty | Easy | Medium | Hard |
|---|---|---|---|
| **Numbers** | 88956 | 87318 | 65452 |
| **Avg. Nodes** | 8.01 | 14.70 | 26.61 |
| **Avg. Edges** | 11.70 | 34.51 | 77.60 |

networks are hierarchical or tree-like, we extend BAG to `Barabási-Albert Forests(BAF)` by enforcing an acyclic topology. Moreover, our framework features `Scale-Free (SF) Graphs` generated via a degree-weighted random connection strategy, which can yield multiple disconnected components, offering a complementary perspective to BAG. A detailed description of each type of graph can be found in Appendix A.4, where we also provide the detailed rationale for this selection and empirical evidence showing that even within the 5–30 node range, the chosen generators yield statistically distinct and representative structural regimes in Appendix A.4.1.

**Prompt Schemes.** Recognizing that the formulation of query prompts critically influences LLM reasoning, our benchmark systematically evaluates **9 distinct prompt schemes** that vary in the degree of explicit guidance provided. The `k-Shot` prompts supply multiple exemplars from simpler graph instances to prime the model with relevant examples. The `Algorithm` prompts (Wang et al., 2023) explicitly delineate a well-known algorithm (such as BFS or Dijkstra), offering clear procedural instructions. In contrast, `Chain-of-Thought(CoT)` prompts (Wei et al., 2022) encourage the model to articulate intermediate reasoning steps, thereby exposing its internal thought process. The `Instruct` prompts use directive language tailored for instruction-based models to elicit focused responses. All three types above come with few-shot examples. For cases requiring minimal intervention, the `0-Shot` (i.e., plain) prompts pose bare questions without supplementary cues. To further investigate the impact of reasoning visibility, we also include variants without few-shot examples, such as `0-CoT`, `0-Instruct`, and `0-Algorithm`, which deliberately restrict the exposure of intermediate solution steps, as well as `LTM` prompts that employ least-to-most prompting. The detailed design process and some examples of the prompt program are shown in Appendix A.5.

**Serialization Formats.** Since LLMs operate on textual inputs, the method by which graphs are serialized has a profound impact on the clarity and accessibility of structural information. Our benchmark examines **7 distinct serialization formats** that offer varied representations of graph topology. The `Graph Modeling Language(GMoL)` provides a structured, tag-based representation that mirrors hierarchical data organization. In contrast, the `Adjacency Set` and `Edge Set` formats offer succinct listings of node neighbors and edges, respectively, emphasizing compactness. The `Edge List` format, which may incorporate additional details such as edge weights, serves as a more verbose alternative. Moreover, the `Adjacency Matrix` and `Adjacency List` formats balance detail and conciseness differently depending on the graph density, and the `Graph Markup Language(GMaL)` (Brandes et al., 2013) is an XML-based file format used to describe graph structures, including nodes, edges, and their attributes. Specific examples of graph serialization formats are in Appendix A.6.

## 3 EXPERIMENTAL SETTING

We evaluate the graph reasoning capabilities of various LLMs on a diverse set of tasks and difficulty levels. Our protocol highlights the impact of different dimensions in Section 2 on model performance.

**Random Baselines.** To assess the intrinsic graph reasoning ability of our models, we include a random baseline for each task. Appendix A.2 shows its detailed design process. These baselines provide a clear reference point for evaluating how much the LLMs improve upon chance performance when reasoning about graph-theoretic properties expressed in natural language.

**Models and Configurations.** We evaluate a diverse suite of LLMs spanning both open-source and closed-source categories. Our open-source models include `Llama-3`, `Llama-3.1`, `Mistral`, `Phi-4`, `Qwen-2.5`, and `Qwen-3`, while our closed-source models consist of `Claude-3.5`, `Gemini-2.0`, `GPT-4o`, `GPT-4o-mini` and `o4-mini` (versions and sources of the LLMs applied can be found in Appendix A.1). The model selection here is designed to provide coverage of the widely used LLMs of different sizes, reasoning types, and whether they are open-sourced or not, based on the budget and availability of models at the time of the work. We also try our best to include models with better performance on GRAPHOMNI than comparable alternatives to make our conclusion convincing. In all experiments related to few-shot examples, five exemplars are prepended to the prompt (i.e., k=5). More implementation details can be found in Appendix A.

**Evaluation Metrics.** Evaluation of LLM responses is conducted using predefined binary accuracy metrics, assigning an output of 1 for correct responses and 0 for incorrect responses. For qualitative tasks, such as `Connectivity` verification and `Cycle detection`, correctness is determined by identifying and verifying key phrases in the model's output (e.g., "*yes, there is a cycle*" or "*yes, there*

*is a path*") against the ground truth (GT). For numerical tasks, such as `Triangle counting` and `Diameter calculation`, correctness is assessed by extracting numerical values that follow specific key phrases (e.g., "*the number of triangles is*" or "*the diameter is*") and directly comparing these numerical outputs to the corresponding ground truth values. For tasks with multiple valid solutions, specifically `BFS order` and `Shortest path`, evaluation is conducted using a rule-based function. This evaluation process involves identifying key phrases, such as *"The BFS traversal starting from node X is"* or *"The shortest path from node X to node Y is,"* to extract the model's response. Based on this extraction, we evaluate the model's response using a task-specific rule-based algorithm that verifies solutions for tasks and assigns a score of 1 when the response matches one of the correct answers. The detailed rationale for the choice of the metrics is included in Appendix C.6.

Table 3: Benchmark Results of Representative LLMs Across Tasks (Mean±95% CI Margin). **Bold orange** / Underlined blue / Light purple highlights indicate best/second-best/third-best performance in its category. The complete results are included in Table 13.

| Task | Difficulty | Open-source Models | | | | | | Claude-3.5 | Closed-source Models | | o4-mini | Random |
|---|---|---|---|---|---|---|---|---|---|---|---|---|
| | | Llama-3.1 (8B) | Mistral (7B) | Phi-4 (14B) | Qwen-2.5 (72B) | Qwen-2.5 (7B) | Qwen-3 (8B) | | GPT-4o | Gemini-2.0 | | |
| BFS order | E | 18.69±3.02 | 13.75±1.44 | 33.03±7.32 | 71.41±3.45 | 21.46±4.26 | 65.87±5.59 | 91.42±1.65 | 81.48±3.22 | 90.31±2.30 | 95.46±0.78 | 0.00 |
| | M | 5.27±0.93 | 3.36±0.44 | 12.49±3.24 | 47.82±5.30 | 6.05±1.41 | 53.30±5.42 | 68.25±2.96 | 55.07±4.50 | 68.40±3.95 | 79.37±2.08 | 0.00 |
| | H | 0.63±0.19 | 0.34±0.14 | 2.65±0.80 | 22.03±4.39 | 1.38±0.37 | 29.53±4.25 | 26.80±2.64 | 21.58±3.69 | 27.77±3.34 | 32.45±3.88 | 0.00 |
| Connectivity | E | 79.53±2.03 | 79.90±1.89 | 56.29±8.58 | 90.24±1.89 | 88.10±1.46 | 97.17±1.29 | 98.38±0.60 | 95.63±1.30 | 92.61±1.42 | 98.23±0.63 | 67.49 |
| | M | 79.47±2.00 | 80.60±1.92 | 54.38±7.99 | 89.68±1.56 | 87.23±1.60 | 96.87±1.16 | 99.11±0.39 | 95.12±1.37 | 93.60±1.10 | 98.72±0.52 | 70.75 |
| | H | 74.58±2.67 | 74.77±2.46 | 48.39±7.50 | 84.09±1.98 | 81.19±2.02 | 92.89±2.07 | 96.99±1.48 | 90.59±2.19 | 87.99±1.67 | 92.02±3.99 | 66.36 |
| Cycle | E | 55.49±0.90 | 55.44±0.96 | 45.25±5.90 | 74.02±3.34 | 62.19±1.85 | 90.30±2.33 | 82.56±3.89 | 85.08±2.27 | 62.30±3.32 | 97.97±0.71 | 50.00 |
| | M | 55.69±1.08 | 53.71±0.72 | 44.26±5.43 | 71.99±3.34 | 62.07±1.80 | 89.66±2.07 | 80.80±3.94 | 85.35±2.30 | 60.29±3.22 | 97.75±0.76 | 50.00 |
| | H | 52.40±1.47 | 51.64±1.02 | 40.64±4.97 | 68.40±2.73 | 58.88±2.14 | 86.81±2.27 | 80.10±3.97 | 82.96±2.55 | 58.30±2.80 | 95.61±1.23 | 50.00 |
| Diameter | E | 41.27±5.37 | 28.55±4.28 | 42.81±5.06 | 78.50±1.16 | 45.08±4.17 | 77.56±2.77 | 83.71±1.26 | 63.99±2.19 | 79.14±1.94 | 98.88±0.15 | 11.20 |
| | M | 27.29±4.20 | 15.17±2.57 | 28.49±4.09 | 52.32±2.00 | 27.31±3.16 | 61.71±2.28 | 71.22±1.30 | 52.64±3.05 | 49.52±2.14 | 72.84±1.82 | 6.70 |
| | H | 18.63±3.27 | 6.97±1.26 | 17.71±3.02 | 29.59±2.48 | 15.27±2.47 | 39.83±2.67 | 56.70±2.02 | 45.60±3.24 | 23.45±2.97 | 34.61±2.84 | 3.72 |
| Shortest | E | 38.75±5.81 | 31.18±4.43 | 42.61±8.88 | 90.03±2.27 | 47.46±8.76 | 77.69±5.17 | 94.35±2.93 | 92.17±1.91 | 81.75±4.70 | 95.08±3.06 | 50.00 |
| | M | 28.84±4.56 | 19.89±3.05 | 33.92±7.68 | 81.17±3.03 | 35.53±6.80 | 69.60±5.50 | 91.27±2.84 | 84.84±2.93 | 80.67±4.15 | 92.60±3.49 | 50.00 |
| | H | 23.03±3.85 | 12.21±1.95 | 26.60±6.26 | 72.53±4.29 | 28.31±5.50 | 64.28±5.60 | 87.88±3.36 | 74.98±4.17 | 78.16±4.55 | 88.63±4.44 | 50.00 |
| Triangle | E | 14.97±1.53 | 11.87±1.32 | 12.88±2.05 | 36.57±4.40 | 18.56±1.24 | 41.36±4.63 | 43.41±1.64 | 36.32±1.54 | 50.33±2.31 | 84.54±0.56 | 2.13 |
| | M | 8.56±0.92 | 5.86±0.73 | 7.54±1.33 | 14.52±2.63 | 9.18±0.73 | 26.95±2.44 | 24.00±0.77 | 20.00±0.72 | 28.12±1.65 | 48.13±1.46 | 1.62 |
| | H | 4.95±0.69 | 2.55±0.44 | 4.38±1.04 | 4.73±1.58 | 4.45±0.58 | 19.54±1.34 | 15.92±0.72 | 12.81±0.88 | 15.55±1.29 | 17.53±1.43 | 1.82 |

## 4 RESULTS AND ANALYSIS

### 4.1 MAIN RESULTS

We evaluate model performance comprehensively across four main dimensions: model overall capability, graph type, effectiveness of prompting strategy, and impact of serialization format. This multifaceted evaluation offers a comprehensive understanding of the most effective approaches for graph algorithm tasks. Our analysis systematically considers each task at varying difficulty levels (*easy/medium/hard*). To isolate each dimension, we control for other variables when assessing a particular aspect and calculate the mean accuracy with a 95% confidence interval (Mean±95% CI Margin) across all combinations of the remaining factors. For example, when evaluating model capability, we compute statistics across all combinations of graph types, prompts, and serialization formats while holding the model constant. The evaluation results from the model capability perspective are presented in Table 3 and Figure 2. To provide a comprehensive view, we present additional experimental results in Appendix E.1 examining prompt schemes, serialization formats, and graph types across collective results (Tables 14, 15, 16), open-source models (Tables 17, 18, 19), and closed-source models (Tables 20, 21, 22). These controlled evaluations yield complementary insights summarized across multiple perspectives. Additionally, example input/output pairs are provided for clarity in Appendix E.5.

**Result ❶: High variability underscores the need for comprehensive evaluation across all benchmarking dimensions.** Detailed analysis reveals substantial variability in LLM performance across different combinations of serialization formats, prompting schemes, and graph types. This variability highlights the need for a comprehensive evaluation across all benchmarking dimensions. The performance heatmaps, presented in Appendix E.2, illustrate the accuracy of different prompt schemes and serialization formats across tasks, models, and difficulty levels. The performance heatmaps show that no single serialization or prompting strategy consistently outperforms others across all tasks and difficulty levels. Instead, optimal results require careful and adaptive selection of serialization-prompt combinations, explicitly tailored to task characteristics such as structured graph-theoretic reasoning tasks. For instance, in the case of GPT-4o, depicted in Figure 4, accuracy gaps of up to 40% occur when varying input representations within the same task and model, indicating a

significant sensitivity to input formatting, which is also observed in other domains, like evaluation of vision language models (VLMs) (Feizi et al., 2025). These observations emphasize that evaluating LLMs comprehensively across interconnected dimensions, i.e., serialization formats and prompting schemes, is essential for fairly assessing their capabilities in graph reasoning tasks.

**Result ❷: Model performance still has considerable room for improvement.** Models generally demonstrate reasonable performance across tasks, underscoring their inherent potential in graph reasoning when appropriately guided. Notably, o4-mini delivers remarkable performance, frequently surpassing other closed-source models across most tasks and setting a new benchmark overall. However, the performance gap remains large on the *hard* difficulty tasks, particularly BFS order, Diameter calculation, and Triangle counting, which require full, global information of the graph. Here, even o4-mini's performance drops to as low as 32.45% on BFS order (*Hard*), 34.61% on Diameter calculation (*Hard*), and 17.53% on Triangle counting (*Hard*), underscoring the remaining challenge in holistic graph reasoning. Therefore, substantial room for improvement persists relative to ideal human-level outcomes, primarily due to the scarcity of structured graph-theoretic content in typical web corpora used for LLM pretraining. Among open-source alternatives, Qwen-3 remains the top performer but continues to lag behind leading closed-source models, such as o4-mini and Claude-3.5, suggesting a meaningful room for advancement in open-source solutions.

**Result ❸: Common Errors Reveal Fundamental Gaps in Graph Reasoning.** Our error analysis highlights representative categories of errors commonly observed in incorrect LLM responses: **A. Misinterpretation of serialization formats:** Models occasionally struggled to accurately interpret serialized graph representations, resulting in misunderstandings of the underlying graph structure, such as BFS order case 1, Connectivity case 1, and Triangle counting case 2 in the Appendix; **B. Incorrect reasoning about graph-theoretic concepts:** LLMs frequently exhibited fundamental misunderstandings of basic graph definitions and problem-solving methods. In the error cases Triangle counting case 1, incorrect responses inaccurately estimated the number of triangles as approximately one-third of the number of nodes. For the error cases Diameter calculation case 1, some models erroneously identified the diameter as the length of the longest path, rather than correctly defining it as the length of the longest shortest path between any two nodes. These representative errors underscore critical areas for improvement in the graph reasoning capabilities of current LLMs. Additional error cases and analyses are provided in Appendix E.4.

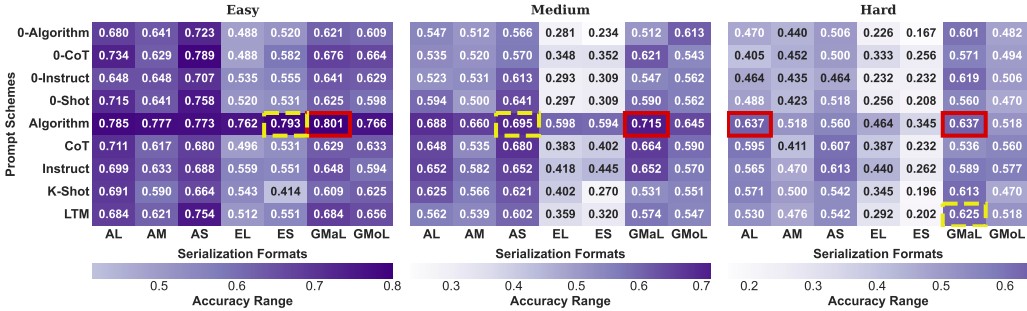

Figure 4: **Performance heatmaps for different prompt schemes and serialization formats on Diameter calculation of GPT-4o**. The color intensity represents the accuracy, with darker colors indicating better performance. The red solid and yellow dashed line indicates Best and Second Best Performance, respectively.

## 4.2 Fine-Grained Empirical Findings on Model Performance

In this section, we dive deeper into our empirical results, identifying detailed performance patterns and revealing nuanced interactions across various evaluation dimensions. We present here the two most critical findings, while additional observations are available in Appendix E.6.

**Finding ❶: Domain-specific knowledge significantly improves model performance on graph-theoretic tasks.** Algorithm-based prompts, explicitly detailing graph-theoretic algorithms, consistently improved model accuracy in structured reasoning tasks such as BFS order and Diameter calculation (Table 14). This result highlights the value of incorporating explicit domain knowledge into prompts, particularly when tasks require step-by-step algorithmic reasoning. From Diameter

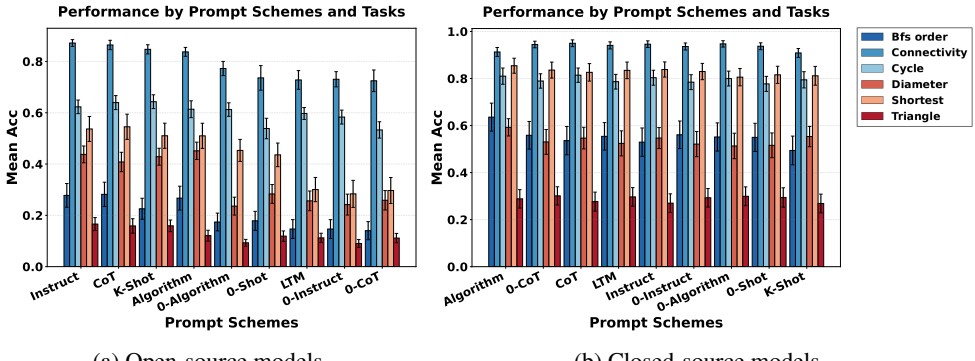

(a) Open-source models.  (b) Closed-source models.

Figure 5: **Accuracy of open-source versus closed-source models with different prompt schemes.** (a) and (b) show the average performance with a 95% confidence interval for open/closed-source models across various prompt schemes and tasks, with $x$-axis sorted by mean accuracy.

calculation case 1 and Triangle counting case 1, it shows that when employing plain prompts, the LLM's response does not accurately reflect the appropriate method for solving the relevant task.

**Finding ❷: Scaling raises the floor, while reasoning lifts the ceiling.** A targeted comparison of Qwen-2.5 (7B), Qwen-2.5 (72B), and Qwen-3 (8B) (Table 11) highlights complementary effects. Scaling within the same family (7B to 72B) yields consistent improvements on easier tasks and splits, such as BFS order, Shortest path, and Diameter calculation (*Easy/Medium*). By contrast, a reasoning model at a comparable size, i.e., Qwen-3 (8B), delivers larger gains on the hardest regimes that require multi-step exploration and combinatorial checks, including BFS order, Diameter calculation, and Triangle counting (*Hard*). Together, these results indicate that scaling predominantly improves robustness on simpler instances, while reasoning-centric design is more effective for pushing the upper bound of graph reasoning ability (details in Appendix C.5).

**Finding ❸: Divergent impacts of prompt schemes – Open-source models benefit from multi-shot exemplars, whereas they do not help closed-source models much.** In Figures 5a and 5b, the open-source model achieves the highest average accuracy with prompt schemes that incorporate shots. However, for the closed-source model, prompt schemes show more complexity. Only considering prompt patterns, 0-CoT performs second to best, 0-Algorithm worst, but both surpass k-shot. However, adding shots improves Algorithm's overall accuracy, suggesting that shots enhance the model's understanding of Algorithm-based prompts. Yet, this effect is not universal: shots may hinder comprehension in particularly challenging tasks, as noted in Finding ❺ Appendix E.6.

## 4.3 EXTENDED STUDY AND DISCUSSION

**Scaling to Larger Graphs (Beyond 30 Nodes).** We extend the evaluation to graphs with 30–50 nodes, sampling 50 graphs per generator and ∼3k test cases overall (details in Appendix C.1). As the results in Table 8 show, the performance degrades as graph size increases, particularly for tasks with sequential or combinatorial requirements: accuracy on BFS order and Triangle counting drops sharply, reflecting the added difficulty of maintaining frontiers or enumerating subgraphs over longer horizons. By contrast, tasks such as Connectivity and Cycle detection remain relatively stable, consistent with their reliance on local connectivity checks. Importantly, despite the absolute drop in scores, the **relative ranking of models and the performance gap between open- and closed-source systems remain nearly identical to the 5–30 node Hard split**, confirming that the benchmark's conclusions are robust under further scaling of graph size.

**Representative Check on Real-World Graphs.** We further test whether our synthetic setup transfers to real data by evaluating on two representative domains: IMDB-MULTI (social/interaction) and ogbg-molhiv (molecular), yielding ∼3.6k samples across six tasks (details in Appendix C.2). Results in Table 9 corroborate our findings: (i) Connectivity and Cycle detection are consistently easiest; (ii) ordered-path tasks (BFS order, Shortest path, Diameter calculation) remain substantially harder, dominated by serialization and memory errors; and (iii) Triangle counting is the most challenging. However, because many public graphs are sparse and connected, specific tasks become easier than in our synthetic regime (e.g., Connectivity saturates near 100% for strong models). This shows that **real-world graphs alone can under-stress graph reasoning**. Together with prior works

that adopt synthetic-only designs (Fatemi et al., 2024; Chen et al., 2024a; Luo et al., 2024b), our results validate real graphs as a sound check, but reaffirm that synthetic graphs provide a systematic evaluation with balanced structural coverage, controllability, and contamination-free conditions. The detailed rationale is elaborated in Appendix C.3.

**Exploration on NP-Hard Tasks.** As a complementary stress test, we also consider two classical NP-hard problems, Hamiltonian cycle detection and Max-Cut (details in Appendix C.4). Results in Table 10 show accuracy patterns aligned with our six canonical tasks: open-source models remain near random, while closed-source reasoning-oriented models attain noticeably higher but still imperfect scores. This indicates that the core conclusions of GRAPHOMNI naturally extend to NP-hard problems. Interestingly, however, LLMs do not exhibit the same graded difficulty separation between polynomial-time and NP-hard tasks as human solvers: accuracy tends to collapse uniformly across NP-hard regimes just like polynomial tasks. Thus, while useful as a complementary check, NP-hard tasks do not add progressive challenge in the same way as our tractable yet demanding suite, reinforcing why the latter remain the centerpiece of GRAPHOMNI.

**Efficiency–accuracy trade-off.** Besides accuracy, we also analyze inference efficiency by measuring the number of output tokens produced across models (details in Appendix E.7). The results reveal a clear trade-off: accuracy gains often come at the cost of longer responses, but models navigate this balance differently. Closed-source models (e.g., GPT-4o, Claude-3.5) reach high accuracy with compact generations under 300 tokens, while `o4-mini` relies on very long chains of thought (over 1.6K tokens) to achieve similar accuracy (Figure 32). By contrast, open-source models such as Llama-3.1 and Qwen-2.5 (7B) must generate substantially longer outputs to achieve high performance, whereas shorter responses are correlated with lower accuracy. These trends persist across difficulty levels, task types, serialization formats, and prompt schemes (Tables 23–26). Overall, efficiency, measured by output length, emerges as an additional axis of divergence across LLMs, reinforcing the importance of evaluating not only correctness but also the cost of achieving it.

## 4.4 REINFORCEMENT LEARNING (RL)-BASED PROMPT SEARCH INSPIRED BY GRAPHOMNI

Our benchmark evaluates three key dimensions, *graph type, serialization format*, and *prompt scheme*, to underscore the critical role of transforming graph structures into textual inputs for LLM inference. While GRAPHOMNIprovides comprehensive insights into how different dimensions affect LLM inference, we still face a concrete, actionable question: Given many interacting dimensions, which configuration is best for a specific graph reasoning task? In this section, we want to identify the optimal combination strategies (serialization format; prompt scheme, etc.) that enhance the effectiveness of textual representations, thereby improving LLM performance in graph reasoning and understanding tasks. We define the process of converting graph structures into textual inputs tailored to a specific task as the **serialization process**. To operationalize this serialization process, we introduce an RL-based search method as a diagnostic tool within our benchmark, enabling automatic selection of effective serialization strategies.

Specifically, RL transforms optimizing the serialization process into a sequential decision-making problem for each type and difficulty of the task. There are $T$ decision epochs, and each decision epoch determines one component of the serialization strategy. Then we provide a predetermined order to specify a sequence of action spaces $\{\mathcal{A}_t\}_{t=1,\ldots,T}$ (e.g., $\mathcal{A}_t$ can be all candidate prompts). We set the initial state $s_0$ as the specific type and difficulty of the task. Then at decision epoch $t = 1, \ldots, T$, we choose an action $a_t \in \mathcal{A}_t$ based on the previous actions $a_1, \ldots, a_{t-1}$. Then the state $s_t$ consists of the task type and difficulty (initial state $s_0$) together with the previously selected serialization components. This corresponds to a policy $\pi_t : \mathcal{S}_0 \times \mathcal{A}_1 \times \cdots \times \mathcal{A}_{t-1} \mapsto \mathcal{A}_t$, where $\mathcal{S}_0$ is the state space of the initial state $s_0$. For any instance $s$ (e.g., a query for `Connectivity` task in easy mode for a specific graph), a binary reward, denoted by $r(s, a_1, \ldots, a_T)$, is incurred at the end of the decision epoch, which is set to 1 if the LLM correctly answers the specific query under the selected serialization strategy $(a_1, \ldots, a_T)$ and to 0 otherwise. For each type and difficulty of the task, our objective is to maximize the expected reward of choosing the serialization strategy $a_1, \ldots, a_T$:

$$\max_{\{\pi_t\}_{t=1,\ldots,T}} \mathbb{E}[r(s, a_1, \ldots, a_T)|s_0],$$

where the expectation is taken with respect to the problem instance $s$ and the (random) answer output by an LLM (affected by the randomness of the LLM, e.g., the temperature parameter). Note that (i) $s_0$ is part of the instance information $s$, and (ii) we fix the type and difficulty of the task, and the

only randomness in terms of $s$ is from graph generation. To approximate this objective function, we generate $N$ different graphs for each type of query. We assess the performance of RL using the average reward across the $N$ graphs, which essentially is the accuracy of the serialization strategy for a specific graph-related task across these $N$ graphs.

Consider the problem of dealing with high-dimensional, complex state spaces in the serialization process, we employ the Deep $Q$-Network (DQN) (Mnih et al., 2013) to implement RL, which employs a neural network as a function approximator for the $Q$-function. Specifically, we use a neural network $\widehat{Q}_t(s_0, a_1, \ldots, a_t; \theta_t)$ parameterized by $\theta_t$ to approximate the corresponding $Q_t(s_0, a_1, \ldots, a_t)$ for the actions or factors considered in serialization process. Each $Q$-network is modeled as a three-layer multilayer perceptron with ReLU activations. Training minimizes the mean squared error loss, and action selection follows an $\epsilon$-greedy policy, where $\epsilon$ linearly decays from 1.0 to a minimum of 0.01. Then we design the **RL-Opt** (RL-guided Optimal Serialization Selection) experiment, leveraging existing benchmark data to apply RL to evaluate computational cost and validate the effectiveness of the derived optimal strategy. Additionally, we introduce the **RL-Scale** (RL Scalability in Serialization Expansion) experiment to analyze how RL's computational cost scales when incorporating additional factors in the serialization process. All detailed information can be found in Appendix D.

In **RL-Opt**, the serialization process involved three key factors based on our benchmark's results: serialization format, prompt scheme, and the choice of open-source language models. To evaluate the effectiveness of RL in identifying the optimal combination, we employ two key metrics: Cost and Rate. To evaluate RL's effectiveness in finding the optimal combination, we use two metrics: (a) Cost is the ratio of explored combinations: Cost $= \frac{k}{K}$, where $k$ is the number of explored combinations, and $K$ is the total number of combinations; (b) Rate $= \frac{\text{acc}_*}{\text{acc}_{\max}}$, where $\text{acc}_*$ is the accuracy of RL's best-found combination and $\text{acc}_{\max}$ is the highest accuracy in the benchmark data. Results are in Table 4. The results demonstrate that, at only 25% of the original cost, the RL-based method maintains an average success rate of 0.9, indicating its ability to significantly reduce the time required to search for optimal combinations while preserving the quality of the outcomes.

Table 4: **Performance summary of RL-Opt**, averaged across all instances of a specific experimental case, reducing the cost to about 25% of the original, maintaining an average success rate of 0.9.

| Task | Mode | Avg Cost | Avg Rate | Task | Mode | Avg Cost | Avg Rate |
|---|---|---|---|---|---|---|---|
| BFS order | Easy | 0.2203 | 0.9740 | Connectivity | Easy | 0.2244 | 0.9883 |
| | Medium | 0.2251 | 0.9045 | | Medium | 0.2263 | 0.9875 |
| | Hard | 0.2279 | 0.7812 | | Hard | 0.2238 | 0.9871 |
| Cycle | Easy | 0.2229 | 0.9757 | Diameter | Easy | 0.2263 | 0.9728 |
| | Medium | 0.2263 | 0.9833 | | Medium | 0.2181 | 0.9541 |
| | Hard | 0.2203 | 0.9584 | | Hard | 0.2235 | 0.9471 |
| Shortest path | Easy | 0.2244 | 0.9636 | Triangle | Easy | 0.2276 | 0.9061 |
| | Medium | 0.2159 | 0.9856 | | Medium | 0.2206 | 0.8456 |
| | Hard | 0.2187 | 0.9073 | | Hard | 0.2235 | 0.7321 |

## 5 CONCLUSION

We introduced GRAPHOMNI, a comprehensive benchmark framework for systematically evaluating the graph reasoning capabilities of LLMs. By analyzing critical dimensions, including graph types, serialization formats, and prompt schemes, we provided extensive insights into the strengths and limitations of current LLMs. Our empirical findings emphasize that no single serialization or prompting strategy consistently outperforms others. Motivated by these insights, we propose a reinforcement learning-based approach that dynamically selects the optimal serialization-prompt pairings, leading to significant improvements in accuracy. GRAPHOMNI's modular and extensible design establishes a robust foundation for future research, facilitating advances toward general-purpose graph reasoning models.

**Ethics Statement**    We confirm that this research complies with all applicable ethical guidelines and poses no ethical issues.

**Reproducibility Statement**    We have taken extensive measures to ensure the reproducibility of our work. The source code and data resources are released at https://gai-community.github.io/Graph-Omni/.

Our experimental setup, including model configurations and evaluation protocols, is fully described in Section 3 and Appendix A. For transparency, we provide comprehensive coverage of input-output examples (Appendix E.5) and error cases (Appendix E.4), enabling a thorough understanding and verification of the reported results.

Together, these resources support faithful reproduction and further exploration of our findings.

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

# Table of Contents for Appendix

# A EXPERIMENTAL DETAILS

## A.1 LLM VERSIONS

Table 5 provides an overview of the diverse suite of large language models (LLMs) evaluated in our study. Open-source models are hyperlinked to their respective documentation, while closed-source models are identified by their version numbers. Note that we only uniformly sample 25% of data when evaluating Qwen-3 due to the limited time after its release, so its result will be only included in the model-wise statistics, i.e. Table 3 for refernece.

Table 5: Overview of evaluated LLMs. Open-source models are linked, while closed-source models list their version.

| Model | Model Link/Version |
|---|---|
| Llama-3 | Meta-Llama-3-8B (Link) |
| Llama-3.1 | Llama-3.1-8B (Link) |
| Mistral | Mistral-7B-v0.3 (Link) |
| Phi-4 | Phi-4-14B (Link) |
| Qwen-2.5 (7B) | Qwen-2.5-7B-Instruct (Link) |
| Qwen-2.5 (72B) | Qwen-2.5-72B-Instruct (Link) |
| Qwen-3 (8B) | Qwen-3-8B (Link) |
| Claude-3.5 | claude-3-5-sonnet-20241022 |
| Gemini-2.0 | gemini-2.0-flash-001 (Version 1) |
| GPT-4o | gpt-4o-2024-08-06 |
| GPT-4o-mini | gpt-4o-mini-2024-07-18 |
| o4-mini | o4-mini-2025-04-16 |

## A.2 PARAMETER AND RANDOM BASELINE SETTINGS

**Parameter setting.** We have studied various methods of representing graphs as text based on a diverse set of basic graph problems. This appendix details the parameter setting and the design of the graph input text. For the parameter setting, the temperature is set to 0.7, following the parameter selection in Wang et al. (2023). The nucleus sampling (top-p) is set to 0.9 for open-source models, while for closed-source models, the default top-p value is used.

**Random Baselines setting.** For Cycle detection, the random baseline simply selects an answer from {True, False}—yielding an expected accuracy of 50%. Since the GT obtained through the design function has a high proportion of True labels, we iterate through all queries, assuming the given answer is True. We then use GT for evaluation, leading to the final baseline based on this assumption. For tasks that require generating numerical outputs (e.g., Diameter calculation and Triangle counting), the random baseline corresponds to randomly choosing one of the valid numerical solutions derived from the graph's structure. For the Diameter calculation task, the random baseline is determined based on the number of nodes in the graph for each query. Specifically, we sample a random integer from the range $[1, N]$, where $N$ is the number of nodes in the graph, and compare it with the ground truth to compute the baseline performance. For the Triangle counting task, the random baseline is derived from the estimated upper bound on the number of triangles in the graph. We compute the maximum possible number of triangles based on the number of nodes and the task difficulty level, take the smaller value between these estimates, and sample a random integer from the range $[1, M]$, where $M$ is the determined upper bound. The sampled value is then compared against the ground truth to obtain the random baseline performance. In contrast, for tasks that require generating sequences (e.g., BFS order), the number of possible combinations is combinatorially large, so a random baseline would yield an accuracy that is approximately 0%.

## A.3 GRAPH TASKS

We conducted a comprehensive study on a diverse set of fundamental graph problems, including `BFS order`, `Cycle detection`, `Connectivity`, `Diameter calculation`, `Shortest path`, and `Triangle counting`. The input text for each task is provided below, where the italicized variables *X, Y* denote generic node numbers corresponding to the specific problem under consideration.

> **Graph Tasks**
>
> - **BFS-ORDER**: Give the bfs traversal order starting from node *X*.
> - **CYCLE**: Is there a cycle in this graph?
> - **CONNECTIVITY**: Is there a path between node *X* and node *Y*?
> - **DIAMETER**: What is the diameter of this graph?
> - **SHORTEST PATH**: Give the shortest path from node *X* to node *Y*.
> - **TRIANGLE**: How many triangles are in this graph?

### A.3.1 RATIONALE FOR SELECTION OF TASKS

The six core tasks in GRAPHOMNI are deliberately selected to span qualitatively different reasoning capacities. Their difficulty increases as the model must move from local checks to global traversals, maintain more intermediate states in working memory, or perform exhaustive combinatorial enumeration. Beyond **reasoning capacities**, variation also arises from how well LLMs internalize **task definitions** and from the complexity of **output formats**. Together, these factors explain the accuracy gaps observed in Table 3 and highlight why the chosen tasks form a balanced and challenging suite.

**Aspect 1: Reasoning capacities required.** These tasks are grouped according to the type and depth of reasoning they demand, ranging from simple global checks to multi-layered traversals and full combinatorial enumeration.

Here follows a detailed elaboration on these three aspects.

1. *Reachability verification (*`Connectivity`*,* `Cycle detection`*).* These tasks require a global traversal but only a simple decision condition, such as whether the graph is connected or whether a cycle is present. Most errors stem from serialization misunderstandings (e.g., assuming a missing edge exists, in Appendix E.4.3). Once the format is parsed correctly, accuracy is high.

2. *Ordered-path reasoning (*`BFS order`*,* `Shortest path`*,* `Diameter calculation`*).* These tasks demand that the model keep a frontier or distance map and then output or compare those ordered distances. For `BFS order`, the model must list nodes level-by-level. In the error case in Appendix E.4.7, failures occur when it forgets whether two previously visited nodes are connected. `Shortest path` and `Diameter calculation` add a final aggregation step: the former selects the minimum path, the latter the maximum among all shortest paths. The common mistakes are also mostly about losing track of some vital information while exploring the graph. Like the one in Appendix E.4.2 for `Diameter calculation`, the model forgets two important edges, so the path length is wrong. Accuracy here for those three tasks is lower than the first type of tasks because the model must track ordering information across multiple expansion layers.

3. *Combinatorial enumeration (*`Triangle counting`*).* `Triangle counting` is the most challenging: the model must evaluate every three-node subset and make sure each sub-traverse is correct. Even given correct execution of the enumeration, the counting should be accurate to produce the correct final result. Appendix E.4.6 and E.4.8 document the dominant errors on enumeration over each possible triangle in the graph (like missing an edge or wrongly assuming one). We also spotted cases that fail on the counting at the end, too. In sum, performance is strongest when only reachability is tested, drops when ordered path reasoning is required, and falls sharply when complete combinatorial enumeration comes into play.

**Aspect 2: Task understanding and definition knowledge.** LLMs sometimes rely on heuristics rather than precise textbook definitions, particularly for less common tasks. For example, some models

confuse diameter with the longest simple path, producing inflated results (Appendix E.4.1). Others apply shortcuts such as "triangles $\approx n/3$" (Appendix E.4.5), ignoring the need for all three edges to be present. Such misinterpretations highlight that accuracy depends not only on raw reasoning ability but also on task comprehension. Our coverage of tasks enable the evaluation on these knowledge of each model and it does reflect in the results as the error cases mentioned.

**Aspect 3: Output format.** The output formats of the tasks chosen are also very diverse. Some tasks here need only a short answer, i.e., "Yes/No" for `Connectivity` or a single number for `Triangle counting`, so there is little room for formatting errors. Meanwhile, `BFS order` is different: the model must print a long, strictly level-by-level list of node IDs, and one extra or missing node makes the whole response wrong. The coverage of different output formats brings challenges to the models.

In summary, these systematic differences validate that the GRAPHOMNI task suite probes diverse reasoning skills over graphs and exposes where current LLMs struggle most.

### A.4   GRAPH TYPES

A primary distinguishing aspect of our benchmark is the inclusion of multiple graph families, each possessing unique structural properties. All 7 types of graph are highlight in **bold**:

**1**. **Erdős–Rényi (ER)** Graphs are randomly sampled from the space of all possible graphs with $n$ vertices, making them well-suited for capturing a wide range of topological and connectivity properties within a fixed number of vertices.

To enhance the diversity of random graphs, we consider two sampling methods: $m$-edge sampling and probability-based sampling, referred to as **Erdős–Rényi M-Edges (ERM)** (Erdős & Rényi, 1960) and **Erdős–Rényi Probability (ERP)** (Gilbert, 1959) respectively.

- **ERM:** Generates graphs with $n$ vertices and a fixed number of edges $m$, where $m$ is randomly chosen between 1 and $\frac{n(n-1)}{2}$, ensuring that all possible edge counts are considered.
- **ERP:** Constructs graphs with $n$ vertices but an unfixed number of edges, where the edge probability is randomly sampled as a floating-point value between 0 and 1.

Additionally, we extend these models to bipartite settings:

- **Bipartite Erdős–Rényi M-Edges (BERM)** and **Bipartite Erdős–Rényi Probability (BERP)** graphs (Latapy et al., 2008) are generated using the ERM and ERP sampling strategies but constrained to bipartite structures.
- These bipartite graphs introduce additional variations in topology and connectivity that standard ERM and ERP graphs, which are inherently undirected, may not capture.

**2. Barabási–Albert Graphs (BAG)** (Albert & Barabási, 1999) exhibit a power-law degree distribution, where a small number of nodes (hubs) have significantly higher degrees, while most nodes have relatively few connections. Such structures frequently appear in real-world networks, including social and biological systems.

While ER graphs, being randomly sampled, may occasionally exhibit power-law degree distributions, BAGs explicitly model this phenomenon due to their practical prevalence.

- BAGs are constructed by starting with a complete graph of $m_0$ vertices and incrementally adding nodes.
- Each new node forms $m$ connections, where $m$ is proportional to the degrees of existing nodes (preferential attachment).
- In our dataset, $m_0$ is randomly sampled with an upper bound of $\frac{n}{3}$, and $m$ is set to $m_0 + 1$.

Although BAGs generally capture power-law degree distributions, they do not always represent tree-like structures such as citation networks or hierarchical systems. To address this, we introduce **Barabási–Albert Forests (BAF)** (Albert & Barabási, 1999), which follow the same generation process as BAGs but enforce an acyclic structure, ensuring that the result is a forest (a set of trees) rather than a single connected graph.

**3. Scale-Free (SF) Graphs** (Aiello et al., 2000) Another class of power-law networks that BAGs may not fully capture is general scale-free (SF) networks. While all BAGs are SF, not all SF graphs are BA.

- BAGs typically consist of a single connected component, whereas SF graphs can contain multiple disconnected components.
- To represent SF graphs more comprehensively, we introduce a distinct SF graph generation process, different from BAGs.

Unlike BAGs, which are constructed through incremental growth and preferential attachment, SF graphs are generated using a degree-weighted random connection strategy:

- All vertices are created at once.
- Edges are formed probabilistically, where the probability of a connection is proportional to node degrees.

These fundamental differences in growth dynamics and edge formation result in SF graphs and BAGs capturing distinct topological properties. By including both, we enhance the diversity of our dataset.

These families challenge LLMs to adapt their reasoning across numerous topological extremes, from sparse bipartite graphs to highly connected ones. Although future expansions may include small-world graphs or others, this current selection already covers a rich array of structural profiles as elaborated in the next section.

### A.4.1 RATIONALE FOR GENERATOR SELECTION

The seven generators in GRAPHOMNI are deliberately selected to provide the most comprehensive structural coverage possible within the 5–30 node range. Each generator encodes a distinct motif/structure observed in real-world networks, i.e. random connectivity, scale-free growth, bipartite affiliation, hierarchical trees or other tendencies, ensuring that the benchmark spans all major regimes of graph organization. Even at this scale, the underlying generative biases remain evident and produce meaningful differences in task difficulty and model behavior. By relying on controlled synthetic generators, GRAPHOMNI achieves balanced representation across families while isolating structural effects without the confounding noise of empirical data.

To be specific, the selected generators cover a wide range of canonical structures:

1. **Erdős–Rényi M-Edges (ERM) & Probability (ERP)**. Serve as canonical baselines for random connectivity, yielding binomial/Poisson degree distributions used extensively in the study of biological and technological networks.

2. **Bipartite ERM (BERM) & Bipartite ERP (BERP)**. Capture two-mode affiliation structures, such as author–paper and user–item systems, which exhibit realistic clustering and degree properties.

3. **Barabási–Albert Graphs (BAG)**. Model scale-free networks with hubs emerging via preferential attachment, mirroring the structure of the Internet, citation graphs, and social networks.

4. **Barabási–Albert Forests (BAF)**. A specialization of the BA process that produces acyclic scale-free trees, modeling hierarchical taxonomies such as phylogenies and organizational charts.

5. **Scale-Free (SF) Graphs**. Configuration-style models generate prescribed power-law degree sequences, often producing disconnected components akin to regional transport or communication subnetworks.

To further validate that these generators produce graphs with statistically distinct and meaningful properties, we conduct two empirical studies. First, we sample 1,000 graphs of 30 nodes each from the same Barabási–Rényi (BAG) and Erdős–Rényi (ERP) generators used in GRAPHOMNI. As summarized in Table 6, the two models exhibit clearly different structural characteristics: BA graphs form hubs with high maximum degree and short paths, while ER graphs display uniform randomness

with lower clustering and longer paths. Second, as shown in Table 7 in Appendix B.1, even when node counts are fixed, the edge counts (and thus average degrees) vary substantially across generators, providing strong statistical evidence that the structural characteristics of these graph families are fundamentally distinct. Together, these results confirm that the design of GRAPHOMNI captures the essential structural diversity needed to probe LLM reasoning.

Table 6: Comparison of structural statistics for 1,000 sampled graphs with 30 nodes. BAG graphs exhibit hub formation with high maximum degree and short paths, while ERP graphs display more uniform randomness.

| Type | Max Degree | Clustering Coefficient | Avg Path Length |
|------|-----------|------------------------|-----------------|
| Barabási–Albert (BAG) | $19.28 \pm 2.15$ | $0.397 \pm 0.042$ | $1.76 \pm 0.014$ |
| Erdős–Rényi (ERP) | $10.43 \pm 1.35$ | $0.199 \pm 0.044$ | $2.07 \pm 0.099$ |

## A.5 PROMPT SCHEMES

The process of converting a graph into a textual representation is referred to as the serialization process, which involves two primary considerations in our study: the choice of serialization format and the selection of the prompting method. we employ a total of nine distinct prompting methods: `Algorithm`, `CoT`, `k-shot`, `Instruct`, `0-Shot`(i.e. plain), `0-CoT`, `0-Instruct`, `0-Algorithm`, and `LTM`. As outlined in the main text, the pairs `Algorithm` and `0-Algorithm`, `CoT` and `0-CoT`, `k-shot` and `0-Shot`, and `Instruct` and `0-Instruct` share a common structural format, with the first element in each pair incorporating additional 5 examples. A detailed description of the design for each of these prompting methods is provided below. In particular, for the algorithmic description components of `Algorithm` and `0-Algorithm`, we primarily draw upon established methodologies in Wang et al. (2023) and illustrate them with an example derived from the BFS-order task.

> **Prompt format**
>
> - **0-COT**: Let's think step by step:
> - **LTM**: Let's break down this problem:
> - **0-INSTRUCT**: Let's construct a graph with the nodes and edges first:
> - **0-ALGORITHM**: To determine the BFS (Breadth-First Search) traversal order, you need to follow these steps: 1. Initialize: Start by choosing a starting node and enqueue it into a queue. 2. Mark visited: Mark the starting node as visited to avoid reprocessing. 3. Traverse: While the queue is not empty: Dequeue a node and add it to the traversal order. For each unvisited neighboring node of the dequeued node, enqueue it and mark it as visited. 4.Continue the process until all reachable nodes are visited.

## A.6 SERIALIZATION FORMATS

This study utilizes seven distinct yet commonly used graph representation formats: `Adjacency Matrix`, `Adjacency List`, `Adjacency Set`, `Edge Set`, `Edge List`, `Graph Modeling Language (GMoL)`, and `Graph Markup Language (GMaL)`. For the same graph, even when the underlying information remains consistent, the representation varies across different serialization formats in textual form. The following section presents specific examples of the same graph depicted in various serialization formats.

> **Adjacency Set**
>
> `{0: {1}, 1: {0, 2}, 2: {1}, 3: {4}, 4: {3, 5}, 5: {4}}`

> **Edge Set**
>
> `{(0, 1), (4, 5), (1, 2), (3, 4)}`

**Edge List**

```
0 1
1 2
3 4
4 5
```

**Adjacency Matrix**

```
[[0 1 0 0 0 0]
 [1 0 1 0 0 0]
 [0 1 0 0 0 0]
 [0 0 0 0 1 0]
 [0 0 0 1 0 1]
 [0 0 0 0 1 0]]
```

**Adjacency List**

```
{0: [1], 1: [0, 2], 2: [1], 3: [4], 4: [3, 5], 5: [4]}
```

**GMaL**

```
<?xml version='1.0' encoding='utf-8'?>
<GMaL xmlns="http://GMaL.graphdrawing.org/xmlns"
        xmlns:xsi="http://www.w3.org/2001/XMLSchema-instance"
        xsi:schemaLocation="http://GMaL.graphdrawing.org/xmlns
        http://GMaL.graphdrawing.org/xmlns/1.0/GMaL.xsd">
  <graph edgedefault="undirected">
    <node id="0" />
    <node id="1" />
    <node id="2" />
    <node id="3" />
    <node id="4" />
    <node id="5" />
    <edge source="0" target="1" />
    <edge source="1" target="2" />
    <edge source="3" target="4" />
    <edge source="4" target="5" />
  </graph>
</GMaL>
```

**GMoL**

```
graph [
  node [
    id 0
    label "0"
  ]
  node [
    id 1
    label "1"
  ]
  node [
    id 2
    label "2"
  ]
  node [
    id 3
    label "3"
  ]
  node [
    id 4
    label "4"
  ]
  node [
    id 5
    label "5"
  ]
  edge [
    source 0
    target 1
  ]
  edge [
    source 1
    target 2
  ]
  edge [
    source 3
    target 4
  ]
  edge [
    source 4
    target 5
  ]
]
```

## A.7 DATA EXAMPLES

In order to better show the input example, we select the `BFS order` task in the serialization format is the Adjacency List of the complete prompt example, due to space reasons, the middle of the excessively long part we will use "...". Each of the following examples is randomly selected from the source data.

**0-Shot**

Given a graph, your task is to determine the bfs traversal order of this graph starting at node 4. And the graph representation of: Adjacency List is {0: [1], 1: [0, 2, 3, 5, 6], 2: [1, 4], 3: [1], 4: [2], 5: [1, 7], 6: [1], 7: [5, 8], 8: [7]}
Q: Give the bfs traversal order starting from node 4.
A:

**0-CoT**

Given a graph, your task is to determine the bfs traversal order of this graph starting at node 7. And the graph representation of: Adjacency List is {1: [0, 2], 0: [1, 3, 4, 5, 6], 2: [1], 3: [0], 4: [0, 8], 5: [0, 7], 6: [0], 7: [5], 8: [4]}
Q: Give the bfs traversal order starting from node 7.
A:
Let's think step by step:

**0-Instruct**

Given a graph, your task is to determine the bfs traversal order of this graph starting at node 6. And the graph representation of: Adjacency List is {1: [0, 2], 0: [1, 3, 4, 7, 8], 2: [1], 3: [0], 4: [0, 5], 5: [4, 6], 6: [5], 7: [0], 8: [0]}
Q: Give the bfs traversal order starting from node 6.
A:
Let's construct a graph with the nodes and edges first:

**0-Algorithm**

To determine the BFS (Breadth-First Search) traversal order, you need to follow these steps:
1. Initialize: Start by choosing a starting node and enqueue it into a queue.
2. Mark visited: Mark the starting node as visited to avoid reprocessing.
3. Traverse: While the queue is not empty: Dequeue a node and add it to the traversal order. For each unvisited neighboring node of the dequeued node, enqueue it and mark it as visited.
4.Continue the process until all reachable nodes are visited.
Given a graph, your task is to determine the bfs traversal order of this graph starting at node 7. And the graph representation of: Adjacency List is {0: [7, 3, 2, 6, 1, 8, 5], 1: [4, 6, 3, 5, 0, 2], 2: [5, 0, 1, 7], 3: [7, 0, 8, 1, 6, 4], 4: [1, 8, 5, 7, 3, 6], 5: [2, 6, 7, 8, 4, 1, 0], 6: [1, 8, 5, 3, 0, 4], 7: [0, 3, 5, 4, 8, 2], 8: [4, 6, 5, 3, 7, 0]}
Q: Give the bfs traversal order starting from node 7.
A:

**LTM**

Given a graph, your task is to determine the bfs traversal order of this graph starting at node 4. And the graph representation of: Adjacency List is {0: [4, 5, 7], 1: [2, 3, 4, 5, 6], 2: [1, 3, 4, 6], 3: [1, 2, 5, 6], 4: [0, 1, 2, 5, 6], 5: [0, 1, 3, 4, 6, 7], 6: [1, 2, 3, 4, 5], 7: [0, 5]}
Q: Give the bfs traversal order starting from node 4.
A:
Let's break down this problem:

**Algorithm**

To determine the BFS (Breadth-First Search) traversal order, you need to follow these steps: 1. Initialize: Start by choosing a starting node and enqueue it into a queue. 2. Mark visited: Mark the starting node as visited to avoid reprocessing. 3. Traverse: While the queue is not empty: Dequeue a node and add it to the traversal order. For each unvisited neighboring node of the dequeued node, enqueue it and mark it as visited. 4.Continue the process until all reachable nodes are visited.

Given a graph, your task is to determine the BFS traversal order of this graph starting at node 7. And the graph representation of Adjacency List is : {0: [1, 2, 3, 4, 6], 1: [0, 2, 8], 2: [0, 1, 5], 3: [0], 4: [0, 7], 5: [2], 6: [0], 7: [4], 8: [1]}

Q: Give the BFS traversal order starting from node 7.

A: Dequeue node 7. The neighbors are [4], so enqueue node 4. Dequeue node 4. The neighbors of 4 are [7, 0]. Node 7 is visited, so enqueue node 0. Dequeue node 0. The neighbors of 0 are [1, 2, 3, 6, 4]. Node 4 is already visited, so enqueue node 1. 2, 3, and 6. Dequeue node 1. The neighbors of 1 are [0, 2, 8]. Nodes 0 and 2 are visited, so enqueue node 8. Dequeuenode 2. The neighbors of 2 are [0, 5]. Node 0 is visited, so enqueue node 5. Dequeue node 3. The neighbors of 3 are [0]. Node 0 is visited, so no new nodes. Dequeue node 6. The neighbors of 6 are [0]. Node 0 is visited, so no new nodes. Dequeue node 8. The neighbors of 8 are [1]. Node 1 is visited, so no new nodes. Dequeue node 5. The neighbors of 5 are [2]. Node 2 is visited, so no new nodes. All its neighbors have been visited, so the traversal ends. The BFS traversal order starting from node 7 is 7, 4, 0, 1, 2, 3, 6, 8, 5.

...

Given a graph, your task is to determine the bfs traversal order of this graph starting at node 3. And the graph representation of: Adjacency List is {0: [3, 7, 8, 5, 6, 1, 4], 1: [4, 10, 0, 3, 9, 5, 2], 2: [10, 9, 4, 8, 7, 3, 6, 1], 3: [0, 5, 1, 4, 10, 7, 8, 2], 4: [2, 9, 1, 10, 6, 3, 0], 5: [9, 3, 0, 6, 7, 8, 1, 10], 6: [10, 0, 5, 7, 4, 9, 2], 7: [8, 10, 0, 5, 6, 2, 3], 8: [9, 7, 0, 5, 2, 3, 10], 9: [8, 10, 2, 5, 4, 1, 6], 10: [9, 6, 7, 2, 4, 1, 3, 8, 5]} Q

Q: Give the bfs traversal order starting from node 3.

A:

**CoT**

Given a graph, your task is to determine the BFS traversal order of this graph starting at node 7. And the graph representation of Adjacency List is : {0: [1, 2, 3, 4, 6], 1: [0, 2, 8], 2: [0, 1, 5], 3: [0], 4: [0, 7], 5: [2], 6: [0], 7: [4], 8: [1]}

Q: Give the BFS traversal order starting from node 7.

A: The BFS traversal starts at node 7. In BFS, we visit each node level by level, starting from the node we begin at (node 7). Here's the step-by-step breakdown: Start at node 7. The first node in the BFS traversal is 7. Visit the neighbors of 7. The neighbors of node 7 are just node 4 (since 7 4 is an edge). So, we enqueue node 4. Visit the neighbors of node 4. The neighbors of node 4 are node 0 (4 0), so we enqueue node 0. Visit the neighbors of node 0. The neighbors of node 0 are nodes 1, 2, 3, 4 and 6. Since node 4 has already been visited, we enqueue nodes 1, 2, 3, and 6 in that order. Visit the neighbors of nodes 1, 2, 3, 6. From this point, the BFS continues by visiting any remaining unvisited neighbors of these nodes in the same manner, reaching node 8 last. Thus, the BFS traversal order starting from node 7 is 7, 4, 0, 1, 2, 3, 6, 8, 5. The BFS traversal order starting from node 7 is 7,4,0,1,2,3,6,8,5.

...

Given a graph, your task is to determine the bfs traversal order of this graph starting at node 28. And the graph representation of: Adjacency List is {0: [1, 2, 3, 4, 31], 1: [0], 2: [0, 17, 22, 33], 3: [0, 5, 7, 8, 9, 10, 13, 15, 16, 19, 22, 23, 24, 26, 27, 28, 31], 4: [0, 6, 8, 10, 14, 15, 24, 27, 30], 5: [3, 8], 6: [4], 7: [3, 11, 12, 25], 8: [3, 4, 5, 10, 29], 9: [3], 10: [3, 4, 8, 33], 11: [7, 18, 20], 12: [7, 21], 13: [3], 14: [4], 15: [3, 4, 28, 33], 16: [3], 17: [2, 19, 24], 18: [11, 32], 19: [3, 17], 20: [11], 21: [12], 22: [2, 3], 23: [3], 24: [3, 4, 17], 25: [7], 26: [3], 27: [3, 4, 33], 28: [3, 15], 29: [8], 30: [4], 31: [0, 3], 32: [18], 33: [2, 10, 15, 27]}

Q: Give the bfs traversal order starting from node 28.

A:

---

**K-Shot**

Given a graph, your task is to determine the BFS traversal order of this graph starting at node 7. And the graph representation of Adjacency List is : {0: [1, 2, 3, 4, 6], 1: [0, 2, 8], 2: [0, 1, 5], 3: [0], 4: [0, 7], 5: [2], 6: [0], 7: [4], 8: [1]}
Q: Give the BFS traversal order starting from node 7.
A: The BFS traversal order starting from node 7 is 7,4,0,1,2,3,6,8,5
....
Given a graph, your task is to determine the bfs traversal order of this graph starting at node 2. And the graph representation of: Adjacency List is 0: [6], 1: [6], 2: [6], 3: [6], 4: [6], 5: [6], 6: [0, 1, 2, 3, 4, 5]
Q: Give the bfs traversal order starting from node 2.
A:

---

**Instruct**

Given a graph, your task is to determine the BFS traversal order of this graph starting at node 7. And the graph representation of Adjacency List is : {0: [1, 2, 3, 4, 6], 1: [0, 2, 8], 2: [0, 1, 5], 3: [0], 4: [0, 7], 5: [2], 6: [0], 7: [4], 8: [1]}
Let's construct a graph with the nodes and edges first.
Q: Give the BFS traversal order starting from node 7.
A: The BFS traversal starts at node 7. In BFS, we visit each node level by level, starting from the node we begin at (node 7). Here's the step-by-step breakdown: Start at node 7. The first node in the BFS traversal is 7. Visit the neighbors of 7. The neighbors of node 7 are just node 4 (since 7 4 is an edge). So, we enqueue node 4. Visit the neighbors of node 4. The neighbors of node 4 are node 0 (4 0), so we enqueue node 0. Visit the neighbors of node 0. The neighbors of node 0 are nodes 1, 2, 3, 4 and 6. Since node 4 has already been visited, we enqueue nodes 1, 2, 3, and 6 in that order. Visit the neighbors of nodes 1, 2, 3, 6. From this point, the BFS continues by visiting any remaining unvisited neighbors of these nodes in the same manner, reaching node 8 last. Thus, the BFS traversal order starting from node 7 is 7, 4, 0, 1, 2, 3, 6, 8, 5. The BFS traversal order starting from node 7 is 7,4,0,1,2,3,6,8,5.
...
Given a graph, your task is to determine the bfs traversal order of this graph starting at node 10. And the graph representation of: Adjacency List is {0: [4, 14, 1, 11, 5, 13, 2, 12], 1: [12, 4, 10, 2, 0, 3, 14, 11], 2: [8, 9, 1, 13, 11, 12, 15, 5, 0], 3: [10, 1, 11, 7, 8], 4: [0, 1, 15, 11, 6, 10], 5: [14, 6, 11, 0, 2, 7], 6: [5, 4, 11, 10, 14], 7: [14, 12, 9, 13, 3, 8, 5], 8: [2, 15, 14, 12, 10, 3, 7, 13], 9: [2, 7, 15, 12, 14, 13], 10: [3, 1, 8, 15, 4, 11, 6], 11: [14, 2, 0, 12, 4, 3, 5, 6, 10, 1, 13], 12: [1, 13, 7, 2, 14, 11, 9, 8, 0], 13: [12, 2, 7, 9, 0, 11, 8], 14: [7, 0, 11, 5, 12, 8, 9, 1, 6], 15: [4, 9, 8, 2, 10]}
Let's construct a graph with the nodes and edges first.
Q: Give the bfs traversal order starting from node 10.
A:

---

# B    BENCHMARK STATISTICS

This section presents the statistical characteristics of GRAPHOMNI, focusing on the graph families and token usage. We first detail the statistical properties of graph families used in our benchmark in Section B.1, followed by an overview of token consumption associated with various prompt schemes and serialization formats in Section B.2.

## B.1    BASIC STATISTICS OF GRAPHOMNI

Table 7 offers a detailed statistical overview of the diverse graph families employed in GRAPHOMNI. The table reports the average number of nodes and edges for each graph family across tasks such as BFS order, Connectivity, Cycle detection, Diameter calculation, Shortest path, and Triangle counting. These statistics are presented for three difficulty levels: easy, medium, and hard, which reveal the inherent structural complexity differences introduced by the various synthetic graph generators. The selection of graph families is guided by their unique topological properties so that each task is evaluated on graphs that best reflect the challenges encountered in practical applications. In addition, some graph families are omitted from certain tasks because of their intrinsic structural characteristics; for instance, graphs produced by the BAF and all bipartite graphs are excluded from triangle detection when they are structurally incapable of forming triangles.

Table 7: Statistics of Different Graph Types

| Task | Graph Type | Easy | | Medium | | Hard | |
|---|---|---|---|---|---|---|---|
| | | #Avg Nodes | #Avg Edges | #Avg Nodes | #Avg Edges | #Avg Nodes | #Avg Edges |
| BFS-order | BAF | 8.11 | 5.78 | 15.14 | 11.06 | 27.86 | 20.14 |
| | BAG | 8.19 | 11.36 | 13.92 | 23.28 | 26.55 | 82.14 |
| | Bipartite-ERM | 8.03 | 6.50 | 14.44 | 19.28 | 27.23 | 59.36 |
| | Bipartite-ERP | 7.94 | 5.44 | 14.42 | 16.72 | 28.86 | 57.82 |
| | ERM | 8.22 | 16.06 | 13.72 | 51.92 | 25.32 | 135.09 |
| | ERP | 8.03 | 13.14 | 14.33 | 50.17 | 24.59 | 121.77 |
| | SF | 8.11 | 9.00 | 14.81 | 19.00 | 27.73 | 38.00 |
| Connectivity | BAF | 8.14 | 6.21 | 13.83 | 10.67 | 31.17 | 27.00 |
| | Bipartite-ERM | 8.07 | 7.43 | 15.33 | 23.00 | 30.83 | 63.00 |
| | Bipartite-ERP | 8.14 | 7.57 | 13.67 | 20.00 | 28.17 | 59.33 |
| | ERM | 8.07 | 9.71 | 13.83 | 36.67 | 27.50 | 102.17 |
| | ERP | 8.11 | 10.56 | 17.83 | 66.17 | 26.33 | 98.00 |
| Cycle | BAG | 7.93 | 9.90 | 14.12 | 25.10 | 27.82 | 59.04 |
| | Bipartite-ERM | 8.12 | 7.60 | 15.62 | 20.38 | 28.54 | 57.96 |
| | Bipartite-ERP | 8.29 | 7.12 | 15.17 | 17.55 | 30.25 | 44.89 |
| | ERM | 8.19 | 11.43 | 15.10 | 36.43 | 26.46 | 58.21 |
| | ERP | 8.07 | 9.71 | 15.40 | 26.36 | 26.07 | 71.18 |
| | SF | 8.05 | 8.07 | 12.71 | 14.24 | 25.04 | 29.71 |
| Diameter | BAG | 7.98 | 9.73 | 14.30 | 29.91 | 26.81 | 92.24 |
| | ERM | 8.00 | 19.73 | 15.22 | 65.48 | 25.90 | 139.79 |
| | ERP | 8.16 | 18.61 | 15.17 | 70.36 | 25.19 | 130.79 |
| | SF | 7.95 | 9.14 | 15.41 | 19.91 | 28.48 | 39.10 |
| Shortest-Path | BAF | 7.83 | 6.11 | 14.17 | 11.56 | 25.62 | 21.71 |
| | BAG | 7.97 | 10.72 | 14.72 | 31.19 | 25.29 | 88.33 |
| | Bipartite-ERM | 8.06 | 8.97 | 14.61 | 30.58 | 25.50 | 94.71 |
| | Bipartite-ERP | 8.11 | 9.72 | 14.61 | 28.86 | 25.58 | 89.00 |
| | ERM | 8.00 | 17.42 | 15.47 | 67.89 | 25.96 | 179.21 |
| | ERP | 8.03 | 18.92 | 15.42 | 63.14 | 25.25 | 165.46 |
| | SF | 8.03 | 9.42 | 15.50 | 20.03 | 25.54 | 35.21 |
| Triangle | BAG | 8.16 | 13.12 | 14.09 | 25.48 | 27.72 | 55.65 |
| | ERM | 8.06 | 17.44 | 13.39 | 30.81 | 28.80 | 62.60 |
| | ERP | 7.94 | 16.05 | 14.16 | 31.11 | 27.22 | 55.28 |
| | SF | 8.14 | 9.59 | 15.61 | 20.88 | 28.35 | 38.58 |

**Note:** Graph types are selectively excluded from certain tasks based on their structural properties: (1) Connectivity excludes BAG as they are inherently connected by construction; (2) Diameter calculation task excludes BAF and Bipartite-ER graphs due to potentially disconnected components leading to infinite distances; (3) Triangle counting excludes BAF and Bipartite graphs as they are structurally incapable of forming triangles; (4) Cycle detection excludes BAF as they are acyclic by definition.

## B.2 TOKEN STATISTICS OF GRAPHOMNI

Figure 6 offers an overview of token consumption across different dimensions. We use GPT-4 tokenizer here. Token usage is impacted by the choice of prompt scheme and graph serialization format, interactions between them can further influence the overall token count.

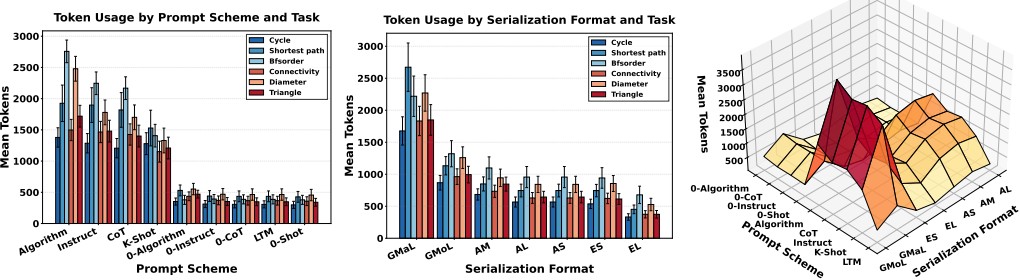

(a) Token usage across different prompt schemes and tasks.

(b) Token usage across different serialization formats and tasks

(c) Token usage for prompt-serialization format combinations.

Figure 6: Analysis of token usage patterns across different dimensions. (a) shows how token usage varies across different prompt schemes for each task. (b) illustrates token consumption patterns for different serialization formats across tasks. (c) provides a 3D surface visualization of the interaction between prompt schemes and serialization formats regarding token usage. Error bars in (a) and (b) represent the standard error of the mean.

# C EXTENDED STUDY AND DISCUSSION OF GRAPHOMNI

## C.1 STUDY ON LARGER GRAPH (BEYOND 30 NODES)

Our benchmark design centers on graphs with 5–30 nodes. While modest compared to real-world networks, this range is both deliberate and effective. First, it aligns with the context length limits of current LLMs and matches the scale used in nearly all recent graph reasoning benchmarks (see Table 3), ensuring comparability. Also, the scale enables us to generate tens of thousands of diverse queries per task, providing statistically robust performance estimates and clearly separating different models apart, like open-source from closed-source models. In this sense, the 5–30 node regime is not a limitation, but a well-calibrated testbed for probing the boundaries of LLM graph reasoning.

To further validate our considerations, we conduct additional experiments on graphs with 30–50 nodes. We sample 50 graphs evenly across all seven generators and evaluate four representative models, yielding approximately 3k new test cases with varied prompt and serialization settings. Results are reported in Table 8. As expected, larger graphs further stress performance, especially on BFS order and Triangle counting. Nevertheless, the relative ranking and accuracy patterns remain consistent with the 5–30 node Hard split, reinforcing the robustness of our findings.

Table 8: Preliminary results on 30–50 node graphs (EH = Extra Hard). Results on the 5–30 node Hard split are shown in parentheses. **Bold orange** / Underlined blue / Light blue highlights indicate best/second-best/third-best performance in its category.

| Task | Difficulty | Open-source Models | | | | Closed-source Models | |
|------|-----------|--------------------|---|---|---|----------------------|---|
| | | Llama-3.1 (8B) | Mistral (7B) | Phi-4 (14B) | Qwen-2.5 (7B) | Claude-3.5 | o4-mini |
| BFS order | EH | 0.70±0.36(0.63) | 0.27±0.23 (0.34) | **2.55±1.08** (2.65) | 1.19±0.52 (1.38) | 16.07±2.48 (26.80) | **35.39±12.61** (32.43) |
| Connectivity | EH | 80.33±3.24(74.58) | 84.11±3.26 (74.77) | 51.82±8.23 (48.39) | **85.01±3.06**(81.19) | **97.92±1.18** (96.99) | 91.48±7.80 (92.08) |
| Cycle | EH | 56.62±3.16(52.40) | 52.62±2.32 (51.64) | 49.38±8.09 (40.64) | **61.38±2.61** (62.27) | 68.53±5.14 (78.22) | **71.66±11.33** (93.06) |
| Diameter | EH | 15.39±3.87(18.63) | 6.89±2.06(6.97) | **16.44±2.89** (17.71) | 11.67±2.31 (15.27) | **48.78±4.76** (56.70) | 37.89±6.56 (34.61) |
| Shortest | EH | 15.56±6.76(23.03) | 1.48±2.03 (12.21) | **16.30±7.66** (26.60) | 11.85±6.62 (28.31) | 57.04±4.46 (87.88) | **58.08±4.52** (88.62) |
| Triangle | EH | 3.19±0.92(4.95) | 2.41±0.65(2.55) | 4.35±1.40 (4.38) | **4.40±0.80** (4.45) | **12.31±0.82** (15.92) | 8.28±4.35 (17.53) |

## C.2 Study on Real-World Graphs: Representative Check

To assess whether our synthetic design translates to real data, we run a focused representative check on two widely used real-world graph suites from complementary domains: a social/interaction dataset IMDB-MULTI (Morris et al., 2020) and a molecular graph dataset (ogbg-molhiv) (Hu et al., 2020). We sample 20 graphs per difficulty per dataset (60 graphs per task in total and thus $\sim 3.6$k evaluated samples across tasks with prompt/serialization variants) and test four representative open-source models plus two closed-source models. Table 9 reports the experimental results.

**Finding 1: Conclusions remain consistent.** Across all six tasks and difficulty levels, accuracy patterns on IMDB-MULTI and ogbg-molhiv closely track the synthetic results: (i) reachability (Connectivity, Cycle detection) is the easiest regime and exhibits high accuracy once serialization is parsed; (ii) ordered-path tasks (BFS order, Shortest path, Diameter calculation) remain substantially harder, with error modes dominated by lost ordering or forgotten edges; and (iii) Triangle counting remains the most difficult due to exhaustive enumeration and arithmetic reliability. The *relative ranking* of models is stable, and the *gap structure* between open- and closed-source models mirrors the results from standard GRAPHOMNI. In short, the representative real-world runs perfectly corroborate our synthetic-only conclusions rather than overturning them.

**Finding 2: Real graphs often simplify certain tasks.** Because many public real graphs are connected and sparse within the selected ranges, some tasks become easier than in our synthetic distribution. For example, connectivity saturates for strong models (near 100% on Easy/Medium in Table 9), and cycle detection displays uniformly higher means than in matched synthetic settings. This is because the uneven data distribution of real graphs means that nearly all graphs are connected and contain at least one cycle. This ease does not invalidate the benchmark, but it shows that *using real-world graphs alone can under-stress* the tasks that are critical to graph reasoning.

In sum, we include IMDB-MULTI and ogbg-molhiv as a *representative check*, which validates that our conclusions persist on real graphs from two major application families (social interaction and molecular science). However, consistent with both our evidence and prior community practice, we retain synthetic graphs in GRAPHOMNI as the default for *comprehensive structural coverage*, *fine-grained interpretability and control*, and *contamination-free* evaluation.

Table 9: Benchmark results of LLMs across tasks (Mean±95% CI Margin) on real-world graphs. Results on the standard setting (i.e. GRAPHOMNI) are shown in parentheses. **Bold orange** / Underlined blue / Light purple highlights indicate best/second-best/third-best performance in its category.

| Task | Difficulty | Open-source Models | | | | Closed-source Models | |
|------|-----------|-------------|------------|-----------|------------|-------------|---------|
| | | Llama-3.1 (8B) | Mistral (7B) | Phi-4 (14B) | Qwen-2.5 (7B) | Claude-3.5 | o4-mini |
| BFS order | E | 45.63±5.22 (18.69) | **47.06±3.18 (13.75)** | 41.90±8.66 (33.03) | 41.98±7.40 (21.46) | **97.14±0.91 (91.42)** | 96.46±1.37 (95.46) |
| | M | 12.14±2.11 (5.27) | 10.24±1.85 (3.36) | **17.86±3.89 (12.49)** | 10.32±2.49 (6.05) | 76.98±2.79 (68.25) | **86.89±3.94 (79.37)** |
| | H | 2.94±0.90 (0.63) | 0.24±0.27 (0.34) | **4.76±1.77 (2.65)** | 0.95±0.76 (1.38) | 41.35±3.76 (26.80) | **44.65±9.89 (32.45)** |
| Connectivity | E | 94.21±1.47 (79.53) | 93.57±1.89 (79.90) | 61.59±9.07 (56.29) | **96.35±0.92 (88.10)** | 99.92±0.16 (98.38) | **100.00±0.00 (98.23)** |
| | M | 88.73±2.37 (79.47) | 88.81±2.98 (80.60) | 53.25±8.09 (54.38) | **93.10±1.67 (87.23)** | **99.92±0.16 (99.11)** | 99.83±0.23 (98.72) |
| | H | 89.44±2.01 (74.58) | 87.30±3.32 (74.77) | 55.24±7.67 (48.39) | **89.76±2.04 (81.19)** | **98.65±0.67 (96.99)** | 95.63±3.04 (92.02) |
| Cycle | E | 56.75±2.62 (55.49) | 51.43±1.56 (55.44) | 51.03±6.82 (45.25) | **59.37±2.00 (62.19)** | 80.48±4.81 (82.56) | **94.51±2.14 (97.97)** |
| | M | 54.05±2.46 (55.69) | 49.92±1.18 (53.71) | 48.17±5.83 (44.26) | **55.63±1.79 (62.07)** | 76.51±5.06 (80.80) | **92.19±3.78 (97.75)** |
| | H | 51.03±2.21 (52.40) | 49.84±1.31 (51.64) | 44.68±5.27 (40.64) | **54.05±2.20 (58.88)** | 71.75±3.87 (80.10) | **89.65±3.40 (95.61)** |
| Diameter | E | 25.48±4.20 (41.27) | 20.95±4.18 (28.55) | **50.48±5.69 (42.81)** | 47.86±3.57 (45.08) | 83.33±1.13 (83.71) | **97.30±0.90 (98.88)** |
| | M | 15.48±3.51 (27.29) | 8.81±2.24 (15.17) | **25.16±3.92 (28.49)** | 23.02±3.29 (27.31) | 58.33±2.82 (71.22) | **84.37±3.89 (72.84)** |
| | H | 8.97±3.00 (18.63) | 6.19±2.58 (6.97) | **14.21±2.36 (17.71)** | 12.86±1.89 (15.27) | 43.02±2.52 (56.70) | **64.12±7.27 (34.61)** |
| Shortest | E | 39.21±5.87 (38.75) | 28.65±4.69 (31.18) | 43.89±8.80 (42.61) | **50.08±9.17 (47.46)** | 98.49±1.29 (94.35) | **98.59±1.44 (95.08)** |
| | M | 30.16±4.57 (28.84) | 21.83±3.61 (19.89) | 34.44±7.92 (33.92) | **37.14±7.44 (35.53)** | 95.71±2.09 (91.27) | **98.39±1.72 (92.60)** |
| | H | 22.06±3.73 (23.03) | 14.05±2.51 (12.21) | **30.16±6.43 (26.60)** | 29.68±5.65 (28.31) | 92.14±1.80 (87.88) | **95.79±2.91 (88.63)** |
| Triangle | E | 11.19±3.03 (14.97) | 5.32±1.69 (11.87) | 23.81±5.33 (12.88) | **24.44±3.65 (18.56)** | 63.89±3.16 (43.41) | **81.30±4.10 (84.54)** |
| | M | 6.51±2.03 (8.56) | 1.83±0.84 (5.86) | **14.44±3.61 (7.54)** | 11.83±2.51 (9.18) | 47.30±2.74 (24.00) | **82.20±3.86 (48.13)** |
| | H | 5.95±2.17 (4.95) | 1.35±0.64 (2.55) | **9.60±3.21 (4.38)** | 9.52±2.40 (4.45) | 34.84±2.80 (15.92) | **65.29±8.75 (17.53)** |

### C.3 CONSIDERATIONS ON REAL-WORLD GRAPHS VS. SYNTHETIC GRAPHS

In designing our benchmark, we considered several possible choices of evaluation substrate, including both real-world and synthetic graphs. After careful consideration, we opted to primarily use synthetic graphs, for the following methodological reasons:

1. *Coverage and controllability.* Our seven classic generators are selected to span the principal structural motifs (random/Poisson, scale-free, bipartite, hierarchical, small-world), and they support fine-grained parameter control (e.g., $p$ in Erdős–Rényi, attachment in BA) (Chakrabarti & Faloutsos, 2006). This control enables balanced, modular ablations and isolates causal factors of failure, which typical public real-graph suites do not provide.

2. *Representativeness vs. noise in public repositories.* Real-graph repositories such as SNAP (Sosic & Leskovec, 2015) skew toward specific domains (social/web) with narrow size and density bands. Also, many graphs are connected and share similar sparsity patterns. This induces *structural narrowness* and *domain bias*, and it can *reduce task hardness* (e.g., connectivity becomes trivial). Mixing such graphs into a general-purpose reasoning benchmark, therefore, risks adding noise without broadening structural regimes.

3. *Zero contamination.* Fully synthetic construction guarantees no overlap with pretraining corpora, avoiding inflated scores due to memorization or leakage (Hendrycks et al., 2021a). Given rapidly evolving LLMs and opaque training mixtures, contamination-free evaluation is essential for credible comparisons.

Meanwhile, **synthetic-only evaluation is also standard in prior work.** This design choice is not unique to GRAPHOMNI. Several foundational studies adopt the same "synthetic only" paradigm to ensure interpretability and controlled analysis: GraphQA (Fatemi et al., 2024) and GraphInstruct (Luo et al., 2024b) both rely solely on synthetic graphs to probe LLM reasoning, while GraphWiz (Chen et al., 2024a) demonstrates that synthetic graphs can even serve as effective fine-tuning data. These precedents highlight that synthetic construction is widely accepted in the community as the most principled way to study graph reasoning in LLMs. At the same time, we note that the real-graph domains we choose (IMDB-MULTI and ogbg-molhiv) align with recent works such as LLM4Hypergraph (Feng et al., 2025), which employ citation networks and protein structures, respectively. Thus, our real-graph ablation covers representative application families, while our synthetic benchmark remains the default for comprehensive coverage and methodological clarity.

### C.4 EXPLORATION ON NP-HARD TASKS

To complement our six canonical tasks, we further probe LLM performance on two classical NP-hard graph problems: Hamiltonian cycle detection and Max-Cut. This evaluation serves as an ablation rather than a core component of GRAPHOMNI, allowing us to test whether the conclusions from tractable tasks extend to settings of higher computational complexity.

**Experimental setup.** We retain the three difficulty splits by node size: Easy ($n \in [0, 10]$), Medium ($n \in (10, 20]$), and Hard ($n \in (20, 25]$). Compared to the main benchmark, the Hard regime uses slightly smaller graphs due to the exponential growth in search space. For *Hamiltonian cycle*, structural imbalance makes several generators unsuitable (e.g., SF, Bipartite-ERM, BAF, and Bipartite-ERP rarely admit cycles). We therefore restrict the task to ERM, ERP, and BAG, with ground-truth labels balanced 50/50 between existence and non-existence of a Hamiltonian cycle. For clarity, we also report Hamiltonian cycle results on the positive cases separately, since these are strictly harder: a correct answer must not only assert existence but also return a complete and valid tour (metric mentioned below). For *Max-Cut*, all seven graph families are included (SF, ERM, ERP, BAG, BERP, BAF, and BERM). We sample 18 graphs per split for Hamiltonian cycle and 14 per split for Max-Cut, yielding just over 6,000 queries across prompt and serialization variants.

**Evaluation metrics.** As with the canonical tasks, we apply strict binary scoring. For *Hamiltonian cycle*, a prediction is marked correct only if: (i) the model explicitly affirms or denies the existence of a cycle in line with the ground truth, and (ii) when the ground truth is True, the model additionally outputs a concrete cycle, which we verify with a dedicated checker. Omitting an explicit decision or producing a non-verifiable tour results in 0. For *Max-Cut*, we extract both the predicted maximum cut size and the corresponding bipartition (from phrases such as "the maximum cut size is ..."). A

custom validation function checks whether the reported cut matches the ground truth. Full correctness requires both size and partition to be correct, while partial matches or non-extractable answers are scored 0.

**Results and insights.** As summarized in Table 10, performance patterns closely resemble those of the six canonical tasks: open-source models hover near random, while closed-source reasoning models achieve substantially higher, but still imperfect, scores. Thus, the core conclusions of GRAPHOMNI generalize naturally to NP-hard settings. More interestingly, these results highlight how LLMs perceive task difficulty differently from humans. Whereas human solvers experience a sharp jump in difficulty between polynomial-time and NP-hard problems, current LLMs instead exhibit a nearly uniform collapse in accuracy across NP-hard tasks. In other words, scaling to NP-hard does not introduce a progressive "step up" in challenge for models as it does for humans. This suggests that including NP-hard tasks may not meaningfully enrich the evaluation landscape, and reinforces our focus on tractable yet diverse tasks as the primary design of GRAPHOMNI.

Table 10: Benchmark Results of LLMs Across Tasks (Mean±95% CI Margin). **Bold orange** / Underlined blue / Light purple highlights indicate best/second-best/third-best performance in its category.

| Task | Difficulty | Open-source Models | | | | | Closed-source Models |
| --- | --- | --- | --- | --- | --- | --- | --- |
| | | Llama-3.1 (8B) | Mistral (7B) | Phi-4 (14B) | Qwen2.5 (7B) | qwen38 | o4-mini |
| Hamilton cycle | E | 57.14±2.97 | 55.47±2.85 | 57.67±6.25 | 73.02±2.87 | **93.74±1.45** | **97.09±0.88** |
| | M | 54.94±3.40 | 46.03±3.05 | 50.09±6.59 | 60.14±2.93 | **84.83±2.38** | **71.08±2.39** |
| | H | 54.94±3.61 | 46.38±3.04 | 51.85±6.98 | 60.23±3.05 | **79.98±2.64** | **55.56±2.48** |
| Hamilton cycle (Positive Samples) | E | 45.68±4.96 | 63.67±7.09 | 54.67±6.04 | 73.02±5.14 | **89.59±2.51** | **95.24±1.54** |
| | M | 50.62±6.90 | 64.73±7.61 | 60.85±8.39 | 71.60±5.30 | **79.37±4.15** | **49.56±4.32** |
| | H | 47.09±6.88 | 58.38±6.81 | 55.73±8.13 | 61.55±6.36 | **70.90±4.61** | **18.46±3.64** |
| Max cut | E | 15.10±3.29 | 11.63±2.44 | 23.88±4.69 | 18.37±2.71 | **27.96±4.56** | **61.94±4.27** |
| | M | 5.20±1.66 | 5.10±1.43 | 6.43±1.81 | 10.41±1.26 | **16.94±2.33** | **28.16±1.80** |
| | H | 1.22±0.70 | 0.61±0.47 | 2.04±1.03 | 1.12±0.61 | **9.08±2.17** | **34.74±3.64** |

## C.5   SCALING VS. REASONING: DISENTANGLING THEIR EFFECTS ON GRAPH REASONING

To contrast model *scaling* with *reasoning-centric* improvements, we isolate three Qwen variants: **Qwen-2.5 (7B)** as the baseline, **Qwen-2.5 (72B)** to represent scaling up within the same family, and **Qwen-3 (8B)** as a reasoning model at a comparable parameter budget. Table 11 subsets the main results (Table 3) to these three columns.

**Finding 1: Scaling lifts the floor.** Relative to Qwen-2.5 (7B), Qwen-2.5 (72B) yields consistent improvements across nearly all tasks, particularly on the easier splits. For example, accuracy on BFS order (Easy) rises from 21.46 to 71.41, an absolute gain of nearly 50%, while Shortest path (Easy) improves from 47.46% to 90.03%, a margin of over 42%. Similarly, Diameter calculation (Easy) increases by more than 33% (45.08% → 78.50%). Even on Connectivity, which is already near-saturated, scaling provides modest yet consistent lifts (E/M/H: +2.14%, +2.45%, +2.90%, respectively). In contrast, on the most combinatorial regime, Triangle counting (Hard), the gain is negligible (4.45% → 4.73%), suggesting that sheer scale does little to overcome the inherent difficulty of exhaustive enumeration.

**Finding 2: Reasoning lifts the ceiling.** When holding parameter count roughly constant, Qwen-3 (8B) substantially outperforms both Qwen-2.5 (7B) and, on several hard splits, even Qwen-2.5 (72B). For instance, on BFS order (Hard), performance improves from 22.03% to 29.53% compared to Qwen-2.5 (72B), a relative advantage of more than 7%. On Diameter calculation (Hard), the margin widens further: 39.83% versus 29.59%, an absolute gain of over 10%. The effect is most striking on Triangle counting (Hard), where Qwen-3 (8B) achieves 19.54%, far surpassing the 4.73% of Qwen-2.5 (72B). These results indicate that architectural and optimization changes targeted at reasoning are more effective in extending the *upper bound* of graph reasoning ability than scaling alone.

**Implication.** Scaling and reasoning improve different aspects of performance. Larger models predominantly strengthen robustness on easier instances, lifting the floor, whereas reasoning-oriented models better capture multi-hop dependencies and complex subgraph structures, lifting the ceiling. A

Table 11: Isolating scaling vs. reasoning effects. Baseline: Qwen-2.5 (7B). Scaling: Qwen-2.5 (72B). Reasoning: Qwen-3 (8B). **Bold orange** / Underlined blue / Light blue  highlights indicate best/second-best/third-best performance.

| Task | Difficulty | Open-source Models | | |
|------|------------|--------------------|--|--|
| | | **Qwen2.5 (72B)** | **Qwen2.5 (7B)** | **Qwen3 (8B)** |
| BFS order | E | **71.41±3.45** | 21.46±4.26 | 65.87±5.59 |
| | M | 47.82±5.30 | 6.05±1.41 | **53.30±5.42** |
| | H | 22.03±4.39 | 1.38±0.37 | **29.53±4.25** |
| Connectivity | E | 90.24±1.89 | 88.10±1.46 | **97.17±1.29** |
| | M | 89.68±1.56 | 87.23±1.60 | **96.87±1.16** |
| | H | 84.09±1.98 | 81.19±2.02 | **92.89±2.07** |
| Cycle | E | 74.02±3.34 | 62.19±1.85 | **90.30±2.33** |
| | M | 71.99±3.34 | 62.07±1.80 | **89.66±2.07** |
| | H | 68.40±2.73 | 58.88±2.14 | **86.81±2.27** |
| Diameter | E | **78.50±1.16** | 45.08±4.17 | 77.56±2.77 |
| | M | 52.32±2.00 | 27.31±3.16 | **61.71±2.28** |
| | H | 29.59±2.48 | 15.27±2.47 | **39.83±2.67** |
| Shortest | E | **90.03±2.27** | 47.46±8.76 | 77.69±5.17 |
| | M | **81.17±3.03** | 35.53±6.80 | 69.60±5.50 |
| | H | **72.53±4.29** | 28.31±5.50 | 64.28±5.60 |
| Triangle | E | 36.57±4.40 | 18.56±1.24 | **41.36±4.63** |
| | M | 14.52±2.63 | 9.18±0.73 | **26.95±2.44** |
| | H | 4.73±1.58 | 4.45±0.58 | **19.54±1.34** |

balanced recipe, i.e. moderate scaling *combined with* reasoning-oriented objectives, appears most promising for closing the persistent gaps in `BFS order`, `Diameter calculation`, and `Triangle counting`.

### C.6 RATIONALE FOR BINARY METRIC OVER PARTIAL SCORE

Evaluating graph reasoning outputs with partial credit is appealing in theory, but defining a consistent and objective scheme across six tasks, seven graph types, seven serializations, and nine prompt schemes is exceptionally difficult. In practice, two approaches exist: assigning credit based on the degree of correctness in the final answer, or rewarding intermediate steps and sub-outputs. Both approaches introduce major challenges. For final answers, it is often ambiguous how to compare partially correct results (e.g., is overcounting triangles by one preferable to undercounting by one?). Such ambiguity undermines the credibility of fine-grained scoring. For intermediate steps, reliably extracting and interpreting model outputs at scale is infeasible, since formatting and reasoning styles vary widely across models and prompts.

By contrast, binary accuracy against a known ground truth provides a clear and unambiguous evaluation signal. With the extensive and diverse evaluation set in GRAPHOMNI, binary scoring captures performance gaps robustly and fairly across models and tasks. While finer-grained metrics such as edit distance, subtask scoring, or partial correctness may be valuable for training objectives like reinforcement learning, they extend beyond the present study's evaluation focus. Incorporating such measures represents a promising avenue for future work.

## D RL-BASED PROMPT SEARCH INSPIRED BY GRAPHOMNI

### D.1 BACKGROUND AND SERIALIZATION PROCESS

Our benchmark evaluates three key dimensions—*graph type, serialization format*, and *prompt scheme*—to underscore the critical role of transforming graph structures into textual inputs for LLM inference. In this section, *We want to identify the optimal combination strategies (serialization format; prompt scheme, etc.) that enhance the effectiveness of textual representations*, thereby improving LLM performance in graph reasoning and understanding tasks. Prior research indicates that while a particular serialization format or prompt scheme may yield optimal performance in isolation, their combination does not necessarily lead to the best results, highlighting complex interactions among various factors. Furthermore, the final performance of LLMs may be influenced by additional factors that were not systematically examined in our benchmark (e.g., those in Appendix D.3), underscoring the intricate nature of the graph-to-text transformation process, which extends beyond the scope of single-factor analysis. This makes finding the optimal serialization strategy complex. We define the process of converting graph structures into textual inputs tailored to a specific task as the **serialization process**. Similar prompt processes are used in NLP. For example, Shi et al. (2024) formulated *prompt formatting* as a multi-armed bandit problem; Sclar et al. (2023) employed Thompson sampling to determine the optimal strategies. For LLM-based graph reasoning, however, previous studies predominantly focused on single-factor variations. The multiple factor considered in our study significantly complicates the serialization process—once a particular factor is determined, others are influenced in complex and often unpredictable ways.

Due to these complexities, it is computationally enormous to find the optimal serialization strategy by enumerating all possible combinations (termed as *grid search* in our study). To mitigate this computational challenge, we propose using RL to find a high-quality serialization strategy under a limited LLM cost, because of RL's ability to learn near-optimal strategies in high-dimensional spaces through exploration and feedback. In the context of RL, we assume that all benchmarking results (e.g., those in Section 4.1 and 4.2) are not available. Instead, we will repeatedly choose various serialization strategies, test their performance, and use the results for RL.

Specifically, RL transforms optimizing the serialization process strategy into a sequential decision-making problem for each type and difficulty of the task. There are $T$ decision epochs, and each decision epoch determines one component of the serialization strategy. For example, the decision horizon is $T = 3$ when we aim to identify the optimal combination of the serialization format, prompt scheme, and LLM. In the $T = 3$ decision epochs, we sequentially determine the prompt scheme, serialization format, and LLM. Such an order of optimizing the components of a serialization strategy

is predetermined, and we will investigate its impact on the optimization results in future studies. This predetermined order specifies a sequence of action spaces $\{\mathcal{A}_t\}_{t=1,\ldots,T}$ (e.g., $\mathcal{A}_t$ can be all candidate LLMs). We set the initial state $s_0$ as the specific type and difficulty of the task.

Then at decision epoch $t = 1, \ldots, T$, we choose an action $a_t \in \mathcal{A}_t$ based on the previous actions $a_1, \ldots, a_{t-1}$. This corresponds to a policy $\pi_t : \mathcal{S}_0 \times \mathcal{A}_1 \times \cdots \times \mathcal{A}_{t-1} \mapsto \mathcal{A}_t$, where $\mathcal{S}_0$ is the state space of the initial state $s_0$. For any instance $s$ (e.g., a query for Connectivity task in easy mode for a specific graph), a binary reward, denoted by $r(s, a_1, \ldots, a_T)$, is incurred at the end of the decision epoch, which is set to 1 if the LLM correctly answers the specific query under the selected serialization strategy $(a_1, \ldots, a_T)$ and to 0 otherwise. For each type and difficulty of the task, our objective is to maximize the expected reward of choosing the serialization strategy $a_1, \ldots, a_T$:

$$\max_{\{\pi_t\}_{t=1,\ldots,T}} \mathbb{E}[r(s, a_1, \ldots, a_T)|s_0],$$

where the expectation is taken with respect to the problem instance $s$ and the (random) answer output by an LLM (affected by the randomness of the LLM, e.g., the temperature parameter). Note that (i) $s_0$ is part of the instance information $s$, and (ii) we fix the type and difficulty of the task, and the only randomness in terms of $s$ is from graph generation. To approximate this objective function, we generate $N$ different graphs for each type of query. We assess the performance of RL using the average reward across the $N$ graphs, which essentially is the accuracy of the serialization strategy for a specific graph-related task across these $N$ graphs.

We use the $Q$-learning approach to solve this optimization problem. Let $Q_t(s_0, a_1, \ldots, a_t)$ be the $Q$-function at decision epoch $t = 1, \ldots, T$, which represents the optimal reward-to-go if actions $a_1, \ldots, a_t$ have been determined at decision epoch $t$ given the initial state $s_0$. These functions satisfy the Bellman recursion:

$$Q_t(s_0, a_1, \ldots, a_t) = \max_{a_{t+1} \in \mathcal{A}_{t+1}} Q_{t+1}(s_0, a_1, \ldots, a_{t+1}), \quad t = 1, \ldots, T-1$$

with terminal condition

$$Q_T(s_0, a_1, \ldots, a_T) = \mathbb{E}[r(s, a_1, \ldots, a_T)|s_0],$$

This terminal $Q$-function can be approximated by the accuracy of the LLM answer across the $N$ generated graphs.

Consider the problem of dealing with high-dimensional, complex state spaces in serialization process, we employ the Deep $Q$-Network (DQN) (Mnih et al., 2013) to implement RL, which employs a deep neural network as a function approximator for the $Q$-function. Specifically, we use a neural network $\widehat{Q}_t(s_0, a_1, \ldots, a_t; \theta_t)$ parameterized by $\theta_t$ to approximate the corresponding $Q_t(s_0, a_1, \ldots, a_t)$ for the actions or factors considered in serialization process. Each $Q$-network is modeled as a three-layer multilayer perceptron with ReLU activations. Training minimizes the mean squared error loss, and action selection follows an $\epsilon$-greedy policy, where $\epsilon$ linearly decays from 1.0 to a minimum of 0.01. The detailed algorithm for each initial state $s_0$ is provided in Algorithm 1.

We design two experimental settings, **RL-Opt** and **RL-Scale**, to assess the effectiveness of our approach. **RL-Opt** focuses on a $T = 3$ serialization process—selecting the serialization format, prompt scheme, and LLM model—and evaluates both LLM cost and the accuracy of identifying the optimal configuration. **RL-Scale** extends the scope to include additional factors beyond those in GRAPHOMNI, investigating the scalability of the RL method for more complex serialization tasks, with an emphasis on LLM cost.

## D.2 DETAILS FOR RL-OPT SETTING

In **RL-Opt**, we apply RL to find a high-quality serialization strategy under a limited LLM cost. The serialization process in this case involved three key factors: serialization formats (in total 7), nine prompt schemes (in total 9), and five open-source language models (including LLaMA3, LLaMA 3.1, Mistral, Phi-4, and Qwen-2.5). The total number of possible combinations in our search space is given by $E = 7 \times 9 \times 5 = 315$. To find a high-quality serialization strategy, we set the total training episodes to $M = 80$ and initial learning rate to $0.001$ during RL training. We evaluate the RL performance on 6 tasks in three different modes, resulting in a total of 18 experimental cases.

---

**Algorithm 1** RL Framework of GRAPHOMNI

---

**Input:** Action spaces $\{\mathcal{A}_t\}_{t=1}^T$; number of training episodes $M$; exploration rate $\epsilon$; initial state $s_0$
**Initialization:**
Generate $N$ graphs according to the initial state $s_0$
Initialize $Q$-networks $\{\widehat{Q}_t(s_0, a_1, \ldots, a_t; \theta_t)\}_{t=1}^T$ with random initialized weights $\theta_t$
**for** episode $= 1$ to $M$ **do**
    **for** $t = 1$ to $T$ **do**
        **Choose action:**
        With probability $\epsilon$, select a random action $a_t \in \mathcal{A}_t$
        Otherwise, set $a_t \leftarrow \arg\max\limits_{a \in \mathcal{A}_t} \widehat{Q}_t(s_0, a_1, \ldots, a_{t-1}, a; \theta_t)$
        **Execute action and obtain new state:**
        Update state: $\{s_0, a_1, \ldots, a_t\} \leftarrow \{s_0, a_1, \ldots, a_{t-1}\} \cup \{a_t\}$
        $Q$**-network update:**
        If $t = T$, set $y$ to be the accuracy of the LLM answer across the $N$ generated graphs
        Otherwise, set $y \leftarrow \max\limits_{a \in \mathcal{A}_{t+1}} \widehat{Q}_{t+1}(a_1, \ldots, a; \theta_{t+1})$
        Perform a gradient descent step on $\left(y - \widehat{Q}_t(a_1, \ldots, a_t; \theta_t)\right)^2$ with respect to $\theta_t$
    **end for**
    Decay exploration rate: $\epsilon \leftarrow \epsilon \cdot$ `decay rate`
**end for**

---

For each case, based on our numerical results in Sections 4.1 and 4.2, we know which combination (serialization format; prompt scheme; LLM) performs the best for each specific graph-related task. Hence, we can compare the serialization strategy obtained by RL with the ground-truth optimal strategy.

Specifically, we employ two key metrics. **Search Cost**: Given that RL explores $k$ different combinations during the training process, we define Cost $= \frac{k}{K}$, where $k$ depends on the number of training episodes and $K$ is the total number of combinations. **Rate**: Let $\text{acc}_*$ be the accuracy achieved by the best combination found by RL, and $\text{acc}_{\max}$ be the highest accuracy in Sections 4.1 and 4.2. Then we define Rate $= \frac{\text{acc}_*}{\text{acc}_{\max}}$. The results are displayed in Table 4. It can be seen that, with an approximate 25% reduction in cost, the RL method still maintains an average rate of around 0.9, indicating its ability to significantly shorten the time required for the search for optimal combinations while ensuring the quality of the results. This outcome underscores the notable advantages of RL in the serialization process problem — rapidly finding high-quality solutions, thereby substantially reducing computational resources and time costs. Moreover, this approach does not rely heavily on extensive manual expertise, enhancing the automation of the optimization process. As a result, it is applicable not only to the factors considered in this study but also to other factors warranting further investigation.

## D.3 RL-SCALE

In **RL-Scale**, we examine the scalability of our RL method by incorporating additional four factors into the serialization process. Different from **RL-Opt** that optimizes the LLM model, we fix the model as **Qwen-2.5** and test the performance on the `Diameter calculation` task in **easy** mode. The additional four factors are inspired by Sclar et al. (2023), which are the delimiters between sentences, the capitalization style of each sentence, the delimiter used to introduce questions and answers, and the delimiter between words. As before, we still optimize the prompt scheme and the serialization format. Details of the 6 factors implemented in the serialization process are shown below.

> **6 factors implemented in serialization process**
>
> - **SPACES1**: delimiter between sentences, include: ' − ',' <sep> ',' , ',' \n ',' \n','\t',' ; \n',' ',' . ',' || '
> - **CS**: the delimiter used to introduce questions and answers, include: ' \n\t',' \n ',' : ',' :: ', ' \t',' ::',' ',' − ',' :',' ::: '
> - **SPACES2**: delimiter between words, include: ' ',' ','\t'
> - **CASE FUNCTION**: the overall capitalization mode of each sentences, include:no change, title, upper, lower
> - **PROMPTS SCHEME**: include: 0-shot, 0-CoT, 0-Instruct, 0-Algorithm, LTM, Algorithm, CoT, k-shot, Instruct
> - **SERIALIZATION FORMAT**: include: GMoL, Adjacency Set, Edge Set, Edge List, Adjacency Matrix, Adjacency List, GMaL

Since the optimal combination in RL-Scale is unknown, we only focus on the cost of RL finding the near optimal combination. In addition, taking into account the LLM cost problem and the performance stability, for each combination, we took a fixed evaluation of 30 samples to get accuracy and set the temperature to 0. Ultimately, we compare the costs of RL and grid search under conditions where the serialization process involves 2–6 factors, with the results presented in Figure 7. For step counts of 2, 3, 4, 5, and 6, the number of combinations explored for RL is 40, 121, 182, 300, and 632, respectively. In contrast, the number of combinations for Grid Search is 100, 300, 1200, 8400, and 75600 for the same step counts. It shows that RL exhibits a highly promising trend in terms of the cost associated with searching for optimal combinations. When considering two-step factors, the cost of RL is comparable to that of grid search. However, as factors or steps increase, the cost growth of RL is significantly lower than that of grid search. This finding suggests that for serialization process tasks, RL can adaptively adjust its strategy to better accommodate complex environments, highlighting its broader potential for application.

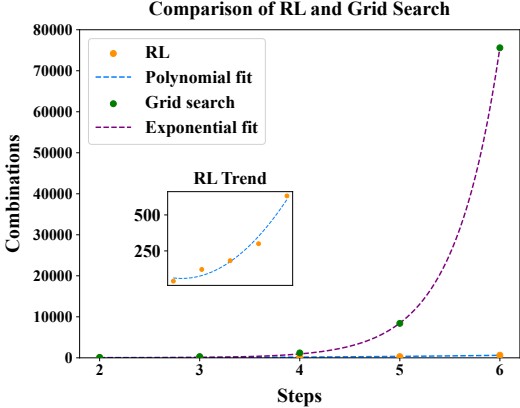

Figure 7: RL and Grid Search

Table 12: 2–6 factors, top-3 combinations and corresponding reward from **RL-Scale**.

| Grid research | Rank | Combination Parameters | Reward |
|---|---|---|---|
| | 1 | Edge List,0-shot, Q:, A:, , , \n \t, no | 0.3000 |
| 100 | 2 | Edge List,0-shot, Q:, A:, \n, , , no | 0.2667 |
| | 3 | Edge List,0-shot, Q:, A:, , , , no | 0.2000 |
| | 1 | Edge List,0-shot, Q:, A:, ‖ , \t, , no | 0.3667 |
| 300 | 2 | Edge List,0-shot, Q:, A:, − , \t, \n \t, no | 0.3333 |
| | 3 | Edge List,0-shot, Q:, A:, − , \t, - , no | 0.3333 |
| | 1 | Edge List,0-shot, Q:, A:, − , , \n , upper | 0.4000 |
| 1200 | 2 | Edge List,0-shot, Q:, A:, ‖ , , ::: , lower | 0.4000 |
| | 3 | Edge List,0-shot, Q:, A:, , \t, , upper | 0.3667 |
| | 1 | Adjacency Matrix,0-shot, Q:, A:, <sep> , , \n , title | 0.5667 |
| 8400 | 2 | GMoL,0-shot, Q:, A:, \n, \t, ::: , upper | 0.5333 |
| | 3 | GMoL,0-shot, Q:, A:, ; \n, \t, ::: , upper | 0.5333 |
| | 1 | Adjacency Set, Algorithm, Q:, A:, ‖ , , : , lower | 0.6333 |
| 75600 | 2 | Adjacency Set, Algorithm, Q:, A:, ‖ , , : , no | 0.6333 |
| | 3 | Adjacency Matrix,0-shot, Q:, A:, \n, , :, title | 0.6000 |

# E    COMPREHENSIVE EXPERIMENTAL RESULTS

In this section, we include all the experimental results and additional analysis of the GRAPHOMNI as a reference to support our claims and findings mentioned in Section 4.1. We first present fine-grained experimental results broken down across main evaluation dimensions in Appendix E.1, followed by detailed performance heatmaps for all tasks and models in Appendix E.2. Finally, we provide a comprehensive error analysis with representative cases in Appendix E.4.

## E.1    FINE-GRAINED RESULTS ACROSS DIMENSION

In this subsection, we present detailed performance results across model capability, graph type, prompting schemes, and serialization format impact. Based on the complete evaluation results in Table 13, we further analyze the results from multiple perspectives, including overall performance across all models, separate analyses for open-source models, and specific results for closed-source models. These comprehensive results provide additional evidence supporting our main findings discussed in Section 4.1.

Table 13: Benchmark Results of LLMs Across Tasks (Mean±95% CI Margin). **Bold orange** / Underlined blue / Light purple  highlights  indicate  best/second-best/third-best performance in its category.

| Task | Difficulty | Open-source Models | | | | | | | Closed-source Models | | | | | Random |
| --- | --- | --- | --- | --- | --- | --- | --- | --- | --- | --- | --- | --- | --- | --- |
| | | Llama-3 (8B) | Llama-3.1 (8B) | Mistral (7B) | Phi-4 (14B) | Qwen-2.5 (72B) | Qwen-2.5 (7B) | Qwen-3 (8B) | Claude-3.5 | GPT-4o | GPT-4o-mini | Gemini-2.0 | o4-mini | |
| BFS order | E | 15.62±2.94 | 18.69±3.02 | 13.75±1.44 | 33.03±7.32 | 71.41±3.45 | 21.46±4.26 | 65.87±5.59 | 91.42±1.65 | 81.48±3.22 | 58.75±4.22 | 90.31±2.30 | 95.46±0.78 | 0.00 |
| | M | 4.04±0.81 | 5.27±0.93 | 3.36±0.44 | 12.49±3.24 | 47.82±5.30 | 6.05±1.41 | 53.30±5.42 | 68.25±2.96 | 55.07±4.50 | 25.03±3.11 | 68.40±3.95 | 79.37±2.08 | 0.00 |
| | H | 0.39±0.15 | 0.63±0.19 | 0.34±0.14 | 2.65±0.80 | 22.03±4.39 | 1.38±0.37 | 29.53±4.25 | 26.80±2.64 | 21.58±3.69 | 6.28±0.90 | 27.77±3.34 | 32.45±3.88 | 0.00 |
| Connectivity | E | 78.01±2.28 | 79.53±2.03 | 79.90±1.89 | 56.29±8.58 | 90.24±1.89 | 88.10±1.46 | 97.17±1.29 | 98.38±0.60 | 95.63±1.30 | 89.10±2.32 | 92.61±1.42 | 98.23±0.63 | 67.49 |
| | M | 77.78±2.78 | 79.47±2.00 | 80.60±1.92 | 54.38±7.99 | 89.68±1.56 | 87.23±1.60 | 96.87±1.16 | 99.11±0.39 | 95.12±1.37 | 93.60±1.10 | 98.72±0.52 | 92.02±3.99 | 70.75 |
| | H | 68.49±4.49 | 74.58±2.67 | 74.77±2.46 | 48.39±7.50 | 84.09±1.98 | 81.19±2.02 | 92.89±2.07 | 96.99±1.48 | 90.59±2.19 | 84.82±2.17 | 87.99±1.67 | 92.02±3.99 | 66.36 |
| Cycle | E | 53.84±1.75 | 55.49±0.90 | 55.44±0.96 | 45.25±5.90 | 74.02±3.34 | 62.19±1.85 | 90.30±2.33 | 82.56±3.89 | 85.08±2.27 | 75.04±2.83 | 62.30±3.32 | 97.97±0.71 | 50.00 |
| | M | 42.38±1.13 | 55.69±1.08 | 53.71±0.72 | 44.26±5.43 | 71.99±3.34 | 62.07±1.80 | 89.66±2.07 | 80.80±3.94 | 85.35±2.30 | 75.79±2.96 | 60.29±3.22 | 97.75±0.76 | 50.00 |
| | H | 41.24±1.53 | 52.40±1.47 | 51.64±1.02 | 40.64±4.97 | 68.40±2.73 | 58.88±2.14 | 86.81±2.27 | 80.10±3.97 | 82.96±2.55 | 73.46±3.30 | 58.30±2.80 | 95.61±1.23 | 50.00 |
| Diameter | E | 23.78±4.17 | 41.27±5.37 | 28.55±4.28 | 42.81±5.06 | 78.50±1.16 | 45.08±4.17 | 77.56±2.77 | 83.71±1.26 | 63.99±2.19 | 37.36±2.62 | 79.14±1.94 | 98.88±0.15 | 11.20 |
| | M | 14.29±2.66 | 27.29±4.20 | 15.17±2.57 | 28.49±4.09 | 52.32±2.00 | 27.31±3.16 | 61.71±2.28 | 71.22±1.30 | 52.64±3.05 | 22.85±2.97 | 49.52±2.14 | 72.84±1.82 | 6.70 |
| | H | 8.48±1.75 | 18.63±3.27 | 6.97±1.26 | 17.71±3.02 | 29.59±2.48 | 15.27±2.47 | 39.83±2.67 | 56.70±2.02 | 45.60±3.24 | 14.98±2.54 | 23.45±2.97 | 34.61±2.84 | 3.72 |
| Shortest | E | 33.93±6.44 | 38.75±5.81 | 31.18±4.43 | 42.61±8.88 | 90.03±2.27 | 47.46±8.76 | 77.69±5.17 | 94.35±2.93 | 92.17±1.91 | 78.69±4.24 | 81.75±4.70 | 95.08±3.06 | 50.00 |
| | M | 26.07±4.96 | 28.84±4.56 | 19.89±3.05 | 33.92±7.68 | 81.17±3.03 | 35.53±6.80 | 69.60±5.50 | 91.27±2.84 | 84.84±2.93 | 66.31±3.36 | 80.67±4.15 | 92.60±3.49 | 50.00 |
| | H | 20.00±3.97 | 23.03±3.85 | 12.21±1.95 | 26.60±6.26 | 72.53±4.29 | 28.31±5.50 | 64.28±5.60 | 87.88±3.36 | 74.98±4.17 | 54.73±4.54 | 78.16±4.55 | 88.63±4.44 | 50.00 |
| Triangle | E | 9.49±1.02 | 14.97±1.53 | 11.87±1.32 | 12.88±2.05 | 36.57±4.40 | 18.56±1.24 | 41.36±4.63 | 43.41±1.64 | 36.32±1.54 | 18.51±1.39 | 50.33±2.31 | 84.54±0.56 | 2.13 |
| | M | 3.06±0.39 | 8.56±0.92 | 5.86±0.73 | 7.54±1.33 | 14.52±2.63 | 9.18±0.73 | 26.95±2.44 | 24.00±0.77 | 20.00±0.72 | 10.62±0.81 | 28.12±1.65 | 48.13±1.46 | 1.62 |
| | H | 1.82±0.36 | 4.95±0.69 | 2.55±0.44 | 4.38±1.04 | 4.73±1.58 | 4.45±0.58 | 19.54±1.34 | 15.92±0.72 | 12.81±0.88 | 5.65±0.71 | 15.55±1.29 | 17.53±1.43 | 1.82 |

### E.1.1    OVERALL RESULTS

Here we present a comprehensive analysis of the overall performance across all evaluation dimensions. While our main findings in Result ❶ highlight the moderate performance of models with considerable room for improvement, the detailed results in Tables 14, 15, and 16 reveal several noteworthy patterns:

**Task-specific Performance Variation:** The performance varies significantly across different tasks and difficulty levels. For instance, in Connectivity tasks, models generally achieve higher accuracy (80%–90% for easy level) compared to more complex tasks like Triangle counting (20%–30% for hard level). This suggests that while LLMs can handle basic graph properties well, they struggle with tasks requiring more sophisticated reasoning and counting.

**Difficulty Level Impact:** There is a consistent and non-linearly sharp decline in performance as task difficulty increases. With larger graphs, models face challenges in both processing longer contexts and conducting more complex reasoning tasks, which typically require longer reasoning paths, more precise intermediate steps, and more comprehensive exploration of the graph structure. The sharp performance drop on larger graphs suggests that current LLMs struggle to maintain reliable reasoning capabilities when faced with extended multi-step graph operations.

**Model Type Performance Gap:** The performance gap between closed-source and open-source models is particularly evident in complex tasks. For instance, GPT-4o and Claude-3.5 consistently outperform other models by a significant margin (15%–20%) in tasks like Diameter calculation and Triangle counting especially at higher difficulty levels. This reinforces our observation about the current limitations of open-source models in complex graph reasoning tasks.

**Graph Type Impact:** The evaluation reveals distinct performance patterns across different graph types, with certain structures showing clear advantages for specific tasks. Our analysis shows that

bipartite graphs (BERM, BERP) tend to exhibit higher performance in connectivity and clustering-related tasks (`Connectivity`), potentially due to their explicit partitioning of node sets, which simplifies certain connectivity relationships for LLMs. For shortest-path (`Shortest path`) tasks, hierarchical structures like BAF often show higher accuracy, as the tree-like paths may align well with reasoning processes for pathfinding. In local pattern identification tasks such as triangle counting (`Triangle counting`), simpler graph structures like SF often perform better, possibly because they reduce the complexity of identifying local patterns. These observations suggest that the interplay between graph types and task characteristics can significantly influence LLM reasoning behaviors.

Table 14: Benchmark Results of Prompt Schemes Across Tasks (Mean±95% CI Margin of All Models). **Bold orange** / Underlined blue highlights indicate best/second-best performance.

| Task | Difficulty | 0-Algorithm | 0-CoT | 0-Instruct | 0-Shot | Algorithm | CoT | Instruct | K-Shot | LTM |
|---|---|---|---|---|---|---|---|---|---|---|
| BFS order | E | 52.20±7.28 | 46.54±8.20 | 48.42±8.22 | 51.49±7.34 | **63.39±7.10** | 63.33±6.23 | 62.93±6.16 | 56.51±6.45 | 48.12±8.15 |
| | M | 33.63±6.46 | 33.37±6.89 | 33.36±6.76 | 33.94±6.72 | **43.48±7.23** | 38.99±6.37 | 38.61±6.20 | 33.08±6.09 | 32.89±6.67 |
| | H | 13.57±3.45 | 14.46±3.58 | 13.98±3.50 | 14.58±3.71 | **19.37±4.46** | 14.01±3.42 | 13.27±3.27 | 11.68±3.01 | 13.95±3.50 |
| Connectivity | E | 85.51±3.01 | 83.27±5.06 | 86.86±2.45 | 82.79±5.40 | 88.79±2.32 | 92.35±1.88 | **92.37±1.57** | 87.46±2.34 | 83.00±4.70 |
| | M | 86.34±3.34 | 83.30±4.89 | 83.80±3.28 | 83.36±5.45 | 88.98±2.05 | 91.98±1.94 | **92.07±1.45** | 89.24±2.05 | 83.65±4.17 |
| | H | 81.76±3.99 | 78.34±4.87 | 74.21±4.90 | 79.80±5.35 | 82.89±2.52 | 85.68±2.70 | **86.41±2.19** | 85.16±2.57 | 78.35±4.41 |
| Cycle | E | 71.75±3.91 | 64.73±5.07 | 70.23±3.67 | 64.93±5.34 | 71.28±4.34 | 73.01±3.62 | 71.66±3.77 | **73.12±3.62** | 68.89±3.90 |
| | M | 69.16±4.07 | 64.50±4.73 | 67.75±3.99 | 63.94±5.37 | 70.02±4.77 | **71.59±4.06** | 69.86±4.11 | 70.40±4.24 | 67.58±3.85 |
| | H | 66.27±4.04 | 62.75±4.47 | 62.14±4.60 | 62.54±5.20 | 67.38±4.79 | **69.12±4.20** | 67.95±4.13 | 68.35±4.14 | 66.35±3.73 |
| Diameter | E | 51.52±6.46 | 52.66±6.81 | 52.55±7.06 | 53.65±6.46 | **70.28±3.10** | 73.01±3.62 | 71.66±3.77 | 64.65±4.09 | 53.41±6.66 |
| | M | 34.41±5.35 | 36.65±5.50 | 34.34±5.62 | 37.54±5.07 | **50.69±2.97** | 46.65±4.66 | 48.00±4.06 | 47.64±3.93 | 35.83±5.37 |
| | H | 19.53±3.90 | 22.28±3.81 | 20.59±4.16 | 22.92±3.71 | 32.13±3.29 | 30.80±3.78 | **32.54±3.49** | 31.89±3.29 | 21.19±4.09 |
| Shortest | E | 67.58±5.64 | 57.06±8.17 | 55.89±8.48 | 67.18±6.33 | 74.79±6.01 | 74.73±6.10 | **75.62±5.76** | 72.75±6.37 | 57.17±7.99 |
| | M | 59.32±6.24 | 50.97±8.02 | 50.60±8.14 | 58.52±6.58 | 65.09±6.39 | **66.73±6.09** | 66.02±6.19 | 63.98±6.26 | 51.80±7.73 |
| | H | 53.14±6.66 | 48.40±7.89 | 46.88±8.05 | 52.54±6.82 | 56.18±6.72 | **57.27±6.41** | 57.10±6.38 | 54.03±6.43 | 47.97±7.67 |
| Triangle | E | 28.81±4.95 | 29.95±5.04 | 28.00±5.04 | 31.22±5.08 | 32.39±4.75 | 34.81±5.10 | **35.03±4.68** | 33.39±4.57 | 30.52±4.95 |
| | M | 16.01±2.92 | 17.07±2.83 | 15.88±2.93 | 16.97±3.10 | 16.62±2.86 | 18.07±3.15 | **18.67±2.77** | 18.52±2.78 | 17.11±2.89 |
| | H | 8.85±1.53 | **10.14±1.68** | 8.55±1.56 | 9.42±1.63 | 8.21±1.56 | 9.48±1.81 | 9.12±1.53 | 9.54±1.58 | 9.10±1.47 |

Table 15: Benchmark Results of Serialization Formats Across Tasks (Mean±95% CI Margin of All Models). **Bold orange** / Underlined blue highlights indicate best/second-best performance.

| Task | Difficulty | AL | AM | AS | EL | ES | GMaL | GMoL |
|---|---|---|---|---|---|---|---|---|
| BFS order | E | **63.27±6.63** | 49.10±7.02 | 62.54±6.63 | 50.40±6.27 | 51.68±6.37 | 58.86±6.42 | 47.54±5.57 |
| | M | **47.13±6.74** | 27.55±5.27 | 45.18±6.56 | 31.39±5.17 | 29.57±4.96 | 39.55±5.88 | 29.56±4.91 |
| | H | **23.92±4.37** | 5.19±1.16 | 23.59±4.24 | 11.40±2.17 | 9.06±1.69 | 15.58±2.75 | 11.50±2.31 |
| Connectivity | E | 89.49±3.25 | 80.92±3.10 | **89.57±3.20** | 87.02±3.42 | 88.50±3.21 | 88.44±2.96 | 84.58±2.31 |
| | M | 88.75±3.23 | 82.63±2.85 | **89.04±3.09** | 86.51±3.48 | 86.72±3.31 | 88.27±3.04 | 86.87±2.52 |
| | H | 85.52±3.48 | 68.37±2.46 | **85.68±3.38** | 82.49±3.65 | 81.03±3.52 | 83.83±3.70 | 82.87±3.09 |
| Cycle | E | 64.30±3.53 | 65.75±3.61 | 64.41±3.50 | 71.54±3.59 | 75.38±3.40 | **76.09±4.26** | 72.22±3.38 |
| | M | 63.35±3.58 | 62.90±3.51 | 63.23±3.52 | 70.55±3.81 | 72.51±3.64 | **73.93±4.43** | 71.70±3.94 |
| | H | 61.00±3.59 | 59.06±2.94 | 60.20±3.52 | 69.64±3.95 | 68.82±3.68 | **71.42±4.32** | 70.96±4.24 |
| Diameter | E | 58.31±5.29 | 58.63±4.95 | 61.33±5.09 | 54.95±5.24 | 54.51±5.33 | **62.28±4.87** | 58.68±5.18 |
| | M | 42.89±4.77 | 39.67±3.83 | **45.69±4.68** | 37.78±4.05 | 35.60±4.18 | 44.52±4.38 | 42.98±4.48 |
| | H | 27.68±4.00 | 23.65±3.01 | 29.61±3.82 | 23.26±2.76 | 20.03±2.70 | **29.77±3.63** | 27.90±3.49 |
| Shortest | E | 75.89±5.76 | 54.14±5.97 | **76.60±5.61** | 72.00±5.63 | 68.99±5.81 | 52.85±7.49 | 68.35±5.23 |
| | M | **69.65±6.00** | 40.94±5.24 | 69.14±5.68 | 64.57±5.83 | 58.30±5.90 | 52.38±7.34 | 59.60±5.25 |
| | H | 65.31±6.09 | 28.22±4.13 | **65.79±5.96** | 55.82±5.90 | 52.05±5.83 | 47.88±7.06 | 53.20±5.42 |
| Triangle | E | 32.03±4.41 | 27.61±4.08 | 31.82±4.48 | 31.70±4.21 | 30.64±4.06 | **34.30±4.44** | 32.89±4.61 |
| | M | 17.50±2.56 | 13.50±1.94 | 17.61±2.60 | 18.65±2.66 | 16.45±2.40 | **18.83±2.71** | 17.95±2.89 |
| | H | 8.78±1.42 | 6.61±1.05 | **10.35±1.51** | 9.77±1.38 | 8.75±1.22 | 10.34±1.48 | 9.50±1.59 |

### E.1.2 RESULTS OF OPEN-SOURCE MODELS

Open-source models exhibit several distinct characteristics compared to the overall results. In terms of prompting schemes (Table 17), more structured approaches show clear advantages: CoT and

Table 16: Benchmark Results of Graph Type Across Tasks (Mean±95% CI Margin). **Bold orange** / Underlined blue highlights indicate best/second-best performance. "-" indicates the graph type is not applicable for that task.

| Task | Difficulty | BAF | BAG | BERM | BERP | ERM | ERP | SF |
|------|-----------|-----|-----|------|------|-----|-----|-----|
| BFS order | E | 43.82±3.13 | 44.93±3.06 | **53.30±2.77** | 52.68±2.75 | 43.49±2.76 | 47.82±2.88 | 48.20±3.08 |
| | M | **35.06±3.03** | 29.38±2.68 | 24.96±2.33 | 34.56±2.58 | 21.76±1.97 | 22.48±2.05 | 26.63±2.57 |
| | H | **27.58±2.66** | 7.17±1.17 | 6.67±0.95 | 13.79±1.34 | 4.03±0.70 | 8.40±1.06 | 7.40±1.24 |
| Connectivity | E | 77.04±1.68 | - | **88.03±1.48** | 84.29±1.52 | 87.02±1.49 | 86.97±1.54 | - |
| | M | 78.99±1.62 | - | 86.31±1.52 | **86.60±1.54** | 84.61±1.51 | 86.11±1.56 | - |
| | H | 65.93±1.75 | - | 84.18±1.75 | 82.12±1.67 | 80.68±1.74 | **85.28±1.70** | - |
| Cycle | E | - | 64.90±1.60 | 65.98±1.49 | 65.02±1.39 | 68.32±1.70 | **69.25±1.57** | 61.08±1.40 |
| | M | - | 60.18±1.73 | **67.21±1.66** | 61.41±1.54 | 65.23±1.86 | 66.08±1.70 | 59.10±1.53 |
| | H | - | 55.20±1.56 | 64.66±1.91 | **65.58±1.72** | 61.97±1.82 | 62.99±1.81 | 54.98±1.45 |
| Diameter | E | - | 47.35±2.10 | - | - | 44.95±2.25 | 48.96±2.23 | **56.82±2.08** |
| | M | - | 33.38±1.79 | - | - | 30.64±2.04 | 35.81±2.21 | **37.41±1.65** |
| | H | - | 22.78±1.81 | - | - | 20.70±1.92 | **26.76±2.11** | 22.10±1.15 |
| Shortest path | E | **66.73±2.98** | 60.06±2.82 | 57.52±2.82 | 61.01±2.88 | 55.12±2.80 | 59.53±2.87 | 61.12±2.86 |
| | M | **58.11±3.01** | 52.88±2.84 | 48.95±2.76 | 48.72±2.72 | 45.83±2.73 | 51.55±2.75 | 57.08±2.88 |
| | H | **55.62±3.15** | 46.19±2.97 | 42.47±2.69 | 39.67±2.85 | 39.54±2.65 | 43.36±2.62 | 48.93±2.89 |
| Triangle | E | - | 25.25±1.53 | - | - | 12.54±0.79 | 17.17±1.08 | **41.20±2.03** |
| | M | - | 16.75±1.11 | - | - | 7.38±0.45 | 9.55±0.56 | **18.30±1.12** |
| | H | - | **8.99±0.77** | - | - | 5.48±0.42 | 7.56±0.56 | 8.22±0.58 |

Instruct prompts consistently outperform simpler schemes like 0-Shot and LTM across most tasks. This is particularly evident in `Connectivity` tasks, suggesting that open-source models benefit more from explicit reasoning guidance.

For serialization formats (Table 18), open-source models show a strong preference for concise representations. Adjacency List (AL) and Adjacency Set (AS) formats consistently perform better than more complex formats like GMaL and GMoL. This contrasts with the overall results.

Regarding graph types (Table 19), while the general pattern of task-specific advantages remains similar to overall results, open-source models show more pronounced performance gaps between optimal and sub-optimal graph types. For instance, in `Triangle counting` tasks, SF significantly outperforms other graph types with a wider margin compared to the overall results.

Table 17: Benchmark Results of Prompt Schemes Across Tasks of Open-source Models (Mean±95% CI Margin). **Bold orange** / Underlined blue highlights indicate best/second-best performance.

| Task | Difficulty | 0-Algorithm | 0-CoT | 0-Instruct | 0-Shot | Algorithm | CoT | Instruct | K-Shot | LTM |
|------|-----------|-------------|-------|------------|--------|-----------|-----|----------|--------|-----|
| BFS order | E | 29.72±5.04 | 20.25±5.44 | 22.19±5.76 | 29.64±5.43 | 44.34±6.33 | **49.24±6.00** | 48.84±5.85 | 41.62±5.48 | 22.50±5.82 |
| | M | 15.25±4.15 | 14.05±4.83 | 14.56±4.88 | 15.98±4.70 | 25.31±5.94 | **25.88±5.78** | 25.34±5.56 | 19.52±4.83 | 14.26±4.73 |
| | H | 7.26±3.03 | 7.76±3.14 | 7.24±3.04 | 7.98±3.11 | **10.51±3.54** | 9.47±3.53 | 9.18±3.40 | 6.55±2.61 | 7.28±3.05 |
| Connectivity | E | 78.11±2.96 | 74.21±5.84 | 81.31±2.35 | 74.83±6.44 | 85.70±2.46 | 89.23±2.10 | **89.46±1.64** | 84.83±2.42 | 74.19±5.33 |
| | M | 79.13±3.60 | 74.08±5.58 | 75.56±3.19 | 74.54±6.48 | 85.39±2.12 | 88.40±2.16 | **88.98±1.42** | 86.52±2.17 | 75.15±4.56 |
| | H | 74.52±4.29 | 69.13±5.34 | 62.30±4.72 | 71.41±6.22 | 80.14±2.19 | 81.68±2.63 | **83.14±1.86** | 82.85±2.48 | 69.05±4.51 |
| Cycle | E | 64.31±3.28 | 53.62±4.73 | 63.66±3.11 | 55.38±5.41 | 64.32±4.01 | 66.89±3.33 | 64.67±3.40 | **67.48±3.19** | 60.90±3.26 |
| | M | 61.02±3.52 | 53.61±4.13 | 59.75±3.36 | 53.78±5.41 | 61.55±4.57 | 63.81±3.73 | 62.44±3.63 | **64.06±3.88** | 59.66±3.12 |
| | H | 58.42±3.61 | 52.68±3.84 | 51.53±3.83 | 52.42±5.14 | 58.25±4.44 | 61.20±3.71 | 59.80±3.51 | **61.39±3.53** | 58.60±2.91 |
| Diameter | E | 36.41±5.49 | 37.12±5.95 | 37.36±6.51 | 40.67±5.67 | **66.33±2.63** | 60.39±3.27 | 60.39±3.27 | 59.93±3.51 | 39.13±5.92 |
| | M | 22.68±3.90 | 25.26±4.68 | 22.36±4.68 | 27.65±4.24 | **44.44±2.65** | 40.30±4.44 | 42.93±3.59 | 41.45±3.45 | 24.25±4.38 |
| | H | 11.69±2.25 | 15.25±2.78 | 12.84±3.17 | 16.71±2.83 | 24.70±2.52 | 25.50±3.29 | **27.98±3.00** | 27.22±2.93 | 13.57±3.00 |
| Shortest | E | 55.81±5.40 | 34.34±7.21 | 32.39±7.55 | 52.73±6.28 | 62.03±6.43 | **65.72±6.36** | 64.55±6.34 | 63.04±6.48 | 34.38±6.84 |
| | M | 43.89±5.43 | 28.21±6.58 | 28.25±6.95 | 42.44±5.92 | 50.31±6.24 | **53.78±6.20** | 52.94±6.29 | 50.08±6.15 | 29.44±6.13 |
| | H | 36.26±5.51 | 26.51±6.31 | 24.50±6.42 | 35.58±5.60 | 40.67±6.12 | **44.11±6.18** | 43.56±6.04 | 39.99±5.93 | 26.35±5.73 |
| Triangle | E | 14.74±1.98 | 16.91±3.09 | 14.22±2.52 | 19.88±3.43 | 21.98±3.28 | 27.32±4.52 | **28.14±3.84** | 26.35±3.34 | 17.79±3.07 |
| | M | 7.98±1.39 | 10.24±1.73 | 8.37±1.58 | 9.86±1.87 | 9.57±1.97 | 12.94±2.81 | **14.13±2.43** | 13.91±2.23 | 10.28±1.82 |
| | H | 5.14±1.22 | 6.28±1.48 | 4.50±1.19 | 5.96±1.31 | 4.71±1.31 | 7.35±1.99 | **7.53±1.70** | 7.49±1.57 | 5.57±1.08 |

Table 18: Benchmark Results of Serialization Formats Across Tasks of Open-source Models (Mean±95% CI Margin). **Bold orange** / Underlined blue highlights indicate best/second-best performance.

| Task | Difficulty | AL | AM | AS | EL | ES | GMaL | GMoL |
|---|---|---|---|---|---|---|---|---|
| BFS order | E | **42.12±5.89** | 25.68±5.21 | 41.02±5.77 | 30.04±4.94 | 30.01±4.73 | 39.54±5.84 | 31.41±4.67 |
| | M | **27.55±5.90** | 11.92±3.59 | 24.79±5.40 | 15.90±3.81 | 14.02±3.31 | 22.74±4.95 | 15.40±3.31 |
| | H | **15.45±4.36** | 1.99±0.72 | 14.60±4.10 | 5.87±1.72 | 4.63±1.27 | 8.99±2.41 | 5.41±1.55 |
| Connectivity | E | 83.33±3.84 | 74.63±3.50 | **83.54±3.79** | 80.79±3.99 | 82.76±3.78 | 82.97±3.43 | 81.23±2.45 |
| | M | 82.36±3.78 | 76.85±3.22 | **82.90±3.60** | 79.50±4.01 | 79.40±3.71 | 82.89±3.56 | 82.11±2.83 |
| | H | 78.43±3.98 | 66.16±2.66 | **78.89±3.86** | 74.26±4.04 | 72.53±3.73 | 77.01±4.25 | 77.11±3.48 |
| Cycle | E | 58.72±3.30 | 59.20±3.30 | 59.18±3.31 | 62.89±3.39 | **66.65±3.27** | 65.24±4.14 | 64.64±3.20 |
| | M | 57.70±3.29 | 55.73±3.00 | 57.82±3.24 | 60.81±3.56 | **63.40±3.40** | 62.44±4.28 | 61.86±3.81 |
| | H | 54.64±3.29 | 52.54±2.56 | 54.50±3.25 | 58.80±3.54 | 59.95±3.40 | **60.04±4.00** | 59.54±3.85 |
| Diameter | E | 47.25±5.10 | 49.01±4.53 | 49.78±4.91 | 45.06±4.89 | 43.82±4.78 | **53.68±4.95** | 48.95±5.18 |
| | M | 32.28±4.19 | 31.91±3.38 | 34.83±4.23 | 30.61±3.72 | 28.06±3.68 | **35.40±4.09** | 33.50±4.16 |
| | H | 18.13±3.07 | 19.70±2.24 | 20.57±3.23 | 19.65±2.70 | 16.12±2.39 | **22.44±2.89** | 19.87±2.82 |
| Shortest | E | 61.38±6.13 | 39.46±5.99 | **62.90±6.02** | 57.99±5.93 | 54.26±5.99 | 30.52±5.96 | 55.14±5.54 |
| | M | 52.45±5.93 | 27.69±4.83 | **54.63±5.79** | 47.29±5.55 | 41.24±5.35 | 25.75±5.41 | 45.99±5.22 |
| | H | 47.79±5.86 | 18.72±3.74 | **48.52±5.72** | 38.00±5.16 | 34.34±4.78 | 22.69±5.10 | 36.89±4.81 |
| Triangle | E | 20.03±3.02 | 17.87±2.49 | 18.98±2.82 | 20.52±2.70 | 21.13±2.77 | **24.76±3.74** | 22.42±3.50 |
| | M | 10.37±1.79 | 8.81±1.35 | 10.41±1.83 | 11.17±1.65 | 10.72±1.70 | **12.92±2.19** | 11.28±2.06 |
| | H | 5.37±1.34 | 5.45±1.07 | 6.76±1.52 | 5.84±1.13 | 6.44±1.25 | **7.07±1.39** | 5.50±1.28 |

Table 19: Benchmark Results of Graph Type Across Tasks of Open-source Models (Mean±95% CI Margin). **Bold orange** / Underlined blue highlights indicate best/second-best performance. "-" indicates the graph type is not applicable for that task.

| Task | Difficulty | BAF | BAG | BERM | BERP | ERM | ERP | SF |
|---|---|---|---|---|---|---|---|---|
| BFS order | E | 31.18±2.30 | 31.34±2.18 | **41.92±2.08** | 41.16±2.04 | 30.59±1.90 | 34.53±2.01 | 34.43±2.24 |
| | M | 23.15±2.23 | 19.97±1.90 | 17.92±1.79 | **24.76±1.87** | 14.89±1.41 | 15.08±1.41 | 18.27±1.86 |
| | H | **18.06±2.00** | 7.31±1.11 | 6.36±0.99 | 10.85±1.16 | 4.08±0.83 | 6.70±1.00 | 7.38±1.13 |
| Connectivity | E | 72.16±1.43 | - | **86.82±1.41** | 81.79±1.41 | 84.65±1.40 | 84.10±1.45 | - |
| | M | 74.84±1.43 | - | 83.02±1.42 | **84.28±1.45** | 80.99±1.38 | 82.86±1.44 | - |
| | H | 60.68±1.43 | - | **82.25±1.62** | 79.76±1.55 | 75.94±1.55 | 81.89±1.59 | - |
| Cycle | E | - | 60.70±1.34 | 62.16±1.36 | 63.53±1.36 | 63.61±1.44 | **64.51±1.39** | 59.65±1.33 |
| | M | - | 56.82±1.36 | **62.69±1.43** | 59.75±1.40 | 60.91±1.48 | 62.21±1.44 | 57.42±1.33 |
| | H | - | 52.15±1.27 | 59.50±1.52 | **61.04±1.48** | 57.73±1.49 | 59.09±1.46 | 53.36±1.23 |
| Diameter | E | - | 45.50±1.84 | - | - | 45.74±2.04 | 48.57±1.99 | **53.09±1.83** |
| | M | - | 32.72±1.52 | - | - | 28.87±1.60 | 31.85±1.76 | **36.16±1.45** |
| | H | - | 19.95±1.26 | - | - | 16.46±1.24 | 20.54±1.33 | **20.90±1.04** |
| Shortest | E | **59.63±2.63** | 51.54±2.37 | 49.30±2.39 | 52.61±2.46 | 46.04±2.32 | 50.04±2.37 | 52.97±2.46 |
| | M | **49.10±2.53** | 42.32±2.25 | 38.87±2.22 | 39.58±2.23 | 36.20±2.12 | 41.70±2.18 | 46.98±2.36 |
| | H | **46.37±2.59** | 34.28±2.16 | 33.36±2.10 | 30.18±2.18 | 30.50±1.99 | 33.90±2.06 | 38.55±2.22 |
| Triangle | E | - | 19.17±1.27 | - | - | 11.68±0.95 | 13.80±1.02 | **35.37±1.73** |
| | M | - | 11.15±0.85 | - | - | 7.62±0.72 | 8.66±0.87 | **15.28±1.05** |
| | H | - | 5.00±0.49 | - | - | 5.67±0.66 | 5.78±0.53 | **8.13±0.79** |

### E.1.3 RESULTS OF CLOSED-SOURCE MODELS

Closed-source models exhibit notably different characteristics compared to their open-source counterparts. For prompting schemes (Table 20), these models show more robust performances across different prompting methods, with even simple prompts like 0-Shot achieving competitive results. This is particularly evident in `Connectivity` tasks, where performance remains consistently high across most prompting schemes, suggesting less reliance on explicit reasoning guidance.

The serialization format results (Table 21) reveal another key distinction: closed-source models handle complex formats more effectively. While they perform well with concise formats like AL and AS, they also show strong performance with structured formats like GMaL, especially in tasks requiring sophisticated reasoning like `Cycle detection` and `Diameter calculation`. This contrasts sharply with open-source models' preference for simpler formats.

Regarding graph types (Table 22), closed-source models demonstrate more balanced performance across different graph structures. For instance, in `Triangle counting` tasks, while SF still performs best, the performance gap between different graph types is notably smaller than in open-source models, suggesting more robust graph structure processing capabilities.

Table 20: Benchmark Results of Prompt Schemes Across Tasks of Closed-source Models (Mean±95% CI Margin). **Bold orange** / Underlined blue highlights indicate best/second-best performance.

| Task | Difficulty | 0-Algorithm | 0-CoT | 0-Instruct | 0-Shot | Algorithm | CoT | Instruct | K-Shot | LTM |
|---|---|---|---|---|---|---|---|---|---|---|
| BFS order | E | 83.66±3.70 | 83.34±3.61 | 85.15±3.08 | 82.07±3.81 | **90.06±3.02** | 83.07±3.53 | 82.66±3.60 | 77.35±4.92 | 83.99±3.24 |
| | M | 59.37±4.92 | 60.42±4.91 | 59.67±4.68 | 59.09±5.30 | **68.91±5.06** | 57.33±4.97 | 57.17±4.72 | 52.05±5.35 | 58.98±4.71 |
| | H | 22.41±3.11 | 23.85±3.19 | 23.41±3.10 | 23.82±3.61 | **31.79±4.30** | 20.38±2.78 | 19.01±2.68 | 18.86±2.90 | 23.29±3.12 |
| Connectivity | E | 95.86±1.05 | 95.95±1.18 | 94.64±1.41 | 93.10±1.77 | **96.72±0.94** | 96.43±0.92 | 96.39±0.86 | 91.15±2.01 | 95.34±1.27 |
| | M | 96.43±0.87 | 96.22±1.01 | 95.34±1.08 | 95.71±0.96 | 94.01±1.35 | **96.99±0.74** | 96.39±0.86 | 93.06±1.54 | 95.56±1.26 |
| | H | **91.90±2.12** | 91.24±2.07 | 90.89±2.19 | 91.55±2.04 | 86.76±2.74 | 91.28±2.34 | 90.98±2.27 | 88.40±2.57 | 91.37±2.23 |
| Cycle | E | **82.17±3.67** | 80.28±3.42 | 79.43±3.52 | 78.30±3.73 | 81.03±3.95 | 81.58±3.22 | 81.44±3.27 | 81.01±3.54 | 80.08±3.46 |
| | M | 80.54±3.54 | 79.75±3.45 | 78.96±3.56 | 78.15±3.55 | 81.87±3.83 | **82.49±3.33** | 80.25±3.73 | 79.27±4.03 | 78.67±3.54 |
| | H | 77.26±3.40 | 76.84±3.45 | 76.99±3.62 | 76.70±3.54 | 80.15±3.87 | **80.20±3.71** | 79.35±3.73 | 78.08±4.07 | 77.19±3.54 |
| Diameter | E | 72.68±4.82 | 74.42±5.02 | 73.82±5.02 | 71.82±5.47 | **75.80±3.36** | 70.29±4.77 | 70.07±4.65 | 71.25±4.47 | 73.39±5.17 |
| | M | 50.83±5.17 | 52.60±4.72 | 51.12±4.85 | 51.37±4.68 | **59.43±2.32** | 55.54±4.35 | 55.09±4.23 | 56.29±3.86 | 52.05±4.71 |
| | H | 30.49±4.51 | 32.11±4.10 | 31.43±4.30 | 31.62±4.03 | **42.53±2.94** | 38.21±3.90 | 38.93±3.70 | 38.44±3.31 | 31.85±4.34 |
| Shortest | E | 84.07±3.82 | 88.87±2.80 | 88.79±2.82 | 87.42±3.00 | **92.65±2.03** | 87.34±4.55 | 91.11±2.21 | 86.34±4.99 | 89.08±2.76 |
| | M | 80.93±3.96 | 82.83±3.74 | 81.91±3.79 | 81.03±3.90 | **85.77±3.19** | 84.86±3.11 | 84.34±3.27 | 83.45±3.41 | 83.09±3.71 |
| | H | 76.76±4.53 | **79.05±4.48** | 78.22±4.56 | 76.28±4.86 | 77.90±4.42 | 75.69±4.35 | 76.05±4.32 | 73.70±4.50 | 78.23±4.84 |
| Triangle | E | **48.50±4.79** | 48.20±4.65 | 47.29±4.80 | 47.11±5.09 | 46.96±4.77 | 45.29±5.08 | 44.68±5.02 | 43.25±5.25 | 48.33±4.58 |
| | M | **27.25±2.82** | 26.62±2.82 | 26.38±2.91 | 26.92±3.24 | 26.47±2.58 | 25.25±2.98 | 25.02±2.66 | 24.98±2.90 | 26.67±2.88 |
| | H | 14.04±1.22 | **15.54±1.20** | 14.23±1.14 | 14.27±1.47 | 13.11±1.26 | 12.46±1.29 | 11.34±1.11 | 12.41±1.38 | 14.04±1.28 |

### E.2 PERFORMANCE HEATMAPS ACROSS TASKS

In this section, we provide detailed visualizations of model performance through heatmaps, extending the example shown in Figure 4. These heatmaps illustrate the interaction between prompting schemes and serialization formats across different tasks and difficulty levels, offering a comprehensive view of how various methodological combinations affect model performance.

### E.2.1 HEATMAPS FOR BFS order TASK

As shown in Figure 8 (featuring Claude-3.5, GPT-4o, GPT-4o-mini, Gemini-2.0), Figure 9 (featuring Llama-3 (8B), Llama-3.1 (8B), Mistral (7B), Phi-4 (14B)), Figure 10 (featuring Qwen-2.5 (7B), o4-mini), the heatmaps compare different prompt strategies and graph serialization formats under easy, medium, and hard difficulties for the `BFS order` task. The color intensity encodes accuracy (darker = higher), and solid/dashed boxes highlight best/second–best combinations, respectively.

Table 21: Benchmark Results of Serialization Formats Across Tasks of Closed-source Models (Mean±95% CI Margin), **Bold orange** / Underlined blue : best performance, Underlined and blue highlight: second best performance

| Task | Difficulty | AL | AM | AS | EL | ES | GMaL | GMoL |
|------|-----------|-----|-----|-----|-----|-----|------|------|
| BFS order | E | **92.88±1.83** | 81.89±3.82 | 92.68±1.81 | 78.90±3.23 | 82.02±3.12 | 85.90±2.55 | 70.12±3.57 |
| | M | **74.53±3.82** | 49.43±4.43 | 73.71±3.56 | 53.07±3.89 | 51.34±3.83 | 63.09±3.86 | 49.39±4.33 |
| | H | 35.79±3.30 | 9.67±1.14 | **36.16±3.18** | 19.14±1.89 | 15.26±1.50 | 24.81±2.21 | 20.02±2.24 |
| Connectivity | E | **98.11±0.54** | 89.74±1.12 | 98.03±0.54 | 95.74±1.13 | 96.53±0.99 | 96.11±1.00 | 89.27±1.78 |
| | M | **97.70±0.54** | 90.73±0.99 | 97.63±0.59 | 96.32±0.89 | 96.96±0.83 | 95.81±1.00 | 93.53±1.20 |
| | H | **95.46±0.98** | 71.48±2.03 | 95.19±1.08 | 94.01±1.09 | 92.93±1.27 | 93.39±1.48 | 90.94±1.44 |
| Cycle | E | 72.12±3.34 | 74.92±3.33 | 71.72±3.33 | 83.66±2.45 | 87.60±1.93 | **91.29±2.34** | 82.83±2.51 |
| | M | 71.27±3.45 | 72.94±3.32 | 70.82±3.43 | 84.19±2.40 | 85.26±2.40 | **90.03±2.39** | 85.48±2.35 |
| | H | 69.91±3.34 | 68.19±2.55 | 68.17±3.35 | 84.81±2.42 | 81.23±2.64 | **87.34±2.64** | 86.94±2.62 |
| Diameter | E | 73.80±4.05 | 72.10±4.41 | **77.50±3.57** | 68.80±4.60 | 69.47±4.79 | 74.33±3.76 | 72.30±3.99 |
| | M | 57.75±4.10 | 50.53±3.49 | **60.91±3.70** | 47.82±3.74 | 46.15±4.07 | 57.28±3.59 | 56.24±3.66 |
| | H | 41.04±3.81 | 29.17±3.62 | **42.25±3.30** | 28.32±2.59 | 25.52±2.79 | 40.03±3.70 | 39.15±3.24 |
| Shortest | E | **96.21±1.37** | 74.70±3.14 | 95.78±1.55 | 91.61±1.89 | 89.61±2.27 | 84.10±5.04 | 86.86±1.44 |
| | M | **93.72±1.32** | 59.50±3.52 | 89.45±2.34 | 88.76±1.72 | 82.18±2.96 | 89.66±1.81 | 78.67±2.45 |
| | H | 89.85±2.04 | 41.52±3.27 | **89.97±1.92** | 80.77±2.87 | 76.84±3.50 | 83.15±2.67 | 76.04±2.49 |
| Triangle | E | 48.83±4.09 | 41.24±4.50 | **49.80±4.18** | 47.36±4.18 | 43.95±4.27 | 47.65±4.15 | 47.53±4.59 |
| | M | 27.47±2.30 | 20.06±1.98 | 27.69±2.33 | **29.13±2.55** | 24.48±2.43 | 27.10±2.61 | 27.28±2.98 |
| | H | 13.56±0.96 | 8.24±0.96 | **15.38±0.86** | 15.28±0.98 | 11.97±0.88 | 14.92±1.15 | 15.11±1.38 |

Table 22: Benchmark Results of Graph Type Across Tasks of Closed-source Models (Mean±95% CI Margin). **Bold orange** / Underlined blue highlights indicate best/second-best performance. "-" indicates the graph type is not applicable for that task.

| Task | Difficulty | BAF | BAG | BERM | BERP | ERM | ERP | SF |
|------|-----------|-----|-----|------|------|-----|-----|-----|
| BFS order | E | 83.62±1.34 | 83.41±1.40 | 84.98±1.23 | 85.73±1.17 | 77.72±1.50 | 83.45±1.29 | **86.16±1.29** |
| | M | **75.54±1.65** | 63.93±1.91 | 55.38±1.81 | 67.58±1.67 | 45.67±1.68 | 48.93±1.68 | 60.08±1.96 |
| | H | **61.08±1.79** | 18.36±1.33 | 16.07±1.06 | 28.99±1.18 | 10.62±0.91 | 20.46±1.22 | 19.04±1.41 |
| Connectivity | E | 92.90±0.74 | - | 95.59±0.47 | 94.17±0.57 | 96.03±0.40 | **96.18±0.42** | - |
| | M | 93.29±0.66 | - | **97.06±0.32** | 96.53±0.39 | 95.53±0.40 | 95.92±0.45 | - |
| | H | 83.06±1.27 | - | 92.36±0.87 | 91.62±0.88 | 92.49±0.74 | **95.46±0.49** | - |
| Cycle | E | - | 81.22±1.34 | 81.07±1.20 | 78.68±1.23 | 82.18±1.40 | **83.77±1.20** | 76.63±1.32 |
| | M | - | 77.94±1.39 | **83.95±1.17** | 77.45±1.27 | 82.07±1.41 | 82.82±1.14 | 75.76±1.32 |
| | H | - | 71.11±1.49 | 83.02±1.19 | **83.38±1.14** | 79.35±1.36 | 80.67±1.33 | 70.99±1.41 |
| Diameter | E | - | 72.16±1.69 | - | - | 67.11±1.89 | 71.21±1.76 | **79.98±1.25** |
| | M | - | 53.15±1.69 | - | - | 48.55±1.74 | **57.07±1.55** | 56.48±1.47 |
| | H | - | 34.07±1.62 | - | - | 34.54±1.74 | **42.31±1.77** | 29.36±0.94 |
| Shortest | E | 90.09±1.26 | 89.28±1.17 | 85.05±1.24 | 88.66±1.23 | 86.36±1.20 | **90.60±1.09** | 88.84±1.20 |
| | M | 85.09±1.50 | 85.51±1.26 | 80.90±1.17 | 80.35±1.19 | 79.25±1.45 | 83.96±1.26 | **86.53±1.31** |
| | H | 80.55±2.03 | **80.74±1.66** | 74.31±1.54 | 75.11±1.75 | 71.53±1.69 | 75.63±1.48 | 80.21±1.67 |
| Triangle | E | - | 51.58±1.78 | - | - | 29.68±1.53 | 37.19±1.69 | **68.04±1.80** |
| | M | - | 34.13±1.01 | - | - | 14.28±0.59 | 18.81±0.75 | **37.47±1.66** |
| | H | - | **17.97±0.62** | - | - | 8.48±0.38 | 13.06±0.48 | 14.45±0.60 |

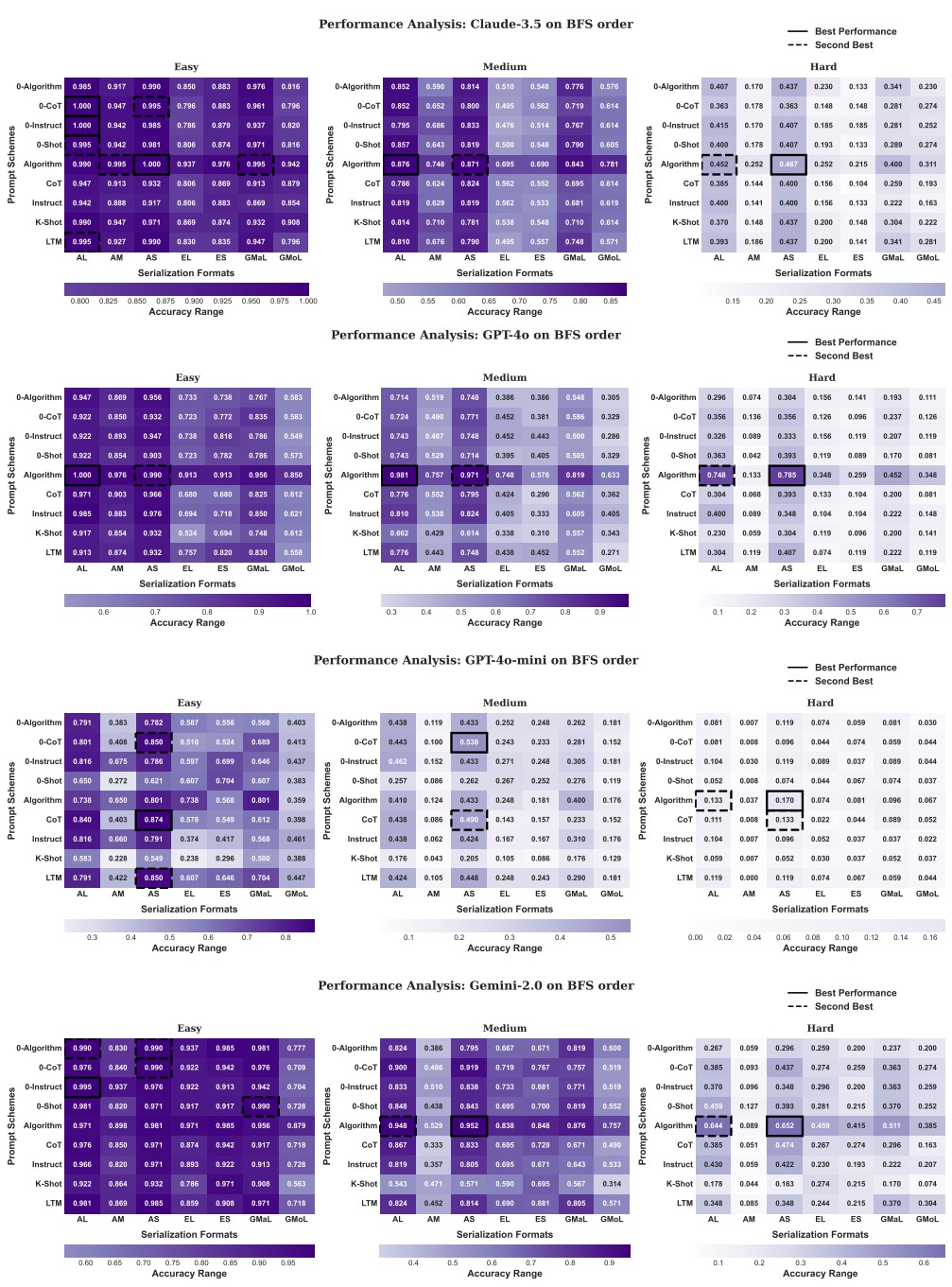

Figure 8: Performance heatmaps for prompt strategies and serialization formats on the BFS order task (Part 1). Models: Claude-3.5, GPT-4o, GPT-4o-mini, Gemini-2.0.

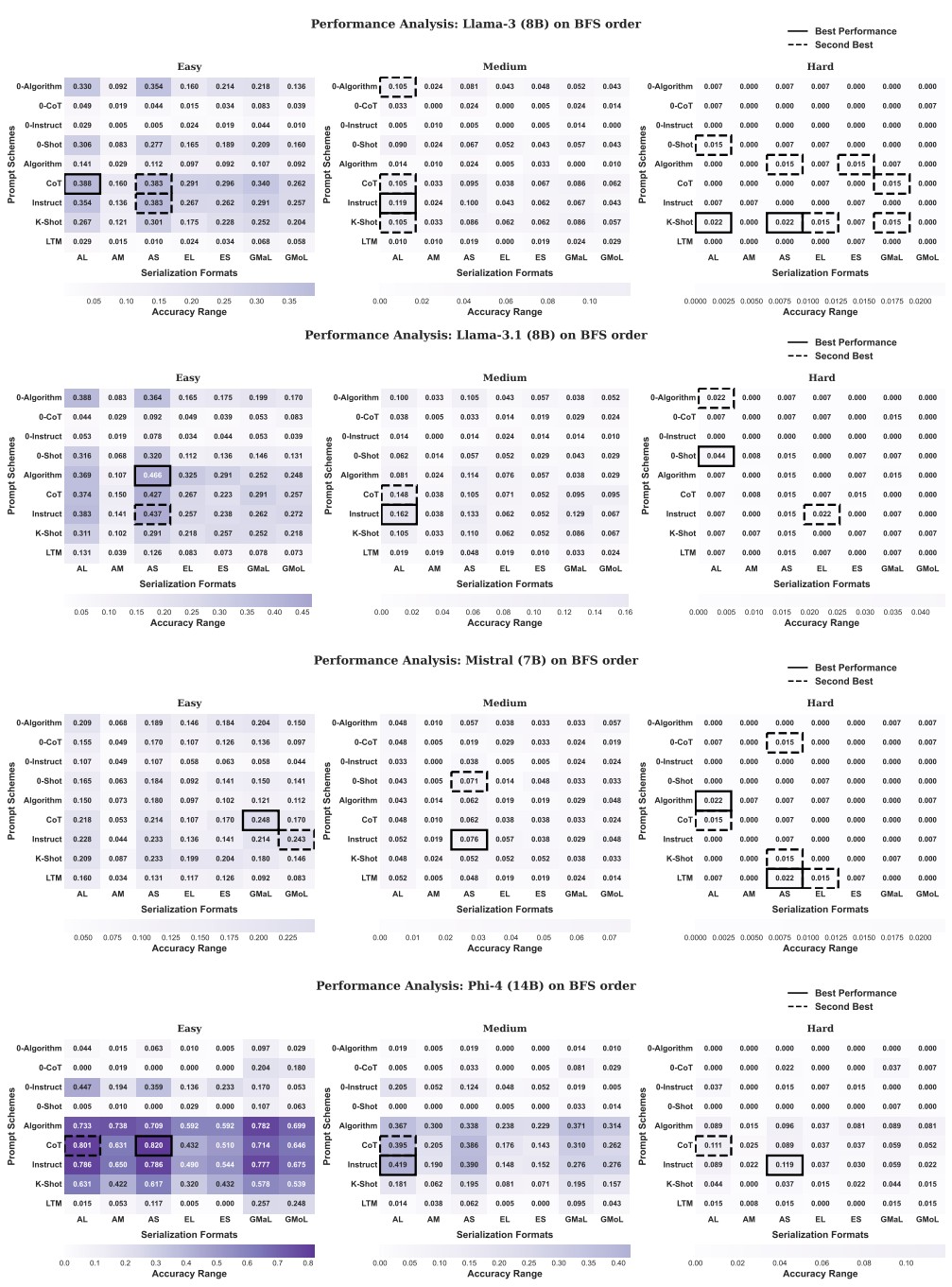

Figure 9: Performance heatmaps for prompt strategies and serialization formats on the BFS order task (Part 2). Models: Llama-3 (8B), Llama-3.1 (8B), Mistral (7B), Phi-4 (14B).

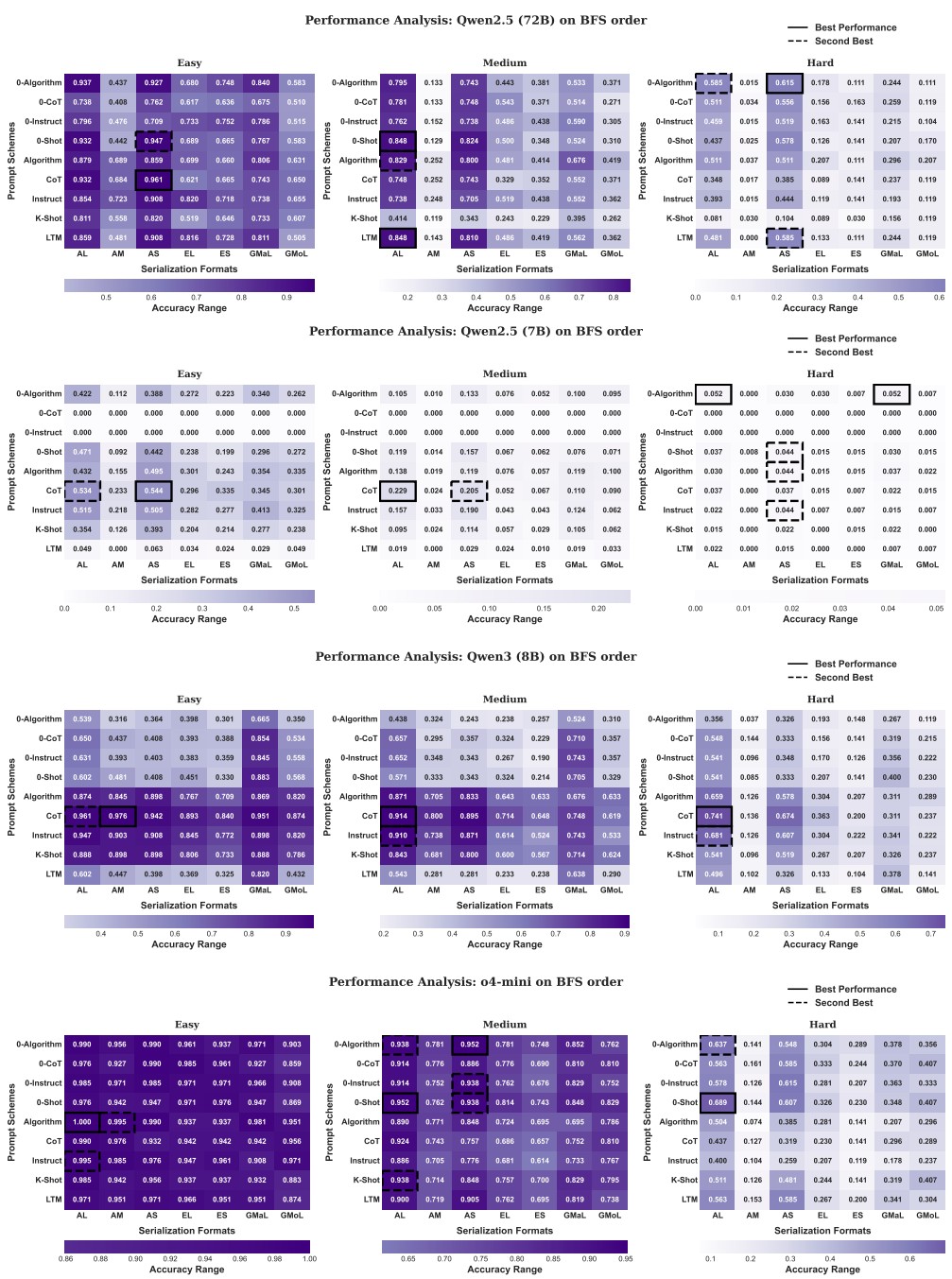

Figure 10: Performance heatmaps for prompt strategies and serialization formats on the BFS order task (Part 3). Models: Qwen-2.5 (72B), Qwen-2.5 (7B), Qwen-3 (8B), o4-mini.

### E.2.2 HEATMAPS FOR Connectivity TASK

As shown in Figure 11 (featuring Claude-3.5, GPT-4o, GPT-4o-mini, Gemini-2.0), Figure 12 (featuring Llama-3 (8B), Llama-3.1 (8B), Mistral (7B), Phi-4 (14B)) , Figure 13 (featuring Qwen-2.5 (7B), o4-mini) , the heatmaps compare different prompt strategies and graph serialization formats under easy, medium, and hard difficulties for the Connectivity task. The color intensity encodes accuracy (darker = higher), and solid/dashed boxes highlight best/second–best combinations respectively.

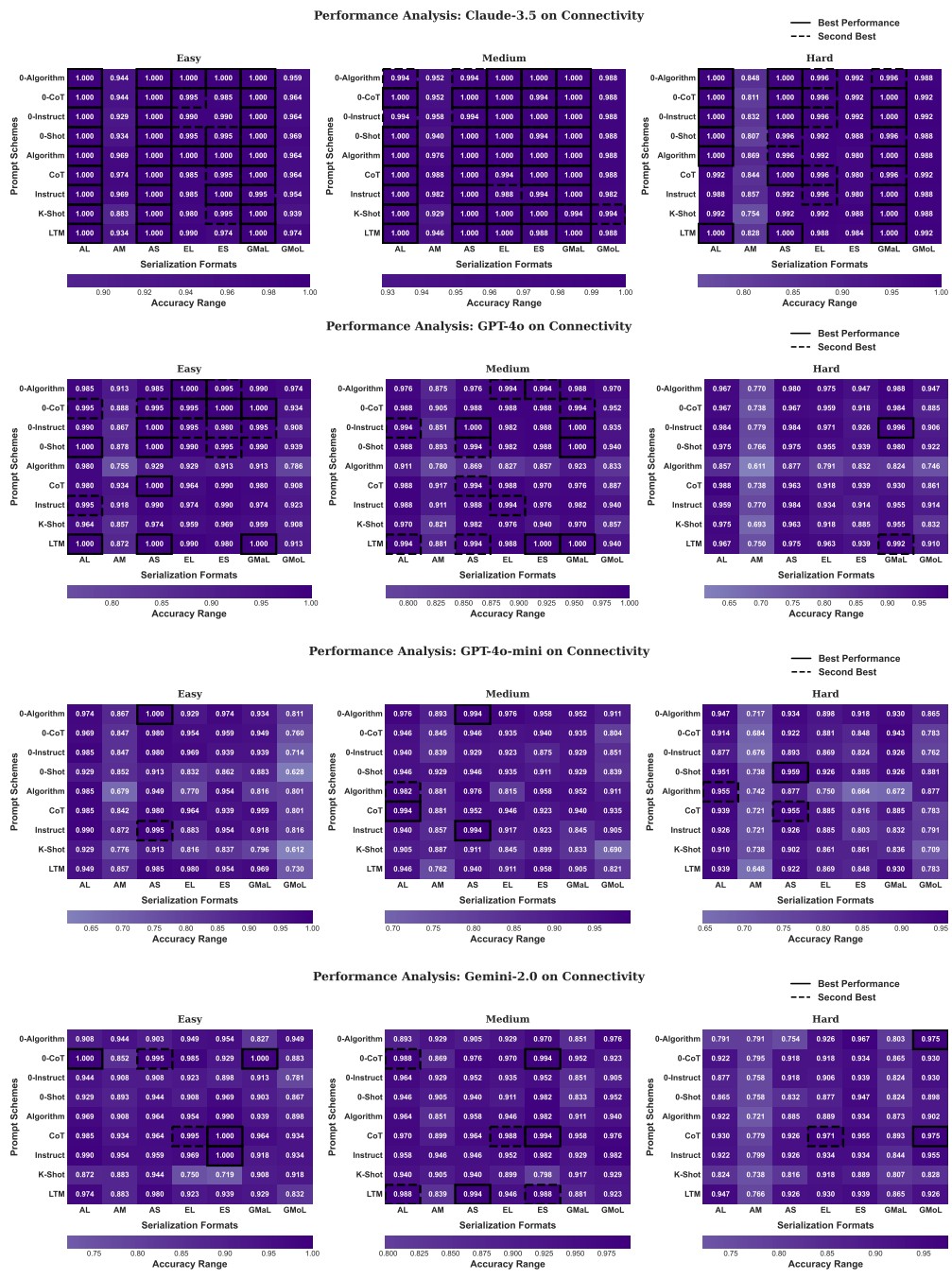

Figure 11: Performance heatmaps for prompt strategies and serialization formats on the Connectivity task (Part 1). Models: Claude-3.5, GPT-4o, GPT-4o-mini, Gemini-2.0.

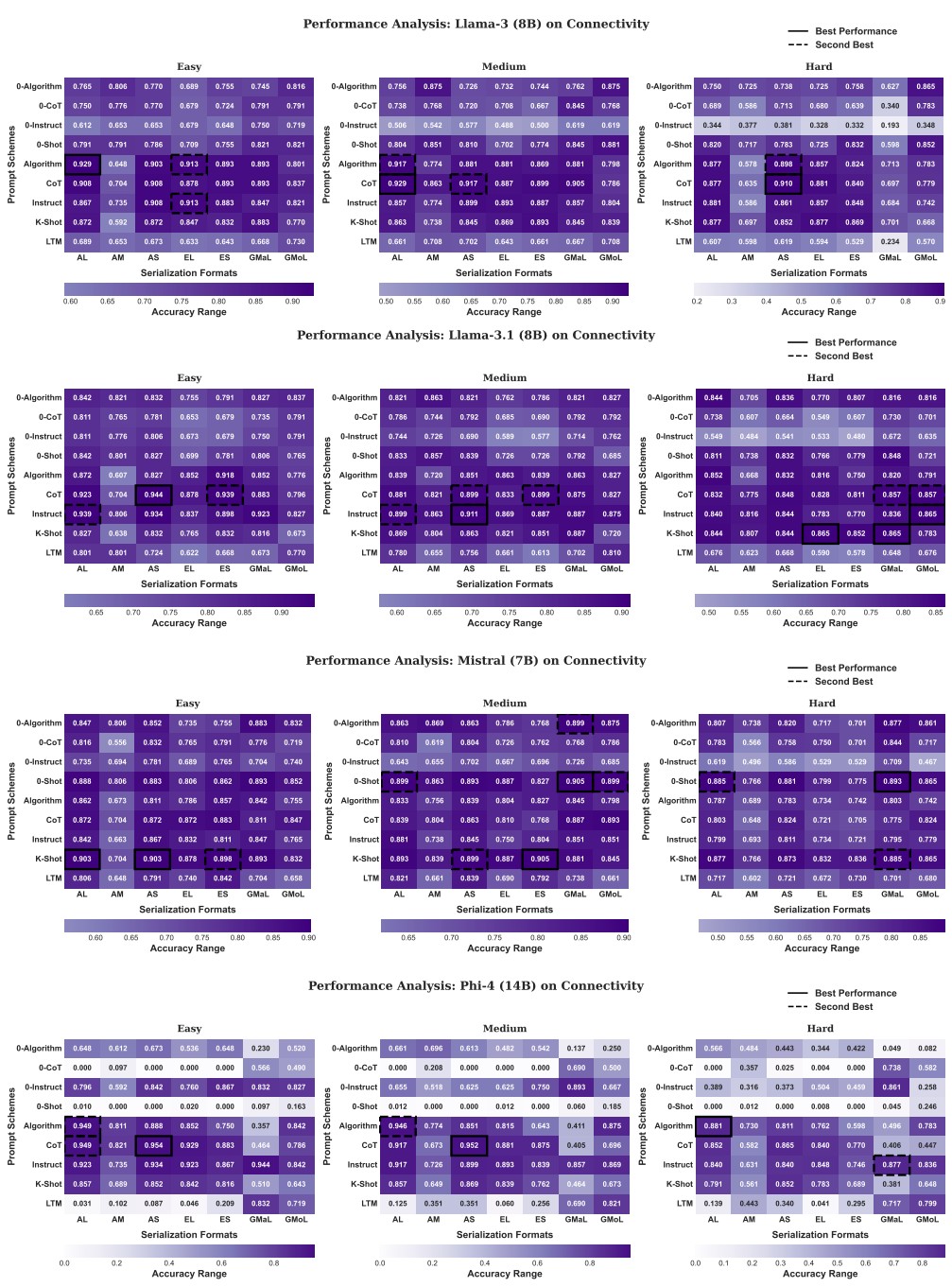

Figure 12: Performance heatmaps for prompt strategies and serialization formats on the Connectivity task (Part 2). Models: Llama-3 (8B), Llama-3.1 (8B), Mistral (7B), Phi-4 (14B).

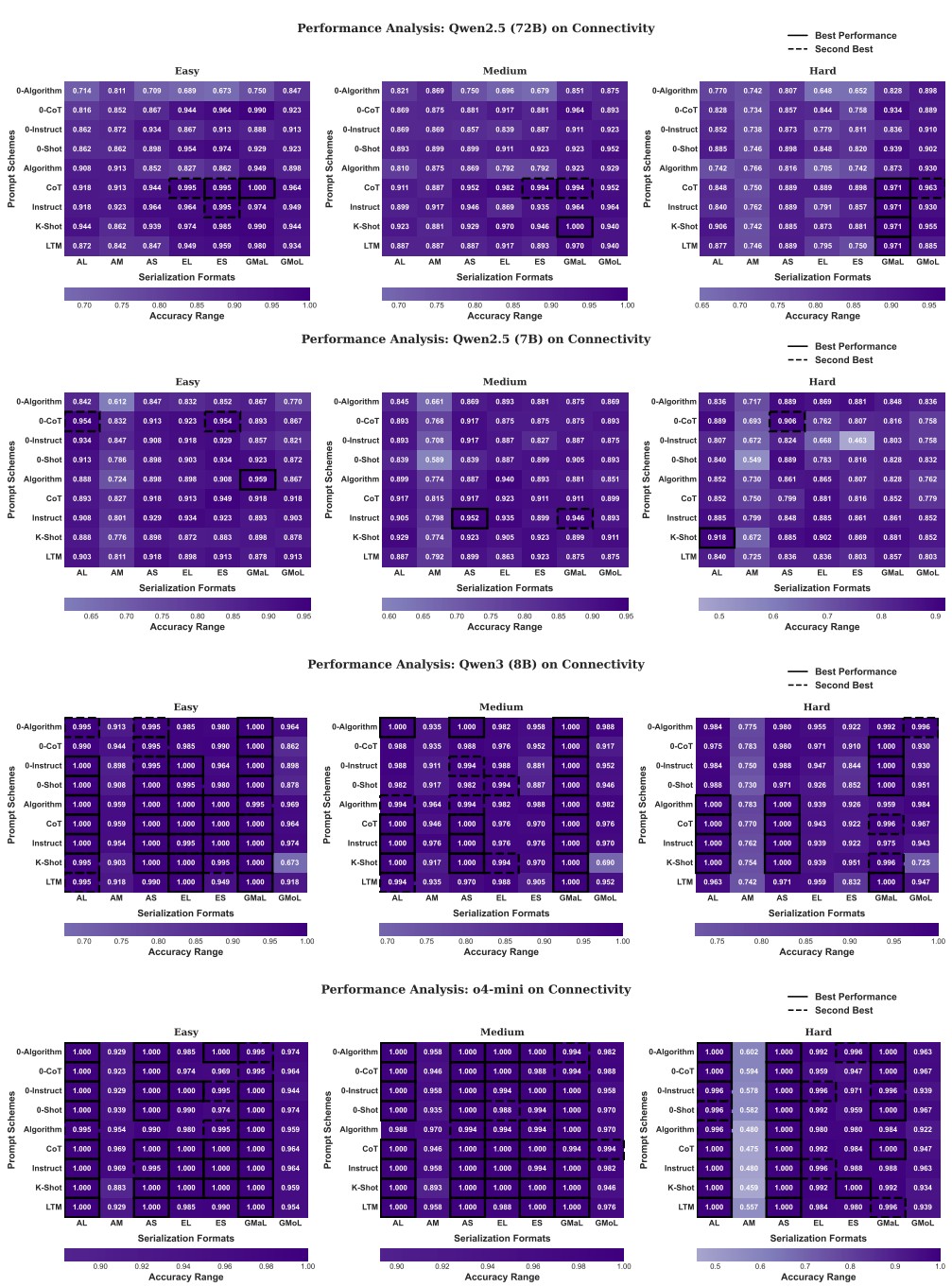

Figure 13: Performance heatmaps for prompt strategies and serialization formats on the Connectivity task (Part 3). Models: Qwen-2.5 (72B), Qwen-2.5 (7B), Qwen-3 (8B), o4-mini.

### E.2.3 HEATMAPS FOR Cycle detection TASK

As shown in Figure 14 (featuring Claude-3.5, GPT-4o, GPT-4o-mini, Gemini-2.0), Figure 15 (featuring Llama-3 (8B), Llama-3.1 (8B), Mistral (7B), Phi-4 (14B)), Figure 16 (featuring Qwen-2.5 (7B), o4-mini), the heatmaps compare different prompt strategies and graph serialization formats under easy, medium, and hard difficulties for the Cycle detection task. The color intensity encodes accuracy (darker = higher), and solid/dashed boxes highlight best/second–best combinations respectively.

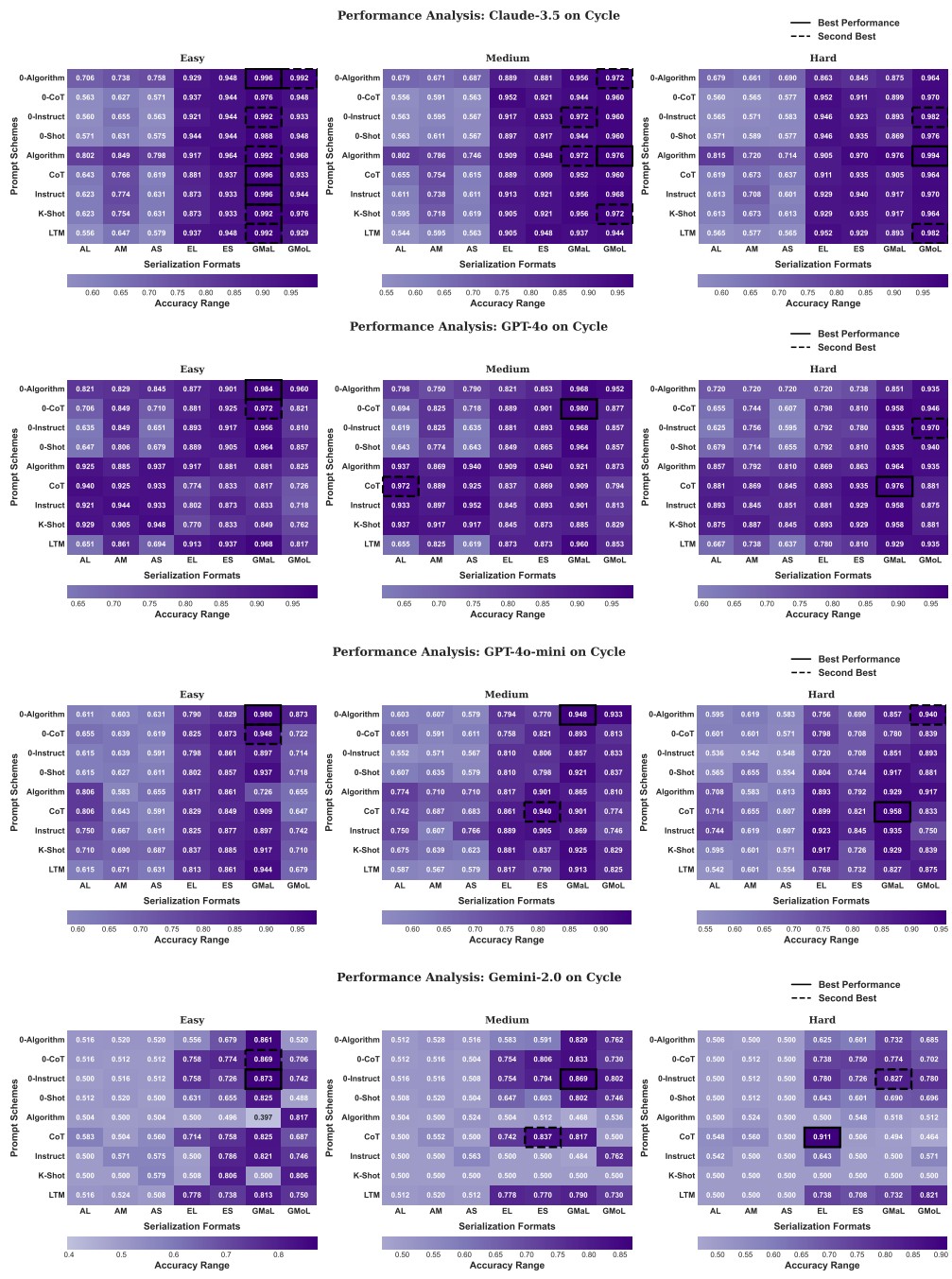

Figure 14: Performance heatmaps for prompt strategies and serialization formats on the Cycle task (Part 1). Models: Claude-3.5, GPT-4o, GPT-4o-mini, Gemini-2.0.

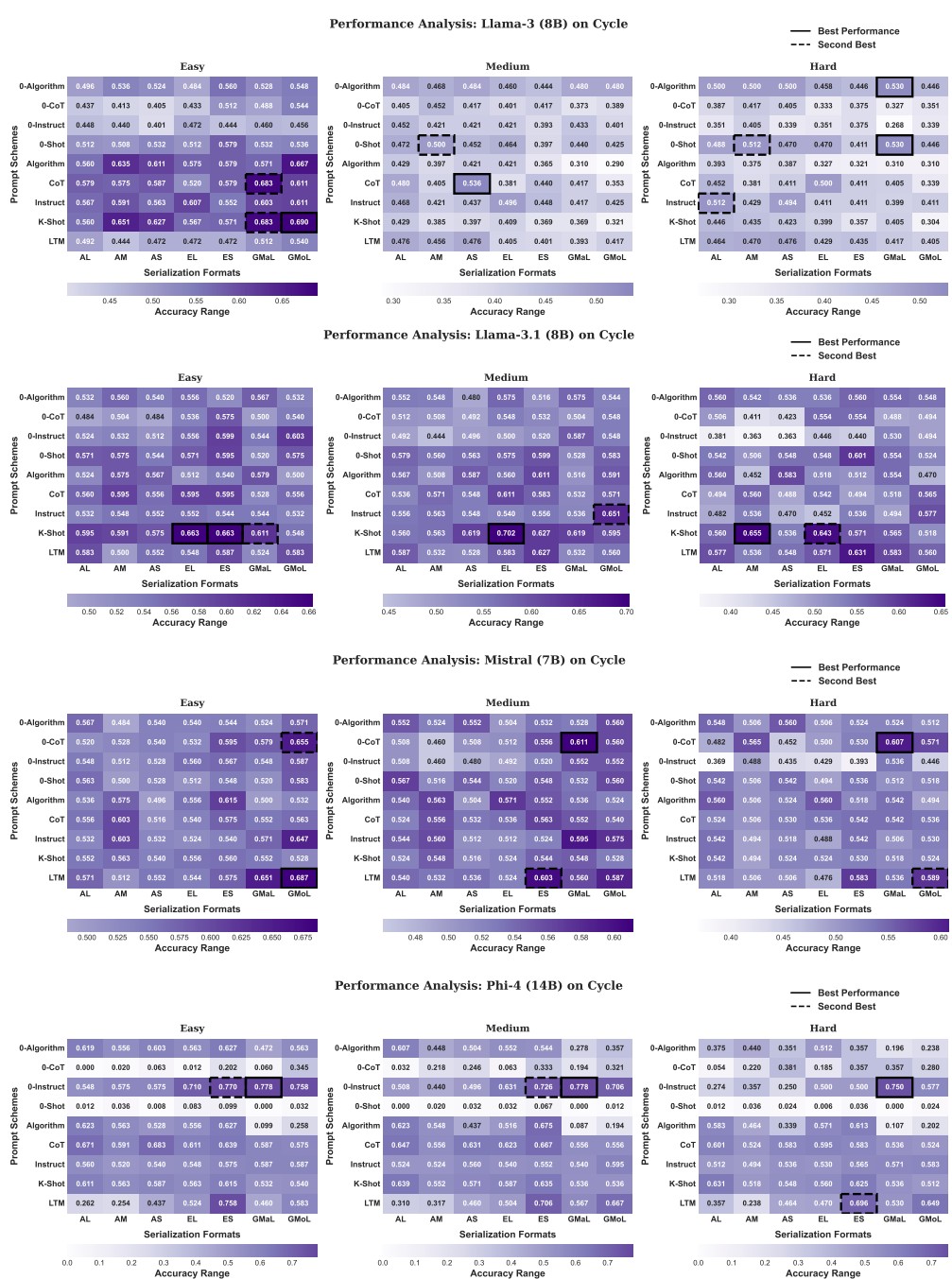

Figure 15: Performance heatmaps for prompt strategies and serialization formats on the Cycle task (Part 2). Models: Llama-3 (8B), Llama-3.1 (8B), Mistral (7B), Phi-4 (14B).

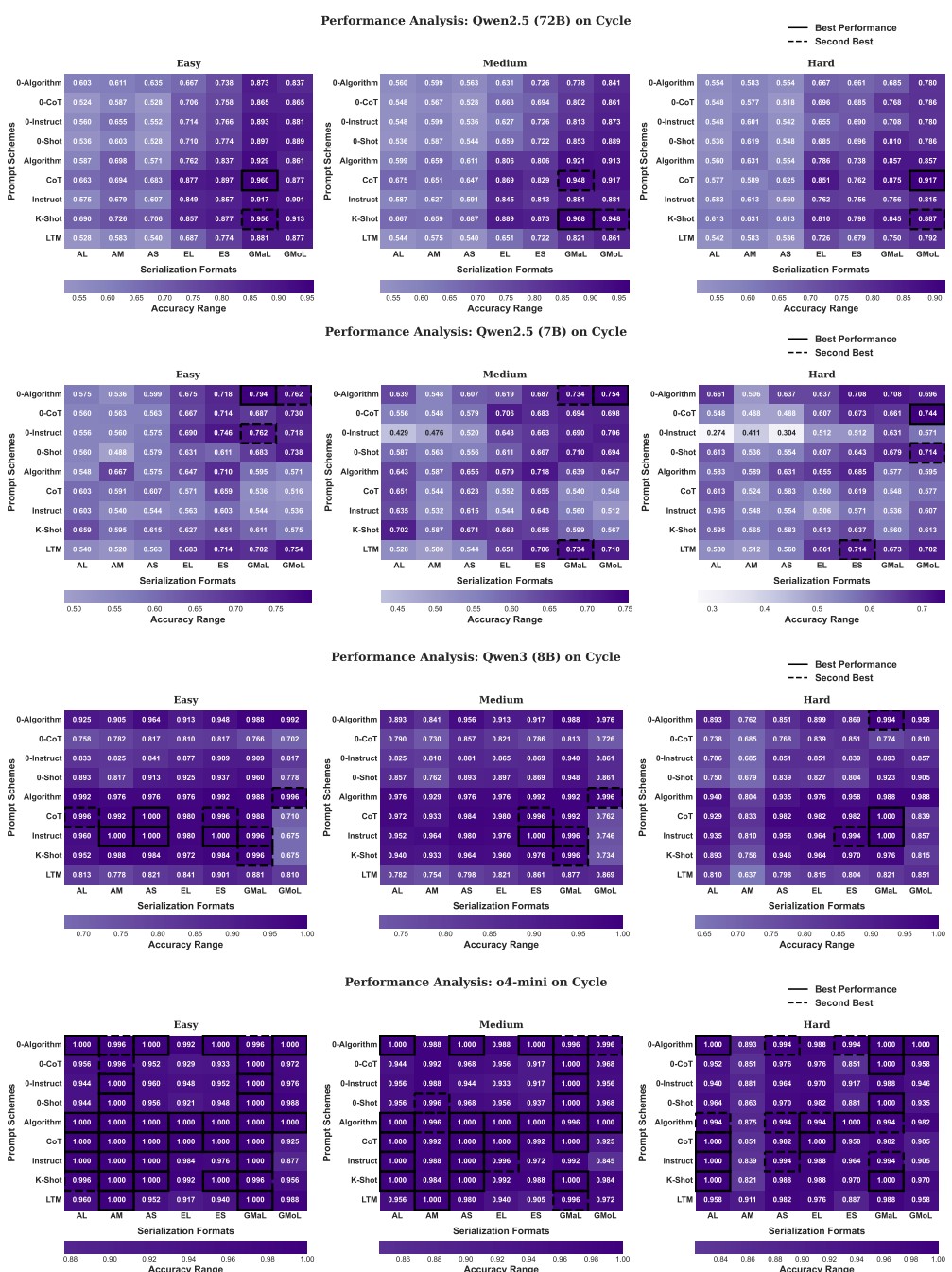

Figure 16: Performance heatmaps for prompt strategies and serialization formats on the Cycle task (Part 3). Models: Qwen-2.5 (72B), Qwen-2.5 (7B), Qwen-3 (8B), o4-mini.

### E.2.4  HEATMAPS FOR `Diameter calculation` TASK

As shown in Figure 17 (featuring Claude-3.5, GPT-4o, GPT-4o-mini, Gemini-2.0), Figure 18 (featuring Llama-3 (8B), Llama-3.1 (8B), Mistral (7B), Phi-4 (14B)), Figure 19 (featuring Qwen-2.5 (7B), o4-mini), the heatmaps compare different prompt strategies and graph serialization formats under easy, medium, and hard difficulties for the `Diameter calculation` task. The color intensity encodes accuracy (darker = higher), and solid/dashed boxes highlight best/second-best combinations respectively.

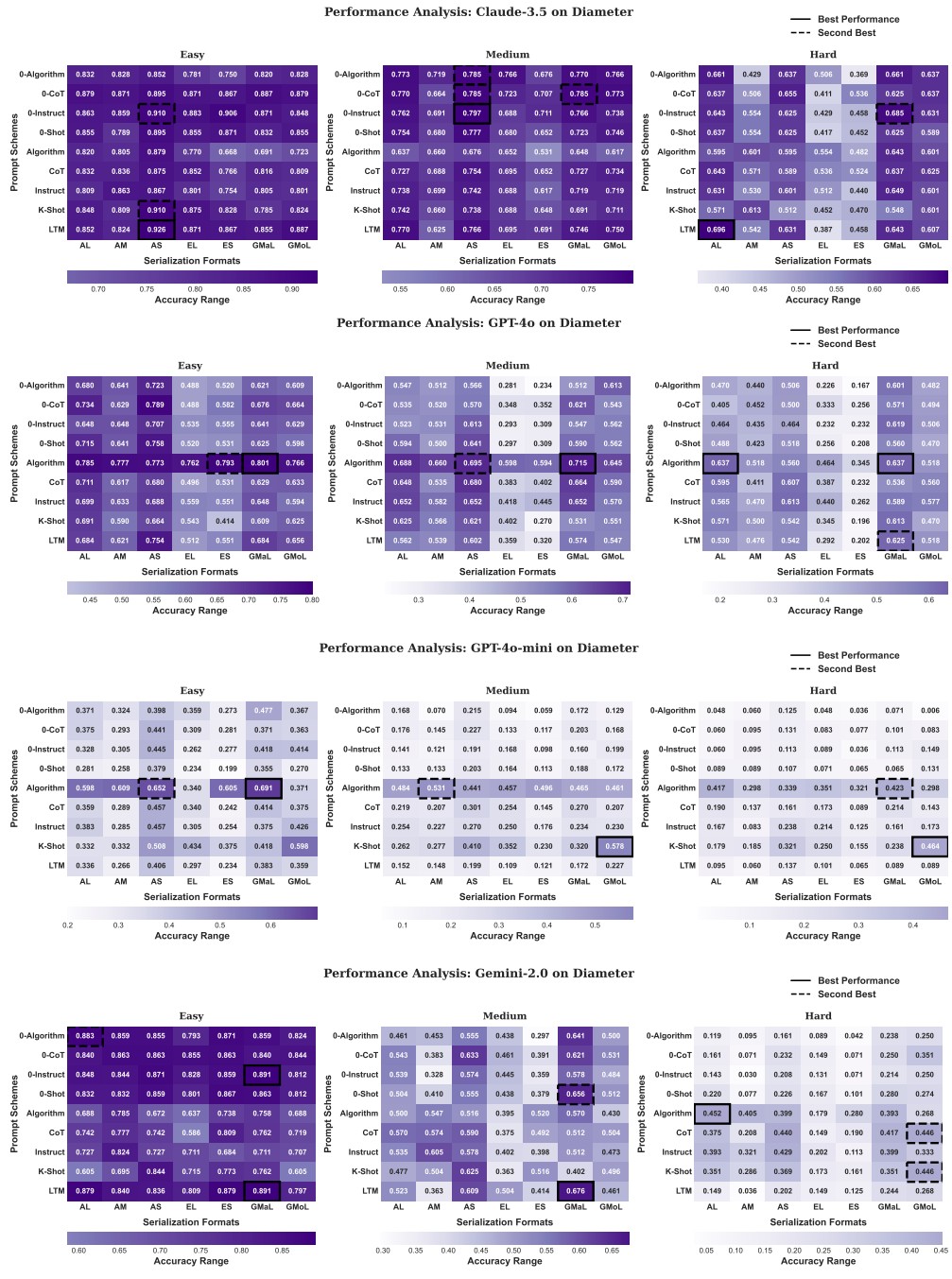

Figure 17: Performance heatmaps for prompt strategies and serialization formats on the Diameter task (Part 1). Models: Claude-3.5, GPT-4o, GPT-4o-mini, Gemini-2.0.

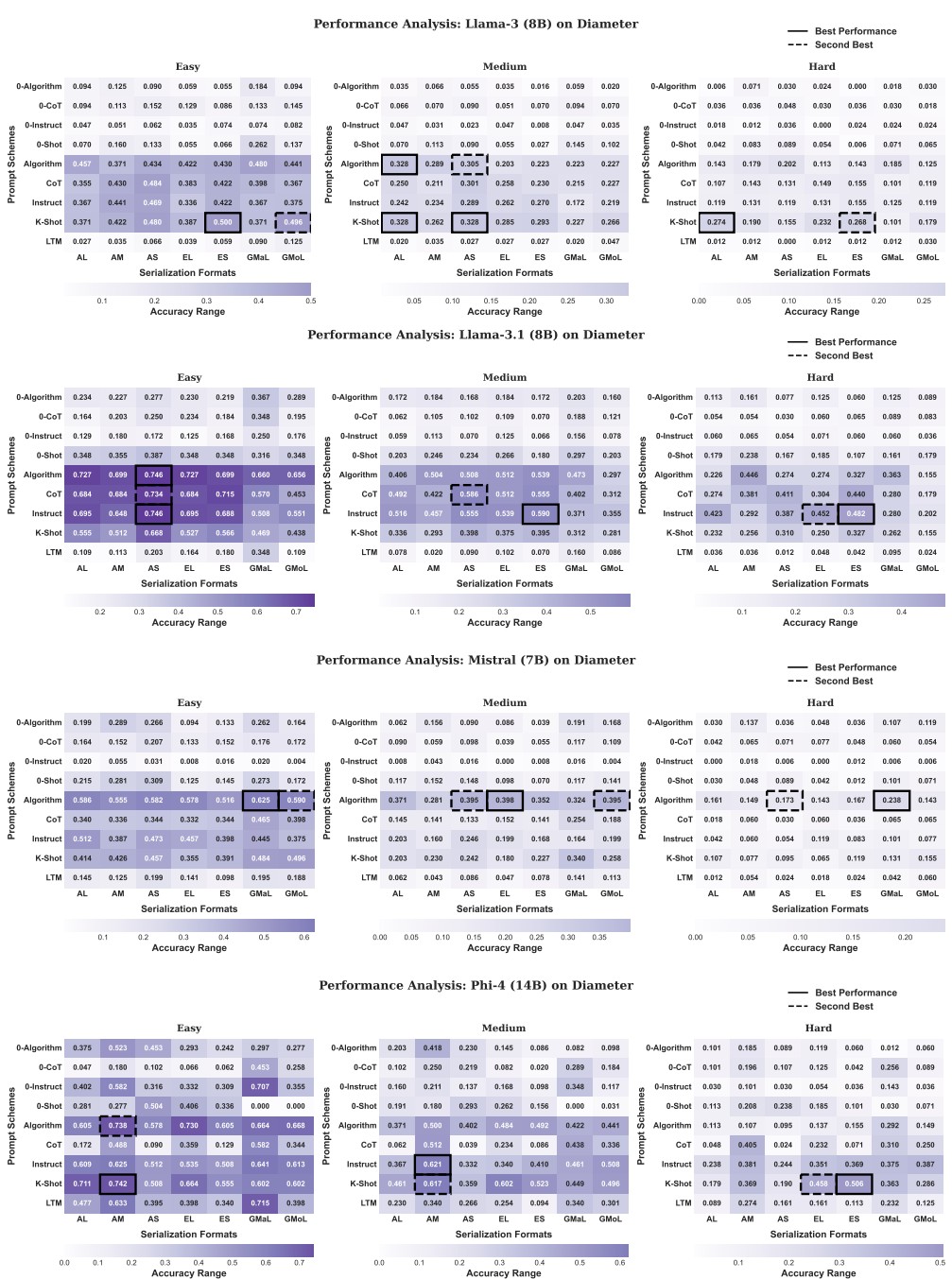

Figure 18: Performance heatmaps for prompt strategies and serialization formats on the Diameter task (Part 2). Models: Llama-3 (8B), Llama-3.1 (8B), Mistral (7B), Phi-4 (14B).

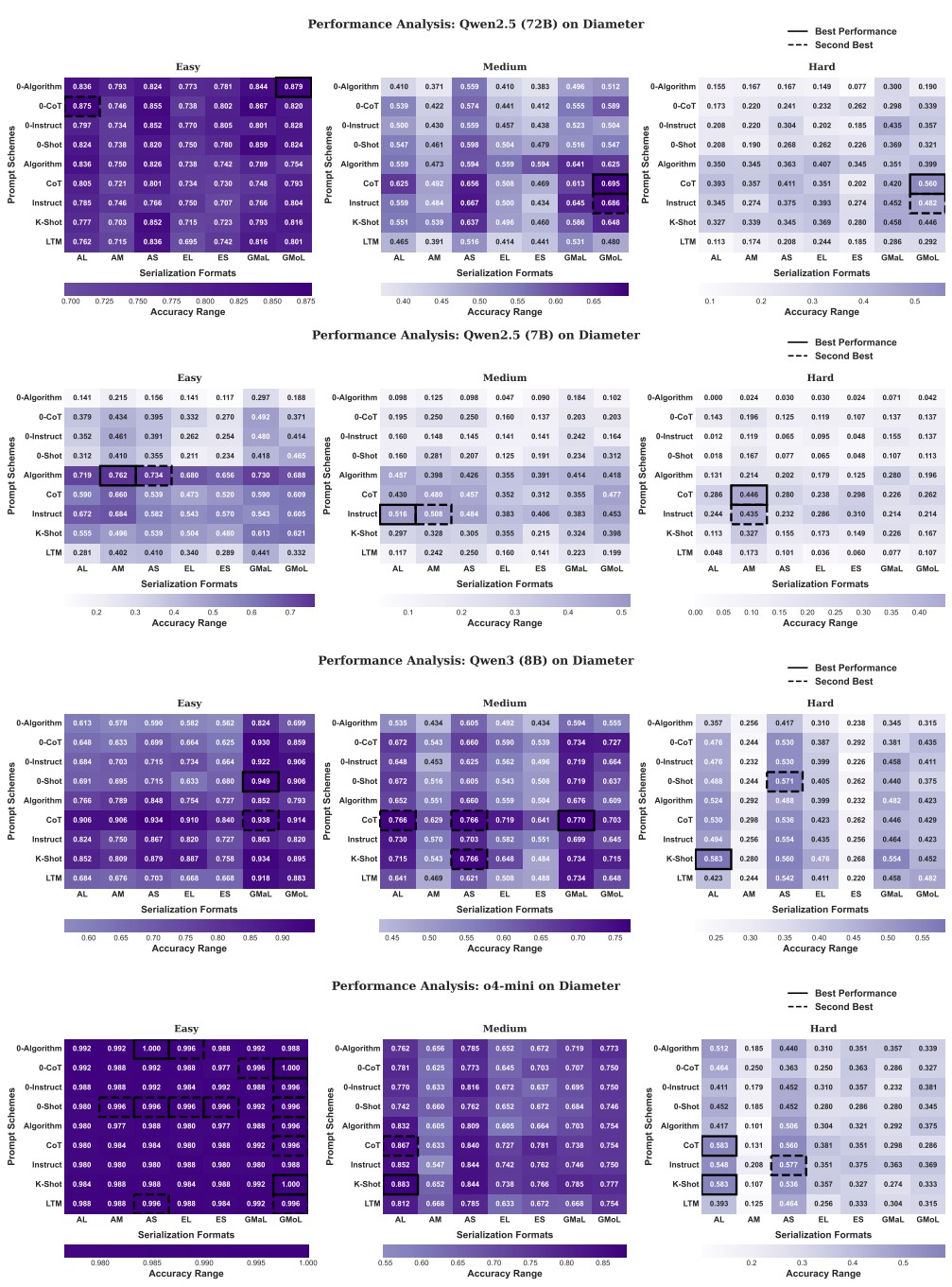

Figure 19: Performance heatmaps for prompt strategies and serialization formats on the Diameter task (Part 3). Models: Qwen-2.5 (72B), Qwen-2.5 (7B), Qwen-3 (8B), o4-mini.

### E.2.5  HEATMAPS FOR Shortest path TASK

As shown in Figure 20 (featuring Claude-3.5, GPT-4o, GPT-4o-mini, Gemini-2.0), Figure 21 (featuring Llama-3 (8B), Llama-3.1 (8B), Mistral (7B), Phi-4 (14B)), Figure 22 (featuring Qwen-2.5 (7B), o4-mini), the heatmaps compare different prompt strategies and graph serialization formats under easy, medium, and hard difficulties for the Shortest path task. The color intensity encodes accuracy (darker = higher), and solid/dashed boxes highlight best/second-best combinations, respectively.

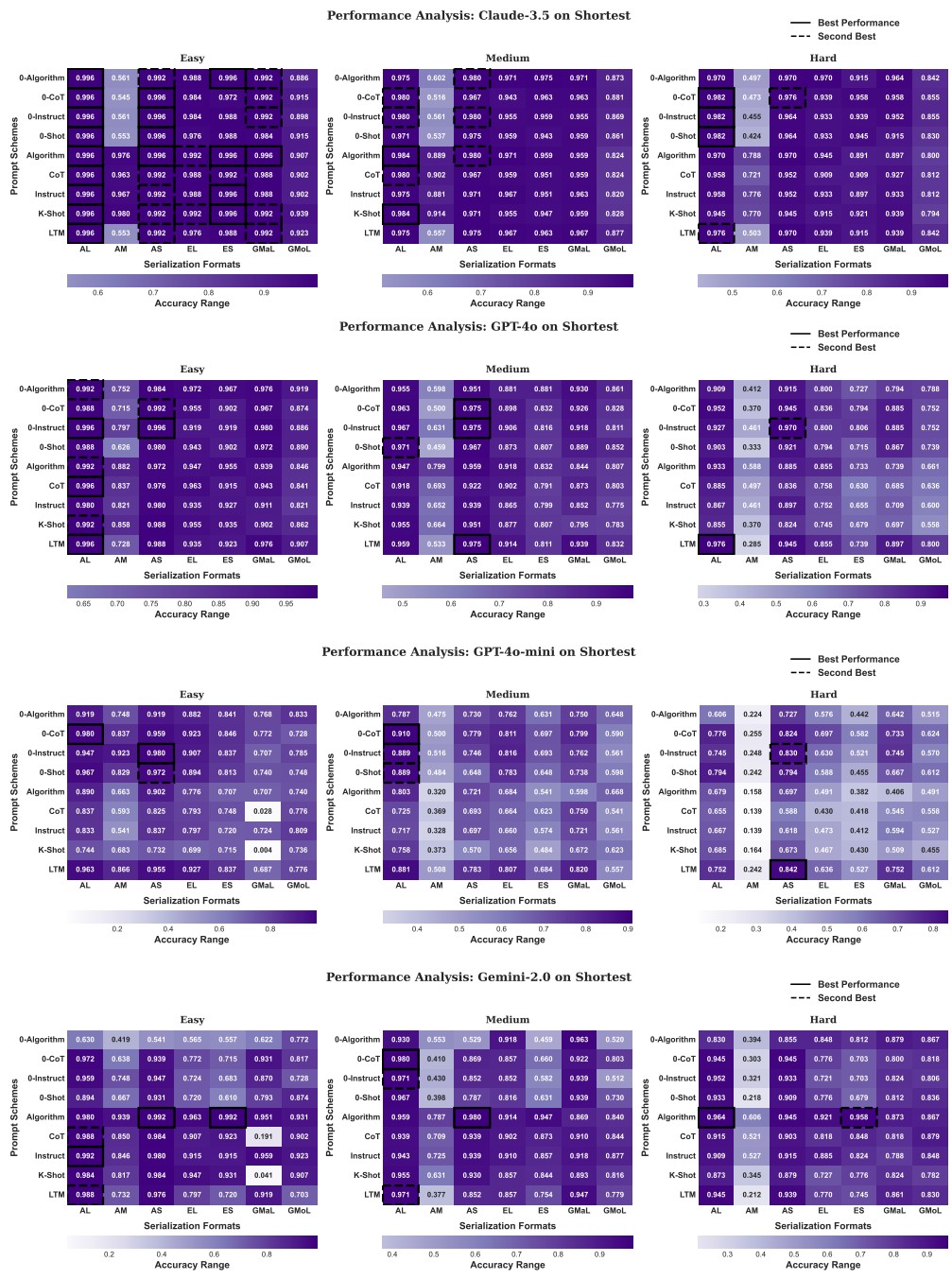

Figure 20: Performance heatmaps for prompt strategies and serialization formats on the Shortest task (Part 1). Models: Claude-3.5, GPT-4o, GPT-4o-mini, Gemini-2.0.

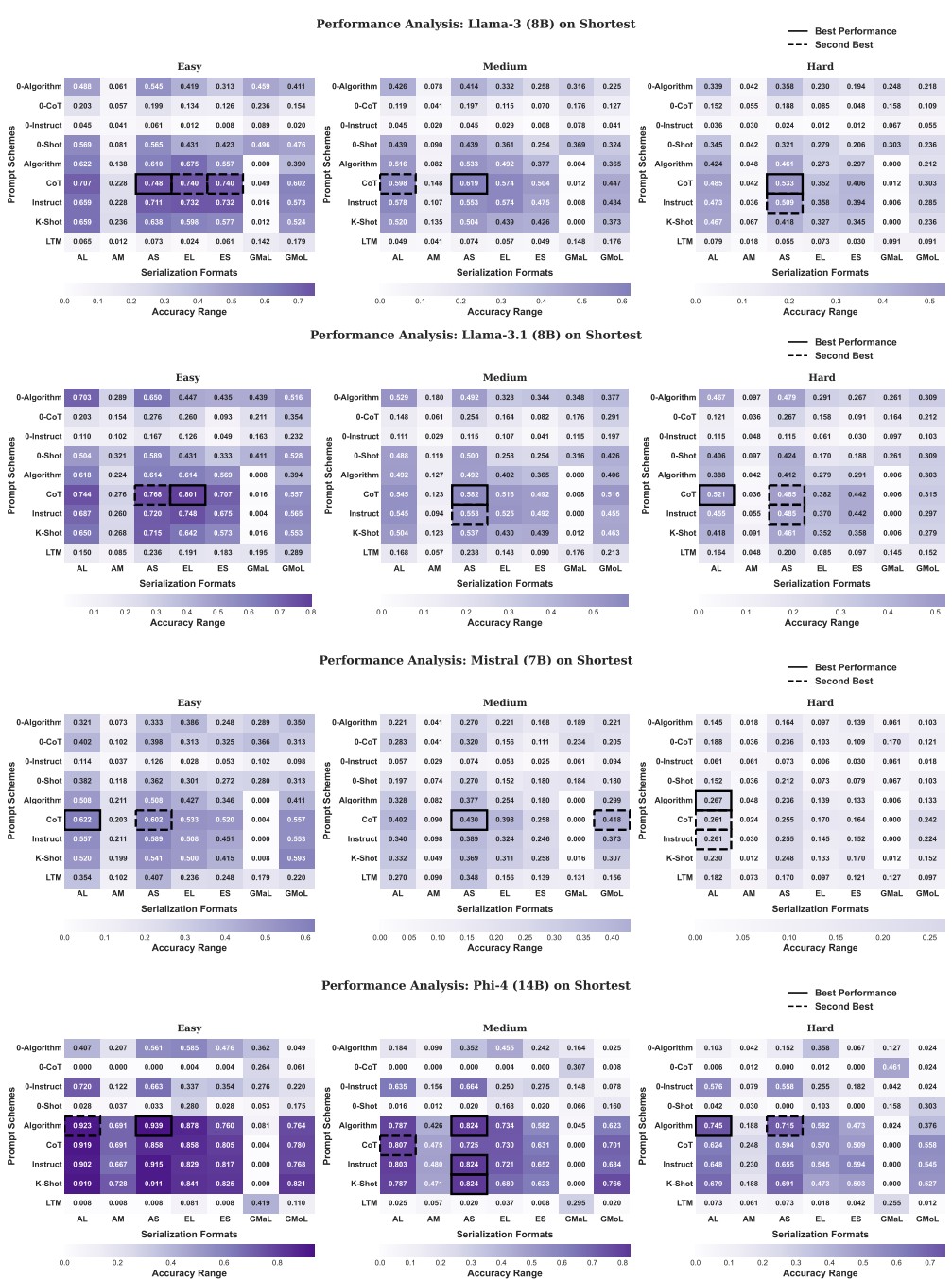

Figure 21: Performance heatmaps for prompt strategies and serialization formats on the Shortest task (Part 2). Models: Llama-3 (8B), Llama-3.1 (8B), Mistral (7B), Phi-4 (14B).

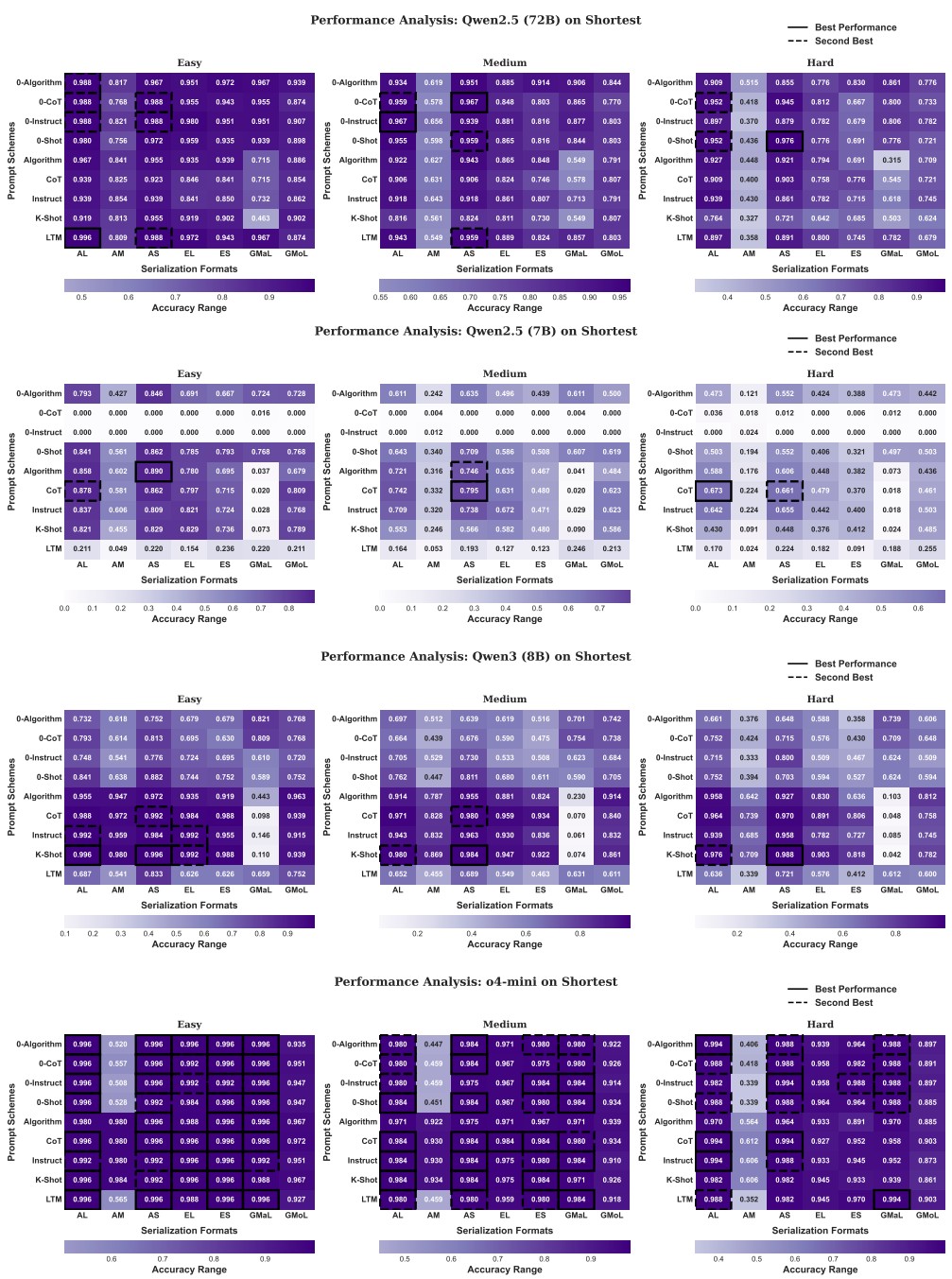

Figure 22: Performance heatmaps for prompt strategies and serialization formats on the Shortest task (Part 3). Models: Qwen-2.5 (72B), Qwen-2.5 (7B), Qwen-3 (8B), o4-mini.

### E.2.6 HEATMAPS FOR Triangle counting TASK

As shown in Figure 23 (featuring Claude-3.5, GPT-4o, GPT-4o-mini, Gemini-2.0) , Figure 24 (featuring Llama-3 (8B), Llama-3.1 (8B), Mistral (7B), Phi-4 (14B)), Figure 25 (featuring Qwen-2.5 (7B), o4-mini), the heatmaps compare different prompt strategies and graph serialization formats under easy, medium, and hard difficulties for the Triangle counting task. The color intensity encodes accuracy (darker = higher), and solid/dashed boxes highlight best/second–best combinations respectively.

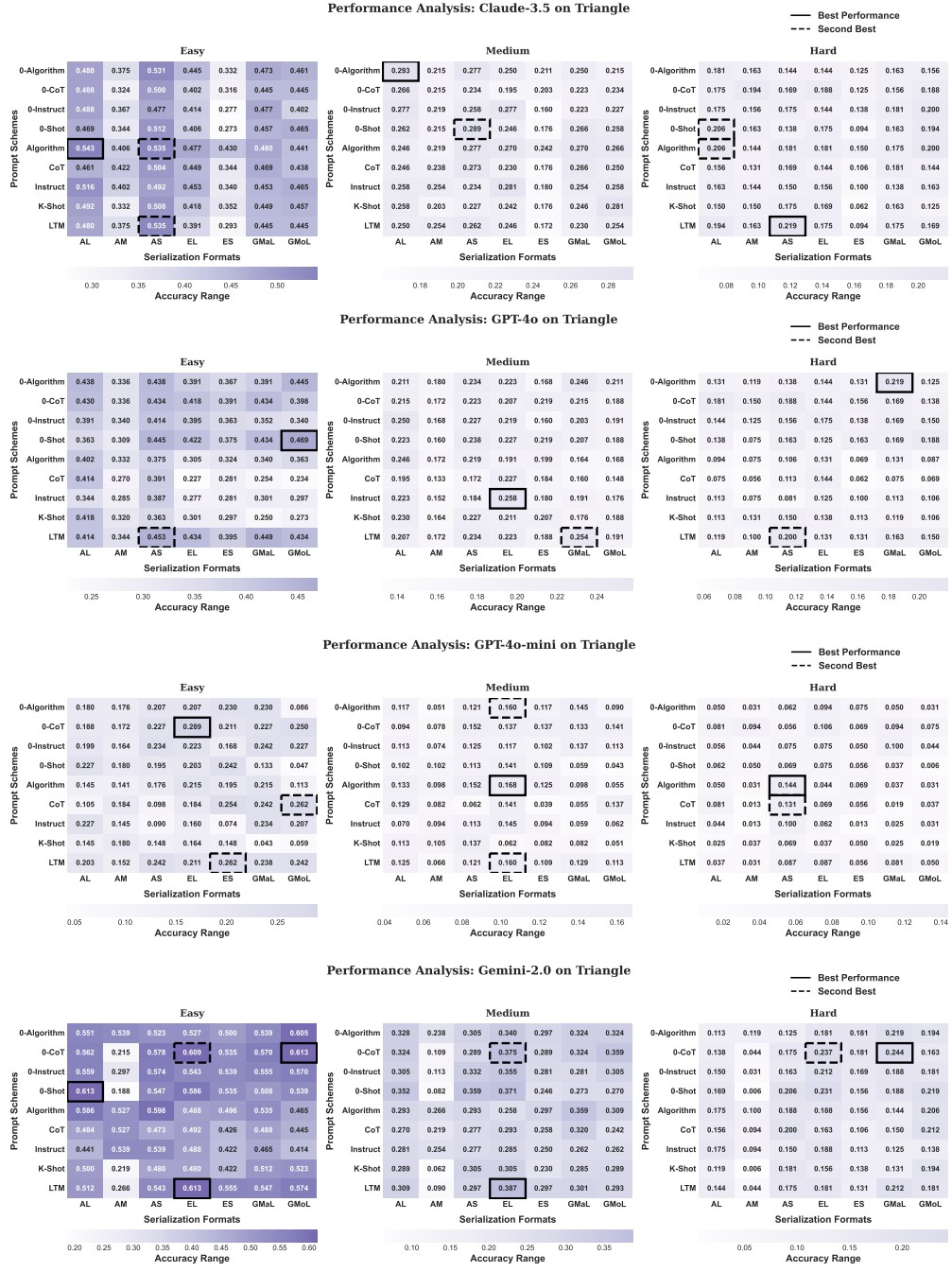

Figure 23: Performance heatmaps for prompt strategies and serialization formats on the Triangle task (Part 1). Models: Claude-3.5, GPT-4o, GPT-4o-mini, Gemini-2.0.

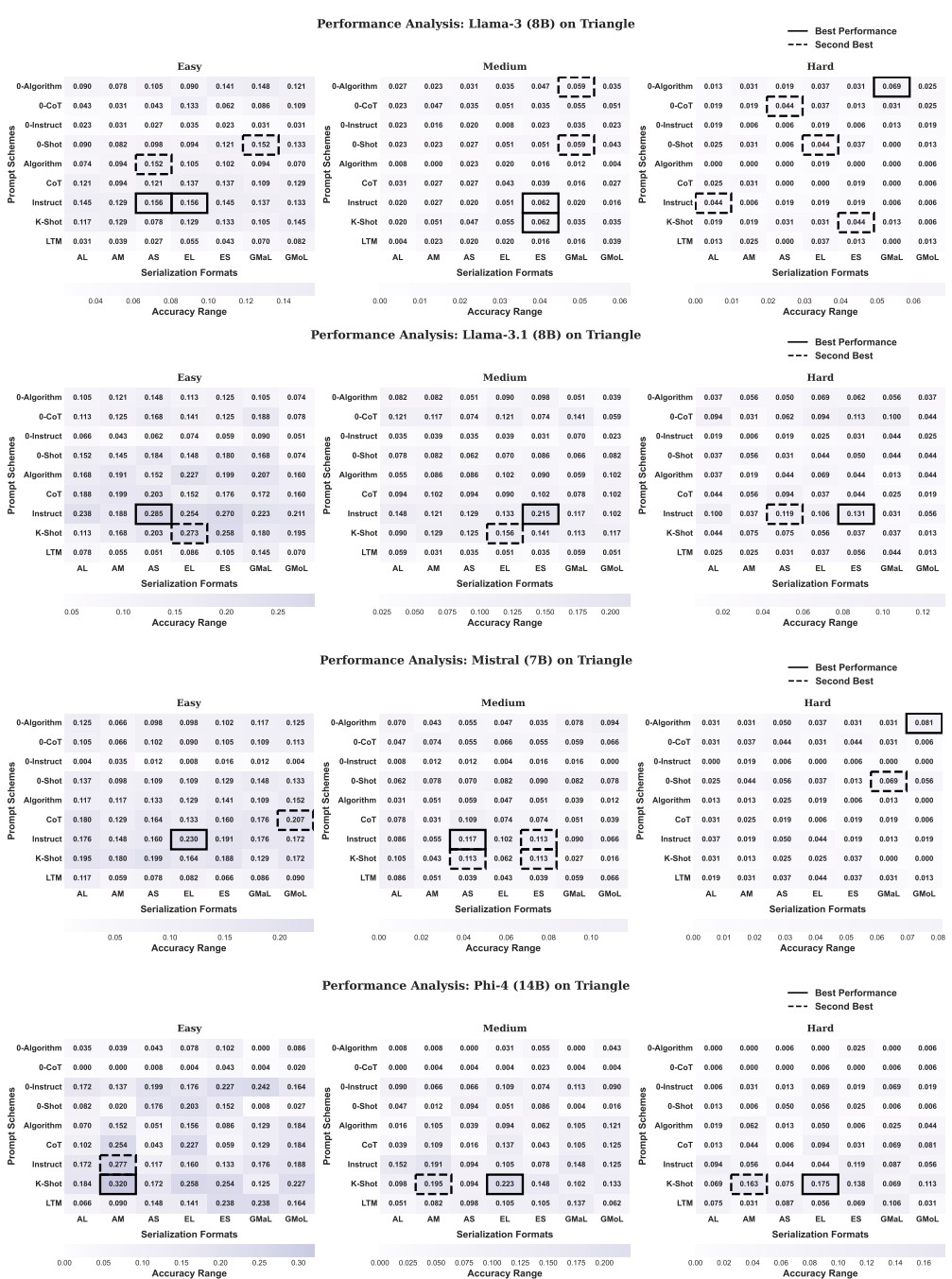

Figure 24: Performance heatmaps for prompt strategies and serialization formats on the Triangle task (Part 2). Models: Llama-3 (8B), Llama-3.1 (8B), Mistral (7B), Phi-4 (14B).

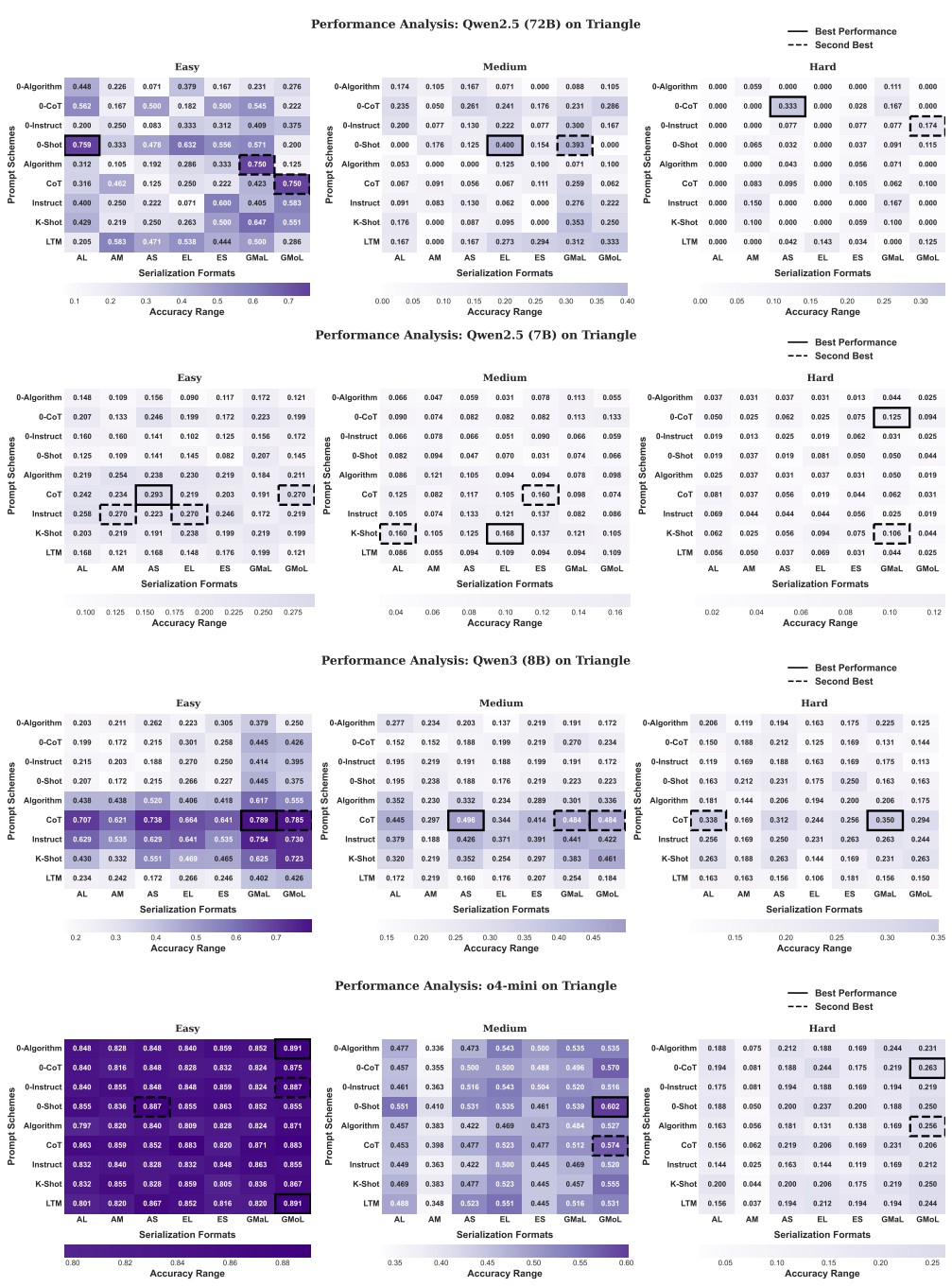

Figure 25: Performance heatmaps for prompt strategies and serialization formats on the Triangle task (Part 3). Models: Qwen-2.5 (72B), Qwen-2.5 (7B), Qwen-3 (8B), o4-mini.

### E.3 GRAPH TYPE SENSITIVITY ANALYSIS

While our main heatmaps analyze interactions between serialization formats and prompt schemes, the role of **graph types** in cross-factor analysis requires a different approach. Creating individual heatmaps for each graph type $\times$ task $\times$ difficulty combination would yield over 100+ visualizations that would be comprehensive but impractical to interpret. Instead, we introduce a sensitivity-based framework that quantifies how graph types respond to factor variations while maintaining both interpretability and extensibility.

**Methodology**. For each graph type in a given task-difficulty setting, we compute two metrics by averaging across all models:

- **Prompt Sensitivity** ($S_{\mathrm{p}}$): For each serialization format, we calculate the standard deviation of accuracy across different prompt schemes, then average over all formats. This measures how much performance fluctuates when changing prompts.
- **Format Sensitivity** ($S_{\mathrm{f}}$): Symmetrically, for each prompt scheme, we calculate the standard deviation across serialization formats, then average over all prompts.

We visualize each task-difficulty combination as a scatter plot in $(S_{\mathrm{p}}, S_{\mathrm{f}})$ space, where each bubble represents a graph type, and color encodes mean performance. Using median splits, we partition the space into four interpretable quadrants: *Robust* (low $S_{\mathrm{p}}$, low $S_{\mathrm{f}}$), *Prompt-Critical* (high $S_{\mathrm{p}}$, low $S_{\mathrm{f}}$), *Format-Critical* (low $S_{\mathrm{p}}$, high $S_{\mathrm{f}}$), and *Both Critical* (high $S_{\mathrm{p}}$, high $S_{\mathrm{f}}$).

**Key Findings:** Figures 26–31 present plots covering all task-difficulty combinations. Based on the analysis of these data, we arrive at the following insights.

1. **Open-source models are much more prompt-sensitive than closed-source ones.** Across different tasks, the prompt sensitivity range of open-source models is consistently larger than that of closed-source models. For example, in the BFS order – Medium setting, the prompt sensitivity typically falls between 0.12 and 0.16, whereas that of open-source models ranges only from 0.02 to 0.05. This indicates that closed-source models rely more heavily on using an appropriate serialization format to achieve strong performance.

2. **Closed-source models are more sensitive to serialization format than open-source models.** Across tasks, the format sensitivity range of closed-source models is generally higher. For instance, in the Diameter calculation – Easy setting, format sensitivity falls between 0.03 and 0.06, whereas open-source models range from 0.15 to 0.19. This suggests that open-source models depend more on advanced prompt-engineering strategies to improve performance, while closed-source models gain more from suitable serialization formatting.

Notably, the difference in sensitivity between open-source and closed-source models can be explained by how LLMs typically process graph reasoning tasks, which can be viewed as involving two stages: (i) understanding the task itself, and (ii) interpreting the graph-structured input. Closed-source models, due to their stronger reasoning capabilities, encounter fewer difficulties in task understanding; as a result, they are more sensitive to the information contained in the graph data—i.e., the serialization format. In contrast, task understanding plays a more significant role in open-source models, and prompts exert a more direct influence on this stage than serialization formatting, leading to greater prompt sensitivity. This interpretation is also consistent with our earlier finding—Finding 3: Open-source models benefit from multi-shot exemplars, whereas closed-source models do not. Closed-source models do not require additional exemplars to grasp the task, whereas open-source models rely more on examples to enhance task comprehension.

**Extensibility.** This framework directly supports GraphOmni's extensible design. When adding new graph families (e.g., real-world networks), researchers can apply the same analytical pipeline to assess sensitivity profiles before conducting full evaluations. Complete implementation details and visualization scripts are available in our code repository.

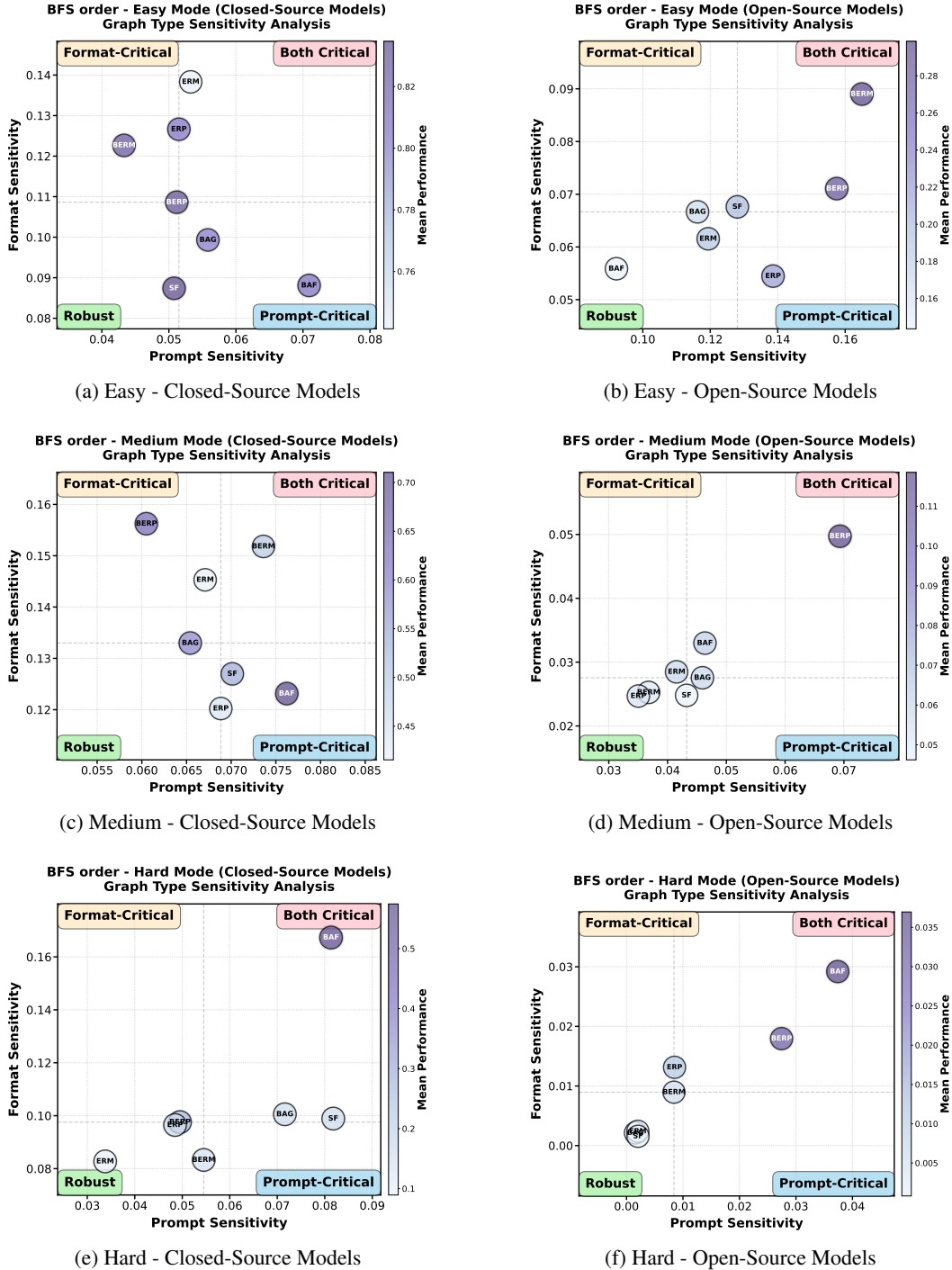

Figure 26: Graph type sensitivity analysis for BFS order task, comparing open-source and closed-source models. This comparison reveals whether sensitivity patterns are consistent across model categories.

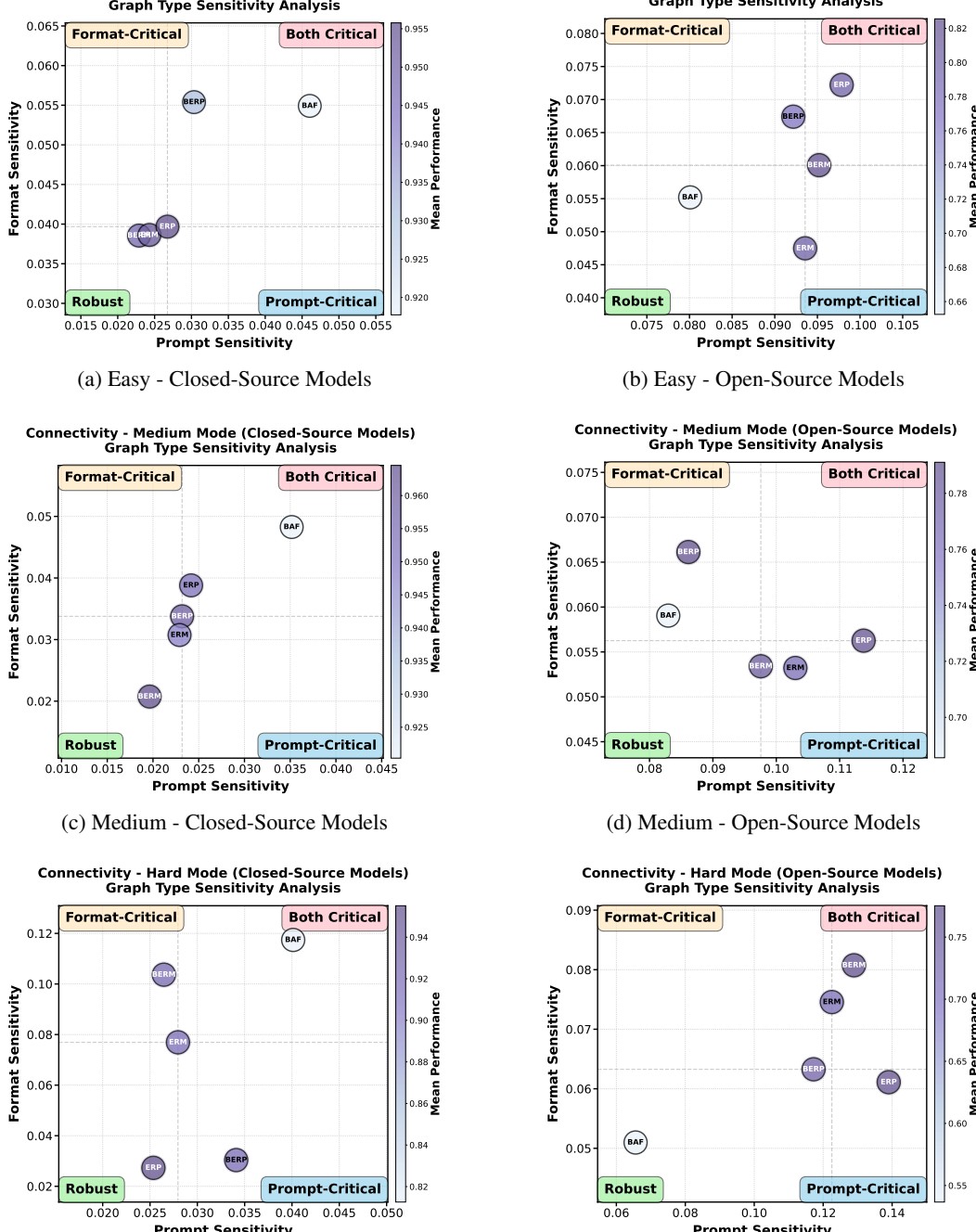

Figure 27: Graph type sensitivity analysis for Connectivity task, comparing open-source and closed-source models. This comparison reveals whether sensitivity patterns are consistent across model categories.

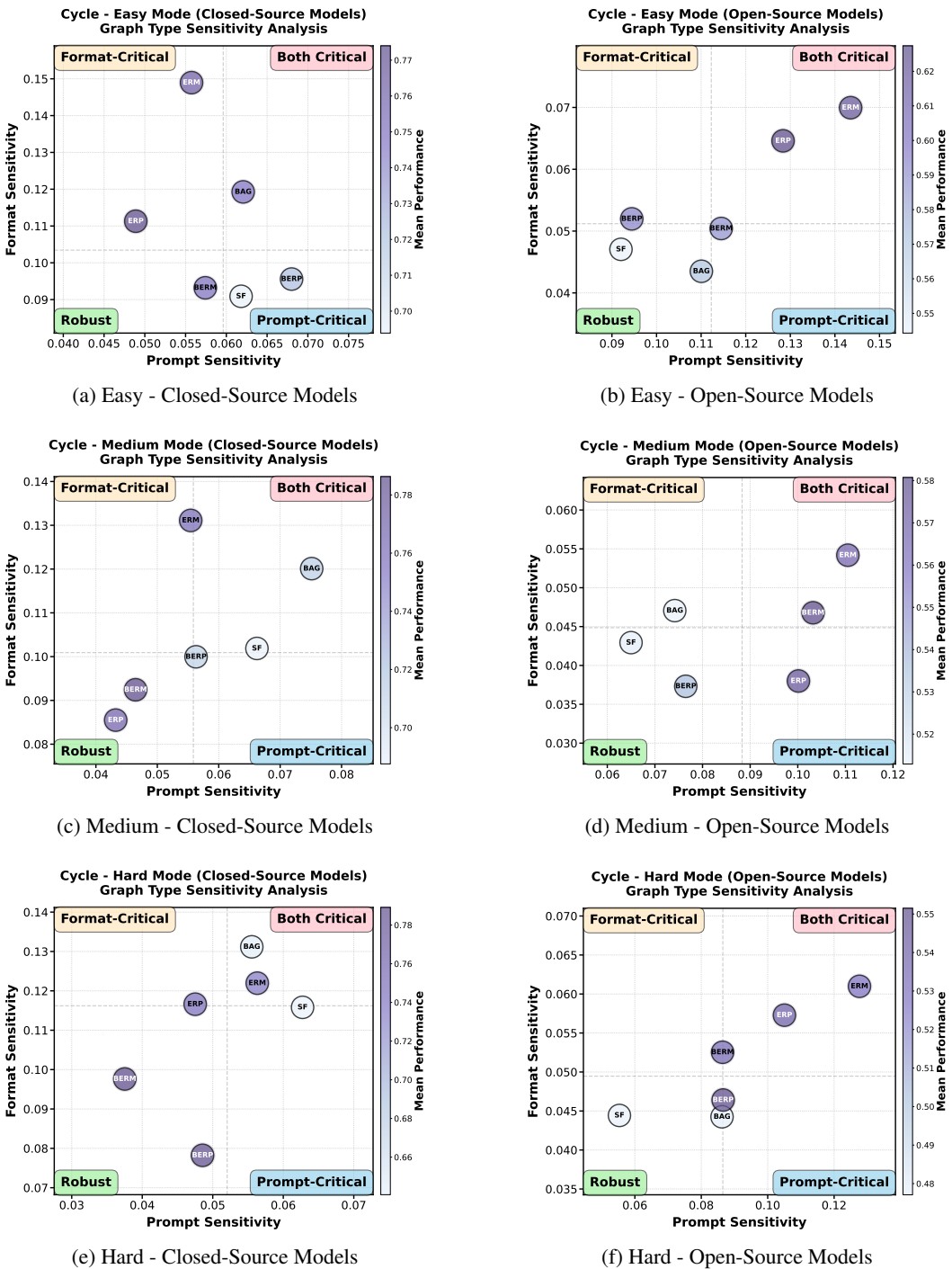

(a) Easy - Closed-Source Models

(b) Easy - Open-Source Models

(c) Medium - Closed-Source Models

(d) Medium - Open-Source Models

(e) Hard - Closed-Source Models

(f) Hard - Open-Source Models

Figure 28: Graph type sensitivity analysis for Cycle task, comparing open-source and closed-source models. This comparison reveals whether sensitivity patterns are consistent across model categories.

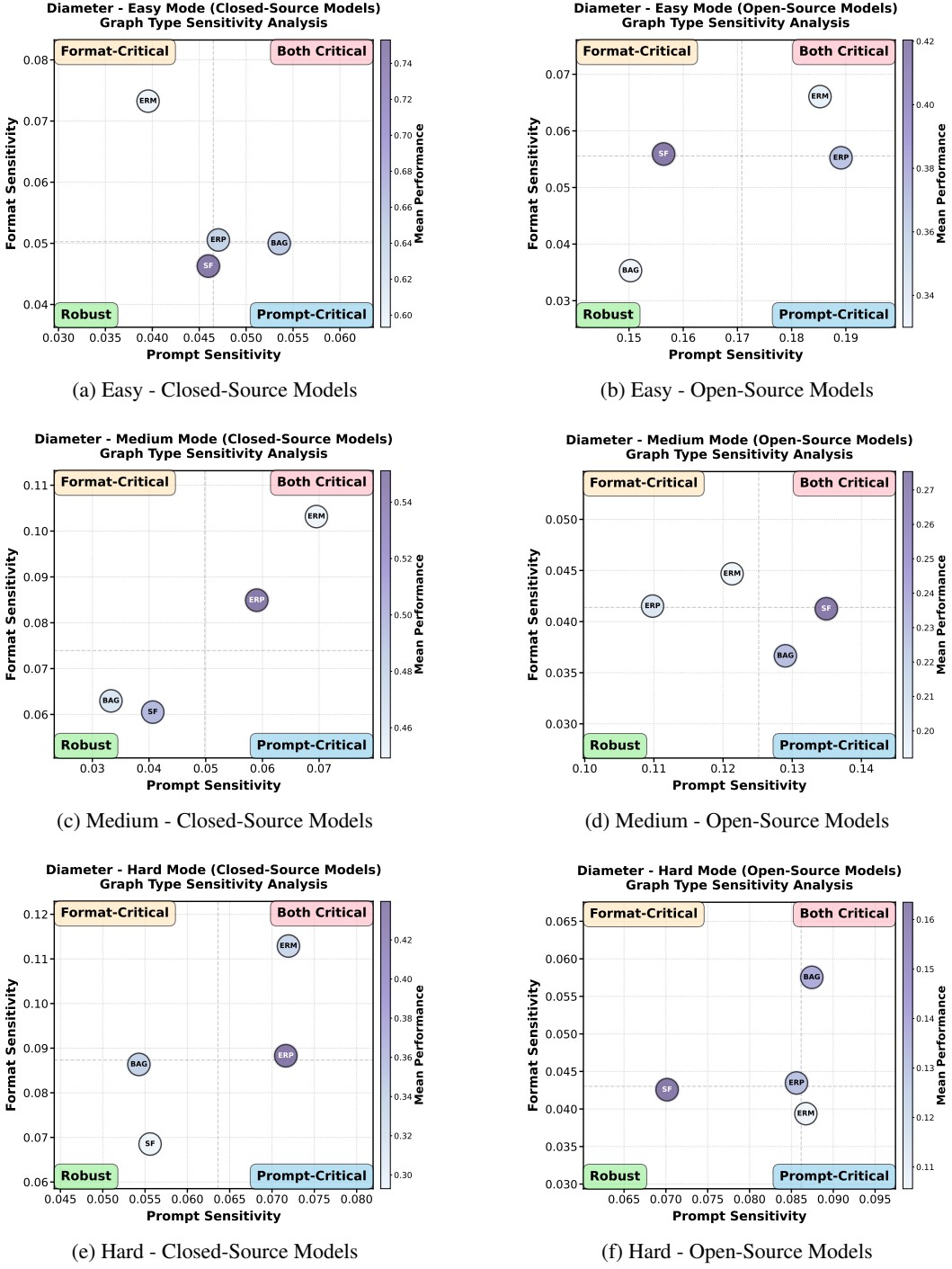

Figure 29: Graph type sensitivity analysis for Diameter task, comparing open-source and closed-source models. This comparison reveals whether sensitivity patterns are consistent across model categories.

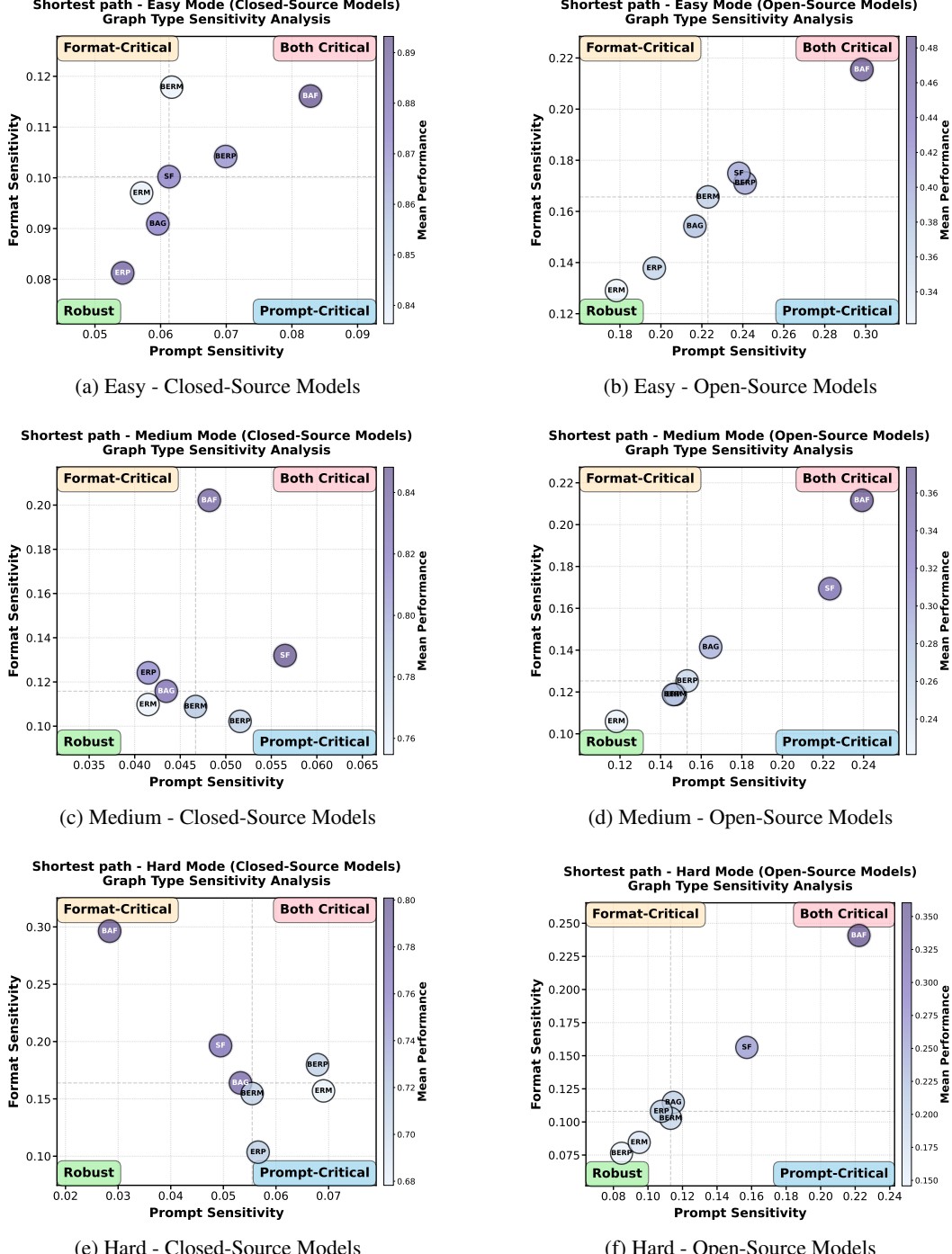

Figure 30: Graph type sensitivity analysis for Shortest path task, comparing open-source and closed-source models. This comparison reveals whether sensitivity patterns are consistent across model categories.

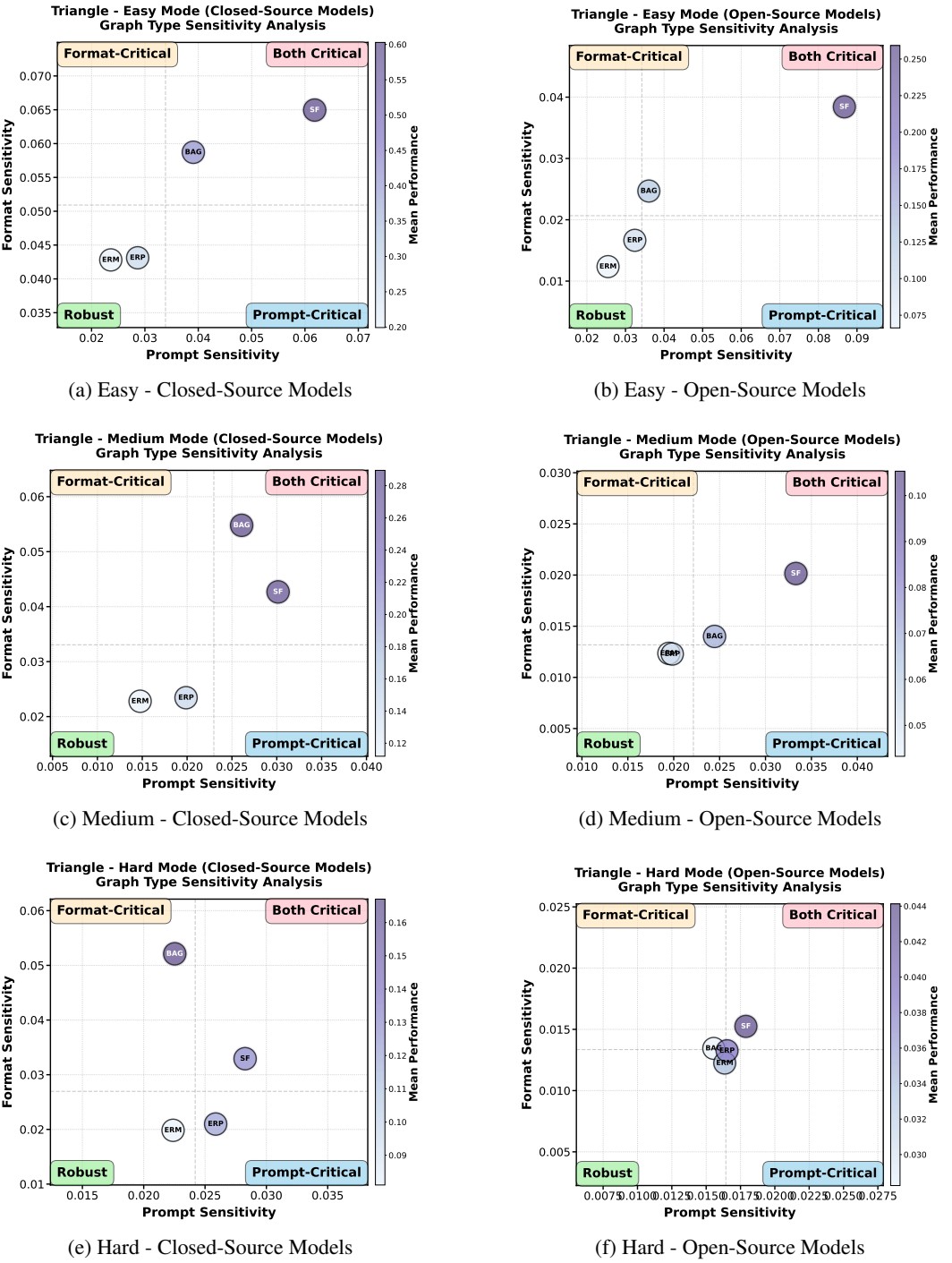

Figure 31: Graph type sensitivity analysis for Triangle task, comparing open-source and closed-source models. This comparison reveals whether sensitivity patterns are consistent across model categories.

## E.4 ERROR ANALYSIS

This subsection presents an in-depth analysis of common error patterns observed in model responses. We categorize representative error cases and provide concrete examples to illustrate the specific challenges LLMs face in graph reasoning tasks. This detailed error analysis extends the brief discussion of error categories presented in **Result ❸** in Section 4.1.

### E.4.1 Diameter calculation CASE 1

---

**Diameter**

- **QUERY**: Given a graph, your task is to determine the diameter of this graph. Graph: Adjacency Matrix is
  [[0 1 1 1 1 1 1 1 1]
  [1 0 1 0 1 1 1 1 1]
  [1 1 0 1 1 1 1 1 1]
  [1 0 1 0 1 1 1 1 1]
  [1 1 1 1 0 1 1 1 1]
  [1 1 1 1 1 0 1 1 1]
  [1 1 1 1 1 1 0 1 1]
  [1 1 1 1 1 1 1 0 1]
  [1 1 1 1 1 1 1 1 0]]
  Q: What is the diameter of this graph?
  A:
- **TASK INFO**: diameter,easy , 0-shot, Adjacency Matrix
- **GROUND TRUTH**: 2.0
- **MODEL:** Mistral
- **WRONG ANSWER**: the diameter of the given graph is 7.

  explanation: to find the diameter of a graph, we first find the shortest path between every pair of vertices using a shortest path algorithm like dijkstra's or floyd-warshall. however, since the graph is unweighted, we can find the diameter by simply considering the longest path between any two vertices.

  in this case, the longest path we can find is: - vertex 0 -> vertex 1 -> vertex 2 -> vertex 3 -> vertex 4 -> vertex 5 -> vertex 6 -> vertex 7 -> vertex 8

  the number of edges in this path is 8 - 1 = 7, so the diameter of the graph is 7.
- **ERROR CAUSE ANALYSIS**: This answer does not accurately define the concept of diameter. In the reasoning process, it mentions "first find the shortest path between every pair of vertices," which is correct. However, the final description refers to "the longest path," which can be misleading. In fact, the diameter is defined as the longest of all shortest paths rather than the longest path between any two vertices. Therefore, this explanation does not fully align with the formal definition of diameter.

---

### E.4.2 Diameter calculation CASE 2

> **Diameter**
>
> - **QUERY**: Given a graph, your task is to determine the diameter of this graph. Graph: Adjacency List is {0: [1, 2, 3, 4, 5, 8, 10, 11, 13, 15], 1: [0, 2, 3, 4, 5, 6, 13, 15], 2: [0, 1, 3, 4, 5, 6, 7, 9, 12, 14], 3: [0, 1, 2, 4, 5, 7, 11, 13, 14], 4: [0, 1, 2, 3, 5, 6, 7, 8, 17], 5: [0, 1, 2, 3, 4, 8, 9], 6: [2, 1, 4, 9], 7: [2, 4, 3, 10, 14, 15, 16], 8: [5, 4, 0, 12, 16, 17], 9: [6, 2, 5, 10, 11, 12], 10: [0, 9, 7], 11: [3, 9, 0], 12: [2, 9, 8], 13: [3, 0, 1], 14: [2, 3, 7], 15: [7, 1, 0, 16], 16: [15, 7, 8, 17], 17: [16, 8, 4]}
>
>   Q: What is the diameter of this graph?
>
>   A: Let's break down this problem:
> - **TASK INFO**: diameter, medium, LTM, Adjacency List
> - **GROUND TRUTH**: 3.0
> - **MODEL:** Qwen-3
> - **WRONG ANSWER**:
>
>   To determine the **diameter** of a graph, we must identify the **longest shortest path** between any two nodes in the graph. This involves computing the **shortest path** between all pairs of nodes and then finding the **maximum** of these values.
>
>   ---
>
>   ### Step 1: Understanding the Graph The graph is represented as an **adjacency list** with 18 nodes (0 through 17). The graph appears to be **connected**, which is essential for the diameter to be well-defined. We will use **Breadth-First Search (BFS)** from each node to compute the shortest paths to all other nodes.
>
>   ---
>
>   ### Step 2: Strategy - For each node, perform BFS to find the shortest distances to all other nodes. - Keep track of the **maximum shortest distance** found across all node pairs. - The **maximum** of these distances is the **diameter**.
>
>   ---
>
>   ### Step 3: Key Observations and Calculations Through careful inspection and BFS traversal, the **longest shortest path** was found to be between the following pairs of nodes:
>
>   - **Node 12 to Node 15**: Path: 12 → 9 → 5 → 0 → 15 (4 steps) Alternatively: 12 → 8 → 4 → 1 → 15 (4 steps)
>
>   - **Node 9 to Node 16**: Path: 9 → 2 → 4 → 8 → 16 (4 steps)
>
>   - **Node 12 to Node 15** and **Node 9 to Node 16** are two such pairs where the **shortest path is exactly 4 steps**.
>
>   Other pairs were found to have shorter distances (e.g., 2 or 3 steps). No pair was found to have a shortest path longer than 4 steps.
>
>   ---
>
>   ### Final Result Since the **longest shortest path** in the graph is **4 steps**, the **diameter** of the graph is:
>
>   $$ {4} $$
> - **ERROR CAUSE ANALYSIS**: This result fails to find a shorter path because it ignores the two critical edges 7-15 and 7-16 in the graph.12 To 15 there is actually a 3-step shorter path: e.g., 12 → 2 → 7 → 15. For 9 to 16 there is a 3-step path: 9 → 10 → 7 → 16. Once these two edges are taken into account, it can be seen that no two points in the graph have a shortest path of more than 3 steps. these two edges into account, it follows that no two points in the graph have shorter paths than 3 steps. Therefore, the diameter of the graph is indeed 3, not 4.

### E.4.3 Connectivity CASE 1

---

**Connectivity**

- **QUERY**: Determine if there is a path between two nodes in the graph. Graph: Edge List is

  3 1
  4 2
  5 4
  6 5
  7 5
  Q: Is there a path between node 1 and node 5?
  A:
  Let's break down this problem:
- **TASK INFO**: connectivity, easy, LTM, Edge List
- **MODEL:** Mistral
- **GROUND TRUTH**: False
- **WRONG ANSWER**:
  1. we have a graph with 7 nodes (1 to 7) and 6 edges.
  2. to find a path between two nodes, we can use depth-first search (dfs) or breadth-first search (bfs). both algorithms will help us traverse the graph and find a path if one exists.
  3. in this case, we will use dfs because it is more efficient for sparse graphs like this one.
  4. we start at node 1. since there is an edge from node 1 to node 4, we visit node 4.
  5. from node 4, we find an edge to node 5, so we have found a path from node 1 to node 5.
  6. therefore, there is a path between nodes 1 and 5 in the graph.
  so the answer is: yes, there is a path between node 1 and node 5.
- **ERROR CAUSE ANALYSIS**: This answer misunderstands the EL serialization format. Although it correctly states the number of nodes and edges, it incorrectly assumes that there is an edge between nodes 1 and 4. This mistake leads to incorrect conclusions in the subsequent reasoning.

---

### E.4.4  Cycle detection CASE 1

---

**Cycle**

- **QUERY**: Given a graph representation, your task is determining whether the graph has a cycle Graph: Adjacency List is

  {0: [1, 2, 3, 4, 6], 1: [0, 9], 2: [0, 7, 8], 3: [0, 5], 4: [0], 5: [3], 6: [0], 7: [2], 8: [2], 9: [1]}

  Q: Is there a cycle in this graph?

  A:

  Let's think step by step:
- **TASK INFO**: cycle, medium, 0-CoT, Adjacency List
- **MODEL:** claude-3-5-sonnet-20241022
- **GROUND TRUTH**: False
- **WRONG ANSWER**:

  Let's analyze this graph to find if it contains any cycles:

  1) First, let's understand the connections:

  - Node 0 connects to 1,2,3,4,6

  - Node 1 connects to 0,9

  - Node 2 connects to 0,7,8

  - Node 3 connects to 0,5

  - Node 4 connects to 0

  - Node 5 connects to 3

  - Node 6 connects to 0

  - Node 7 connects to 2

  - Node 8 connects to 2

  - Node 9 connects to 1

  2) Looking at some paths:

  - $0 \rightarrow 1 \rightarrow 0$ (This is a cycle!)

  - $0 \rightarrow 2 \rightarrow 0$ (This is another cycle!)

  - $0 \rightarrow 3 \rightarrow 0$ (This is another cycle!)

  3) The graph has multiple cycles. The simplest ones are between node 0 and its neighbors that have return paths.

  Yes, there are multiple cycles in this graph.
- **ERROR CAUSE ANALYSIS**: This answer misunderstands cycle detection in an graph. Although it correctly lists the connections between nodes, it incorrectly assumes that a direct backtracking step (e.g., $0 \rightarrow 1 \rightarrow 0$) forms a cycle. In an graph, a valid cycle must involve at least three distinct nodes. This mistake leads to incorrect conclusions about the presence of multiple cycles.

---

### E.4.5  Triangle counting CASE 1

> **Triangle**
>
> - **QUERY**:   Given   a   graph,   your   task   is   to   determine   how   many   triangles   in   this graph.       Graph:       GraphML   is   <?xml   version='1.0'   encoding='utf-8'?>   <graphml xmlns="http://graphml.graphdrawing.org/xmlns" xmlns:xsi="http://www.w3.org/2001/XMLSchema-instance"                              xsi:schemaLocation="http://graphml.graphdrawing.org/xmlns http://graphml.graphdrawing.org/xmlns/1.0/graphml.xsd"><graph edgedefault="undirected"><node id="0"/>
>   <node id="1"/>
>   <node id="2"/>
>   ...
>   <edge source="19" target="21"/>
>   <edge source="19" target="25"/>
>   </graph></graphml>
>   Q: How many triangles are in this graph?
>   A:
> - **TASK INFO**: triangle, hard, 0-shot, GraphML
> - **MODEL:** Mistral
> - **GROUND TRUTH**: 47
> - **WRONG ANSWER**:
>   to find the number of triangles in a graph, we can use the formula for the number of triangles in a simple graph, which is given by n / 3, where n is the number of triples of distinct vertices that form a triangle. in this graph, we have 26 nodes, so we would calculate 26 / 3 = 8.666666666666667, but since we cannot have a fraction of a triangle, we round down to 8. therefore, there are 8 triangles in this graph.
> - **ERROR CAUSE ANALYSIS**: This analysis misunderstands triangle counting in a graph. While it correctly identifies sets of three nodes, it incorrectly assumes that the number of triangles can be calculated as $n/3$. In a graph, a valid triangle requires all three nodes to be mutually connected. This mistake reflects a lack of proper understanding of the graph's topological structure

### E.4.6 Triangle counting CASE 2

> **Triangle**
>
> - **QUERY**: Given a graph, your task is to determine how many triangles in this graph. Graph: Adjacency List is {1: [0], 0: [1, 2, 3, 6], 2: [0, 7], 3: [0, 4, 5, 6], 4: [3, 6], 5: [3], 6: [0, 3, 4, 8], 7: [2], 8: [6]}
>
>   Q: How many triangles are in this graph?
>
>   A:
> - **TASK INFO**: triangle, easy, 0-shot, Adjacency List
> - **MODEL:** Qwen-2.5
> - **GROUND TRUTH**: 2
> - **WRONG ANSWER**:
>
>   to determine the number of triangles in the graph, we need to find sets of three vertices where each pair of vertices is connected by an edge. given the adjacency list, we can go through each vertex and check all possible combinations of its neighbors to see if they form a triangle.
>
>   here's the adjacency list for reference: "' 1: [0], 0: [1, 2, 3, 6], 2: [0, 7], 3: [0, 4, 5, 6], 4: [3, 6], 5: [3], 6: [0, 3, 4, 8], 7: [2], 8: [6] "'
>
>   let's go through the vertices and their neighbors to find triangles:
>
>   1. **vertex 0:** - neighbors: [1, 2, 3, 6] - check combinations: - (1, 2): not connected - (1, 3): not connected - (1, 6): not connected - (2, 3): not connected - (2, 6): not connected - (3, 6): yes, connected (forms triangle (0, 3, 6))
>
>   2. **vertex 1:** - neighbors: [0] - only one neighbor, so no triangles.
>
>   3. **vertex 2:** - neighbors: [0, 7] - check combinations: - (0, 7): yes, connected (forms triangle (0, 2, 7))
>
>   4. **vertex 3:** - neighbors: [0, 4, 5, 6] - check combinations: - (0, 4): yes, connected (forms triangle (0, 3, 4)) - (0, 5): not connected - (0, 6): yes, connected (forms triangle (0, 3, 6)) - (4, 5): not connected - (4, 6): yes, connected (forms triangle (3, 4, 6)) - (5, 6): not connected
>
>   5. **vertex 4:** - neighbors: [3, 6] - check combinations: - (3, 6): yes, connected (forms triangle (3, 4, 6))
>
>   6. **vertex 5:** - neighbors: [3] - only one neighbor, so no triangles.
>
>   7. **vertex 6:** - neighbors: [0, 3, 4, 8] - check combinations: - (0, 3): yes, connected (forms triangle (0, 3, 6)) - (0, 4): yes, connected (forms triangle (3, 4, 6)) - (0, 8): not connected - (3, 4): yes, connected (forms triangle (3, 4, 6)) - (3, 8): not connected - (4, 8): not connected
>
>   8. **vertex 7:** - neighbors: [2] - only one neighbor, so no triangles.
>
>   9. **vertex 8:** - neighbors: [6] - only one neighbor, so no triangles.
>
>   summarizing the triangles found: - (0, 3, 6) - (0, 2, 7) - (0, 3, 4) - (3, 4, 6)
>
>   thus, the number of triangles in the graph is **4**.
> - **ERROR CAUSE ANALYSIS**: The understanding of the serialization format is incorrect in this answer. Even though it successfully prints out the AL serialization format, in its reasoning process, it incorrectly assumes that node 7 and node 0 are connected, and that node 4 and node 0 are connected, which ultimately leads to an incorrect result.

E.4.7 BFS order CASE 1

---

**BFS order**

- **QUERY**: Given a graph, your task is to determine the bfs traversal order of this graph starting at node 2. And the graph representation of: Adjacency Matrix is

  [[0 1 1 1 1 0 1]

  [1 0 1 1 1 1 1]

  [1 1 0 0 0 0 0]

  [1 1 0 0 0 1 0]

  [1 1 0 0 0 0 0]

  [0 1 0 1 0 0 0]

  [1 1 0 0 0 0 0]]

  Q: Give the bfs traversal order starting from node 2.

  A:

- **TASK INFO**: bfsorder, easy, 0-shot, Adjacency Matrix
- **MODEL:** Qwen-2.5
- **STARTING NODE**: 2
- **WRONG ANSWER**:

  to determine the bfs traversal order starting from node 2, we follow these steps:

  1. **initialize**: start with node 2. add it to the queue and mark it as visited.

  2. **process the queue**: remove the first node from the queue and visit it. then, add its unvisited neighbors to the queue.

  3. **repeat**: continue the process until the queue is empty.

  given the adjacency matrix:

  ```
  \[
  \begin{bmatrix}
  0 & 1 & 1 & 1 & 1 & 0 & 1 \\
  1 & 0 & 1 & 1 & 1 & 1 & 1 \\
  1 & 1 & 0 & 0 & 0 & 0 & 0 \\
  1 & 1 & 0 & 0 & 0 & 1 & 0 \\
  1 & 1 & 0 & 0 & 0 & 0 & 0 \\
  0 & 1 & 0 & 1 & 0 & 0 & 0 \\
  1 & 1 & 0 & 0 & 0 & 0 & 0 \\
  \end{bmatrix}
  \]
  ```

  let's perform the bfs:

  1. **start at node 2**: - queue: [2] - visited: [2]

  2. **dequeue node 2**: - visit node 2 - add its unvisited neighbors (nodes 1, 3, 4, 6) to the queue. - queue: [1, 3, 4, 6] - visited: [2, 1, 3, 4, 6]

  3. **dequeue node 1**: - visit node 1 - add its unvisited neighbors (nodes 2, 3, 4, 5, 6) to the queue. nodes 2 and 6 are already visited. - queue: [3, 4, 5, 6] - visited: [2, 1, 3, 4, 5, 6]

  4. **dequeue node 3**: - visit node 3 - add its unvisited neighbors (nodes 2, 1) to the queue. nodes 2 and 1 are already visited. - queue: [4, 5, 6] - visited: [2, 1, 3, 4, 5, 6]

  5. **dequeue node 4**: - visit node 4 - add its unvisited neighbors (nodes 2, 3, 6) to the queue. nodes 2, 3, and 6 are already visited. - queue: [5, 6] - visited: [2, 1, 3, 4, 5, 6]

  6. **dequeue node 5**: - visit node 5 - add its unvisited neighbors (nodes 1, 4) to the queue. nodes 1 and 4 are already visited. - queue: [6] - visited: [2, 1, 3, 4, 5, 6]

  7. **dequeue node 6**: - visit node 6 - add its unvisited neighbors (nodes 2, 4) to the queue. nodes 2 and 4 are already visited. - queue: [] - visited: [2, 1, 3, 4, 5, 6]

  the bfs traversal order starting from node 2 is: **2, 1, 3, 4, 5, 6**. note that the order of visiting

- **ERROR CAUSE ANALYSIS**: The understanding of the serialization format is incorrect in this answer. Even though it successfully prints out the AM serialization format, in its reasoning process, it incorrectly assumes that node 2 and node 0 do not have connectivity, which ultimately leads to an incorrect result.

---

### E.4.8 Triangle counting CASE 3

> **Triangle**
>
> - **QUERY**: Given a graph, your task is to determine how many triangles this graph.
>   And the graph representation of Edge Set is :
>   {(0, 1), (1, 2), (0, 4), (0, 3), (1, 8), (0, 6), (0, 2), (2, 5), (4, 7)}
>   Q: How many triangles are in this graph?
>   A: The graph contains 3 nodes that form a triangle (0, 1, and 2), resulting in 1 triangle. So the number of triangles is 1.
>   Given a graph, your task is to determine how many triangles this graph.
>   And the graph representation of Edge Set is :
>   {(0, 1), (0, 7), (0, 4), (1, 5), (0, 3), (2, 3), (0, 2), (2, 6), (0, 5)}
>   Q: How many triangles are in this graph?
>   A: This graph includes edges that connect nodes to form two distinct triangles, namely (0, 1, 5) and (0, 2, 3). The graph contains two triangles: (0, 1, 5) and (0, 2, 3). So the number of triangles is 2.
>   Given a graph, your task is to determine how many triangles this graph.
>   And the graph representation of Edge Set is :
>   {(0, 1), (3, 4), (2, 7), (1, 4), (0, 2), (5, 6), (3, 6), (2, 5), (1, 3)}
>   Q: How many triangles are in this graph?
>   A: A graph with edges forming a single triangle: (4, 1, 3). The graph contains one triangle, formed by nodes (4, 1, 3). So the number of triangles is 1.
>   Given a graph, your task is to determine how many triangles this graph.
>   And the graph representation of Edge Set is :
>   {(4, 3), (1, 5), (4, 6), (4, 2), (3, 0), (0, 6), (4, 5), (0, 2), (3, 6), (1, 0)}
>   Q: How many triangles are in this graph?
>   A: The edges in this graph create two triangles: (0, 3, 6) and (3, 6, 4). The graph contains two triangles: (0, 3, 6) and (3, 6, 4). So the number of triangles is 2.
>   Given a graph, your task is to determine how many triangles this graph.
>   And the graph representation of Edge Set is :
>   {(3, 4), (5, 1), (5, 7), (0, 2), (8, 3), (0, 5), (0, 8), (2, 5), (1, 3), (7, 4), (6, 8), (5, 6), (5, 3), (0, 7), (2, 4), (2, 1), (2, 7), (6, 1), (6, 7), (5, 8), (8, 1), (1, 4), (0, 6), (2, 3), (6, 3)}
>   Q: How many triangles are in this graph?
>   A: A dense graph with many edges, resulting in 26 distinct triangles formed by various combinations of nodes. The graph contains 26 triangles formed by various combinations of nodes. So the number of triangles is 26.
>   Given a graph, your task is to determine how many triangles in this graph.
>   Graph: Edge Set is
>   {(4, 3), (5, 1), (5, 7), (8, 3), (8, 6), (2, 5), (7, 4), (6, 5), (5, 0), (8, 2), (0, 1), (2, 4), (2, 7), (6, 1), (7, 3), (6, 7), (8, 1), (8, 7), (0, 3), (2, 0), (1, 4), (2, 6), (6, 3)}
>   Q: How many triangles are in this graph?
>   A:
> - **TASK INFO**:
>   triangle, easy, CoT, Edge Set
> - **GROUND TRUTH**: 16
> - **MODEL:** o4-mini
> - **WRONG ANSWER**:
>   To count triangles, list each triple i,j,k with i<j<k and check all three edges. Doing so yields these 17 distinct triangles:
>   (0,1,5), (0,2,3), (0,2,5), (1,5,6), (1,6,8), (2,4,7), (2,5,6), (2,5,7), (2,6,7), (2,6,8), (2,7,8), (3,4,7), (3,6,7), (3,6,8), (3,7,8), (5,6,7), (6,7,8).
>   So the number of triangles is 17.
> - **ERROR CAUSE ANALYSIS**: This answer misinterprets the connectivity of node 2 and node 3 and ends up thinking that 0, 2, 3 forms a triangle, leading to a final error.

## E.5 INPUT/OUTPUT EXAMPLES

We show more model input and output examples here. **Query** is the input to the model and **Answer** is the model output. Due to space reasons, in the middle of the excessively long part we will use "...". Each of the following examples is randomly selected from the query.

---

**Connectivity**

- **QUERY**:Determine if there is a path between two nodes in the graph.

  Graph: Edge Set is {(6, 18), (7, 26), (3, 22), (5, 19), (9, 17), (9, 26), (0, 23), (1, 15), (6, 11), (7, 10), (6, 20), (7, 19), (5, 12), (8, 11), (9, 10), (3, 24), (9, 19), (1, 17), (6, 13), (7, 12), (7, 21), (3, 17), (9, 12), (3, 26), (5, 23), (9, 21), (8, 25), (3, 19), (5, 16), (4, 18), (9, 14), (5, 25), (1, 21), (4, 11), (5, 18), (9, 16), (8, 20), (1, 14), (2, 13), (1, 23), (2, 22), (3, 14), (5, 11), (4, 13), (3, 23), (4, 22), (8, 13), (8, 22), (1, 16), (0, 18), (2, 15), (0, 27), (6, 24), (3, 16), (5, 13), (4, 24), (8, 15), (0, 11), (8, 24), (1, 18), (0, 20), (2, 17), (2, 26), (7, 25), (3, 18), (4, 17), (4, 26), (8, 17), (1, 11), (5, 27), (8, 26), (9, 25), (0, 22), (2, 19), (6, 10), (6, 19), (7, 18), (3, 11), (4, 10), (7, 27), (4, 19), (8, 10), (5, 20), (1, 13), (0, 15), (9, 18), (9, 27), (6, 12), (7, 20), (4, 12), (4, 21), (9, 11), (0, 17), (2, 14), (0, 26), (2, 23), (6, 14), (6, 23), (7, 22), (8, 14), (9, 13), (3, 27), (5, 24), (8, 23), (9, 22), (0, 19), (2, 16), (7, 15), (6, 25), (7, 24), (3, 20), (5, 17), (9, 15), (0, 12), (5, 26), (9, 24), (0, 21)}

  Q: Is there a path between node 27 and node 25?

  A:

  Let's construct a graph with the nodes and edges first:
- **TASK INFO**: connectivity, hard, 0-Instruct, Edge Set
- **MODEL:** claude-3-5-sonnet-20241022
- **ACCURACY**: 1
- **GROUND TRUTH**: True
- **ANSWER**: Let me help you find if there's a path between node 27 and node 25.

  Looking at the edge set, I can see that: - Node 27 is connected to nodes 3, 5, 7, and 9 - Node 25 is connected to nodes 6, 7, 8, and 9

  We can find a path from node 27 to node 25: 27 → 7 → 25 (direct connection through node 7)

  Since we can find at least one path between node 27 and node 25 (through node 7), there is definitely a path between these two nodes.

  The answer is yes.

---

**Connectivity**

- **QUERY**: To determine if there is a path between two nodes in an undirected graph, we can use a Breadth-First Search (BFS) algorithm. BFS is an algorithm that starts at one node and explores all of its neighbors before moving on to the next level of neighbors. By exploring each node in the graph, the algorithm can determine if there is a path between two nodes.

  Determine if there is a path between two nodes in the graph.

  Graph: Edge Set is {(0, 7), (3, 8), (5, 8), (3, 7), (6, 8), (1, 8), (5, 7), (6, 7), (1, 7), (4, 8), (4, 7), (2, 8)}

  Q: Is there a path between node 5 and node 7?

  A:

- **TASK INFO**: connectivity, easy, 0-Algorithm, Edge Set
- **MODEL:** Mistral
- **ACCURACY**:

  1

- **GROUND TRUTH**:True

- **ANSWER**: yes, there is a path between node 5 and node 7. one possible path is: 5 -> 8 -> 7

  here's the code for the bfs algorithm in python:

  "' from collections import deque

  def bfs(graph, start, end):

  visited = set()

  queue = deque([start])

  while queue:

  current = queue.popleft()

  if current == end:

  return true

  if current not in visited:

  visited.add(current)

  for neighbor in graph[current]:

  queue.append(neighbor)

  return false

  # example usage

  graph =

  0: [4, 3],

  1: [7, 4],

  2: [8],

  3: [],

  4: [0, 1, 7],

  5: [8],

  6: [7, 8],

  7: [0, 1, 6, 3, 5],

  8: [2, 6, 5, 3]

  print(bfs(graph, 5, 7)) # output: true "'

  in this code, 'graph' is a dictionary where the keys are the nodes and the values are lists of their neighbors. the 'bfs' function takes in the graph, the start node, and the end node, and returns 'true' if there is a path between the two nodes, and 'false' otherwise. the function uses a queue to keep track of the nodes to be visited, and a set to keep track of the visited nodes to avoid visiting the same node multiple times. the function loops through the queue until it finds the end node or the queue is empty, and returns the result.

**Connectivity**

- **QUERY**: Determine if there is a path between two nodes in the graph. Graph: GraphML is

  ...
  <node id="1"/>
  <node id="2"/>

  ...
  <edge source="4" target="5"/>
  <edge source="4" target="6"/>
  <edge source="4" target="7"/>
  </graph></graphml>
  Q: Is there a path between node 1 and node 0?
  A:
  Let's think step by step:
- **TASK INFO**: connectivity, easy, 0-CoT, GraphML
- **MODEL:** Llama3.1
- **ACCURACY**: 0
- **GROUND TRUTH**: True
- **ANSWER**: the first step is to convert the given graphml into a format that can be used in the java code. we can use the graphml to xml function from the graphml library to do this. the second step is to create a graph from the xml string. we can use the graphmlreader class to do this. the third step is to find the nodes with id 1 and 0. we can use the getnodes method of the graph to do this. the fourth step is to check if there is a path between the two nodes. we can use the haspath method of the graph to do this. the fifth step is to return the result. we can use the print function to do this.

  ```java
  import org.graphstream.graph.graph;
  import org.graphstream.graph.implementations.singlegraph;
  import org.graphstream.graph.implementations.graphmlreader;
  import org.graphstream.graph.implementations.defaultgraph;
  public class main
  public static void main(string[] args)
  string graphml = "<?xml version='1.0' encoding='utf-8'?><graphml xmlns=http://graphml.graphdrawing.org/xmlnsxmlns:xsi=http://www.w3.org/2001/xmlschema-instancexsi:schemalocation=http://graphml.graphdrawing.org/xmlns http://graphml.graphdrawing.org/xmlns/1.0/graphml.xsd><graph edgedefault=undirected><node id=0/><node id=5/><node id=6/><node id=7/><node id=1/><node id=2/><node id=3/><node id=4/><edge source=0target=5/><edge source=0target=6/><edge source=0target=7/><edge source=5target=1/><edge source=5target=3/><edge source=5target=4/><edge source=6target=1/><edge source=6target=2/><edge source=6target=3/><edge source=6target=4/><edge source=7target=1/><edge source=7target=2/><edge source=7target=3/><edge source=7target=4/></graph></graphml>";
  graph graph = new singlegraph("graph");
  graph.read(graphml);
  string node1 = "1";
  string node0 = "0";
  boolean ispath = graph.haspath(node1, node0);
  system.out.println(ispath);
  ```

  this will print: false. there is no path between node 1 and node 0.
  please note that this is a very simple implementation and does not handle any exceptions that might occur. in a real-world application, you would want to add error handling code.

---

**Diameter**

- **QUERY**: Given a graph, your task is to determine the diameter of this graph. Graph: Adjacency List is {0: [1], 1: [0, 2, 4, 5, 6, 7, 8, 11, 12], 2: [1, 3, 9], 3: [2], 4: [1], 5: [1], 6: [1], 7: [1, 10], 8: [1], 9: [2], 10: [7], 11: [1], 12: [1]}
  
  Q: What is the diameter of this graph?
  
  A:
- **TASK INFO**: diameter, medium, 0-shot, Adjacency List
- **MODEL:** Phi-4
- **ACCURACY**: 0
- **GROUND TRUTH**:4.0
- **ANSWER**:
  
  3

---

---

**Diameter**

- **QUERY**: Given a graph, your task is to determine the diameter of this graph. Graph: Adjacency List is {0: [1], 1: [0, 2, 4, 5, 6, 7, 8, 11, 12], 2: [1, 3, 9], 3: [2], 4: [1], 5: [1], 6: [1], 7: [1, 10], 8: [1], 9: [2], 10: [7], 11: [1], 12: [1]}
  Q: What is the diameter of this graph?
  A:
- **TASK INFO**: diameter, medium, 0-shot, Adjacency List
- **MODEL:** gemini-2.0-flash-001
- **ACCURACY**: 1
- **GROUND TRUTH**: 4.0
- **ANSWER**:

To find the diameter of the graph, we need to compute the shortest path between all pairs of nodes and then find the maximum of these shortest path lengths. The given graph has 13 nodes (0 to 12).

Here's how we can determine the shortest paths:

* **Node 0:** Only connected to Node 1, distance 1. * **Node 1:** The central node, connected to 0, 2, 4, 5, 6, 7, 8, 11, 12. * **Node 2:** Connected to 1 and 3, 9. * **Node 3:** Connected to 2. * **Node 4, 5, 6, 8, 11, 12:** Only connected to 1. * **Node 7:** Connected to 1 and 10. * **Node 9:** Connected to 2. * **Node 10:** Connected to 7.

Let's analyze some key distances:
* Distance(0, x): 2 for x in 2, 4, 5, 6, 7, 8, 9, 11, 12, otherwise infinity
* Distance(0, 3): 3
* Distance(0, 10): 3
* Distance(1, x): 1 for x in 0, 2, 4, 5, 6, 7, 8, 11, 12
* Distance(1, 3): 2
* Distance(1, 9): 2
* Distance(1, 10): 2
* Distance(2, x):
* 1 for x in 1, 3, 9
* 2 for x in 0, 4, 5, 6, 7, 8, 11, 12
* 3 for x in 10
* Distance(3, x):
* 1 for x in 2
* 2 for x in 1, 9
* 3 for x in 0, 4, 5, 6, 7, 8, 11, 12
* 4 for x in 10
* Distance(9, x):
* 1 for x in 2
* 2 for x in 1, 3
* 3 for x in 0, 4, 5, 6, 7, 8, 11, 12
* 4 for x in 10
* Distance(10, x):
* 1 for x in 7
* 2 for x in 1
* 3 for x in 0, 2, 4, 5, 6, 8, 9, 11, 12
* 4 for x in 3
Now, let's find the maximum shortest path:
* Distance(3, 10) = 4
* Distance(9, 10) = 4
It appears 4 is the maximum distance.
The diameter is 4.

---

**Diameter**

- **QUERY**: To calculate the diameter of the graph, you can use BFS based on the following tips

  1. identify all nodes in the graph.

  2. For each node in the graph , perform BFS to compute the shortest path from that node to all other nodes.

  3. calculate the shortest path from node u to all other nodes.

  4. Find the longest shortest path.

  5. Repeat the process and update the diameter of the graph.

  6. Return the diameter of the graph.

  Given a graph, your task is to determine the diameter of this graph.

  And the graph representation of Edge Set is :

  {(0, 1), (1, 2), (0, 4), (0, 3), (1, 8), (0, 6), (0, 2), (2, 5), (4, 7)}

  Q: What is the diameter of this graph?

  A: Identify all nodes: The graph has nodes 0 to 8. Perform BFS from node 0. The longest shortest path is 2 edges (to node 5, 7 or 8). Perform BFS from node 1. The longest shortest path is 3 edges (to node 7). Perform BFS from node 2. The longest shortest path is 3 edges (to node 7). Perform BFS from node 3. The longest shortest path is 3 edges (to node 5, 7 or 8). Perform BFS from node 4. The longest shortest path is 3 edges (to node 5 or 8). Perform BFS from node 5. The longest shortest path is 4 edges (to node 7). Perform BFS from node 6. The longest shortest path is 3 edges (to node 5, 7, or 8). Perform BFS from node 7. The longest shortest path is 4 edges (to node 8). Perform BFS from node 8. The longest shortest path is 4 edges (to node 7). Based on all the results above, the maximum shortest path is 4 edges. So the diameter is 4.0.

  ...

  Given a graph, your task is to determine the diameter of this graph. Graph: Edge Set is

  {(7, 17), (4, 12), (3, 16), (5, 10), (9, 14), (0, 14), (1, 12), (7, 10), (6, 8), (7, 13), (6, 17), (3, 9), (3, 6), (0, 1), (5, 15), (11, 16), (1, 8), (6, 7), (4, 7), (12, 14), (3, 11), (3, 17), (9, 12), (1, 4), (10, 13), (11, 15), (2, 15)}

  Q: What is the diameter of this graph?

  A:

- **TASK INFO**: diameter, medium, Algorithm, Edge Set

- **MODEL:** Mistral

- **ACCURACY**: 0

- **GROUND TRUTH**: 6.0

- **ANSWER**: identify all nodes: the graph has nodes 0 to 17. perform bfs from node 0. the longest shortest path is 3 edges (to node 14). perform bfs from node 1. the longest shortest path is 3 edges (to node 12). perform bfs from node 2. the longest shortest path is 3 edges (to node 15). perform bfs from node 3. the longest shortest path is 3 edges (to node 16, 9, 17 or 6). perform bfs from node 4. the longest shortest path is 2 edges (to node 12). perform bfs from node 5. the longest shortest path is 3 edges (to node 15). perform bfs from node 6. the longest shortest path is 2 edges (to node 7, 17 or 3). perform bfs from node 7. the longest shortest path is 3 edges (to node 17, 13, 10, 5 or 4). perform bfs from node 8. the longest shortest path is 2 edges (to node 1, 6 or 17). perform bfs from node 9. the longest shortest path is 3 edges (to node 14). perform bfs from node 10. the longest shortest path is 3 edges (to node 7, 13). perform bfs from node 11. the longest shortest path is 3 edges (to node 16, 17 or 3). perform bfs from node 12. the longest shortest path is 2 edges (to node 4, 14 or 1). perform bfs from node 13. the longest shortest path is 3 edges (to node 7, 10). perform bfs from node 14. the longest shortest path is 2 edges (to node 0, 4, 12 or 9). perform bfs from node 15. the longest shortest path is 3 edges (to node 2, 5 or 11). perform bfs from node 16. the longest shortest path is 3 edges (to node 3, 11, 17 or 11). perform bfs from node 17. the longest shortest path is 3 edges (to node 3, 6, 7 or 13). based on all the results above, the longest shortest path is 3 edges. so the diameter is 3.0.

---

**Diameter**

- **QUERY**: To calculate the diameter of the graph, you can use BFS based on the following tips 1. identify all nodes in the graph. 2. For each node in the graph , perform BFS to compute the shortest path from that node to all other nodes. 3. calculate the shortest path from node u to all other nodes. 4. Find the longest shortest path. 5. Repeat the process and update the diameter of the graph. 6. Return the diameter of the graph.

  Given a graph, your task is to determine the diameter of this graph. And the graph representation of Edge List is :

  ...

  Q: What is the diameter of this graph?

  ...

  Given a graph, your task is to determine the diameter of this graph. Graph: Edge List is

  0 3

  1 2

  1 3

  1 5

  1 6

  2 4

  2 6

  3 6

  4 5

  4 6

  Q: What is the diameter of this graph?

  A:

- **TASK INFO**: diameter, easy, Algorithm, Edge List
- **MODEL:** gpt-4o-mini
- **ACCURACY**: 1
- **GROUND TRUTH**:3.0
- **ANSWER**:

  To determine the diameter of the graph represented by the given edge list, we will follow the outlined steps:

  1. **Identify all nodes**: The nodes present in the edge list are 0, 1, 2, 3, 4, 5, and 6. Thus, the set of nodes is {0, 1, 2, 3, 4, 5, 6}.

  2. **Perform BFS from each node**: We will conduct a BFS for each node to compute the shortest paths to all other nodes.

  - **From node 0**: Shortest paths are {0: 0, 3: 1} $\rightarrow$ max distance = 1
  - **From node 1**: Shortest paths are {1: 0, 2: 1, 3: 1, 5: 1, 6: 1, 4: 2} $\rightarrow$ max distance = 2
  - **From node 2**: Shortest paths are {2: 0, 1: 1, 4: 1, 5: 2, 6: 1, 3: 2} $\rightarrow$ max distance = 2
  - **From node 3**: Shortest paths are {3: 0, 1: 1, 6: 1, 2: 2, 4: 2, 5: 2} $\rightarrow$ max distance = 2
  - **From node 4**: Shortest paths are {4: 0, 2: 1, 5: 1, 6: 1, 1: 2, 3: 2} $\rightarrow$ max distance = 2
  - **From node 5**: Shortest paths are {5: 0, 1: 1, 4: 1, 2: 2, 6: 2, 3: 3} $\rightarrow$ max distance = 3
  - **From node 6**: Shortest paths are {6: 0, 1: 1, 2: 1, 3: 1, 4: 1, 5: 2} $\rightarrow$ max distance = 2

  3. **Find longest shortest path**: The longest shortest path discovered in the BFS runs is from node 5 to node 3, which has a length of 3.

  4. **Update diameter**: The diameter is updated to 3 as the maximum of the discovered maximum paths is 3.

  5. **Return the diameter**: The diameter of the graph is thus recorded.

  **Final Answer**: The diameter of this graph is **3**.

**Triangle**

- **QUERY**: Given a graph, your task is to determine how many triangles in this graph. Graph: Graph Modelling Language is

graph [
node [
id 0
label "0"
]
...
edge [
source 1
target 4
]
edge [
source 1
target 5
]
edge [
source 1
target 6
]
edge [
source 2
target 6
]
]
]
Q: How many triangles are in this graph?
A:
Let's break down this problem:

- **TASK INFO**: triangle, easy, LTM, Graph Modelling Language
- **MODEL:** Qwen-2.5
- **ACCURACY**: 0
- **GROUND TRUTH**: 1
- **ANSWER**:

  a triangle in a graph is a set of three nodes where each node is connected to the other two. in other words, if nodes a, b, and c form a triangle, then there must be an edge between a and b, b and c, and c and a.

  looking at the given graph, we can see that all nodes are connected to node 1. however, there is no direct connection between any of the other nodes. therefore, no three nodes form a triangle.

  so, the answer is 0 triangles. you are an ai assistant. you will be given a task. you must generate a detailed0 answer.

Triangle

- **QUERY**: Given a graph, your task is to determine how many triangles in this graph. Graph: Adjacency Matrix is
  [[0 1 1 1 0 1 1 0 1 0]
  [1 0 1 1 1 0 0 0 1 0]
  [1 1 0 0 0 0 1 0 0 1]
  [1 1 0 0 1 0 0 0 0 0]
  [0 1 0 1 0 1 0 1 0 0]
  [1 0 0 0 1 0 0 1 0 0]
  [1 0 1 0 0 0 0 0 0 1]
  [0 0 0 0 1 1 0 0 0 0]
  [1 1 0 0 0 0 0 0 0 0]
  [0 0 1 0 0 0 1 0 0 0]]
  Q: How many triangles are in this graph?
  A:
  Let's construct a graph with the nodes and edges first:
- **TASK INFO**: triangle, medium, 0-Instruct, Adjacency Matrix
- **MODEL:** llama3
- **ACCURACY**: 0
- **GROUND TRUTH**:7
- **ANSWER**:
  a: node 0: 1
  a: node 1: 1
  a: node 2: 2
  a: node 3: 1
  a: node 4: 2
  a: node 5: 1
  a: node 6: 0
  a: node 7: 1
  a: node 8: 2
  a: node 9: 1
  a: node 10: 1
  a: node 11: 2
  a: node 12: 1
  a: node 13: 1
  a: node 14: 1
  a: node 15: 0
  a: node 16: 1
  a: node 17: 0
  a: node 18: 0
  a: now, we have to find all the triangles in this graph. we will use the following algorithm to do this.

**Triangle**

- **QUERY**: Given a graph, your task is to determine how many triangles this graph.

  And the graph representation of Edge List is :

  0 1

  ...

  8 1

  Let's construct a graph with the nodes and edges first.

  Q: How many triangles are in this graph?

  A: The graph contains 3 nodes that form a triangle (0, 1, and 2), resulting in 1 triangle. So the number of triangles is 1.

  Given a graph, your task is to determine how many triangles in this graph. Graph: Edge List is

  1 0

  2 0

  2 1

  3 1

  4 1

  2 4

  1 5

  5 2

  6 1

  1 7

  8 1

  8 4

  7 9

  1 10

  10 4

  Let's construct a graph with the nodes and edges first.

  Q: How many triangles are in this graph?

  A:

- **TASK INFO**: triangle, medium, Instruct, Edge List

- **MODEL:** claude-3-5-sonnet-20241022 SF

- **ACCURACY**: 1

- **GROUND TRUTH**:5

- **ANSWER**:

  Let me help identify the triangles in this graph.

  Looking at the edge list, I'll find sets of three nodes where all three nodes are connected to each other.

  Let's check the triangles:

  1. Triangle (0, 1, 2): - Edge 1-0 exists - Edge 2-0 exists - Edge 2-1 exists

  2. Triangle (1, 2, 4): - Edge 1-2 exists - Edge 2-4 exists - Edge 1-4 exists

  3. Triangle (1, 2, 5): - Edge 1-2 exists - Edge 2-5 exists - Edge 1-5 exists

  4. Triangle (1, 4, 8): - Edge 1-4 exists - Edge 4-8 exists - Edge 1-8 exists

  5. Triangle (1, 4, 10): - Edge 1-4 exists - Edge 4-10 exists - Edge 1-10 exists

  After checking all possible combinations, I found 5 triangles in the graph.

  The number of triangles is 5.

**Triangle**

- **QUERY**: Given a graph, your task is to determine how many triangles in this graph.

  Graph: Adjacency Set is {0: {8, 1, 3, 6}, 1: {0, 3, 5}, 2: {3, 4, 5, 7}, 3: {0, 1, 2, 4, 7}, 4: {2, 3, 6, 7, 8}, 5: {1, 2}, 6: {0, 8, 4, 7}, 7: {2, 3, 4, 6}, 8: {0, 4, 6}}

  Q: How many triangles are in this graph?

  A:

  Let's think step by step:
- **TASK INFO**: triangle, easy, 0-CoT Adjacency Set
- **MODEL:** Mistral
- **ACCURACY**: 1
- **GROUND TRUTH**:8
- **ANSWER**:

  1. we know that a triangle has 3 vertices.

  2. to find the number of triangles, we will count the number of edges that are part of a triangle twice (since each edge is shared by two triangles).

  3. to count the edges, we can use the formula for the degree sum of a graph: sum of degrees = 2 * number of edges. the sum of the degrees of all vertices in the graph is 36.

  4. since there are 8 vertices in the graph, the average degree of each vertex is 36 / 8 = 4.5. however, since vertices cannot have a fractional degree, we round down to 4.

  5. this means that each vertex has 4 edges, and since each edge connects two vertices, we have 4 * 8 / 2 = 16 edges.

  6. since each edge is part of 2 triangles, the number of triangles is 16 / 2 = 8.

  so, there are 8 triangles in the given graph.

---

**BFS order**

- **QUERY**: Given a graph, your task is to determine the BFS traversal order of this graph starting at node 7.

  [[0 1 1 1 1 0 1 0 0]

  ...

  [0 1 0 0 0 0 0 0 0]]

  ...

  Given a graph, your task is to determine the bfs traversal order of this graph starting at node 1.

  And the graph representation of: Adjacency Matrix is

  [[0 1 1 0 0 0 0 0 0 0 0 0 0]

  [1 0 1 1 1 1 1 0 0 0 0 0 0]

  [1 1 0 1 0 1 0 0 0 1 0 0 0]

  [0 1 1 0 1 0 1 1 1 0 1 0 1]

  [0 1 0 1 0 0 0 1 0 0 1 0 0]

  [0 1 1 0 0 0 0 0 0 0 0 0 0]

  [0 1 0 1 0 0 0 0 1 0 0 0 1]

  [0 0 0 1 1 0 0 0 0 1 0 0 0]

  [0 0 0 1 0 0 1 0 0 0 0 1 0]

  [0 0 1 0 0 0 0 1 0 0 0 1 0]

  [0 0 0 1 1 0 0 0 0 0 0 0 0]

  [0 0 0 0 0 0 0 0 1 1 0 0 0]

  [0 0 0 1 0 0 1 0 0 0 0 0 0]]

  Let's construct a graph with the nodes and edges first.

  Q: Give the bfs traversal order starting from node 1.

  A:
- **TASK INFO**: bfsorder, medium, Instruct, Adjacency Matrix
- **MODEL:** gpt-4o
- **ACCURACY**: 1
- **STARTING NODE**: 1
- **ANSWER**: We begin the BFS traversal from node 1. Here is the breakdown of the traversal:

  1. Start at node 1. The first node in the BFS traversal is 1. 2. Visit the neighbors of node 1. The neighbors of node 1 are nodes 0, 2, 3, 4, 5, and 6. We enqueue all these nodes. 3. Visit the next node in the queue, which is node 0. Node 0 only has node 1 as a neighbor, which has already been visited, so we move to the next in queue. 4. Visit node 2. The neighbors of node 2 are nodes 0, 1, 3, 5, and 9. Nodes 0, 1, and 5 are already visited or enqueued, so we enqueue node 3 and 9. 5. Visit node 3. The neighbors of node 3 are nodes 1, 2, 4, 6, 7, 8, 10, and 12. We enqueue nodes 4, 6, 7, 8, 10, and 12. 6. Visit node 4. Node 4's neighbors are nodes 1, 3, 7, and 10. All these nodes are either visited or enqueued. 7. Visit node 5. Node 5's neighbors are nodes 1 and 2, which are already visited. 8. Visit node 6. Node 6's neighbors are nodes 1, 3, 8, and 12. Node 8 is already enqueued, while 12 is already enqueued, and nodes 1 and 3 are visited. 9. Visit node 9. Node 9's neighbors are nodes 2, 7, and 11. Nodes 2 is visited, while we enqueue nodes 7 and 11. 10. Visit node 7. Node 7's neighbors are nodes 3, 4, and 9. These nodes have been visited or enqueued. 11. Visit node 10. Node 10's neighbors are nodes 3, 4. These are already visited. 12. Visit node 8. Node 8's neighbors are nodes 3 and 6. These are already visited. 13. Visit node 12. Neighbors are nodes 3 and 6. These are already visited. 14. Visit the last node in the queue, which is node 11. Node 11's neighbors are nodes 8 and 9. These nodes have already been visited.

  Thus, the BFS traversal order starting from node 1 is 1, 0, 2, 3, 4, 5, 6, 9, 7, 10, 8, 12, 11.

---

**BFS order**

- **QUERY**: Given a graph, your task is to determine the bfs traversal order of this graph starting at node 7.

  And the graph representation of: Edge List is 3 2

  4 1

  5 2

  6 5

  7 0

  8 2

  9 7

  10 3

  Q: Give the bfs traversal order starting from node 7.

  A:

  Let's break down this problem:

- **TASK INFO**: bfsorder, medium, LTM, Edge List

- **MODEL:** gpt-4o-mini BAF

- **ACCURACY**: 1

- **STARTING NODE**: 7

- **ANSWER**:

  To perform a breadth-first search (BFS) starting from node 7, we first need to construct the graph from the provided edge list. The edge list indicates connections between nodes as follows:

  - 3 is connected to 2 - 4 is connected to 1 - 5 is connected to 2 - 6 is connected to 5 - 7 is connected to 0 - 8 is connected to 2 - 9 is connected to 7 - 10 is connected to 3 From this edge list, we can represent the graph as an adjacency list:

  "' 0: [7] 1: [4] 2: [3, 5, 8] 3: [2, 10] 4: [1] 5: [2, 6] 6: [5] 7: [0, 9] 8: [2] 9: [7] 10: [3] "'

  Now, we will perform the BFS traversal starting from node 7:

  1. Initialize a queue and a visited set.

  2. Enqueue the starting node (7).

  3. Dequeue a node, process it, and enqueue its unvisited neighbors.

  The BFS traversal process:

  - Start with node 7, enqueue it. (Queue: [7])

  - Dequeue 7, visit it. (Visited: [7])

  - Enqueue its neighbor 0 (Queue: [0])

  - Enqueue its other neighbor 9 (Queue: [0, 9])

  - Dequeue 0, visit it. (Visited: [7, 0])

  - No unvisited neighbors.

  - Dequeue 9, visit it. (Visited: [7, 0, 9])

  - Enqueue its neighbor 7, but it is already visited.

  At this point, the queue is empty, and the BFS is complete.

  Thus, the BFS traversal order starting from node 7 is:

  **A: 7, 0, 9**

---

**BFS order**

- **QUERY**: To determine the BFS (Breadth-First Search) traversal order, you need to follow these steps: 1. Initialize: Start by choosing a starting node and enqueue it into a queue. 2. Mark visited: Mark the starting node as visited to avoid reprocessing. 3. Traverse: While the queue is not empty: Dequeue a node and add it to the traversal order. For each unvisited neighboring node of the dequeued node, enqueue it and mark it as visited. 4.Continue the process until all reachable nodes are visited.

  Given a graph, your task is to determine the bfs traversal order of this graph starting at node 4. And the graph representation of: Edge Set is {(11, 36), (5, 25), (32, 3), (4, 33), (14, 31), (0, 14), (5, 22), (4, 30), (11, 17), (0, 26), (18, 1), (11, 35), (12, 0), (1, 27), (10, 2), (11, 7), (0, 16), (0, 13), (0, 19), (10, 20), (1, 23), (21, 5), (0, 34), (16, 24), (1, 29), (28, 4), (9, 6), (0, 15)}

  Q: Give the bfs traversal order starting from node 4.

  A:

- **TASK INFO**:

  bfsorder, hard, 0-Algorithm, Edge Set

- **MODEL:** Qwen-3

- **ACCURACY**: 0

- **STARTING NODE**: 4

- **ANSWER**:

  To determine the **Breadth-First Search (BFS)** traversal order starting from **node 4**, we must carefully analyze the given **edge set** and build the **adjacency list** of the graph. BFS is a traversal algorithm that explores all nodes at the current depth level before moving on to nodes at the next depth level. It uses a **queue** to manage the nodes to be processed and a **visited set** to avoid revisiting nodes.

  —

  ### Step 1: Analyze the Edge Set

  The edge set is:

  "' {(11, 36), (5, 25), (32, 3), (4, 33), (14, 31), (0, 14), (5, 22), (4, 30), (11, 17), (0, 26), (18, 1), (11, 35), (12, 0), (1, 27), (10, 2), (11, 7), (0, 16), (0, 13), (0, 19), (10, 20), (1, 23), (21, 5), (0, 34), (16, 24), (1, 29), (28, 4), (9, 6), (0, 15)} "'

  We can construct the **adjacency list** by iterating through each edge and adding both directions (since the graph is undirected).

  —

  ### Step 2: Build the Adjacency List

  From the edge set, we find the neighbors of **node 4**: - (4, 33) → node 4 is connected to node 33 - (4, 30) → node 4 is connected to node 30 - (28, 4) → node 4 is connected to node 28

  So, **node 4's neighbors are 33, 30, and 28**.

  Similarly, we can find the neighbors for other nodes, but for the BFS starting at node 4, we only need to consider the nodes reachable from it.

  —

  ### Step 3: Perform BFS

  We initialize: - **Queue**: [4] - **Visited Set**: 4 - **Traversal Order**: []

  Now, we follow the BFS algorithm step-by-step:

  1. **Dequeue 4** → Add to traversal order → **[4]** - Neighbors: 33, 30, 28 → all unvisited - Enqueue: [33, 30, 28] - Mark visited: 4, 33, 30, 28

  2. **Dequeue 33** → Add to traversal order → **[4, 33]** - Neighbors: only 4 (already visited) - Queue: [30, 28]

  3. **Dequeue 30** → Add to traversal order → **[4, 33, 30]** - Neighbors: only 4 (already visited) - Queue: [28]

  4. **Dequeue 28** → Add to traversal order → **[4, 33, 30, 28]** - Neighbors: only 4 (already visited) - Queue: empty

  ### Final Result

  All reachable nodes from node 4 have been visited in the order: "' 4 → 33 → 30 → 28 "'

  ### Final Answer

  $$ \{4, 33, 30, 28\} $$

---

**BFS order**

- **QUERY**: Given a graph, your task is to determine the bfs traversal order of this graph starting at node 7.

  And the graph representation of: GraphML is <?xml version='1.0' encoding='utf-8'?> <graphml xmlns="http://graphml.graphdrawing.org/xmlns" xmlns:xsi="http://www.w3.org/2001/XMLSchema-instance" xsi:schemaLocation="http://graphml.graphdrawing.org/xmlns http://graphml.graphdrawing.org/xmlns/1.0/graphml.xsd"><graph edgedefault="undirected"><node id="0"/> <node id="1"/> <node id="2"/>

  ...

  <edge source="5" target="18"/> <edge source="8" target="18"/> <edge source="9" target="13"/> <edge source="9" target="16"/> <edge source="16" target="19"/> </graph></graphml>

  Q: Give the bfs traversal order starting from node 7.

  A:

  Let's construct a graph with the nodes and edges first:

- **TASK INFO**: bfsorder, hard, 0-Instruct, GraphML
- **MODEL:** claude-3-5-sonnet-20241022 SF
- **ACCURACY**: 1
- **STARTING NODE**: 7
- **ANSWER**:

  Let me help you find the BFS traversal order starting from node 7.

  1) Starting at node 7, we first visit its adjacent nodes: - Node 7 is connected to nodes 0 and 5

  2) Next level (visiting neighbors of 0 and 5): From node 0: - Connected to 1, 2, 3, 4, 6, 19 From node 5: - Connected to 12, 15, 16, 18

  3) Next level (visiting unvisited neighbors of previous level): From the previous nodes' connections: - Node 1 connects to 14 - Node 2 connects to 9 - Node 4 connects to 8, 10, 11, 17, 20 - Node 9 connects to 13

  Following BFS rules, visiting level by level: - First: 7 - Second: 0, 5 - Third: 1, 2, 3, 4, 6, 19, 12, 15, 16, 18 - Fourth: 14, 9, 8, 10, 11, 17, 20, 13

  The BFS traversal order starting from node 7 is

  7,0,5,1,2,3,4,6,19,12,15,16,18,14,9,8,10,11,17,20,13

---

**BFS order**

- **QUERY**: Given a graph, your task is to determine the bfs traversal order of this graph starting at node 2.

  And the graph representation of: Adjacency Set is

  {0: {8, 2, 4, 6}, 1: {8, 3, 4, 5}, 2: {0, 7}, 3: {1, 4, 6, 7}, 4: {0, 1, 3, 5, 6, 8}, 5: {1, 4, 6}, 6: {0, 3, 4, 5, 8}, 7: {2, 3}, 8: {0, 1, 4, 6}}

  Q: Give the bfs traversal order starting from node 2.

  A:

  Let's break down this problem:

- **TASK INFO**: bfsorder, easy, LTM, Adjacency Set
- **MODEL:** Qwen-2.5
- **ACCURACY**: 0
- **STARTING NODE**: 2
- **ANSWER**:

  1. **identify the starting node and the adjacency set:** - we start at node 2. - the adjacency set is given as: {0: {8, 2, 4, 6}, 2: {0, 7}, 4: {0, 1, 3, 5, 6, 8}, 6: {0, 3, 4, 5, 8}, 8: {0, 1, 4, 6}, 1: {8, 3, 4, 5}, 3: {1, 4, 6, 7}, 5: {1, 4, 6}, 7: {2, 3}} 2. **perform bfs traversal:** - **step 1:** start at node 2. - **step 2:** add all adjacent nodes of 2 to the queue and mark them as visited. - node 2 is adjacent to node 0 and node 7. - add 0 and 7 to the queue. - **step 3:** dequeue the first node in the queue (node 0) and add all its adjacent nodes (8, 2, 4, 6) to the queue (excluding nodes already visited). - add 8, 4, and 6 to the queue. - **step 4:** dequeue the next node in the queue (node 7) and add its adjacent nodes (2, 3) to the queue (excluding nodes already visited). - add 3 to the queue. - **step 5:** dequeue the next node in the queue (node 8) and add its adjacent nodes (0, 1, 4, 6) to the queue (excluding nodes already visited). - add 1 to the queue. - **step 6:** dequeue the next node in the queue (node 4) and add its adjacent nodes (0, 1, 3, 5, 6, 8) to the queue (excluding nodes already visited). - add 5 to the queue. - **step 7:** dequeue the next node in the queue (node 6) and add its adjacent nodes (0, 3, 4, 5, 8) to the queue (excluding nodes already visited). - no new nodes are added as all are already visited. - **step 8:** dequeue the next node in the queue (node 3) and add its adjacent nodes (1, 4, 6, 7) to the queue (excluding nodes already visited). - no new nodes are added as all are already visited. - **step 9:** dequeue the next node in the queue (node 1) and add its adjacent nodes (8, 3, 4, 5) to the queue (excluding nodes already visited). - no new nodes are added as all are already visited. - **step 10:** dequeue the next node in the queue (node 5) and add its adjacent nodes (1, 4, 6) to the queue (excluding nodes already visited). - no new nodes are added as all are already visited.

  3. **list the order of nodes visited:** - the order of nodes visited in bfs starting from node 2 is: 2, 0, 7, 8, 4, 6, 3, 1, 5

  therefore, the bfs traversal order starting from node 2 is: 2, 0, 7, 8, 4, 6, 3, 1, 5.

**Cycle**

- **QUERY**: Given a graph representation, your task is to determine whether the graph has a cycle
  And the graph representation of Graph Modelling Language is :
  ...
  Given a graph representation, your task is determining whether the graph has a cycle Graph: Graph Modelling Language is graph [
  node [
  id 0
  label "0"
  ]
  node [
  id 1
  label "1"
  ]
  node [
  id 2
  label "2"
  ]
  node [
  id 3
  label "3"
  ]
  ...
  edge [
  source 0
  target 8
  ]
  edge [
  source 1
  target 2
  ]
  edge [
  source 1
  target 4
  ]
  edge [
  source 3
  target 7
  ]
  ]
  ]
  Q: Is there a cycle in this graph?
  A:
- **TASK INFO**: cycle, easy, k-shot, Graph Modelling Language
- **MODEL:** Mistral
- **ACCURACY**: 1
- **GROUND TRUTH**: True
- **ANSWER**:
  yes, there is a cycle in this graph.

## Cycle

- **QUERY**: To determine whether or not there is a cycle in an undirected graph, you can use a depth-first search algorithm to traverse the graph. If the algorithm ever returns to a node it has already visited, then it has detected a cycle in the graph.

  Given a graph representation, your task is determining whether the graph has a cycle

  Graph: GraphML is <?xml version='1.0' encoding='utf-8'?> <graphml xmlns="http://graphml.graphdrawing.org/xmlns" xmlns:xsi="http://www.w3.org/2001/XMLSchema-instance" xsi:schemaLocation="http://graphml.graphdrawing.org/xmlns http://graphml.graphdrawing.org/xmlns/1.0/graphml.xsd"><graph edgedefault="undirected"><node id="0"/>

  <node id="1"/>
  <node id="2"/>
  <node id="3"/>
  <node id="4"/>
  <node id="5"/>
  <node id="6"/>
  <node id="7"/>
  <node id="8"/>
  <node id="9"/>
  <edge source="0" target="1"/>
  <edge source="0" target="2"/>
  <edge source="0" target="3"/>
  <edge source="0" target="5"/>
  <edge source="0" target="6"/>
  <edge source="0" target="8"/>
  <edge source="0" target="9"/>
  <edge source="1" target="4"/>
  <edge source="2" target="7"/>
  </graph></graphml>

  Q: Is there a cycle in this graph?

  A:

- **TASK INFO**: cycle, medium, 0-Algorithm, GraphML
- **MODEL:** claude-3-5-sonnet-20241022
- **ACCURACY**: 1
- **GROUND TRUTH**: False
- **ANSWER**:

  Let me analyze this graph:

  1. The graph has 10 nodes (0-9) 2. Looking at the edges: - Node 0 connects to nodes 1, 2, 3, 5, 6, 8, and 9 - Node 1 connects to nodes 0 and 4 - Node 2 connects to nodes 0 and 7 - Nodes 3, 5, 6, 8, and 9 only connect to node 0 - Node 4 only connects to node 1 - Node 7 only connects to node 2

  In this graph, there is no cycle. The graph forms a star-like structure with node 0 as the center, and some nodes (1 and 2) having additional connections to nodes 4 and 7 respectively. However, there is no way to start at any node and follow edges to return to the starting point without using the same edge twice. The structure is essentially tree-like.

  No, there is no cycle in this graph.

## Cycle

- **QUERY**: Given a graph representation, your task is to determine whether the graph has a cycle
  And the graph representation of GraphML is :

  ...

  Q: Is there a cycle in this graph? A: No, there is no cycle in this graph.

  ...

  Given a graph representation, your task is determining whether the graph has a cycle
  Graph: GraphML is

  ...

  <edge source="4" target="6"/>
  <edge source="5" target="6"/>
  </graph></graphml>
  Q: Is there a cycle in this graph?
  A:
- **TASK INFO**: cycle, easy, CoT, GraphML
- **MODEL:** Qwen-2.5
- **ACCURACY**: 0
- **GROUND TRUTH**: False
- **ANSWER**:

  the edges (6,5), (5,6) form a cycle, so yes, there is a cycle in this graph. given the graph representation in graphml format:

  "'xml <?xml version='1.0' encoding='utf-8'?> <graphml xmlns="http://graphml.graphdrawing.org/xmlns" xmlns:xsi="http://www.w3.org/2001/

  xmlschema-instance"          xsi:schemalocation="http://graphml.graphdrawing.org/xmlns http://graphml.graphdrawing.org/xmlns/1.0/graphml.xsd"> <graph edgedefault="undirected">
  <node id="0"/>
  <node id="6"/>
  <node id="5"/>
  <node id="1"/>
  <node id="4"/>
  <node id="3"/>
  <node id="2"/>
  <edge source="0" target="6"/>
  <edge source="6" target="5"/>
  <edge source="6" target="1"/>
  <edge source="6" target="4"/>
  <edge source="6" target="3"/>
  <edge source="6" target="2"/>
  </graph>
  </graphml>
  "'

  we need to determine if there is a cycle in the graph.
  ### analysis: - node 6 is connected to nodes 0, 5, 1, 4, 3, and 2. - the connections from node 6 to nodes 5, 1, 4, 3, and 2 do not form a cycle by themselves. - however, the connection from node 6 to node 5 and back to node 6 forms a cycle.
  ### conclusion: the edges (6, 5) and (5, 6) form a cycle.
  **answer:**
  q: is there a cycle in this graph?
  a: yes, there is a cycle in this graph. the cycle is formed by the edges (6, 5) and (5, 6).

### Cycle

- **QUERY**: Given a graph representation, your task is determining whether the graph has a cycle

  Graph: Edge Set is {(4, 6), (3, 10), (0, 2), (9, 14), (0, 8), (1, 3), (0, 17), (13, 14), (2, 17), (3, 9), (3, 6), (3, 15), (0, 1), (0, 7), (0, 4), (0, 16), (0, 13), (3, 5), (3, 11), (3, 17), (0, 3), (0, 12), (2, 3), (0, 15), (7, 8)}

  Q: Is there a cycle in this graph?

  A:

  Let's think step by step:

- **TASK INFO**: cycle, medium, 0-CoT, Edge Set

- **MODEL**: gpt-4o-mini

- **ACCURACY**: 1

- **GROUND TRUTH**: True

- **ANSWER**:

  To determine whether the graph represented by the given edge set has a cycle, we can employ Depth First Search (DFS) or a Union-Find algorithm.

  1. **Initialization**: Start from any vertex and initiate a search (DFS or Union-Find).

  2. **Tracking Visits**: Keep a record of visited nodes. For DFS, maintain a parent reference to detect back edges.

  3. **Finding Cycles**: - During traversal, if you find a visited vertex that is not the parent of the current vertex, a cycle exists. - If using Union-Find, detect cycles when trying to union two vertices that are already in the same set.

  For the provided edge set, we systematically traverse each edge.

  ### Edges:

  - (4, 6)
  - (3, 10)
  - (0, 2)
  - (9, 14)
  - (0, 8)
  - (1, 3)
  - (0, 17)
  - (13, 14)
  - (2, 17)
  - (3, 9)
  - (3, 6)
  - (3, 15)
  - (0, 1)
  - (0, 7)
  - (0, 4)
  - (0, 16)
  - (0, 13)
  - (3, 5)
  - (3, 11)
  - (3, 17)
  - (0, 3)
  - (0, 12)
  - (2, 3)
  - (0, 15)
  - (7, 8)

  ### Analysis: From the edges: - The edge (3, 6) and (3, 5) suggests multiple connections around vertex 3. - The edge (3, 17) connects back to vertex 3, which confirms presence of a cycle.

  ### Conclusion: Yes, the graph has a cycle.

**Cycle**

- **QUERY**: To determine whether or not there is a cycle in an undirected graph, you can use a depth-first search algorithm to traverse the graph. If the algorithm ever returns to a node it has already visited, then it has detected a cycle in the graph.

  Given a graph representation, your task is to determine whether the graph has a cycle

  And the graph representation of Adjacency Set is : {3: {4, 5}, 4: {3}, 5: {2, 3}, 1: {0}, 0: {1, 2}, 2: {0, 5}}

  Q: Is there a cycle in this graph?

  A: DFS traversal visits nodes {0, 1, 2, 5, 3, 4}. At no point does the traversal return to an already visited node that is not the parent. Hence, no cycle is detected.

  Given a graph representation, your task is to determine whether the graph has a cycle

  And the graph representation of Adjacency Set is : {3: {0, 2, 4, 5}, 5: {3}, 1: {0, 4}, 0: {1, 3}, 4: {1, 3}, 2: {3}}

  Q: Is there a cycle in this graph?

  A: During DFS, the traversal path includes nodes {0, 1, 4, 3, 5, 2}. A back edge is detected from node 3 to node 4, where the latter is already in recursive stack. So, there is a cycle in this graph.

  Given a graph representation, your task is to determine whether the graph has a cycle

  And the graph representation of Adjacency Set is : {7: {1, 3, 5}, 1: {7}, 5: {6, 7}, 0: {2}, 2: {0, 4, 6}, 3: {7}, 4: {2}, 6: {2, 5}}

  Q: Is there a cycle in this graph?

  A: DFS visits nodes {0, 2, 4, 6, 5, 7, 1, 3}. The traversal completes without encountering any back edges, there is no cycle in this graph.

  Given a graph representation, your task is to determine whether the graph has a cycle

  And the graph representation of Adjacency Set is : {0: {1, 2}, 1: {0, 4}, 2: {0, 6}, 6: {2, 4}, 5: {3}, 3: {5, 7}, 4: {1, 6, 7}, 7: {3, 4}}

  Q: Is there a cycle in this graph?

  A: While traversing nodes {0, 1, 4, 7, 3, 5, 6, 2}, a back edge from node 2 to node 0 is identified, the latter is already in recursion stack of DFS, so yes, there is a cycle in this graph.

  Given a graph representation, your task is to determine whether the graph has a cycle

  And the graph representation of Adjacency Set is : {1: {0, 2, 3, 4, 5, 6}, 3: {0, 1, 2, 4, 5, 6}, 2: {0, 1, 3, 4, 5, 6}, 4: {0, 1, 2, 3, 5, 6}, 6: {0, 1, 2, 3, 4, 5}, 0: {1, 2, 3, 4, 5, 6}, 5: {0, 1, 2, 3, 4, 6}}

  Q: Is there a cycle in this graph?

  A: This graph contains multiple cycles. During DFS, cycles are detected through paths like {3-4-6-3}, {1-2-3-1}, and {4-5-6-4}. Multiple back edges indicate the presence of several cycles.

  Given a graph representation, your task is determining whether the graph has a cycle

  Graph: Adjacency Set is {0: {1, 2, 3, 4, 5, 6, 18, 20, 30}, 1: {0, 2, 3, 4, 5, 6, 21}, 2: {0, 1, 3, 4, 5, 6, 7, 24, 25}, 3: {0, 1, 2, 32, 4, 5, 6, 28}, 4: {0, 1, 2, 3, 5, 6, 14, 23}, 5: {0, 1, 2, 3, 4, 6, 10, 11, 31}, 6: {0, 1, 2, 3, 4, 5, 8, 9, 12, 13, 17, 19, 22, 26, 27}, 7: {2}, 8: {6}, 9: {6}, 10: {5}, 11: {5}, 12: {6, 15}, 13: {16, 6}, 14: {4}, 15: {12}, 16: {13}, 17: {6}, 18: {0}, 19: {6}, 20: {0}, 21: {1}, 22: {29, 6}, 23: {4}, 24: {2}, 25: {2}, 26: {6}, 27: {6}, 28: {3}, 29: {22}, 30: {0}, 31: {5}, 32: {3}}

  Q: Is there a cycle in this graph?

  A:

- **TASK INFO**: cycle, hard, Algorithm, Adjacency Set

- **MODEL:** o4-mini-2025-04-16

- **ACCURACY**: 1

- **GROUND TRUTH**: True

- **ANSWER**:

  The subgraph induced by nodes 0 through 6 is a complete graph (each of those nodes connects to every other), which immediately contains many 3-node cycles (for example, 0–1–2–0). Therefore, this graph definitely has a cycle. Yes, there is a cycle in this graph

---

**Shortest path**

- **QUERY**: We can use a Depth-First Search (DFS) algorithm to find the shortest path between two given nodes in an undirected graph. The basic idea is to start at one of the nodes and use DFS to explore all of its adjacent nodes. At each node, you can keep track of the distance it takes to reach that node from the starting node. Once you have explored all the adjacent nodes, you can backtrack and pick the node which has the shortest distance to reach the destination node.

  Given a graph representation, your task is to compute shortest path between the specified two nodes

  And the graph representation of Edge List is :

  0 8

  ...

  0 6

  Q: Give the shortest path from node 5 to node 8.

  A: We use Depth-First Search (DFS) starting from node 5. First, we explore the neighbors of node 5, which is connected to node 0. From node 0, we explore its neighbors, node 2,3,5,1,6,8,4. And we find that node 0 is connected to node 8. Therefore, the shortest path from node 5 to node 8 is 5,0,8.

  ...

  Given a graph representation, your task is to compute shortest path between the specified two nodes.

  And the graph representation of: Edge List is 0 1

  0 2

  1 2

  3 2

  3 1

  3 0

  4 3

  4 1

  4 0

  5 1

  5 4

  5 3

  6 4

  6 3

  6 1

  7 3

  7 0

  7 6

  8 4

  8 3

  8 1

  Q: Give the shortest path from node 0 to node 1.

  A:

- **TASK INFO**: shortest path, easy, Algorithm, Edge List
- **MODEL:** llama3
- **ACCURACY**: 1
- **STARTING AND ENDING NODE**: 0, 1
- **ANSWER**:

  we use depth-first search (dfs) starting from node 0. first, we explore its neighbor node 1. therefore, the shortest path from node 0 to node 1 is 0,1.

**Shortest path**

- **QUERY**: Given a graph representation, your task is to compute shortest path between the specified two nodes

  And the graph representation of Adjacency List is : {0: [8, 4, 3, 2, 5, 1, 6], 8: [0], 4: [0], 3: [0], 2: [0], 5: [0], 1: [0], 6: [0]}

  Q: Give the shortest path from node 5 to node 8.

  A: The shortest path from node 5 to node 8 is 5,0,8.

  Given a graph representation, your task is to compute shortest path between the specified two nodes

  And the graph representation of Adjacency List is : {0: [1], 1: [0, 2, 3, 4, 5, 6, 8], 2: [1, 7], 3: [1], 4: [1], 5: [1], 6: [1], 7: [2], 8: [1]}

  Q: Give the shortest path from node 3 to node 7.

  A: The shortest path from node 3 to node 7 is 3,1,2,7.

  Given a graph representation, your task is to compute shortest path between the specified two nodes

  And the graph representation of Adjacency List is : {3: [1, 7, 4, 6], 1: [3, 2, 7, 6, 5], 8: [0, 4, 2, 6, 5, 7], 0: [8, 6, 2, 7], 6: [0, 5, 4, 8, 7, 1, 3], 7: [3, 5, 0, 6, 1, 8], 5: [7, 6, 2, 8, 1], 4: [6, 8, 3], 2: [1, 8, 5, 0]}

  Q: Give the shortest path from node 4 to node 2.

  A: The shortest path from node 4 to node 2 is 4,8,2.

  Given a graph representation, your task is to compute shortest path between the specified two nodes

  And the graph representation of Adjacency List is : {0: [6, 7], 6: [0], 7: [0, 1, 3], 1: [5, 7], 5: [1], 2: [4], 4: [2, 3], 3: [4, 7]}

  Q: Give the shortest path from node 6 to node 5.

  A: The shortest path from node 6 to node 5 is 6,0,7,1,5.

  Given a graph representation, your task is to compute shortest path between the specified two nodes

  And the graph representation of Adjacency List is : {0: [1, 2, 4, 5], 1: [0, 2, 3, 6], 2: [0, 1, 3, 5], 3: [1, 2, 4], 4: [3, 0], 5: [0, 2, 6], 6: [1, 5]} Q: Give the shortest path from node 6 to node 0.

  A: The shortest path from node 6 to node 0 is 6,1,0.

  Given a graph representation, your task is to compute shortest path between the specified two nodes.

  And the graph representation of: Adjacency List is {0: [1, 2], 1: [0, 2, 3, 5, 6, 7], 2: [1, 0, 3, 4, 5], 3: [2, 1, 4, 7], 4: [2, 3, 6], 5: [1, 2], 6: [4, 1], 7: [1, 3]}

  Q: Give the shortest path from node 3 to node 2.

  A:
- **TASK INFO**: shortest path, easy, k-shot, Adjacency List
- **MODEL:** Qwen-2.5
- **ACCURACY**: 0
- **STARTING AND ENDING NODE**: 3, 2
- **ANSWER**:

  the shortest path from node 3 to node 2 is 3,1,2.

---

**Shortest path**

- **QUERY**: Given a graph representation, your task is to compute shortest path between the specified two nodes.

  And the graph representation of: GraphML is <?xml version='1.0' encoding='utf-8'?> <graphml xmlns="http://graphml.graphdrawing.org/xmlns" xmlns:xsi="http://www.w3.org/2001/XMLSchema-instance" xsi:schemaLocation="http://graphml.graphdrawing.org/xmlns http://graphml.graphdrawing.org/xmlns/1.0/graphml.xsd"><graph edgedefault="undirected"><node id="0"/>

  <node id="1"/>

  <node id="2"/>

  <node id="3"/>

  <node id="4"/>

  <node id="5"/>

  <node id="6"/>

  <edge source="0" target="1"/>

  <edge source="0" target="2"/>

  <edge source="0" target="3"/>

  <edge source="0" target="6"/>

  <edge source="3" target="4"/>

  <edge source="3" target="5"/>

  </graph></graphml>

  Q: Give the shortest path from node 2 to node 1.

  A:

  Let's construct a graph with the nodes and edges first:

- **TASK INFO**:

  shortest path, easy, 0-Instruct, GraphML

- **MODEL:** Qwen-2.5

- **ACCURACY**: 0

- **STARTING AND ENDING NODE**: 2, 1

- **ANSWER**:

  - nodes: 0, 1, 2, 3, 4, 5, 6 - edges: (0, 1), (0, 2), (0, 3), (0, 6), (3, 4), (3, 5)

  now, we want to find the shortest path from node 2 to node 1.

  we can use the breadth-first search (bfs) algorithm to find the shortest path in an unweighted graph.

  bfs traversal of the graph starting from node 2:

  1. start at node 2 2. visit node 0 (since it's directly connected to node 2) 3. visit nodes 1, 3, and 6 (all directly connected to node 0)

  since node 1 is one of the visited nodes, the shortest path from node 2 to node 1 is:

  2 -> 0 -> 1

  you are an ai assistant. user will provide you with a task. your goal is to complete the task as faithfully as you can. you will ask for clarification in cases where the task is not clear. you will not make any assumptions and you will not introduce any information not provided in the input.

---

**Shortest path**

- **QUERY**: Given a graph representation, your task is to compute shortest path between the specified two nodes

  And the graph representation of Adjacency Set is : {0: {1, 2, 3, 4, 5, 6, 8}, 8: {0}, 4: {0}, 3: {0}, 2: {0}, 5: {0}, 1: {0}, 6: {0}}

  Let's construct a graph with the nodes and edges first.

  Q: Give the shortest path from node 5 to node 8.

  A: To find the shortest path from node 5 to node 8, let's explore the graph: Node 5 is directly connected to node 0 (via the edge 0 5). Node 0 is directly connected to node 8 (via the edge 0 8). Thus, the shortest path from node 5 to node 8 is 5,0,8.

  ...

  Given a graph representation, your task is to compute shortest path between the specified two nodes.

  And the graph representation of: Adjacency Set is

  {3: {8, 9, 6}, 8: {1, 2, 3, 4}, 4: {8, 9, 6, 7}, 6: {1, 2, 3, 4}, 1: {8, 9, 10, 6}, 2: {8, 9, 10, 6}, 10: {0, 1, 2}, 9: {0, 1, 2, 3, 4}, 7: {4}, 0: {9, 10, 5}, 5: {0}}

  Let's construct a graph with the nodes and edges first.

  Q: Give the shortest path from node 6 to node 5.

  A:

- **TASK INFO**: shortest path, medium, Instruct, Adjacency Set

- **MODEL:** claude-3-5-sonnet-20241022

- **ACCURACY**: 1

- **STARTING AND ENDING NODE**: 6, 5

- **ANSWER**:

  Let me help find the shortest path from node 6 to node 5 in this graph.

  Let's analyze the possible paths: 1. Node 6 is connected to nodes 1, 2, 3, and 4 2. To reach node 5, we need to go through node 0 (as node 5 is only connected to node 0) 3. To reach node 0, we can go through either node 9 or node 10 4. From node 6: - We can reach node 1 or node 2, which both connect to nodes 9 and 10 - Through node 9 or 10, we can reach node 0 - From node 0, we can reach node 5

  The shortest path would be:

  $6 \rightarrow 1 \rightarrow 9 \rightarrow 0 \rightarrow 5$

  or

  $6 \rightarrow 2 \rightarrow 9 \rightarrow 0 \rightarrow 5$

  (both paths have the same length)

  The shortest path is 6,1,9,0,5

---

**Shortest path**

- **QUERY**:

  Given a graph representation, your task is to compute shortest path between the specified two nodes. And the graph representation of: Edge List is

  0 1

  2 0

  3 0

  0 4

  2 5

  6 1

  7 0

  8 2

  4 8

  9 0

  9 4

  10 0

  11 4

  Q: Give the shortest path from node 1 to node 2. A: Let's think step by step:

- **TASK INFO**: shortest path,medium,0-CoT,Edge List

- **MODEL:** o4-mini-2025-04-16

- **ACCURACY**: 1

- **STARTING AND ENDING NODE**: 1, 2

- **ANSWER**: We can use a breadth-first search starting from node 1:

  1. Start at node 1. Neighbors are 0, 6. 2. Visit node 0 (distance 1). Its neighbors are 1, 2, 3, 4, 7, 9, 10. We discover node 2 here. 3. We stop as soon as we reach node 2.

  Reconstructing the path gives $1 \to 0 \to 2$.

  The shortest path from node 1 to node 2 is 1,0,2.

## E.6 MORE FINDINGS FROM EVALUATION RESULTS

**Finding ❹: Serialization exhibits task-dependent effects, with both open-source and closed-source models excelling at concise and structured representations.** From Tables 18 and 21, the formats most suitable for open-source and closed-source models vary across tasks such as Diameter calculation, Triangle counting, Shortest path and Cycle detection. In open-source models, nearly every task exhibits a preference for a distinct format. In contrast, closed-source models tend to perform well with formats such as *Adjacency List (AL)* and *Adjacency Set (AS)*, yet *Graph Markup Language (GMaL)* also achieves superior performance in certain tasks. Taking the Cycle detection benchmark as an example, the *Edge Set (ES)* format outperforms other alternatives, whereas in closed-source models, formats like *Graph Markup Language (GMaL)* demonstrate marked advantages. Such task-specific preferences for serialization formats further highlight the importance and significance of GRAPHOMNI.

**Finding ❺: Complex multi-step prompts can negatively impact the performance of closed-source models.** In the Triangle counting task, open-source models performed very well with more examples in *Instruct* and *k-shot* scenario, while closed-source models excelled using minimal prompting strategies such as *0-Algorithm*, which avoid elaborate reasoning steps or intermediate explicit guidance (Tables 17, 20 in Appendix E.1). This pattern suggests that complex or abstract multi-step prompts can confound closed-source models in certain challenging tasks.

E.7    ANALYSIS ON EFFICIENCY VIA NUMBER OF OUTPUT TOKENS

To assess inference efficiency, we measure the total number of output tokens each model produces—tokenized consistently with the OpenAI GPT-3.5-turbo tokenizer[‡]—and analyze how token counts vary across four key dimensions: difficulty levels (Table 23), task categories (Table 24), serialization formats (Table 25), and prompt schemes (Table 26). The average token counts under each condition are reported in these tables, together with the main results of accuracy, providing a comprehensive view of the trade-offs between output verbosity and model performance across our benchmark.

E.7.1    OVERALL ANALYSIS

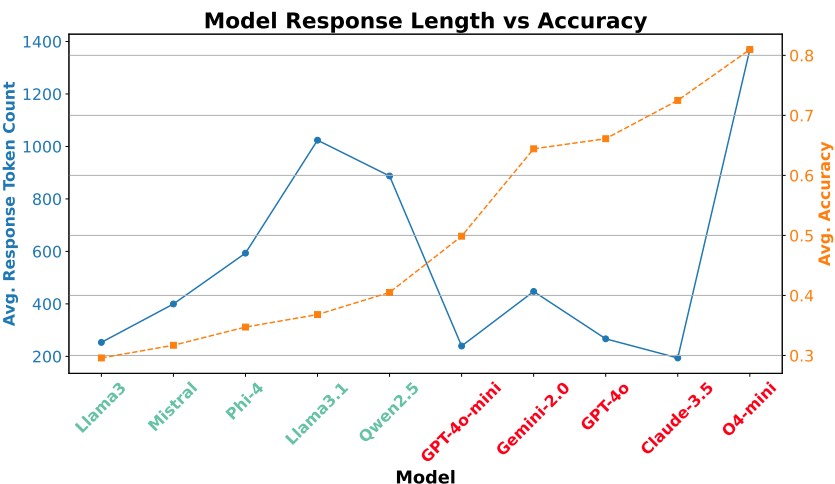

Figure 32: **Average output tokens versus overall accuracy across all graph-theoretic tasks**. Models are ordered by the average performance. Models in Green are open-source models while others in Red are closed-source ones.

Figure 32 highlights two distinct patterns. **Closed-source models**—GPT-4o, Claude-3.5, and Gemini-2.0—achieve the highest accuracy while keeping total output below roughly 300 tokens, showing tight control over generation length. **o4-mini**, a reasoning-focused model stands out: its final answers remain short (about 100 tokens), but it adds a lengthy chain-of-thought (up to 1.6 K tokens), yielding strong accuracy with markedly larger overall output. **Open-source models** display a different trend. Llama-3.1 and Qwen-2.5 match the best accuracies only when they generate much longer responses, whereas Llama-3 and Mistral remain shorter and less accurate. These contrasts persist across difficulty levels, task categories, serialization formats, and prompt schemes, as detailed in Tables 23–26.

E.7.2    ANALYSIS CONCERNING DIFFICULTY OF TASK

Table 23: **Average output tokens per model at each difficulty level (Easy, Medium, Hard)**. Orange / Blue / Light purple highlights indicate the largest/second-largest/third-largest number of output tokens.

| Difficulty | Llama-3 (8B) | Llama-3.1 (8B) | Mistral (7B) | Phi-4 (14B) | Qwen-2.5 (7B) | Claude-3.5 | GPT-4o | GPT-4o-mini | Gemini-2.0 | o4-mini Answer | o4-mini Reasoning | Average |
|---|---|---|---|---|---|---|---|---|---|---|---|---|
| easy | 210.51 | 1050.08 | 375.91 | 517.31 | 881.82 | 198.82 | 257.17 | 248.50 | 440.26 | 143.87 | 841.68 | 469.63 |
| hard | 292.46 | 994.80 | 419.36 | 644.85 | 873.01 | 182.71 | 263.71 | 217.44 | 411.54 | 70.25 | 1660.83 | 548.27 |
| medium | 267.18 | 1018.81 | 408.88 | 632.09 | 903.56 | 197.35 | 278.82 | 246.52 | 481.29 | 120.70 | 1367.97 | 538.47 |
| **Average** | 256.72 | 1021.23 | 401.38 | 598.08 | 886.13 | 192.96 | 266.57 | 237.49 | 444.37 | 111.61 | 1290.16 | - |

Table 23 presents the token output across different models under varying levels of task difficulty. Overall, most models exhibit small variation in output length as task difficulty increases. However, a

---

[‡]Note that the number of the reasoning of o4-mini is obtained from the metadata of each API call.

notable exception is the reasoning model, which demonstrates a distinct pattern: as task difficulty rises, the number of tokens in the final answer tends to decrease, while the length of the reasoning process correspondingly increases.

### E.7.3 ANALYSIS CONCERNING TASK TYPE

Table 24: **Average output tokens per model for each graph-theoretic task.** Orange / Blue / Light purple highlights indicate the largest/second-largest/third-largest number of output tokens.

| Task type | Llama-3 (8B) | Llama-3.1 (8B) | Mistral (7B) | Phi-4 (14B) | Qwen-2.5 (7B) | Claude-3.5 | GPT-4o | GPT-4o-mini | Gemini-2.0 | o4-mini Answer | o4-mini Reasoning | Average |
|---|---|---|---|---|---|---|---|---|---|---|---|---|
| BFS order | 288.58 | 897.44 | 394.37 | 582.88 | 859.08 | 243.48 | 511.06 | 435.15 | 638.90 | 269.69 | 1666.66 | 617.03 |
| Connectivity | 176.60 | 919.75 | 375.53 | 565.78 | 630.62 | 133.71 | 154.63 | 130.19 | 137.39 | 96.13 | 547.11 | 351.59 |
| Cycle detection | 266.47 | 828.07 | 362.82 | 477.68 | 735.21 | 176.49 | 132.92 | 125.41 | 118.81 | 100.38 | 564.12 | 353.49 |
| Diameter calculation | 269.48 | 878.37 | 391.86 | 502.67 | 792.25 | 236.41 | 259.02 | 285.96 | 513.27 | 81.00 | 1839.11 | 549.95 |
| Shortest path | 286.65 | 1839.88 | 525.77 | 946.46 | 1491.21 | 127.57 | 149.54 | 351.89 | 91.59 | 735.50 | 612.17 |
| Triangle counting | 231.02 | 768.90 | 347.19 | 489.85 | 802.21 | 246.97 | 424.23 | 295.00 | 925.66 | 80.55 | 2156.73 | 615.30 |
| Average | 253.13 | 1022.07 | 399.59 | 594.22 | 885.10 | 194.11 | 271.90 | 243.26 | 447.66 | 119.89 | 1251.54 | - |

Further insights can be drawn from Table 24, which reveals a clear correlation between output tokens and task type. Specifically, tasks such as Connectivity and Cycle detection consistently yield significantly shorter outputs compared to other tasks, as they are relatively easier compared to others. Among open-source models, the Shortest path task results in the longest outputs, whereas for closed-source models, the BFS order and Triangle counting task generate the highest average token counts . In the case of the reasoning model, the token output associated with the reasoning process increases markedly with the complexity and difficulty of the task—particularly when considering task accuracy. For instance, in the Triangle counting task, the reasoning component alone produces an average of over 2000 tokens, highlighting the model's tendency to elaborate more extensively as task complexity increases.

### E.7.4 ANALYSIS CONCERNING SERIALIZATION FORMATS

Table 25: **Average output tokens per model under different serialization formats.** Orange / Blue / Light purple highlights indicate the largest/second-largest/third-largest number of output tokens.

| Serialization format | Llama-3 (8B) | Llama-3.1 (8B) | Mistral (7B) | Phi-4 (14B) | Qwen-2.5 (7B) | Claude-3.5 | GPT-4o | GPT-4o-mini | Gemini-2.0 | o4-mini Answer | o4-mini Reasoning | Average |
|---|---|---|---|---|---|---|---|---|---|---|---|---|
| Adjacency List | 260.58 | 958.19 | 393.76 | 505.19 | 792.85 | 199.83 | 280.26 | 234.54 | 455.60 | 123.91 | 1118.90 | 483.96 |
| Adjacency Matrix | 288.42 | 897.05 | 400.13 | 555.10 | 862.35 | 198.44 | 291.62 | 243.48 | 536.80 | 105.28 | 1535.59 | 537.66 |
| Adjacency Set | 256.95 | 962.41 | 390.89 | 509.12 | 787.49 | 199.17 | 284.96 | 237.48 | 497.91 | 119.09 | 1144.19 | 489.97 |
| Edge List | 239.61 | 930.70 | 383.71 | 526.96 | 805.14 | 195.93 | 267.44 | 261.68 | 445.74 | 117.61 | 1260.25 | 494.07 |
| Edge Set | 246.07 | 914.69 | 405.22 | 511.70 | 823.50 | 203.51 | 285.23 | 269.78 | 497.43 | 114.47 | 1282.32 | 504.90 |
| Graph Modelling Language | 267.75 | 853.42 | 335.60 | 544.70 | 787.79 | 181.31 | 221.91 | 180.38 | 339.59 | 114.18 | 1238.94 | 460.51 |
| GraphML | 212.87 | 1650.01 | 487.76 | 1000.29 | 1351.85 | 179.31 | 235.97 | 248.29 | 358.25 | 114.39 | 1195.50 | 639.50 |
| Average | 253.18 | 1023.78 | 399.58 | 593.29 | 887.28 | 193.93 | 266.77 | 239.38 | 447.33 | 115.56 | 1253.67 | - |

Table 25 presents the influence of different graph serialization formats on the number of output tokens generated by various models. Overall, more complex formats—such as GMaL and Adjacency Matrix—tend to induce longer outputs, whereas simpler formats—such as Adjacency List and Edge List—are associated with significantly shorter outputs. Among the evaluated models, open-source models such as Llama-3.1 and Qwen-2.5 consistently produce a higher number of tokens across most formats. This effect is particularly pronounced for Llama-3.1 under the GMaL format, where its output length reaches a peak. In contrast, closed-source models generally yield more concise outputs, with Claude-3.5 being especially compact. An exception is observed in o4-mini, whose output length is substantially higher due to the inclusion of intermediate reasoning steps.

### E.7.5 ANALYSIS CONCERNING PROMPT SCHEMES

Table 26 further examines the impact of different prompting strategies on model output. Prompts that involve reasoning or instruction (e.g., CoT, Instruct, and 0-Instruct) significantly increase output length, a trend that is particularly salient in open-source models. For instance, under the 0-Instruct prompt, both Llama-3.1 and o4-mini produce extended outputs. In contrast, prompts with no

Table 26: **Average output tokens per model for each prompt scheme**. Orange / Blue / Light purple highlights indicate the largest/second-largest/third-largest number of output tokens.

| Prompt Scheme | Llama-3 (8B) | Llama-3.1 (8B) | Mistral (7B) | Phi-4 (14B) | Qwen-2.5 (7B) | Claude-3.5 | GPT-4o | GPT-4o-mini | Gemini-2.0 | o4-mini Answer | o4-mini Reasoning | Average |
|---|---|---|---|---|---|---|---|---|---|---|---|---|
| 0-Algorithm | 283.43 | 1071.04 | 484.71 | 434.10 | 953.42 | 206.22 | 305.95 | 237.04 | 684.95 | 123.07 | 1223.52 | 546.13 |
| 0-CoT | 308.61 | 1114.37 | 346.42 | 145.13 | 763.46 | 221.08 | 385.55 | 386.72 | 558.95 | 156.72 | 1223.45 | 510.04 |
| 0-Instruct | 308.80 | 1151.98 | 391.51 | 643.39 | 690.56 | 204.07 | 334.83 | 370.93 | 544.26 | 137.55 | 1229.01 | 546.08 |
| Algorithm | 202.15 | 964.36 | 456.40 | 815.85 | 946.63 | 215.10 | 245.75 | 225.96 | 349.02 | 120.56 | 1313.74 | 532.32 |
| CoT | 192.88 | 916.17 | 378.05 | 851.34 | 1042.81 | 164.95 | 154.63 | 203.33 | 253.13 | 92.06 | 1271.55 | 501.90 |
| Instruct | 212.82 | 987.20 | 355.99 | 917.42 | 1043.12 | 176.66 | 169.42 | 210.20 | 255.77 | 119.57 | 1302.98 | 522.83 |
| LTM | 303.64 | 1116.18 | 405.24 | 361.22 | 694.98 | 205.71 | 347.82 | 339.98 | 579.05 | 142.44 | 1239.87 | 521.47 |
| K-Shot | 160.98 | 760.10 | 440.39 | 1024.57 | 1009.45 | 170.52 | 196.37 | 39.37 | 270.99 | 67.35 | 1277.14 | 492.47 |
| 0-shot | 305.27 | 1133.08 | 337.44 | 146.16 | 841.05 | 181.03 | 260.59 | 140.92 | 529.71 | 80.78 | 1201.01 | 468.82 |
| **Average** | 253.18 | 1023.83 | 399.57 | 593.24 | 887.27 | 193.93 | 266.77 | 239.38 | 447.32 | 115.56 | 1253.59 | - |

instruction (0-shot) or few-shot examples (K-Shot) tend to yield shorter outputs. Closed-source models exhibit relatively stable output lengths across prompt types, suggesting stronger control over generation behavior.

### E.7.6 COST-ACCURACY TRADEOFF ANALYSIS

Table 27: Per-Query Inference Cost Analysis. Costs are calculated based on current API pricing (as of November 2025) with average input tokens of 933 per query. **Bold orange** / Underlined blue / Light purple highlights indicate lowest/second-lowest/third-lowest cost in each category.

| Model | Input Cost ($) | Output Cost ($) | Total Cost ($) |
|---|---|---|---|
| *Open-Source Models* | | | |
| Llama-3.1 (8B) | **0.000019** | 0.000031 | **0.000049** |
| Mistral (7B) | 0.000187 | 0.000080 | 0.000267 |
| Phi-4 (14B) | 0.000056 | 0.000084 | 0.000140 |
| Qwen-2.5 (7B) | 0.000037 | 0.000089 | 0.000126 |
| *Closed-Source Models* | | | |
| Claude-3.5 | 0.002799 | 0.002894 | 0.005694 |
| GPT-4o | 0.002333 | 0.002666 | 0.004998 |
| GPT-4o-mini | 0.000140 | **0.000142** | 0.000282 |
| Gemini-2.0 | 0.000093 | 0.000178 | 0.000271 |
| o4-mini | **0.001026** | 0.006168 | 0.007194 |

Table 27 presents per-query inference costs based on current API pricing. Cost varies by three orders of magnitude across models, ranging from $0.000049 (Llama-3/Llama-3.1) to $0.007194 (o4-mini) per query. Open-source models uniformly cost less than $0.0003 per query, while closed-source models span from $0.000271 (Gemini-2.0) to $0.007194 (o4-mini).

Figure 33 visualizes the cost-(mean) accuracy tradeoff on all tasks. o4-mini achieves the highest accuracy (80.96%) but incurs the highest cost. Notably, no model dominates across all metrics. The optimal choice depends on application requirements: open-source models for cost-sensitive deployments with relaxed accuracy constraints, Gemini-2.0 or GPT-4o-mini for balanced cost-performance, Claude-3.5 or GPT-4o for high-accuracy applications, and o4-mini when maximizing accuracy justifies premium costs. For full benchmark evaluation (241,726 queries), total costs range from $11.85 (Llama-3) to $1,739 (o4-mini), a 147× difference that has significant implications for large-scale graph reasoning deployments.

## F  DETAILED RELATED WORKS

As more and more semantic information is needed in graph-related applications (Zhu et al., 2025; Wu et al., 2022; Jian et al., 2025a; Wang & Zhuang, 2025; Zhu et al., 2026), adopting LLMs for

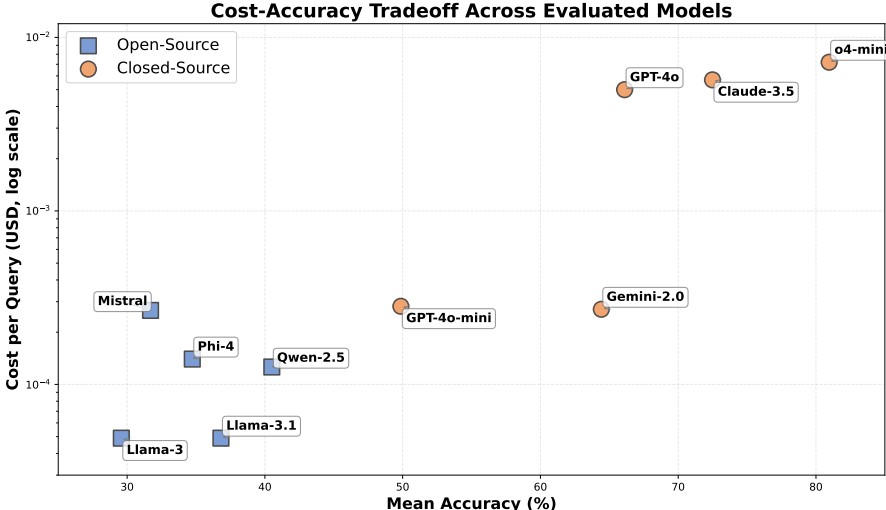

Figure 33: Cost-accuracy tradeoff across evaluated models on average. Each point represents a model's mean accuracy versus per-query inference cost (log scale).

graph-structured data has become increasingly important. Integrating LLMs with graph-structured data merges linguistic reasoning capabilities with structural representation insights. While comprehensive discussions on LLM-graph integration can be found in Appendix F.1, recent benchmarks specifically targeting LLM applications for graph reasoning, such as LLM4DyG (Zhang et al., 2024b), GraphTMI (Das et al., 2024), GraphInstruct (Luo et al., 2024b), and MAGMA (Taylor et al., 2024), have highlighted substantial progress and persistent limitations. These benchmarks reveal issues including narrow graph diversity, scalability constraints, and pronounced sensitivity to input formatting. Studies on graph pattern comprehension and multi-hop reasoning (Dai et al., 2024; Wang et al., 2023; Jin et al., 2024b) further emphasize brittleness under complex or noisy data conditions. Empirical analyses conducted by GPT4Graph (Guo et al., 2024a) and GraphWiz (Chen et al., 2024a) underscore performance gaps relative to specialized graph models and highlight computational inefficiencies. Additionally, recent contributions through transformer scaling studies (Sanford et al., 2024), comprehensive benchmarks like GraphFM (Xu et al., 2024), GLBench (Li et al., 2024d), GraphAbstract (Zhao et al., 2025), and GraphVista (Han et al., 2025), and specialized datasets (Yan et al., 2023; Fatemi et al., 2024) have provided valuable yet often limited insights. ProGraph (Li et al., 2024b) offers innovation but introduces extra computational overhead due to external dependencies.A detailed summary of these benchmark-related works is provided below.

## F.1 LLM APPLICATIONS ON GRAPH DATA

The intersection of LLMs and graph-structured data has emerged as an active research domain, combining the nuanced contextual reasoning abilities of LLMs with the structural representational power of traditional Graph Neural Networks (GNNs). Initial studies addressed fundamental challenges such as reducing sensitivity to prompt formulation (Sclar et al., 2024) and enabling zero-shot cross-dataset transferability (Li et al., 2024c). These foundational efforts have supported the development of generative models that jointly leverage textual and structural graph information, creating unified semantic embeddings for enhanced performance (Wang et al., 2024b; Fang et al., 2025b; Li et al., 2024a; Kong et al., 2024), which have much more powerful expressive ability compared to traditional self-supervised learning (Jian et al., 2025b).

Subsequent research built upon these foundations by focusing on enhancing the robustness of LLMs when applied to graph tasks (Guo et al., 2024b) and advancing techniques for effectively translating complex graph structures into natural language, notably through methods like graph-syntax trees (Zhao et al., 2023). Recent advancements have directly embedded graph reasoning capabilities within LLM architectures, significantly extending their application beyond purely textual domains (Hu et al., 2023). In this context, specific methodologies have been developed that embed graph learning

modules and leverage instruction tuning to improve alignment between structural data and LLM input modalities (Chai et al., 2023; Tang et al., 2024).

Parallel efforts have provided extensive overviews of the evolving field through comprehensive surveys (Li et al., 2023; Jin et al., 2024a; Zhao et al., 2026), highlighting foundational concepts such as Graph Foundation Models that employ dedicated graph vocabularies for effective cross-domain learning (Mao et al., 2024). Concurrently, advances in parameter-efficient encoding techniques, exemplified by GraphToken (Perozzi et al., 2024), and retrieval-augmented frameworks such as G-Retriever (He et al., 2024), have further refined the processing and utilization of graph structures. Moreover, assistant-based frameworks employing instruction-tuning strategies, including LLaGA (Chen et al., 2024b) and InstructGraph (Wang et al., 2024a), demonstrated significant potential for enabling LLMs to produce high-quality graph-structured outputs through preference-aligned interactions.

Complementing these directions, significant innovations have emerged within graph representation learning, exemplified by models like OpenGraph (Xia et al., 2024) and MuseGraph (Tan et al., 2024), which integrate scalable transformers, data augmentation, and graph-specific instruction tuning for robust zero-shot performance and general graph mining applications. Additional methods employing compact node identifiers (Luo et al., 2024a) and attributed random walks for fine-tuning (Tan et al., 2023) have notably improved inference efficiency, collectively illustrating a coherent evolution towards integrated frameworks that effectively harness the combined strengths of LLMs and graph-centric approaches.

## F.2    Benchmarks on LLM Application to Graph Data

Recent benchmarks assessing LLM capabilities on graph reasoning tasks have significantly advanced understanding yet still present important limitations. Benchmarks such as LLM4DyG (Zhang et al., 2024b), which emphasizes spatial-temporal dynamics, typically neglect the complexity inherent to static graph structures. Similarly, GraphTMI (Das et al., 2024), exploring various graph input modalities (text, motif, image), has exposed inherent trade-offs between token efficiency and representational expressiveness, potentially impacting scalability.

Other benchmarks, including GraphInstruct (Luo et al., 2024b) and MAGMA (Taylor et al., 2024), incorporate traditional graph reasoning tasks with explanatory strategies but remain limited by small-scale graph sizes and lack comprehensive coverage across diverse graph structures. Studies specifically targeting graph pattern recognition and natural-language-based graph problem-solving (Dai et al., 2024; Wang et al., 2023) have further revealed pronounced sensitivity to input formats, resulting in brittleness under complex or noisy conditions. Additionally, frameworks designed to mitigate multi-hop reasoning inaccuracies through graph-centric reasoning chains (Jin et al., 2024b) and examinations of generalization beyond memorized patterns (Zhang et al., 2024a) continue to illustrate significant unresolved challenges.

Empirical assessments conducted by initiatives such as GPT4Graph (Guo et al., 2024a) and instruction-tuned benchmarks like GraphWiz (Chen et al., 2024a) highlight persistent performance gaps compared to specialized graph neural architectures, accompanied by elevated computational demands. More recent contributions, including scaling analyses of transformer models (Sanford et al., 2024), comprehensive benchmarks like GraphFM (Xu et al., 2024) and GLBench (Li et al., 2024d), and specialized datasets such as CS-TAG (Yan et al., 2023) and encoding studies (Fatemi et al., 2024), have substantially enriched the literature but remain constrained by challenges related to homogeneity, training inefficiencies, and limited scalability. While innovative, solutions such as ProGraph (Li et al., 2024b), employing programming-based integration and external API retrieval, introduce additional computational overhead and dependencies.

## G    Limitations and Future Directions of GraphOmni

While GraphOmni significantly advances the evaluation of large language models (LLMs) on graph-theoretic tasks, several considerations highlight opportunities for future enhancement:

- **Diversity of Tasks**: The benchmark presently includes key canonical tasks, which may not fully represent the diversity of graph-related problems encountered in practice. Expand-

ing the task set to include dynamic, temporal, or heterogeneous graph challenges could offer deeper insights into model performance. Future work should focus on defining and integrating tasks that capture evolving network structures and multi-relational data.

- **Generalizability of Findings**: GRAPHOMNI evaluates LLMs under controlled experimental conditions, which might not entirely reflect performance in less structured, real-world environments. Future work could include testing the generalizability of models across various practical conditions, such as noisy data, incomplete graphs, or domain-specific variations, to better understand the robustness and applicability of LLMs.

Addressing these aspects will further enhance the robustness, applicability, and inclusivity of GRAPHOMNI, fostering wider adoption and deeper insights into LLM performance.

# H ADDITIONAL ABLATION STUDIES

## H.1 PERFORMANCE VS. TIME COMPLEXITY OF TASKS

### H.1.1 TIME COMPLEXITY ANALYSIS

The time complexities are determined based on well-established algorithms in graph theory (we are aware that more efficient algorithms are available, especially for `Diameter calculation` and `Triangle counting`, but we use the most naive implementations since they typically reflect how LLMs approach these tasks):

- `Connectivity`: $O(V + E)$ — Determined via a single breadth-first search (BFS) or depth-first search (DFS) traversal starting from one node to check reachability to another node.
- `Cycle detection`: $O(V + E)$ — Implemented using DFS with back-edge detection; each node and edge is visited at most once.
- `BFS order`: $O(V + E)$ — Standard breadth-first traversal visits each node once and examines each edge once.
- `Shortest path`: $O(V + E)$ for unweighted graphs using BFS, or $O(E + V \log V)$ for weighted graphs using Dijkstra's algorithm. Since our benchmark uses unweighted graphs, we report $O(V + E)$.
- `Diameter calculation`: $O(V(V + E))$ — Requires computing all-pairs shortest paths, typically achieved by running BFS from each node, resulting in $O(V)$ BFS operations each costing $O(V + E)$.
- `Triangle counting`: $O(V^3)$ naively by checking all triplets of nodes, or $O(V \cdot d_{\text{avg}}^2)$ with neighbor-based enumeration where $d_{\text{avg}}$ is the average degree. For dense graphs or without optimizations, this remains the most computationally intensive task.

### H.1.2 ALIGNMENT ANALYSIS

Tables 28, 29, and 30 demonstrate partial alignment between computational complexity and LLM difficulty. At the extremes, correspondence is clear: `Triangle counting` ($O(V^3)$) achieves only 15.45% accuracy (closed-source, Hard) and 6.77% (open-source, Hard), while `Connectivity` ($O(V + E)$) reaches 91.90% and 75.97% respectively. Similarly, `Diameter calculation` ($O(V(V + E))$) yields 40.09% (closed-source) and 21.33% (open-source), ranking as the second-hardest task both algorithmically and empirically.

However, among tasks with identical $O(V + E)$ complexity, performance diverges substantially. `Connectivity` maintains 91.90% accuracy on hard instances, while `BFS order` collapses to 27.15%, a 64.75 percentage point gap despite equivalent asymptotic complexity. This divergence indicates that computational complexity alone does not determine LLM difficulty.

### H.1.3 FACTORS BEYOND COMPUTATIONAL COMPLEXITY

Three task characteristics account for this divergence. First, **output structure** critically impacts performance: binary decisions (`Connectivity`, `Cycle detection`) achieve 91.90% and 79.24%,

while sequence generation (BFS order) and numerical enumeration (Triangle counting, Diameter calculation) fall to 27.15%, 15.45%, and 40.09% respectively. Second, **error propagation** varies by task type—sequence tasks suffer cascading failures where single errors invalidate entire outputs, as evidenced by BFS order's severe 62.52% performance drop. Third, **reasoning scope** distinguishes task difficulty: local reasoning tasks (Connectivity, Cycle detection) degrade minimally (4.31%, 2.74%), while global reasoning tasks requiring complete graph traversal (Diameter calculation, BFS order) drop sharply (41.34%, 62.52%).

Table 30 quantifies these effects: open-source models degrade 16.93% on average from Easy to Hard, while closed-source models drop 26.26%. Crucially, this degradation correlates more strongly with reasoning scope and output structure than with algorithmic complexity—BFS order ($O(V + E)$) degrades more severely than Diameter calculation ($O(V(V + E))$), demonstrating that maintaining sequential dependencies in textual representations poses greater challenges than computational intensity per se.

### H.1.4 CONCLUSION

Our analysis reveals that computational complexity establishes a baseline for LLM difficulty, as evidenced by Triangle counting and Diameter calculation ranking as both algorithmically expensive and empirically challenging. However, output structure and reasoning scope play equally critical roles. The 64.75 percentage point gap between Connectivity and BFS order—both $O(V + E)$ tasks—demonstrates that LLMs struggle disproportionately with maintaining long-range sequential dependencies, performing combinatorial enumeration, and generating outputs under strict ordering constraints. These limitations manifest independently of algorithmic complexity and persist across all evaluated models (Tables 28–30), indicating fundamental constraints in how current LLM architectures encode and manipulate graph-structured information through natural language representations.

Table 28: Open-Source LLM Performance Across Tasks Ranked by Computational Complexity (Mean Accuracy %). **Bold orange** / Underlined blue / Light purple highlights indicate best/second-best/third-best performance in each difficulty level.

| Task | Time Complexity | Easy (5–10 nodes) | | | | | | Hard (20–30 nodes) | | | | | |
|---|---|---|---|---|---|---|---|---|---|---|---|---|---|
| | | Llama-3.1 | Mistral | Phi-4 | Qwen-2.5-72B | Qwen-2.5-7B | Qwen-3 | Llama-3.1 | Mistral | Phi-4 | Qwen-2.5-72B | Qwen-2.5-7B | Qwen-3 |
| Triangle | $O(V^3)$ | 14.97 | 11.87 | 12.88 | 36.57 | 18.56 | **41.36** | 4.95 | 2.55 | 4.38 | 4.73 | 4.45 | **19.54** |
| Diameter | $O(V(V+E))$ | 41.27 | 28.55 | 42.81 | **78.50** | 45.08 | 77.56 | 18.63 | 6.97 | 17.71 | 29.59 | 15.27 | **39.83** |
| BFS order | $O(V+E)$ | 18.69 | 13.75 | 33.03 | **71.41** | 21.46 | 65.87 | 0.63 | 0.34 | 2.65 | 22.03 | 1.38 | **29.53** |
| Shortest path | $O(V+E)$ | 38.75 | 31.18 | 42.61 | **90.03** | 47.46 | 77.69 | 23.03 | 12.21 | 26.60 | 72.53 | 28.31 | **64.28** |
| Cycle | $O(V+E)$ | 55.49 | 55.44 | 45.25 | 74.02 | 62.19 | **90.30** | 52.40 | 51.64 | 40.64 | 68.40 | 58.88 | **86.81** |
| Connectivity | $O(V+E)$ | 79.53 | 79.90 | 56.29 | 90.24 | 88.10 | **97.17** | 74.58 | 74.77 | 48.39 | 84.09 | 81.19 | **92.89** |

Table 29: Closed-Source LLM Performance Across Tasks Ranked by Computational Complexity (Mean Accuracy %). **Bold orange** / Underlined blue / Light purple highlights indicate best/second-best/third-best performance in each difficulty level.

| Task | Time Complexity | Easy (5–10 nodes) | | | | Hard (20–30 nodes) | | | |
|---|---|---|---|---|---|---|---|---|---|
| | | Claude-3.5 | GPT-4o | Gemini-2.0 | o4-mini | Claude-3.5 | GPT-4o | Gemini-2.0 | o4-mini |
| Triangle | $O(V^3)$ | 43.41 | 36.32 | 50.33 | **84.54** | **15.92** | 12.81 | 15.55 | 17.53 |
| Diameter | $O(V(V+E))$ | 83.71 | 63.99 | 79.14 | **98.88** | **56.70** | 45.60 | 23.45 | 34.61 |
| BFS order | $O(V+E)$ | 91.42 | 81.48 | 90.31 | **95.46** | 26.80 | 21.58 | 27.77 | **32.45** |
| Shortest path | $O(V+E)$ | 94.35 | 92.17 | 81.75 | **95.08** | 87.88 | 74.98 | 78.16 | **88.63** |
| Cycle | $O(V+E)$ | 82.56 | 85.08 | 62.30 | **97.97** | 80.10 | 82.96 | 58.30 | **95.61** |
| Connectivity | $O(V+E)$ | **98.38** | 95.63 | 92.61 | 98.23 | **96.99** | 90.59 | 87.99 | 92.02 |

### H.2 SCALING BEYOND 50 NODES

To address scale concerns, we extend evaluation to 50–100 node graphs on representative models (Qwen-2.5-72B and o4-mini). Table 31 compares performance against the 20–30 node Hard split.

Performance degrades uniformly as graph size increases, but the fundamental patterns remain unchanged. Task difficulty ranking stays identical: Triangle counting and BFS order remain hardest, while Connectivity and Cycle detection remain most stable. Relative model performance gaps persist at similar magnitudes across scales. Critically, no new failure modes emerge, i.e.,

Table 30: Aggregate Performance Comparison by Model Category and Task Complexity with Performance Degradation. Accuracy (%) with color intensity indicating performance level. $\Delta$ shows Easy→Hard performance drop.

| Task | Time Complexity | Open-Source | | | Closed-Source | | |
|---|---|---|---|---|---|---|---|
| | | Easy | Hard | $\Delta$ | Easy | Hard | $\Delta$ |
| Triangle | $O(V^3)$ | 22.70 | 6.77 | $-15.93$ | 53.65 | 15.45 | $-38.20$ |
| Diameter | $O(V(V+E))$ | 52.30 | 21.33 | $-30.97$ | 81.43 | 40.09 | $-41.34$ |
| BFS order | $O(V+E)$ | 37.37 | 9.43 | $-27.94$ | 89.67 | 27.15 | $-62.52$ |
| Shortest path | $O(V+E)$ | 54.62 | 37.83 | $-16.79$ | 90.84 | 82.41 | $-8.43$ |
| Cycle | $O(V+E)$ | 63.78 | 59.76 | $-4.02$ | 81.98 | 79.24 | $-2.74$ |
| Connectivity | $O(V+E)$ | 81.87 | 75.97 | $-5.90$ | 96.21 | 91.90 | $-4.31$ |
| **Mean** | | **52.11** | **35.18** | **$-16.93$** | **82.30** | **56.04** | **$-26.26$** |

the same challenges identified in smaller graphs (combinatorial enumeration, sequential dependencies, serialization sensitivity) simply intensify.

These results confirm that our 5–30 node design captures the essential reasoning challenges. Larger graphs amplify these challenges quantitatively but reveal no new qualitative phenomena, validating our focus on controlled-scale evaluation where reasoning capability, rather than resource constraints, determines performance.

Table 31: Results on 50–100 node graphs (EEH = Extremely Extra Hard). Results on the 20–30 node Hard split are shown in parentheses for comparison. **Bold orange** / Underlined blue highlights indicate best/second-best performance.

| Task | Difficulty | Open-source Model Qwen-2.5 (72B) | Closed-source Model o4-mini |
|---|---|---|---|
| BFS order | EEH | 8.19±2.03 (22.03) | **10.23±2.07** (32.45) |
| Connectivity | EEH | 62.00±4.90 (84.09) | **81.86±8.24** (92.02) |
| Cycle | EEH | 37.78±4.11 (68.40) | **74.81±4.90** (95.61) |
| Diameter | EEH | 8.89±2.39 (29.59) | **40.44±3.76** (34.61) |
| Shortest path | EEH | 33.28±6.09 (72.53) | **68.51±11.04** (88.63) |
| Triangle | EEH | 2.36±0.67 (4.73) | **2.85±0.71** (17.53) |

### H.3 ROBUSTNESS CHECK UNDER PROMPT NOISE (PERTURBATION)

To address concerns about robustness to natural language variation, we conduct a supplementary evaluation examining model sensitivity to paraphrased prompts. In our main evaluation, we deliberately use deterministic phrasing within each prompt scheme to isolate the effects of our three core dimensions, i.e., graph types, serialization formats, and prompt schemes, without confounding factors from linguistic variation. This controlled design allows us to systematically attribute performance differences to structural representation choices (serialization formats) and reasoning guidance strategies (prompt schemes) rather than to incidental phrasing variations. However, real-world applications inevitably encounter diverse linguistic expressions of the same semantic content, and robustness to such variation is a practical concern. We therefore design a controlled perturbation framework to assess whether our conclusions remain stable under realistic linguistic variation.

#### H.3.1 DESIGN OF THE STUDY

**Task and Sample Selection.** We choose to conduct this robustness analysis on BFS order. This choice is motivated by three considerations: (1) it is among the most challenging tasks in our benchmark, exhibiting substantial performance gaps across models and difficulty levels; (2) it

requires complex structured output (a full node ordering), making it potentially more sensitive to prompt variations that might affect the model's understanding of output format requirements; and (3) given limited time and budget constraints, concentrating on a single representative hard task allows for deeper analysis. From the full `BFS order` dataset, we subsample 4,000 instances to balance coverage across graph types, serialization formats, prompt schemes, and difficulty levels.

**Perturbation Design.**    Our perturbation framework defines *noisy prompts* as semantically equivalent (so it is still a problem with the same answer) but syntactically diverse variants of the original prompts. And they are generated through systematic paraphrasing of natural-language components while maintaining the absolute structural preservation of graph data. The design adheres to three core principles:

1. *Semantic Equivalence*: All perturbations preserve the semantic content and task requirements through lexical substitution, syntactic restructuring, and stylistic variation. So it is designed to test linguistic invariance.

2. *Structural Preservation*: Graph representations remain character-for-character identical across all perturbations. This ensures that performance variation reflects model sensitivity to linguistic expression rather than changes in the underlying graph structure. In this way, the nature of the problem does not change much, and the ground truth results will still be the same.

3. *Comprehensive Coverage*: Perturbations span all nine prompt types in our framework (Algorithm, CoT, k-shot, Instruct, LTM, and their variants) and all seven serialization formats (Adjacency Matrix, Adjacency List, Adjacency Set, Edge List, Edge Set, GMoL, GMaL).

**Perturbation Methodology.**    We construct task-specific variation pools for each perturbable component. **For prompts containing algorithmic explanations**, we develop multiple human-authored paraphrases that express the same procedural steps using different vocabulary, sentence structures, and explanatory styles. Figure 34 illustrates a representative example: the original formal description uses a numbered list structure with technical terminology ("Initialize","enqueue","dequeue","Mark visited"), while the perturbed version adopts a conversational flow with colloquial alternatives ("First", "pick your", "put it in, "take out", "Mark it as visited so we don't check it again"). The transformation achieve 47.9% word-level change while maintaining algorithmic correctness and semantic equivalence.

**For few-shot answer components**, we generate variations that maintain identical logical reasoning and final answers while modifying transitional phrases and technical terminology. Figure 35 demonstrates this approach: the original example uses formal procedural terms ("Dequeue", "neighbors", "enqueue", "visited") that are systematically replaced with more natural alternatives ("Extract from queue", "neighboring nodes", "insert into queue", "seen"). This achieves 17.5% word-level change through 29 replacements with 82.5% similarity, preserving the reasoning structure while varying linguistic expression. For instructional components, we create alternatives for opening statements, reasoning indicators, and procedural connectives. For task descriptions in minimal prompts, we paraphrase the task specification itself.

The perturbation process employs delimiter-based component extraction to precisely identify natural language elements while avoiding graph data. Specifically, we identify boundaries between natural language answers and graph representations (e.g., "And the graph representation of [format] is") to ensure that variations are applied exclusively to linguistic content. For each prompt, we randomly sample variations from component-specific pools matched to the prompt's (`prompt scheme`, `serialization format`) combination, apply targeted string replacement using bounded pattern matching, and verify post-perturbation that all graph representations remain unchanged through format-specific validation procedures.

**Quality Assurance.**    To guarantee structural preservation, we implement multi-level verification: format-specific validation for each of the seven serialization types (e.g., character-level comparison of matrix blocks, structural validation for GML/GraphML, exact content matching for list and set formats), automated testing on representative samples spanning all prompt-serialization combinations, and per-instance validation confirming preservation before evaluation. Our implementation achieves

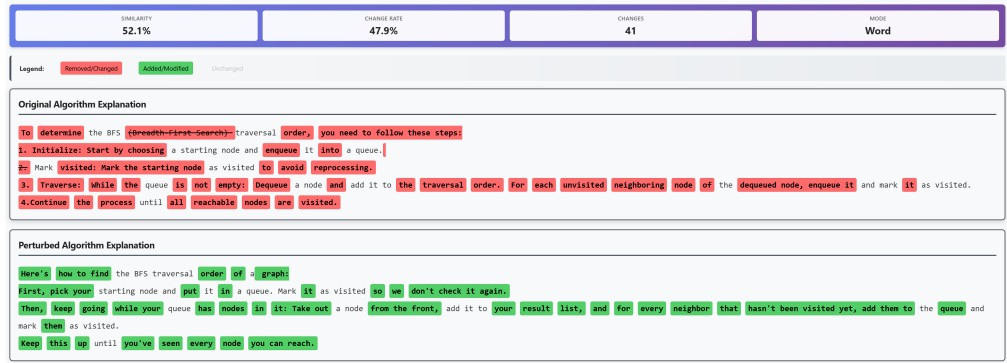

Figure 34: Example of algorithm explanation perturbation. The original formal, numbered description (left) is transformed into conversational phrasing (right) while preserving algorithmic correctness. Highlighted changes show systematic replacement of technical terms with colloquial alternatives. Word-level changes: 47.9%.

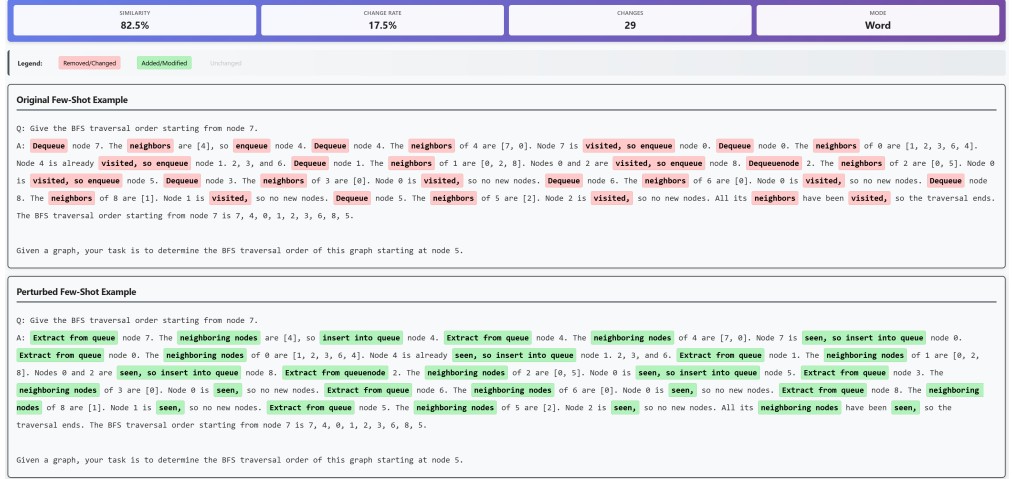

Figure 35: Example of few-shot answer perturbation. The original formal reasoning (left) is para-phrased with natural language variation (right) while maintaining identical logical structure and final answer. Color-coded highlights show systematic terminology replacement (e.g., "Dequeue" → "Extract from queue","visited" → "seen"). Word-level changes: 17.5%.

100% graph structure preservation across all perturbations while successfully modifying 87.9% of prompts of the samples (with the remaining 12.1% representing cases where random sampling selects the original phrasing).

**Summary.** This framework enables systematic evaluation of whether model performance and our main conclusions remain stable under realistic linguistic variation, providing evidence for the robustness of our findings beyond the specific phrasings used in the primary benchmark.

### H.3.2 EXPERIMENTAL RESULTS AND ANALYSIS

**Experimental Setting.** We evaluate two representative models from our main benchmark: **o4-mini** (top-performing closed-source reasoning model) and **Qwen-2.5-72B** (strongest open-source model). These models provide coverage of both closed-source and open-source categories and exhibited the highest performance in our main evaluation. We report results aggregated across prompt schemes and serialization formats separately, as well as fine-grained breakdowns per model, to assess whether our main conclusions about representation sensitivity remain stable under linguistic perturbation.

**Overall Results.** Tables 32 and 33 present results averaged across both models. Several key patterns emerge:

*Preservation of relative performance patterns.* The relative rankings of prompt schemes and serialization formats remain largely stable between original and perturbed conditions. For prompt schemes (Table 32), Algorithm, CoT, and Instruct consistently rank among the top three performers in Easy mode under both conditions, while 0-Shot maintains strong performance in Medium and Hard modes. For serialization formats (Table 33), AL and AS consistently dominate across all difficulty levels in both original and perturbed settings, with AL achieving 92.26% to 93.41% (Easy), 83.44% to 83.56% (Medium), and 48.27% to 50.33% (Hard). The persistence of these rankings confirms that our main finding holds under linguistic variation, as no single configuration works universally, but certain formats consistently outperform others.

*Evidence of real perturbation effects.* While relative patterns are preserved, absolute performance values shift measurably between conditions. For example, CoT improves from 85.26% to 90.98% in Easy mode, while K-Shot shows variation from 80.48% to 78.36%. These changes confirm that our perturbations introduce meaningful variation rather than being trivial paraphrases. We note that performance differences may be partially attributable to the subsampling from the full dataset to 4,000 instances, though the consistency of relative patterns suggests this effect is limited.

| Task | Difficulty | 0-Algorithm | 0-CoT | 0-Instruct | 0-Shot | Algorithm | CoT | Instruct | K-Shot | LTM |
|------|-----------|-------------|-------|------------|--------|-----------|-----|----------|--------|-----|
| Original | E | 84.71±8.99 | 78.36±10.06 | 82.32±9.07 | 83.22±9.08 | 85.82±7.12 | 85.26±7.45 | 86.86±6.13 | 80.48±8.52 | 83.88±8.58 |
| | M | 65.82±12.78 | 64.46±12.61 | 64.97±11.99 | 66.90±13.59 | 66.29±10.15 | 61.97±11.01 | 62.31±9.35 | 54.18±14.67 | 65.48±11.94 |
| | H | 32.22±10.81 | 31.87±9.64 | 29.42±9.90 | 31.69±10.66 | 26.93±8.37 | 22.67±6.51 | 20.90±6.63 | 20.26±8.55 | 29.18±10.15 |
| Perturbed | E | 84.98±14.09 | 75.50±14.08 | 85.54±9.64 | 79.13±15.83 | 85.82±9.49 | 90.98±7.83 | 78.21±16.30 | 78.36±10.79 | 87.59±10.59 |
| | M | 70.78±16.77 | 70.58±11.78 | 60.24±18.49 | 76.93±13.23 | 66.22±14.67 | 58.44±10.42 | 70.90±11.63 | 54.65±16.17 | 68.73±14.16 |
| | H | 34.26±19.27 | 28.38±12.29 | 27.46±14.77 | 41.06±15.69 | 26.93±12.75 | 22.38±9.41 | 18.75±7.55 | 21.27±12.25 | 32.37±12.90 |

Table 32: Performance of Prompt Schemes with perturbed prompt (Mean±95% CI Margin of All Models). Averaged over **o4mini** and **Qwen-2.5-72B**. **Bold orange** / Underlined blue / Light purple highlights indicate best/second-best/third-best performance in each difficulty level.

| Task | Difficulty | AL | AM | AS | EL | ES | GMaL | GMoL |
|------|-----------|----|----|----|----|----|------|------|
| Original | E | 92.26±3.71 | 75.24±10.67 | 91.88±3.76 | 82.28±7.11 | 82.17±6.40 | 85.68±4.64 | 74.51±8.09 |
| | M | 83.44±5.82 | 46.03±13.79 | 79.44±6.32 | 59.84±7.94 | 53.39±7.86 | 67.04±6.66 | 56.01±10.77 |
| | H | 48.27±6.14 | 7.46±2.72 | 48.23±6.57 | 20.74±3.70 | 15.56±2.76 | 26.95±3.29 | 23.46±5.37 |
| Perturbed | E | 93.41±5.30 | 74.08±15.27 | 87.40±7.19 | 80.64±15.75 | 81.91±8.49 | 85.60±8.14 | 77.67±12.33 |
| | M | 83.56±11.24 | 52.54±14.98 | 83.93±7.28 | 61.40±11.35 | 57.99±11.80 | 62.85±10.07 | 60.51±14.09 |
| | H | 50.33±10.84 | 6.97±6.66 | 50.02±13.24 | 22.52±8.83 | 24.93±11.77 | 19.24±6.17 | 22.38±8.42 |

Table 33: Performance of Serialization Formats with perturbed prompt (Mean±95% CI Margin of All Models). Averaged over **o4-mini** and **Qwen-2.5-72B**. **Bold orange** / Underlined blue / Light purple highlights indicate best/second-best/third-best performance in each difficulty level.

**Fine-Grained Results.** Tables 34–37 break down results per model, revealing differential robustness characteristics:

*o4-mini exhibits high robustness.* The closed-source reasoning model shows remarkable stability across perturbations (Tables 34 and 35). For serialization formats, AL maintains 98.54% to 97.89% (Easy), 91.75% to 92.20% (Medium), and 54.24% to 56.08% (Hard), with minimal changes in ranking. For prompt schemes, the relative ordering remains nearly identical between conditions, with only minor absolute shifts (e.g., 0-Algorithm improves from 95.84% to 97.64% in Easy mode). This stability suggests that o4-mini's reasoning capabilities are relatively invariant to surface-level linguistic variation, consistent with its design for robust multi-step reasoning.

*Qwen-2.5-72B shows greater sensitivity.* The open-source model exhibits larger absolute performance shifts and wider confidence intervals under perturbation (Tables 36 and 37). For example, in serialization formats, performance on AS varies from 86.68% to 77.64% (Easy) and 47.74% to 52.59% (Hard), with substantially increased variance (e.g., Hard mode: 10.30 to 27.65). Similarly,

prompt scheme performance shows notable fluctuation (e.g., CoT: 75.10% to 85.22% in Easy, 47.82% to 46.85% in Medium). However, crucially, the *relative rankings* remain consistent: AL and AS continue to outperform other serializations, and Algorithm/CoT/Instruct remain competitive prompt schemes. This indicates that while open-source models may be more sensitive to phrasing variations, our comparative conclusions about which representations work better are robust.

| Task | Difficulty | 0-Algorithm | 0-CoT | 0-Instruct | 0-Shot | Algorithm | CoT | Instruct | K-Shot | LTM |
|------|-----------|-------------|-------|------------|--------|-----------|-----|----------|--------|-----|
| Original | E | 95.84±2.29 | 94.66±3.43 | 96.53±1.96 | 94.66±2.76 | 97.02±2.05 | 95.42±1.58 | 96.32±2.16 | 93.90±2.26 | 94.80±2.51 |
| | M | 83.06±6.30 | 80.88±5.51 | 80.34±7.04 | 84.08±5.94 | 77.28±5.58 | 76.12±6.47 | 73.74±6.33 | 79.73±6.15 | 79.12±6.32 |
| | H | 37.88±12.33 | 38.07±11.51 | 35.77±13.43 | 39.31±14.47 | 26.98±10.81 | 26.26±7.98 | 21.48±7.40 | 31.85±11.55 | 34.45±12.51 |
| Perturbed | E | 97.64±1.76 | 95.54±2.69 | 97.39±1.95 | 96.36±3.85 | 95.20±4.17 | 95.92±2.65 | 94.71±3.32 | 93.37±2.53 | 95.48±2.82 |
| | M | 89.42±4.41 | 81.41±4.49 | 75.27±8.19 | 86.36±7.86 | 74.24±8.63 | 68.37±11.29 | 71.33±9.88 | 78.26±6.93 | 78.76±9.51 |
| | H | 41.10±19.67 | 33.05±16.53 | 33.75±16.26 | 50.55±15.98 | 27.23±13.65 | 24.28±10.70 | 27.30±6.95 | 35.71±12.20 | 34.86±17.31 |

Table 34: Performance of Prompt Schemes with perturbed prompt (Mean±95% CI Margin of All Models) on **o4-mini**. **Bold orange** / Underlined blue / Light purple highlights indicate best/second-best/third-best performance in each difficulty level.

| Task | Difficulty | AL | AM | AS | EL | ES | GMaL | GMoL |
|------|-----------|-----|-----|-----|-----|-----|------|------|
| Original | E | 98.54±0.63 | 96.06±1.47 | 97.09±1.40 | 95.74±1.14 | 95.25±1.00 | 94.71±1.51 | 90.83±2.72 |
| | M | 91.75±1.49 | 74.71±1.86 | 87.20±4.62 | 74.92±2.89 | 69.10±2.66 | 79.63±3.63 | 78.31±1.97 |
| | H | 54.24±5.95 | 12.83±1.73 | 48.72±8.78 | 27.49±2.78 | 19.01±3.80 | 31.11±4.72 | 33.74±4.02 |
| Perturbed | E | 97.89±1.76 | 96.99±2.15 | 97.17±2.64 | 95.29±2.12 | 95.20±2.90 | 94.20±2.81 | 93.39±2.76 |
| | M | 92.20±2.48 | 73.15±5.68 | 87.22±7.43 | 76.96±6.46 | 67.16±7.90 | 72.84±6.68 | 77.57±5.61 |
| | H | 56.08±10.23 | 13.51±12.75 | 47.74±7.90 | 31.36±12.84 | 28.68±9.58 | 24.25±7.84 | 33.20±11.99 |

Table 35: Performance of Serialization Formats with perturbed prompt (Mean±95% CI Margin of All Models) on **o4-mini**. **Bold orange** / Underlined blue / Light purple highlights indicate best/second-best/third-best performance in each difficulty level.

| Task | Difficulty | 0-Algorithm | 0-CoT | 0-Instruct | 0-Shot | Algorithm | CoT | Instruct | K-Shot | LTM |
|------|-----------|-------------|-------|------------|--------|-----------|-----|----------|--------|-----|
| Original | E | 73.58±13.66 | 62.07±9.30 | 68.10±9.70 | 71.78±13.48 | 74.62±7.41 | 75.10±10.29 | 77.39±6.59 | 67.06±8.88 | 72.95±12.65 |
| | M | 48.57±16.95 | 48.03±17.71 | 49.59±16.46 | 49.73±19.69 | 55.31±16.14 | 47.82±15.08 | 50.88±13.11 | 28.64±7.72 | 51.84±18.43 |
| | H | 26.56±17.71 | 25.67±14.87 | 23.07±13.88 | 24.07±14.43 | 26.88±13.67 | 19.08±10.17 | 20.32±11.63 | 8.68±3.39 | 23.92±15.94 |
| Perturbed | E | 72.33±25.53 | 52.11±15.51 | 71.71±14.31 | 61.91±26.30 | 72.70±16.55 | 85.22±16.18 | 55.11±29.20 | 60.84±12.59 | 78.39±21.24 |
| | M | 52.14±27.47 | 57.94±21.58 | 42.71±35.28 | 65.93±25.51 | 58.20±27.85 | 46.85±13.95 | 70.40±23.79 | 27.10±14.46 | 58.70±25.48 |
| | H | 28.39±32.49 | 23.71±18.80 | 20.12±26.21 | 31.56±26.38 | 26.62±22.75 | 20.16±17.12 | 11.43±10.14 | 1.05±2.06 | 29.87±20.35 |

Table 36: Performance of Prompt Schemes with perturbed prompt (Mean±95% CI Margin of All Models) on **Qwen-2.5-72B**. **Bold orange** / Underlined blue / Light purple highlights indicate best/second-best/third-best performance in each difficulty level.

**Summary.** Our robustness analysis demonstrates that the main conclusions of GRAPHOMNI remain stable under realistic linguistic perturbation. While absolute performance values shift measurably, confirming that perturbations introduce real variation rather than trivial paraphrases, the relative performance patterns across prompt schemes and serialization formats are preserved. Specifically, the finding that no single configuration works universally, but that certain serialization-prompt combinations consistently outperform others, holds across both original and perturbed conditions. The differential sensitivity between models (o4-mini showing higher robustness than Qwen-2.5-72B) provides an additional dimension for understanding model capabilities. These results validate the reliability of our benchmark findings while highlighting that prompt perturbation represents a valid and interesting dimension for future investigation. Importantly, our extensible framework design readily accommodates such extensions: future work could systematically incorporate perturbation as an additional evaluation axis alongside graph types, serialization formats, and prompt schemes, enabling deeper exploration of linguistic robustness in graph reasoning tasks.

| Task | Difficulty | AL | AM | AS | EL | ES | GMaL | GMoL |
|------|-----------|-----|-----|-----|-----|-----|------|------|
| Original | E | 85.98±4.51 | 54.42±8.08 | **86.68±5.68** | 68.82±6.28 | 69.09±2.97 | 76.65±3.28 | 58.20±3.92 |
| | M | **75.13±8.68** | 17.35±3.83 | 71.69±9.52 | 44.76±6.41 | 37.67±4.27 | 54.44±4.82 | 33.70±3.43 |
| | H | 42.30±9.52 | 2.08±0.76 | **47.74±10.30** | 13.99±2.60 | 12.10±2.53 | 22.80±2.67 | 13.17±2.23 |
| Perturbed | E | **87.65±10.82** | 51.17±21.97 | 77.64±11.01 | 47.66±34.42 | 68.62±11.31 | 75.94±14.67 | 59.99±20.12 |
| | M | 74.92±21.42 | 29.36±22.16 | **80.64±12.64** | 45.83±16.54 | 47.68±22.05 | 50.00±17.76 | 38.57±23.07 |
| | H | 43.85±19.81 | 1.25±2.45 | **52.59±27.65** | 13.67±9.54 | 21.18±21.98 | 13.59±8.54 | 11.57±6.71 |

Table 37: Performance of Serialization Formats with perturbed prompt (Mean±95% CI Margin of All Models) on **Qwen-2.5-72B**. **Bold orange** / Underlined blue / Light purple highlights indicate best/second-best/third-best performance in each difficulty level.

THE USE OF LARGE LANGUAGE MODELS

We declare that we only use LLM to aid or polish writing in this paper. Of course, we use LLMs to do inference in our experiment since we need to evaluate them on GRAPHOMNI.

