# OpenReview forum: "GraphOmni: A Comprehensive and Extensible Benchmark Framework for Large Language Models on Graph-theoretic Tasks"
_ICLR.cc/2026/Conference — ICLR 2026 Poster_

### Official Review · Reviewer_CVe5 · 2025-10-15

**Soundness:** 3
**Presentation:** 3
**Contribution:** 2
**Rating:** 4
**Confidence:** 5

**Summary:**

The paper introduces a comprehensive benchmark for evaluating LLMs' ability on graph-theoretic tasks. The evaluation is conducted on six graph-theoretic problems, which are encoded with different serialization formats and prompting schemes. The main results show that current LLMs have substantial room for improvement on graph-theoretic problems, and there is no consistently good serialization and prompting strategies among different tasks and models. The paper further introduces a reinforcement learning-based optimizer that adaptively selects factor combinations.

**Strengths:**

The benchmark is comprehensive with multiple aspects of encoding strategies of graph-theoretic problems, covering both open-sourced and closed-sourced LLMs. The workload is heavy, and the experimental results are detailed.

**Weaknesses:**

1. Each part of the key components (Benchmark Tasks, Graph Types, Prompt Schemes, and Serialization Formats) was proposed and researched in previous works. Although this paper considers all four components, the experiments and analysis are still conducted individually, and more importantly, do not bring substantially **new** insights or findings.

2. For most experiments, results are reported without **in-depth interpretation**, and therefore, the inspiration provided is limited.

3. The **conclusions are limited to six graph-theoretic tasks** (Connectivity, Cycle detection, Diameter, BFS, Triangle, Shortest path). What are the differences among the six tasks, and do they measure different aspects of LLMs' graph reasoning ability? It's unknown whether the current conclusion changes when evaluating on other graph-theoretic tasks.

**Questions:**

1. Which main results are first found in this paper or are different from previous benchmarking papers?

2. What's the real reason for the LLM's unsatisfying performance on graph-theoretic problems? In Result 3, it says representative categories of errors are commonly the misinterpretation of serialization formats and incorrect reasoning about graph-theoretic concepts. However, I think this can not be claimed as "fundamental" and can be solved by prompt engineering.

3. Why do open-source models benefit from multi-shot exemplars, whereas they do not help closed-source models much?

4. What is the complexity of the six tasks (e.g., time complexity related to node and edge numbers), and does the ranking align with the performance of LLMs?

---

> ### Author Response · Authors · 2025-11-20
> **Response from the authors (1/n)**
>
> We thank the reviewers for their valuable feedback. Here are the one-to-one responses to the reviewer's questions.
>
> ***
> # Response to Q1 (main results and findings):
>
> We appreciate the reviewer’s comment for asking which main results are truly new compared to previous benchmarking papers. The core contribution of **GraphOmni** is that, to the best of our knowledge, it provides the first **comprehensive and systematic empirical landscape** of graph-theoretic reasoning with LLMs **across multiple representation factors**. Prior works have already observed that individual dimensions such as encoding, prompting, or graph structure can affect performance. However, **evaluating models under specific, often limited, representation choices might overlook critical interactions among these factors**. Building on these insights, GraphOmni jointly varies graph type, serialization format, and prompt scheme within a single unified framework and evaluates a broad pool of models under this design. As discussed in our related work section and Table 1, existing benchmarks do not expose this combined three-dimensional design or provide a modular framework with a random baseline. As a result, many of the patterns that we report were not previously visible in a single, controlled setting, and our study makes them explicit. Within this landscape, the following results stand out as especially clear and actionable.
>
> 1. **Strong and quantitatively characterized representation sensitivity with no single best configuration**. Unlike prior works that typically vary only one factor or report encoding and prompting effects qualitatively, GraphOmni systematically sweeps graph types, serializations, and prompt schemes and measures how accuracy changes for the same underlying graph and task, showing that no single serialization or prompt works best across models, tasks, and difficulty levels. Within this shared representation space, we compare strong open-source and closed-source models across six canonical graph-theoretic tasks and three difficulty levels **(Figure 2)**, highlighting persistent gaps even for the strongest closed-source models and more pronounced weaknesses in open-source models. Prior works typically focus on one side or use different setups and evaluation pipelines, making such direct, like-for-like contrasts difficult. These effects are backed by comprehensive quantitative results across our tables and figures, which make the impact of representation choices explicit and directly comparable. We also further extend the experiments to larger graphs (30–50 nodes), real-world datasets (such as IMDB-MULTI and ogbg-molhiv), and NP-hard problems (including Max-Cut and Hamiltonian Cycle). The more complete and fine-grained experimental results are in **Appendix C (pp. 28-33)**, which provides additional details and supplementary data, further supporting and enriching the core conclusions of the main paper.
>
> 2. **Finding 1: Explicit domain algorithms in prompts significantly help graph reasoning.**
> GraphOmni shows that prompts that explicitly encode graph algorithms, such as BFS or diameter computation, substantially improve accuracy on structured reasoning tasks compared to plain prompts. This indicates that many models do not reliably recover the correct algorithm on their own but can benefit greatly from domain-specific algorithmic guidance. This finding is consistent with and extends prior observations about algorithmic prompting, but here it is evaluated systematically across multiple graph types, difficulty levels, and representation choices.
>
> 3. **Finding 2: Scaling raises the floor, while reasoning lifts the ceiling.**
>  Using Qwen-2.5 (7B), Qwen-2.5 (72B), and Qwen-3 (8B) under the same GraphOmni setting, we disentangle the effects of scaling and reasoning-centric design. Scaling within the same family (7B to 72B) mainly improves easy and medium regimes (for example, BFS order and shortest path in Easy and Medium), while a reasoning-oriented model of comparable size, Qwen-3 (8B), produces much larger gains on the hardest regimes, such as BFS order, diameter, and triangle counting in the Hard split. This leads to the conclusion that scaling tends to raise the floor, whereas reasoning-focused design is more effective at lifting the ceiling of graph reasoning ability. To the best of our knowledge, previous benchmarks have not provided such a controlled decomposition of scaling versus reasoning specifically for graph-theoretic tasks.
>
> **(to be continued in the next comment)**

---

> ### Author Response · Authors · 2025-11-20
> **Response from the authors (2/n)**
>
> # Remaining part of Response to Q1 (main results and findings):
>
> 4. **Efficiency as an additional axis and RL-based optimization of representations.**
>  GraphOmni measures not only correctness but also output length, and finds that different models realize very different accuracy–efficiency trade-offs. For example, some closed-source models achieve high accuracy with relatively short responses, while other models, especially some open-source ones, require much longer chains of thought to reach similar performance. This pattern holds across tasks, difficulty levels, serializations, and prompts. Motivated by the strong representation sensitivity revealed by the benchmark, we then treat the choice of serialization and prompt scheme as an optimization problem and introduce RL-Opt and RL-Scale. These RL-based methods learn effective combinations of formats, prompts, and (for RL-Opt) models, and achieve near-best accuracy while using only about 25% of the exhaustive grid-search cost. This provides a concrete example of how the representation landscape exposed by GraphOmni can be turned into practical improvements.
>
>
> Finally, apart from these empirical findings, **GraphOmni’s modular and extensible design** is a separate contribution. The benchmark is built so that new graph generators, serialization formats, prompt schemes, and models can be added easily, and it includes a random baseline and a unified evaluation protocol. This makes GraphOmni not only a one-time study but also a reusable landscape for consistently and systematically evaluating future work on prompt engineering, automated serialization, or graph-aware architectures.

---

> ### Author Response · Authors · 2025-11-20
> **Response from the authors (3/n)**
>
> # Response to Q2 (fundamental reason for graph reasoning performance):
>
>
> We thank the reviewer for raising the question of what truly underlies LLMs' unsatisfying performance on graph-theoretic problems and for discussing our characterization of the observed errors as "fundamental" gaps. We agree that there is no single "real" cause. Instead, we understand that there are two essential dimensions for graph reasoning of LLMs:
>
> 1. **Perception/input representation**: how graph-structured data, which is not intrinsically of text modality, is serialized and presented to a text-based LLM (choice of serialization format, prompt scheme, and other formatting factors);
>
> 2. **Model-side graph ability**: the model's inherent graph-theoretic knowledge, conceptual understanding, and multi-step reasoning ability, once it receives a suitable textual representation.
>
> We consider both dimensions to be fundamental to successful graph-theoretic problem-solving, and this understanding is supported by some related works as well [1-3].
>
> ### **Point 1: Why GraphOmni Focuses on the First Dimension**:
> GraphOmni is designed primarily to probe the first dimension (Perception/input representation) because we identified a large space for optimization that is pseudo-training-free (i.e., requiring no LLM-level training). The community's attention has mainly been on the second aspect (improving model capabilities through training or architectural changes), neglecting this fundamental aspect, which is cheap to optimize yet **still quite effective**. By systematically varying serialization formats and prompt schemes while keeping the underlying graph instance and task fixed, we are the first work to explicitly and systematically show that performance can shift dramatically across representations (e.g., up to 40% accuracy gaps for the same model and task when changing only input formatting). **GraphOmni comprehensively demonstrates this optimization opportunity**.
>
> Our error analysis further reveals that many failures arise because the model either misinterprets the serialized graph structure or misapplies basic definitions (e.g., treating the diameter as the length of the longest path rather than the longest shortest path), before higher-level reasoning even begins. In this sense, when we talk about "fundamental gaps," we mean that perception and concept-level understanding are prerequisite stages in the graph reasoning pipeline: If the model cannot reliably reconstruct the graph or apply fundamental concepts, no downstream reasoning strategy can fully compensate. We are happy to update and refine the relevant description accordingly based on the reviewer’s suggestion.
>
> ### **Point 2: From Understanding to Action:**
> Based on our evidence-supported understanding of perception, we fully agree with the reviewer that prompt engineering is a natural and powerful direction for improving perception. Sophisticated prompt design and automated prompt search methods (such as the AutoPrompt-style [4] techniques) directly target the sensitivity that GraphOmni exposes. **Our RL-Opt/RL-Scale experiments can be viewed as a concrete instantiation of this idea**: building on our most salient empirical findings about the interaction among serialization formats, prompt schemes, and model choice, we use RL to learn effective combinations that recover near-optimal performance at substantially reduced search cost (75% cost reduction, 90% accuracy retention, **Table 4, p. 10**). We see this not as the final solution, but as an example of how our benchmark's insights can be turned into actionable improvements.
>
>
> ### **Point 3: Broader Contribution**:
>
> **Our goal is to provide a landscape for the field rather than claim a single root cause or a unique solution**. GraphOmni **(i)** systematically maps how graph type, serialization, and prompting jointly affect performance; **(ii)** highlights that input representation is a critical, and often underappreciated, part of graph reasoning with LLMs; and **(iii)** offers a modular, extensible framework where future work can plug in new serialization strategies, prompt-engineering methods, or even graph-aware architectures and tokenizations, and evaluate them fairly on a shared testbed. In the revision, we clarify this perspective by explicitly stating that we view both perception and model-side reasoning as fundamental aspects of graph-theoretic performance, and that our contribution is to rigorously dissect and quantify the perception side, thereby providing actionable insights that can inform prompt engineering, RL-based optimization (as we demonstrate), and more advanced architectural improvements going forward.

---

> > ### Author Response · Authors · 2025-11-20
> > **Reference for "Response from the authors (3/n)"**
> >
> > [1] Bahare Fatemi, Jonathan Halcrow, Bryan Perozzi. Talk like a Graph: Encoding Graphs for Large Language Models. ICLR 2024.
> >
> > [2] Zihan Luo et al. GraphInstruct: Empowering Large Language Models with Graph Understanding and Reasoning Capability. Arxiv preprint.
> >
> > [3] Jiayan Guo, Lun Du, Hengyu Liu, Mengyu Zhou, Xinyi He, Shi Han. GPT4Graph: Can Large Language Models Understand Graph Structured Data? An Empirical Evaluation and Benchmarking. Arxiv preprint.
> >
> > [4] Taylor Shin, Yasaman Razeghi, Robert L. Logan IV, Eric Wallace, Sameer Singh. AutoPrompt: Eliciting Knowledge from Language Models with Automatically Generated Prompts. EMNLP 2020.

---

> ### Author Response · Authors · 2025-11-20
> **Response from the authors (4/n)**
>
> # Response to Q3 (Why do open-source models benefit from multi-shot exemplars, whereas they do not help closed-source models much?):
>
>
> We appreciate this insightful question from the reviewer. Our empirical observations indeed reveal a divergent impact of multi-shot exemplars across model categories, and we offer the following explanation grounded in our findings and model characteristics:
>
> - **Closed-source models demonstrate more sophisticated instruction-following capabilities** through extensive alignment training (instruction-tuning and RLHF), enabling them to understand task requirements with minimal prompting. As shown in our results, closed-source models achieve strong performance with zero-shot prompts (particularly 0-CoT), indicating they can already infer the desired output format and reasoning pattern without explicit examples. For these models, adding few-shot exemplars provides diminishing returns and may even introduce redundancy or noise, particularly when the examples add substantial token overhead without commensurate informational value (as noted in **Finding 5** in **p. 101, Appendix E.6**).
>
> - **Open-source models, by contrast, often lack comparable alignment training** and benefit significantly from concrete demonstrations of the expected response format and reasoning structure. The multi-shot exemplars provide explicit guidance that helps these models calibrate their outputs to meet the task requirements. This is evidenced by our findings (Figure 5a), which show that open-source models achieve their highest accuracy when prompt schemes incorporate shots.
>
>
> In this rebuttal, we also provide additional sensitivity analysis results in **Appendix E.3 (pp.61-67)**, and the new evidence further corroborates our conclusions. The results show that open-source models exhibit stronger **prompt sensitivity**, relying more on explicit in-context guidance to understand the task, whereas closed-source models display greater **format sensitivity**, benefiting more from information-rich serialization schemes due to their stronger intrinsic reasoning capabilities. These findings align with **Finding 3 (p. 8)**, providing additional evidence that multi-shot examples substantially improve the performance of open-source models. We will clarify the explanation in the revised manuscript to better articulate the underlying mechanisms.
>
>
> ***
>
> # Response to Q4 (time complexity of task with respect to LLM’s performance):
>
> We appreciate this excellent question! In the revised manuscript, we have added **Appendix H.1 (p. 107)** to address this point, including improved visualizations that illustrate the relative difficulty of the six tasks. For completeness, **we also provide the main results in Table 1 below**. Our analysis reveals a partial alignment between computational complexity and LLM performance. **At the extremes, the correspondence is clear**: Triangle counting ($O(V^3)$) and Diameter ($O(V(V+E))$) are both algorithmically expensive and empirically difficult for LLMs, achieving only 15.45% and 40.09% accuracy on hard instances, respectively, with closed-source models. However, **among the four tasks with identical $O(V+E)$ complexity, we observe dramatic performance divergence**. Connectivity maintains 91.90% accuracy while BFS order collapses to 27.15%, a gap of 64.75% points despite equivalent asymptotic complexity. This reveals three additional factors beyond computational complexity: output structure (binary decisions vs. sequences vs. numerical values), error propagation (how single mistakes cascade in sequential tasks), and reasoning scope (whether the task requires local or global graph traversal). **Tables 28 to 30 (p. 108)** provide detailed evidence that output format and reasoning requirements are at least as necessary as algorithmic complexity in determining LLM performance on graph reasoning tasks.
>
> **(Table 1 is attached in the next comment due to limited space)**

---

> > ### Author Response · Authors · 2025-11-20
> > **Table 1 used in Response to Q4 in message "Response from the authors (4/n)"**
> >
> > **Table 1: Aggregate performance by task and complexity for open-source vs. closed-source models with respect to the time complexity of each task.**  Accuracy is averaged over models in each category. Δ denotes the Easy to Hard performance drop.
> >
> > | Task          | Time Complexity       | Open-Source Easy | Open-Source Hard | Open-Source Δ | Closed-Source Easy | Closed-Source Hard | Closed-Source Δ |
> > |---------------|-----------------------|------------------|------------------|---------------|--------------------|--------------------|-----------------|
> > | Triangle      | \(O($V^3$)\)            | 22.70            | 6.77             | −15.93        | 53.65              | 15.45              | −38.20          |
> > | Diameter      | \(O(V(V+E))\)         | 52.30            | 21.33            | −30.97        | 81.43              | 40.09              | −41.34          |
> > | BFS order     | \(O(V+E)\)            | 37.37            | 9.43             | −27.94        | 89.67              | 27.15              | −62.52          |
> > | Shortest path | \(O(V+E)\)            | 54.62            | 37.83            | −16.79        | 90.84              | 82.41              | −8.43           |
> > | Cycle         | \(O(V+E)\)            | 63.78            | 59.76            | −4.02         | 81.98              | 79.24              | −2.74           |
> > | Connectivity  | \(O(V+E)\)            | 81.87            | 75.97            | −5.90         | 96.21              | 91.90              | −4.31           |
> > | **Mean**      |                       | **52.11**        | **35.18**        | **−16.93**    | **82.30**          | **56.04**          | **−26.26**      |

---

> ### Author Response · Authors · 2025-11-20
> **Response from the authors (5/n)**
>
> # Response to W1 (new insights and findings):
>
> Thank the reviewer for this observation. While prior works have studied individual dimensions, to the best of our knowledge, there has been no unified framework that systematically examines how these components interact. This leaves a gap in understanding whether LLM performance on graph reasoning is driven by the prompt, the serialization format, or their combined effects. **GraphOmni fills this gap by introducing the first comprehensive benchmark that jointly spans multiple graph types, prompting schemes, and serialization formats**. We further clarify the roles of each component and the new insights they provide.
>
> ***
> ### **Point 1: Component Roles and Contributions**:
> The four components serve distinct purposes in GraphOmni:
> - **Graph Types**: These are not aimed at completely separate evaluation, but rather provide **comprehensive coverage of graph structures that no prior work has achieved**. As shown in **Table 1** in the Introduction (p. 2), existing benchmarks typically use 1-4 graph types. By spanning 7 diverse graph types across three difficulty levels, we ensure our findings about representation choices are robust and generalizable rather than artifacts of a narrow setup.
> - **Benchmark Tasks, Prompt Schemes, and Serialization Formats**: These are the core focus dimensions where we provide **new empirical insights through systematic joint evaluation**. As detailed in our **Response to Q1**, prior work examined these factors individually, but GraphOmni reveals their critical **interaction effects**. We provide **all fine-grained results in comprehensive heatmaps for all models across all tasks (Figure 4, Appendix E.2)**, quantitatively demonstrating that no single configuration works universally across models, tasks, and difficulties (**Result 1** in Section 4.1 (p. 6)).
>
> ***
> ### **Point 2: Concrete Actionable Insights**:
>
> Beyond identifying interaction effects, we provide:
> - **Quantitative performance heatmaps** showing up to 40% accuracy gaps across serialization-prompt combinations for the same model and task, enabling practitioners to identify optimal configurations.
> - **Targeted findings** with direct implications: **Finding 1 (p. 7)** shows that embedding explicit graph algorithms in prompts substantially improves structured reasoning; **Finding 2 (p. 7)** demonstrates that reasoning-centric design (Qwen-3 8B) outperforms pure scaling (Qwen-2.5 72B) on hard tasks.
> - **RL-based optimization method (Section 4.4, Table 4, p. 10)** that reduces evaluation cost by 75% (from 100% to 25%) while maintaining 90% accuracy, providing a practical pseudo-training-free solution (as mentioned in Response to Q2 above) for navigating the large configuration space without LLM-level training.
>
> ***
> ### **Point 3: Acknowledgment of scope.**
>
> The combinatorial space of configurations is very large, so an exhaustive fine-grained analysis of every individual combination is infeasible. Nevertheless, our central conclusion that optimal configurations depend on complex interactions rather than simple universal rules has direct implications for how future evaluations should be designed. Additional findings enabled by this framework are summarized in our **Response to Q1**.

---

> ### Author Response · Authors · 2025-11-20
> **Response from the authors (6/n)**
>
> # Response to W2 (in-depth interpretation):
>
> We thank the reviewer for this feedback. We clarify that GraphOmni provides substantial interpretations and actionable insights while maintaining scientific rigor.
>
> ### **Point 1: Comprehensive Findings with Clear Interpretations**:
>
> **Sections 4.2** and **4.3** are dedicated to interpreting our results. We provide analysis at multiple levels:
>
> 1. **High-level insights (Results 1-2)**: We identify critical interaction effects, quantify representation sensitivity (40% performance gaps), and reveal persistent limitations even in SOTA models.
>
> 2. **Fine-grained findings (Findings 1-5)**: We interpret consistent patterns—algorithmic prompts improve structured reasoning, scaling vs. reasoning have different effects, and few-shot examples impact open-source vs. closed-source models differently.
>
> 3. **Task-by-task heatmaps (Figure 4, Appendix E.2 p. 41)**: We present a comprehensive quantitative analysis across all models and tasks, enabling specific conclusions for particular use cases.
>
> 4. **Sensitivity analysis (Appendix E.3 p. 61)**: We present a sensitivity-driven framework that measures how different graph types react to variations in prompts and serialization formats, while preserving both interpretability and extensibility.
>
> ***
>
> ### **Point 2: Rigorous Yet Insightful Analysis and Conclusion**
>
> Given the large combinatorial space (7 graph types × 7 serializations × 9 prompts × 6 tasks × 3 difficulties), we provide confident interpretations when the evidence is strong and consistent, while acknowledging where complexity limits definitive conclusions. Our key insight, that optimal configurations depend on **interaction effects**rather than universal rules, is itself a meaningful finding that challenges prior assumptions. This measured approach provides more reliable guidance than oversimplified claims about "best practices”. (Please see our **Response to W1** for how we systematically reveal these interactions through joint evaluation.)
>
> ***
>
> ### **Point 3: Concrete Actionable Contributions:**
>
> GraphOmni translates empirical observations into practical value:
> - **RL-based optimization** (**Section 4.4**, Table 4): Reduces evaluation cost by 75% while maintaining 90% accuracy.
> - **Performance heatmaps**: Direct guidance for selecting optimal configurations for specific model-task combinations.
> - **Modular framework**: Enables systematic future research on representation optimization and graph-aware architectures.
>
> As we discussed in **Response to Q1**, these contributions span empirical insights, methodological guidance, and practical tools, all grounded in the first comprehensive multi-dimensional evaluation of LLM graph reasoning.

---

> ### Author Response · Authors · 2025-11-20
> **Response from the authors (7/7)**
>
> # Response to W3 (rationale for task selection):
>
>
> We appreciate the reviewer’s concern that our conclusions are drawn from six graph-theoretic tasks and whether they really probe different aspects of graph reasoning.
>
> Our choice of tasks is deliberate rather than arbitrary. As detailed in **Appendix A.3.1 (p. 18)** (“Rationale for Selection of Tasks”), the six core tasks are designed to cover three complementary aspects of graph reasoning:
>
> 1.**Reasoning capacity**. The tasks form a progression from simple global checks to multi-step traversals and full combinatorial enumeration:
> >(1) **Connectivity and Cycle detection require a global traversal, but only a simple decision condition** (connected or not, cycle or not). Once the serialization is parsed correctly, these mainly test whether the model can explore the whole graph and apply a basic predicate.
>
> >(2) **BFS order, Shortest path, and Diameter require ordered-path reasoning**. The model must maintain a frontier or distance map across multiple layers, preserve ordering information, and then perform a final aggregation step (minimum distance for Shortest path, maximum over all shortest paths for Diameter). Our error cases show that models often “forget” edges or previously discovered nodes while expanding the frontier.
>
> >(3) **Triangle counting is the most demanding, since it requires combinatorial enumeration over all triples and then accurate aggregation**. We see errors both in checking individual triples (missing or hallucinating edges) and in the final counting step.
> Empirically, this hierarchy is reflected in the results: performance is highest for simple reachability, lower for ordered-path reasoning, and lowest for complete combinatorial enumeration.
>
>
> 2.**Task understanding and definition knowledge.** The tasks also differ in how much they rely on precise textbook definitions. For example, some models confuse diameter with the longest simple path, and others use crude heuristics for triangles (such as treating the count as roughly proportional to the number of nodes rather than checking all three edges). These errors show that our suite probes not only search or traversal ability but also whether the model has correctly internalized basic graph-theoretic concepts.
>
>
> 3.**Output format and robustness to formatting constraints.** The output requirements range from very short answers to long, highly structured sequences. Connectivity and Triangle counting require only “Yes/No” or a single number, so there is little room for formatting errors. In contrast, BFS order demands a level-by-level listing of all nodes, where a single missing or extra node makes the entire answer incorrect. By mixing strict and straightforward formats, the tasks test whether models can remain correct under different output constraints, not only different graph structures.
>
>
> In addition to these six core tasks, we also report results on **two NP-hard tasks** in **Appendix C.4 (p. 30)** (**Hamiltonian cycle** and **Max-Cut**). These extended experiments show that open-source models remain close to random performance, while closed-source reasoning models achieve higher but still far-from-perfect scores, and that the human vs. LLM difficulty gap does not exhibit a sharp jump exactly at NP-hardness. This suggests that the qualitative trends we highlight in the main paper are not limited to a single “toy” task, although we agree that a broader set of NP-hard and application-specific problems would be valuable future extensions.
>
> **Also, we appreciate the reviewer’s insightful question about task coverage, as it aligns perfectly with our extensibility design philosophy**. Our goal is not to cover all possible graph-theoretic problems in one paper, but to construct a diverse and well-motivated suite that spans different reasoning regimes, definition requirements, and output formats, while providing a framework that the community can build upon. Importantly, GraphOmni is specifically designed to be extensible: new graph-theoretic tasks can be plugged into the same framework and evaluated under the same graph types, serializations, and prompt schemes with minimal effort. We see expanding the task set by adding additional NP-hard, application-specific, or domain-tailored tasks as a natural and valuable next step that our benchmark is built to support, and we would like to thank the reviewer for highlighting this critical direction for future work.
>
> ***
>
> **We greatly appreciate the reviewers' time in reading our rebuttal and considering our responses. And we are happy to provide further details or answer any additional questions the reviewers may have.**

---

### Official Review · Reviewer_61MW · 2025-10-17

**Soundness:** 4
**Presentation:** 3
**Contribution:** 3
**Rating:** 6
**Confidence:** 5

**Summary:**

The paper proposes GraphOmni, a comprehensive benchmark for LLMs on Graph-theoretic tasks.
The benchmark makes breakthroughs in the number of instances, the number of investigated schemes, and the number of serializations.
Key findings from using GRAPHOMNI include: No single serialization format or prompt works best for all tasks; performance varies widely, and even state-of-the-art LLMs show significant room for improvement on graph reasoning tasks.
Motivated by these findings, the authors also propose an RL-inspired selector to dynamically choose the best settings for a given task, thereby improving performance.

**Strengths:**

1. The benchmark is comprehensive in both scale and involved methods, providing a solid exploration in graph reasoning.
2. The analysis is very solid and presents interesting viewpoints. The findings in Section 4.2 provide meaningful insights.
3. The paper is very clearly written overall. The presentation is excellent and the logic is complete.

**Weaknesses:**

1. The significance of comparing LLMs on graph reasoning is questionable. The challenges LLMs face in graph reasoning seem largely constrained by context length, rather than being focused on core reasoning abilities like mathematical reasoning. There is a lack of discussion on what specific deep capabilities of LLMs the study of graph reasoning is meant to reflect.
2. Expanding on point #1, a key characteristic of graph reasoning is that even humans often rely on external tools (e.g., writing code or drawing diagrams) to solve problems like finding shortest paths or Hamiltonian cycles. The inability of a human to solve these problems entirely "in their head" is not typically taken as a sign of deficient reasoning ability. Therefore, the practical significance of evaluating a single LLM's isolated reasoning capability on these tasks is limited.
3. The study only considers graphs with up to 50 nodes, which is still too small in scale.
4. The description of the RL-based method is too brief. While I understand space constraints, the current description fails to adequately explain in the main text: 1) the core limitation it addresses, 2) the experimental setting, and 3) it lacks a comprehensive comparison with baselines. These elements should be fully described in the main body to ensure the paper is self-contained and coherent.

**Questions:**

See Weaknesses 1 & 2

---

> ### Author Response · Authors · 2025-11-20
> **Response from the authors (1/n)**
>
> We thank the reviewers for their valuable feedback and positive comments regarding our work. Here are the one-to-one responses to the reviewer's questions.
>
> ***
>
> # Response to the first part of W1 (significance of comparing LLMs on graph reasoning):
>
> We thank the reviewer for this insightful observation. We agree that graph representation length grows quadratically with the number of nodes, and context length poses practical constraints for very large graphs. However, this observation precisely motivates our work's primary focus: **GraphOmni is positioned as Graph4LLM rather than LLM4Graph**. We utilize graph problems as a set of well-defined, commonly seen, yet challenging tasks to systematically evaluate and expose the capabilities and limitations of current LLMs.
>
> Graph-theoretic tasks serve as excellent diagnostic tools for several reasons, as noted in a series of works [1-5]. They provide **verifiable ground truth and controllable complexity**, enabling precise evaluation unlike open-ended generation tasks. They represent **ubiquitous structural reasoning patterns** that appear across domains, from dependency analysis to hierarchical inference. Most importantly, they allow us to **isolate reasoning capability from resource constraints** through careful experimental design.
>
> Our key methodological choice directly addresses the context length concern: we deliberately use small graphs (5-30 nodes) with compact serializations consuming approximately 900 input tokens on average, well within modern context limits (128k+ tokens). **This design intentionally eliminates context as a confounding factor. Under these controlled conditions, where information access is unconstrained, we observe systematic failures that reveal fundamental limitations in reasoning**. The state-of-the-art o4-mini achieves only 17.53% accuracy in triangle counting on fully accessible 30-node graphs. Performance on BFS order collapses from 89.67% to 27.15% as difficulty increases, despite all information remaining comfortably within context. Most tellingly, identical graphs yield accuracy gaps up to 40% when serialized differently (Figure 4), demonstrating **sensitivity to surface form rather than ‘deeper’ understanding**.
>
> These empirical results confirm our diagnostic objective: **even on small graphs, we successfully challenge LLMs and expose their weaknesses in structural reasoning**. The failures do not stem from models lacking sufficient access to graph data, but from their difficulty in internally representing and manipulating graph-structured relationships. This reveals that current LLMs have fundamental limitations in compositional, multi-step inference over structured graph data, independent of context constraints.
> Beyond diagnosis, our findings offer **actionable insights with broader applicability**. The RL-Opt method demonstrates that perception augmentation through format optimization can reduce evaluation cost by 75% while maintaining accuracy. This suggests a **potentially generalizable pseudo-training-free (i.e., no training at the LLM level) perception-augmentation strategy** for any domain where structured information must be linearized for language model consumption, offering insights beyond graphs to tables, code, formal logic, and other structured reasoning tasks where perception is a bottleneck.
>
> We emphasize that while our primary contribution is diagnostic (using graphs to understand LLM capabilities), **graph reasoning itself remains an essential capability** for foundation models, as many real-world applications naturally involve graph-structured data (knowledge graphs, social networks, routing problems), as we continue to mention in the following **Response to W1-2**. Our work characterizes both the current state and the substantial room for improvement in how LLMs handle such structured information. We will clarify this dual perspective in the revision: graphs serve as evaluation instruments for understanding LLM reasoning while also representing a practically important capability domain in their own right.
>
> **Reference**:
>
> [1] Bahare Fatemi, Jonathan Halcrow, Bryan Perozzi. Talk like a Graph: Encoding Graphs for Large Language Models. ICLR 2024.
>
> [2] Zihan Luo et al. GraphInstruct: Empowering Large Language Models with Graph Understanding and Reasoning Capability. Arxiv preprint.
>
> [3] Jiayan Guo, Lun Du, Hengyu Liu, Mengyu Zhou, Xinyi He, Shi Han. GPT4Graph: Can Large Language Models Understand Graph Structured Data? An Empirical Evaluation and Benchmarking. Arxiv preprint.
>
> [4] Nuo Chen, Yuhan Li, Jianheng Tang, and Jia Li. Graphwiz: An instruction-following language model for graph computational problems. In Proceedings of KDD 2024.
>
> [5] Heng Wang, Shangbin Feng, Tianxing He, Zhaoxuan Tan, Xiaochuang Han, and Yulia Tsvetkov. Can language models solve graph problems in natural language? NeurIPS 2023.

---

> ### Author Response · Authors · 2025-11-20
> **Response from the authors (2/n)**
>
> # Response to the second part of W1 (discussion on what specific deep capabilities of LLMs the study of graph reasoning is meant to reflect):
>
> We thank the reviewer for this insightful question and for allowing us to provide a more thorough description. **The graph reasoning ability itself is what we want to reflect in this work**. We are not claiming that GraphOmni directly measures a single, general “deep reasoning” factor; instead, our goal is to **understand how current LLMs handle graph-structured information as a capability in its own right**.
>
> We fully agree with the reviewer that graph reasoning is more specialized than broad mathematical reasoning. **However, it is still essential for foundation models, as it supports other modalities beyond plain text**. Many real-world problems naturally take a graph-like form (e.g., knowledge graphs, networks, routing), and handling them well requires models to parse non-textual structure, follow graph-specific procedures (e.g., traversals, combinatorial checks), and apply precise definitions. Our findings cover the strong dependence on serialization, the significant gains from making graph algorithms explicit in the prompt, and the different roles of scaling versus reasoning-centric design, all of which point to graph reasoning having its own characteristic strengths and failure modes. So, we believe they are not just “math in disguise.”
>
> **A helpful analogy may be pedestrian detection versus general object detection**: pedestrian detection is more specific than general object detection, but it still involves distinct patterns and challenges that practitioners study separately. In the same spirit, GraphOmni is intended to reflect and characterize graph reasoning as a specialized but essential capability of LLMs, complementary to (rather than a replacement for) more general mathematical reasoning benchmarks.
>
> ***
>
> # Response to W2 (practical significance of evaluating):
>
> Thank the reviewer for the question. We agree that for many graph problems, especially harder ones like Hamiltonian cycles or large-scale shortest paths, humans naturally use external tools such as code or diagrams, and that failing to solve them entirely “in one’s head” is not evidence of poor reasoning. **Our goal in GraphOmni is not to claim that LLMs should solve all graph problems without any tools, but to understand what they can already do before tools are involved**. Even for humans, there is a sequence: first recognizing the nature and difficulty of the graph task, then understanding which concepts and algorithms apply, and only then deciding to write code or draw a diagram. For LLM-based systems, we currently have very limited knowledge of this **“intrinsic understanding”** layer for graphs. GraphOmni is designed to characterize that layer on small, context-friendly graphs, so we can see **where models already show meaningful graph understanding and where they fail even before tool use enters the picture**.
>
>
> This intrinsic capability is practically vital because it **underpins effective tool use**. An LLM that has no stable grasp of graph structure, shortest paths, or triangles would struggle not only to solve such tasks directly, but also to (i) recognize when a task is hard enough to require an external tool, (ii) write correct code or formulate the right tool call, and (iii) sanity-check or interpret the tool’s output. Our findings, such as strong dependence on serialization, significant gains from making algorithms explicit in the prompt, and different roles of scaling versus reasoning-centric design, show that there are still sizable gaps in this base layer of graph understanding. In that sense, **GraphOmni plays a role similar to “needle-in-a-haystack” in long-context reasoning benchmarks**: even if humans would also prefer tools for such cases, these controlled setups are valuable for mapping the model’s capabilities and limits, which is a prerequisite for designing robust tool-augmented graph agents. We will clarify this perspective in the revision and note that studying how to best combine these intrinsic abilities with external tools is a natural direction for future work.

---

> ### Author Response · Authors · 2025-11-20
> **Response from the authors (3/3)**
>
> # Response to W3 (small graph size):
>
> We thank the reviewer for this comment. As discussed in **Response to W1-1**, our benchmark intentionally uses small graphs to isolate reasoning capability from context constraints. Nevertheless, we have conducted additional experiments on 50-100 node graphs to address this concern.
> **Table 1** below presents results for representative models on 50-100 node graphs compared to our Hard (20-30 nodes) split. Performance degrades as expected, but **task difficulty rankings remain identical**, and **relative model performance gaps persist at similar magnitudes**. Critically, **no new failure modes emerge**, i.e., the same fundamental challenges we identified (combinatorial enumeration, sequential dependencies, serialization sensitivity) simply intensify with scale.
> **These results confirm that our 5-30 node design captures the essential reasoning challenges. Larger graphs amplify these challenges quantitatively but reveal no new qualitative phenomena**, validating our focus on controlled-scale evaluation where reasoning capability, rather than resource constraints, determines performance. We have added this analysis to the revised manuscript in **Appendix H.2 (p. 109)**.
>
> **Table 1: Results on 50-100 node graphs. Values in parentheses show Hard (20-30 nodes) performance for comparison.**
>
> | Task | Qwen-2.5 (72B) | o4-mini |
> |------|----------------|---------|
> | **BFS order** | 8.19% (22.03%) | 10.23% (32.45%) |
> | **Connectivity** | 62.00% (84.09%) | 81.86% (92.02%) |
> | **Cycle** | 37.78% (68.40%) | 74.81% (95.61%) |
> | **Diameter** | 8.89% (29.59%) | 40.44% (34.61%) |
> | **Shortest path** | 33.28% (72.53%) | 68.51% (88.63%) |
> | **Triangle** | 2.36% (4.73%) | 2.85% (17.53%) |
>
> ***
>
> # Response to W4 (description of the RL-based method):
>
> We sincerely thank the reviewer for this constructive suggestion. In addition to providing full RL details (state/action/reward formulation and experimental setup) in **Appendix D (pp. 33-36)**, we will include a more detailed description of our RL method in **Section 4.4** of the main paper. Below, we address the reviewer’s three points.
>
> 1. **Core limitation the RL method addresses**: We agree that the paper should more clearly explain the motivation for introducing the RL method. The main contribution of GraphOmni is to unify and jointly analyze multiple dimensions that affect LLM reasoning in graph-structured tasks, moving beyond prior work that has studied only isolated factors. This joint analysis provides rich insights, and **Section 4.2** reveals meaningful empirical findings. Moreover, our benchmark is flexible and can incorporate additional factors in the future. We also include this content in the main body of the article.
> However, during this process, we also need to answer an actionable question directly, i.e., **given the many interacting dimensions, what is the best prompt configuration for a specific graph reasoning task?** To address this quantitative problem, we introduce an RL-based approach that **searches the large combinatorial serialization space as a diagnostic tool to provide an actionable solution within our benchmark**.
>
>
> 2. **Experimental setting**: We have updated the main text with additional details. Specifically, we employ RL with a neural network to approximate the Q-function. At step $t$, a three-layer ReLU MLP estimates $\widehat{Q}_t$, trained using mean squared error loss. Action selection follows an $\epsilon$-greedy strategy with $\epsilon$ linearly decaying from 1 to 0.01. These details are now summarized in **Section 4.4**, with complete specifications remaining in **Appendix D**.
>
>
> 3. **Comparison with baselines**: We appreciate the reviewer’s suggestion for additional baseline comparisons. We acknowledge that a comprehensive algorithmic comparison would strengthen this component. However, we note that the primary contribution of the RL method is diagnostic rather than algorithmic: it demonstrates that serialization selection can be formulated as a decision-making problem and that optimization within this framework yields consistent improvements. Given the paper's breadth (spanning 6 tasks, 7 graph types, 7 serializations, 9 prompts, and 11 models) and the extensive analyses already presented (including complexity analysis, cost-accuracy tradeoffs, and extended studies on larger graphs, real-world datasets, and NP-hard tasks), **we focused on establishing the feasibility and utility of this approach**. **We agree that systematic comparison with baselines would be valuable future work to establish the relative efficiency of different search strategies within our framework**.
>
> ***
>
> **We greatly appreciate the reviewers' time in reading our rebuttal and considering our responses. And we are happy to provide further details or answer any additional questions the reviewers may have.**

---

> > ### Comment · Reviewer_61MW · 2025-11-26
> >
> > Thank you for the authors' detailed response.
> >
> > Regarding my concerns about W1 and W2, I performed additional tests using advanced LLMs—including GPT, Gemini, and DeepSeek—on shortest-path problems. I observed that these models already manage to apply Dijkstra’s algorithm.
> > In the subsequent steps, they use tools or perform manual reasoning, which appear to be of secondary importance.
> > Do you think these models have already understood graphs?
> >
> > I think this somehow suggests a distinction between an LLM’s ability to solve graph problems and its deeper understanding of graph structure. Since the models demonstrate awareness of relevant algorithms, the bottleneck may lie not in graph understanding but in general reasoning capabilities.
> >
> > Additionally, regarding scalability, I believe the current experiments are comprehensive and do not require additional experiments on larger graphs.
> > I'm worried that, when graphs are provided via prompts, there is an inherent upper bound on input size. As a result, for weaker models, it remains unclear whether their performance is constrained by context length or by genuine difficulties in graph structure comprehension—a point also touched upon in the last paragraph of the discussion.
> >
> > To sum up, I feel the overall impact of this benchmark would be somewhat limited due to the design methodology and practical applicability.
> > However, I fully recognize the authors' contributions in LLM-based graph reasoning and will maintain my positive score for now. I remain open to further discussion.
> >
> > I would also like to offer two minor suggestions:
> >
> > * The rebuttal was quite lengthy, which made it somewhat challenging for all reviewers to understand. A slightly more concise rebuttal and future replies would be helpful.
> > * As per ICLR guidelines, the revised PDF should adhere to the page limit. It would be helpful to shorten the paper to align with the anonymous version requirements (9 pages).

---

> > > ### Author Response · Authors · 2025-11-26
> > >
> > > Thank you again for your thoughtful review and helpful suggestions. We appreciate your constructive comments and will address the specific points in our follow-up response.
> > >
> > > Regarding the page limit concern, we would like to clarify that according to the ICLR guidelines: “*During the discussion/rebuttal phase and for the camera-ready, the page limit will be increased to 10 pages to allow for new results/discussions.*” Our revised PDF adheres to this rule.

---

> > > > ### Comment · Reviewer_61MW · 2025-11-26
> > > >
> > > > Thanks for your clarification.
> > > >
> > > > I'm sorry for my misunderstanding about the guidelines, and I look forward to your further discussion and updates.

---

> > > > > ### Author Response · Authors · 2025-11-28
> > > > >
> > > > > The authors would like to thank the reviewer for the continued engagement and for positive recognition of our work. We want to discuss the two questions the reviewer mentioned concisely and directly as follows.
> > > > >
> > > > > 1. **On Graph Understanding vs. General Reasoning.**
> > > > >
> > > > > Thank the reviewer for the insightful comments and the additional tests. We believe these results highlight a crucial distinction. Your tests confirm that some models have good task understanding. They successfully identify the intent (e.g., "Shortest Path"), recall the correct procedure (Dijkstra), and execute tool calls. This indicates they are well-tuned to instruction and possess graph-theoretic knowledge.
> > > > >
> > > > > However, we argue that understanding the **task** is distinct from understanding the **graph structure** itself. GraphOmni intentionally evaluates the tool-free setting to test the models' actual structural perception. This is the ability to "read" and internalize the topology from the text before performing reasoning upon it.
> > > > >
> > > > > To use an analogy, counting apples in an image requires not just the mathematical ability to count (general reasoning) but the visual perception to locate the apples in the pixels first. Similarly, an LLM must correctly "perceive" the content of the graph before it can reason about it. Our sensitivity analysis (**Appendix E.3**) demonstrates that performance varies drastically based solely on serialization format. If the bottleneck were purely task understanding or general reasoning, performance should remain stable across equivalent formats. This sensitivity confirms that models struggle to robustly "perceive" the graph structure itself.
> > > > >
> > > > > We highly appreciate the reviewer's tests as they effectively isolate the factor of "task understanding" from structural perception. We will incorporate this valuable distinction into our discussion in the revised manuscript.
> > > > >
> > > > > 2. **On Scalability and Context Limits.**
> > > > >
> > > > > We’d like to thank the reviewer for recognizing our comprehensive experiments here. And we fully agree that context length imposes an inherent upper bound when scaling to very large graphs (e.g., thousands of nodes or more). However, GraphOmni is explicitly designed to isolate graph structure comprehension from long-context retrieval capabilities by operating well within these limits.
> > > > >
> > > > > > **Not much about Input Size**: As detailed in our updated **Table 27 (Appendix E.7.6)**, the average input length across the benchmark is only **933 tokens**. This is substantially below the 8k+ token context windows of even the smaller open-weight models evaluated (e.g., Llama-3-8B). Since the data fits comfortably within the window, performance drops in weaker models stem from genuine failures in structural comprehension, not context exhaustion.
> > > > >
> > > > > > **But more on Reasoning Depth**: Furthermore, the "length" discussed in our analysis refers to output generation rather than input capacity. Our findings in Figure 32 indicate that the average number of output tokens is far from the limit of all the models, i.e., the average output token is less than 600. This suggests that the constraint for weaker models is insufficient **reasoning depth**, as they fail even with ample token space, rather than hitting a hard input or output ceiling, at least under GraphOmni’s settings.
> > > > >
> > > > > ***
> > > > > We genuinely appreciate the reviewer’s insights regarding general reasoning and scalability, which suggest valuable future directions. Given that our current study already spans ~110 pages, covering foundational experiments on perception ability and efficiency, we believe we have thoroughly addressed the core bottleneck in structural "reading" and “understanding” capabilities. If the reviewer finds that our clarifications above adequately alleviate your concerns, we would be grateful if you would consider raising your score.

---

> > > > > > ### Comment · Reviewer_61MW · 2025-11-28
> > > > > >
> > > > > > Thank you again for the clarifications.
> > > > > >
> > > > > > Although I am currently unable to modify my score for technical reasons, I am inclined to **accept** the paper. I believe the work provides valuable observations on graph reasoning with LLMs.
> > > > > >
> > > > > > That said, I would like to offer several suggestions—mainly to better delineate the scope and limitations of the work, particularly regarding the distinction between graph understanding and graph tasks.
> > > > > >
> > > > > > While the paper emphasizes graph understanding, the evaluation tasks are still largely tied to graph tasks. For these tasks, as LLMs continue to advance, the solution pipeline is often:
> > > > > >
> > > > > > > recognize that the query is a graph task → apply appropriate graph-solving heuristics/tools → generate the answer.
> > > > > >
> > > > > > This naturally raises a conceptual question:
> > > > > >
> > > > > > > Does improved performance on graph tasks necessarily imply deeper graph understanding by the LLM?
> > > > > >
> > > > > > I believe this is an important open issue. High scores on explicit graph tasks—e.g., handling larger graphs or improving accuracy by a few percentage points—do not automatically indicate an LLM’s conceptual grasp of graphs.
> > > > > > If one already knows how to solve, I tend to believe they have already understood the problem. Even if fails in some steps, it is not due to its failure to understand the graph but other aspects.
> > > > > >
> > > > > > To more fully reflect “graph understanding,” it might be valuable in future work to explore graph reasoning embedded implicitly in natural language, rather than scenarios where the model is explicitly told “this is a graph task.”
> > > > > > For instance, directions such as “Talk Like a Graph” [1] highlight cases where graph structures are woven into everyday language. These settings reduce explicit task cues and instead test whether the model can abstract and internalize graph relational structures.
> > > > > >
> > > > > > In this sense, I would argue: The ability to abstract graph-related structures from natural language could be as important—if not more—than improving performance on explicit graph reasoning tasks.
> > > > > >
> > > > > > This does not diminish the contributions of the present work. I simply hope the community can also push toward understanding how and whether LLMs conceptually internalize graphs, beyond just graph reasoning.
> > > > > > I would be happy to see the authors reflect on these distinctions or consider incorporating such perspectives in future developments. And I would certainly welcome further discussion on these points.
> > > > > >
> > > > > > [1] Talk Like a Graph: Encoding Graphs for Large Language Models

---

> > > > > > > ### Author Response · Authors · 2025-11-29
> > > > > > >
> > > > > > > We sincerely thank the reviewer for their strong support and willingness to accept the paper. And we are very encouraged by the positive evaluation. We want to include our understanding of the two points that the reviewer mentioned in the comment:
> > > > > > >
> > > > > > > **1. On Task Performance vs. Understanding.** We appreciate the reviewer raising the conceptual question: *"Does improved performance on graph tasks necessarily imply deeper graph understanding?"* While we have stated our perspective in the previous rebuttal, i.e., emphasizing that **structural perception** is a necessary prerequisite for understanding, we fully agree that a final conclusion on this definition has not yet been drawn for the whole community. Determining exactly what constitutes "conceptual grasp" versus "task execution" remains a critical open challenge.
> > > > > > >
> > > > > > > **2. On Implicit Reasoning and GraphOmni's Role.** This connects directly to the reviewer's valuable point about implicit graph reasoning. In many real-world scenarios, where graph structures are implicit in natural language, precise algorithms or external tools are often unavailable. Consequently, the model must possess the intrinsic ability to handle the entire pipeline: from **perception** to **internalization**, and finally to **reasoning**. We believe this reality validates the GraphOmni approach. By focusing on a tool-free setting, we aim to evaluate specifically whether models can internalize structure when "shortcuts" are removed. We view this work as a necessary step on the long road toward the deep internalization of graph-structured data in foundation models. And yes, it is still a long way to go at this stage.
> > > > > > > ***
> > > > > > > We want to thank the reviewer again for these deeply insightful and constructive comments. They have significantly helped us shape the positioning of our work and clarify the path forward.

---

### Official Review · Reviewer_nDLz · 2025-10-18

**Soundness:** 3
**Presentation:** 3
**Contribution:** 3
**Rating:** 6
**Confidence:** 2

**Summary:**

The paper introduces GraphOmni, a comprehensive and extensible benchmark for evaluating large language models on graph-theoretic reasoning tasks expressed in natural language. It systematically varies three critical dimensions—graph type, serialization format, and prompt scheme—across six canonical tasksand three difficulty levels. Experiments reveal substantial variability: no single model or prompt-serialization combination consistently dominates. The authors further introduce a reinforcement learning–based optimizer that adaptively selects optimal factor combinations.

**Strengths:**

S1. The benchmark framework is comprehensive, jointly considering three critical dimensions — graph types, serialization formats, and prompting schemes — to provide a multidimensional evaluation of LLMs’ graph reasoning abilities.

S2. The paper explores the use of reinforcement learning to search for optimal combination strategies, achieving significant cost reduction while maintaining high performance.

**Weaknesses:**

W1. The benchmark mainly focuses on classical graph-theoretic problems, which limits its applicability to real-world large-scale or labeled graphs that involve complex attributes, heterogeneous structures, or task-specific supervision.

W2. While GraphOmni reports accuracy across various tasks and factor combinations, the paper lacks a systematic analysis of how different task types interact with specific factor combinations.

**Questions:**

Q1. How do the authors ensure that the comparison between LLM-generated outputs and the ground truth is accurate? Given that the answers are evaluated from natural-language responses, what mechanisms or verification procedures are used to avoid misparsing or misjudging correctness during evaluation?

Q2. Could the authors provide a cross-factor analysis showing how different graph-theoretic tasks align with combinations of graph type, serialization format, and prompt scheme?

---

> ### Author Response · Authors · 2025-11-20
> **Response from the authors (1/n)**
>
> We thank the reviewers for their valuable feedback and positive comments regarding our work. Here are the one-to-one responses to the reviewer's questions.
>
> ***
>
> # Response to W1 (clarification on scope of the work):
>
> Thank the reviewer for this comment. We want to clarify the **scope and extensibility of GraphOmni**.
>
> GraphOmni intentionally focuses on the **topological structure** of graphs, which forms the **foundational layer of graph reasoning**.  Any reasoning over graphs must ultimately rely on understanding the underlying connectivity and structural properties, whether or not it involves semantic attributes, heterogeneous structures, or domain-specific tasks. This focus complements existing work on semantic-rich graphs such as GraphInstruct[1], GLBench[2], and GraphQA[3], which examine graphs with rich node/edge attributes and domain-specific semantics. Our work addresses a different but equally important question: Can LLMs reason about topological graph structure itself when expressed in natural language? This topological foundation is essential, as **models that struggle with basic structural reasoning (as our results show) will be highly likely to face similar challenges even when additional semantic complexity is introduced**.
>
> What’s more, GraphOmni is designed as an **extensible benchmark framework rather than a fixed dataset**. The modular architecture naturally accommodates future extensions to large-scale graphs, labeled graphs, and tasks beyond classical graph-theoretic problems. Our framework already demonstrates this extensibility in **Section 4.3**, where **we extend to larger graphs (30-50 nodes), real-world datasets (IMDB-MULTI, ogbg-molhiv), and NP-hard problems (Max cut, Hamilton cycle)**. The same systematic methodology and evaluation pipeline can readily incorporate attributed graphs, heterogeneous networks, or domain-specific graph reasoning tasks. The key contribution is establishing a rigorous foundation for topological reasoning while providing the infrastructure for the community to build upon with more complex graph scenarios.
>
>
> [1] Zihan Luo, Xiran Song, Hong Huang, Jianxun Lian, Chenhao Zhang, Jinqi Jiang, Xing Xie, Hai Jin. GraphInstruct: Empowering Large Language Models with Graph Understanding and Reasoning Capability. arXiv preprint, 2024.
>
> [2] Yuhan Li, Peisong Wang, Xiao Zhu, Aochuan Chen, Haiyun Jiang, Deng Cai, Victor W. Chan, Jia Li. GLBench: A Comprehensive Benchmark for Graphs with Large Language Models. NeurIPS 2024.
>
> [3] Bahare Fatemi, Jonathan Halcrow, Bryan Perozzi. Talk like a Graph: Encoding Graphs for Large Language Models. ICLR 2024.

---

> > ### Author Response · Authors · 2025-11-20
> > **Response from the authors (3/3)**
> >
> > # Response to Q1 (result parsing and evaluation):
> >
> > We thank the reviewer for the valuable comment. Ensuring that comparisons between LLM-generated outputs and ground-truth answers are accurate is crucial for a benchmarking study. To guarantee reliability, per the most common practice of QA tasks of LLMs, we designed a two-stage evaluation pipeline consisting of **(1) robust answer extraction from natural-language responses and (2) accurate correctness checking**. Both stages were refined through extensive manual verification.
> >
> > **1. Robust extraction of answers from natural-language outputs**
> >
> > For each task, we manually inspected a large sample of model responses to identify the typical surface forms used by LLMs (e.g., “BFS traversal starting from node X is …”, “The BFS order would be …”, etc.). Based on these observations, we implemented task-specific rule-based extractors.
> > We then iteratively evaluated these extractors, manually reviewed all unparsed or incorrectly parsed cases, and expanded the rules to cover newly observed response formats. This refinement loop continued until no additional rule changes altered the extraction results.
> >
> > **2. Reliable correctness verification**
> >
> > After extracting the structured answer, correctness was assessed using task-specific evaluation functions (details provided in the Evaluation Metrics in **Section 3** of the main content). These functions were refined using the same iterative process: inspect failures → fix edge cases → re-evaluate → confirm stability.
> >
> > After refinement, the final extraction accuracy was around 95% across all tasks. **For reference:
> > Triangle Counting: 93.77%; Diameter: 96.79%; BFS Order: 92.45%; Shortest Path: 94.82%; Connectivity: 94.41%; Cycle: 96.47%** (extraction accuracy is the ratio of generations that we can actually extract a valid output that is not None). The extraction failure rate is therefore very small, and we verified that further refinement does not meaningfully change the benchmark scores, indicating the **errors are basically due to the bad model generation itself**.
> >
> > Through **task-specific extraction rules**, **iterative manual verification**, and **convergence checking**, we ensure that the comparison between LLM-generated outputs and the ground truth is accurate and stable. This process guarantees the reliability of all reported benchmark results.
> >
> > ***
> >
> > **We greatly appreciate the reviewers' time in reading our rebuttal and considering our responses. And we are happy to provide further details or answer any additional questions the reviewers may have.**

---

> ### Author Response · Authors · 2025-11-20
> **Response from the authors (2/n)**
>
> # Response to W2 and Q2 (cross-factor analysis):
>
> We sincerely thank the reviewer for this insightful question regarding cross-factor analysis. Our extensive heatmaps in the original manuscript (**Figures 8-25,  pp. 43-60**) systematically **examine interactions between serialization formats and prompt schemes across all tasks, difficulty levels, and models**, revealing how different factor combinations affect performance under varying conditions. Furthermore, we examine **how different graph types perform across tasks** (**Appendix E.1.1 and Tables 15, 18, 21, pp. 37-41**), showing that certain structural properties present challenges while others facilitate specific reasoning tasks.
> To provide a more complete cross-factor picture, we **systematically integrate graph types with prompt schemes and serialization formats**.  While creating heatmaps for each graph type × task × difficulty combination would be comprehensive, this would generate over 100+ visualizations, making systematic interpretation impractical. Therefore, we introduce a **sensitivity-based analysis framework for scalable and interpretable cross-factor analysis**.
>
>
> **Sensitivity-based analysis framework**. For each graph type, we compute: (1) **Prompt Sensitivity ($S_p$)**: standard deviation of accuracy across different prompts under each format, averaged over all formats; and (2) **Format Sensitivity ($S_f$)**: standard deviation across different formats under each prompt, averaged over all prompts. We visualize each task-difficulty pair as a scatter plot in (**$S_p$, $S_f$**) space, where each point corresponds to a graph type and is colored by its mean performance. Median splits partition the space into four interpretable quadrants (Robust, Prompt-Critical, Format-Critical, Both Critical).
>
> **Key findings**. Our full results (**Figures 26-31, pp.61-67 in Appendix E.3**) span all 18 task-difficulty settings and reveal consistent patterns:
> 1. Open-source models exhibit stronger prompt sensitivity (a larger **$S_p$** range).
> 2. Closed-source models exhibit higher format sensitivity (**$S_f$**) but lower **$S_p$**.
>
> These sensitivity patterns **connect naturally to underlying LLM reasoning behaviors**: open-source models rely more on prompts for task comprehension, while closed-source models benefit more from informative serialization formats due to their stronger intrinsic reasoning capabilities. This aligns well with **Finding 3** in our main text (open-source models benefit from multi-shot examples while closed-source models do not), with additional supporting evidence provided in the newly added **Appendix E.3**.
>
>
> **Extensibility.** When researchers introduce new graph families, they can apply this sensitivity-based framework to quickly characterize how these graph types respond to different prompts and formats, helping prioritize configurations for more extensive evaluation. Complete visualization code and implementation details will be made available in our public repository.

---

### Official Review · Reviewer_xfYv · 2025-10-31

**Soundness:** 2
**Presentation:** 3
**Contribution:** 3
**Rating:** 6
**Confidence:** 3

**Summary:**

The paper introduces a large-scale benchmark (241,726 queries) for evaluating LLMs’ reasoning ability on graph-theoretic problems expressed in natural language. The framework systematically varies three dimensions—graph types, serialization formats, and prompt schemes—and assesses model performance across multiple difficulty levels and tasks (e.g., connectivity, BFS, cycle detection, triangle counting). The authors analyze open- and closed-source models, report detailed results, and propose a reinforcement learning–based optimization method (RL-Opt, RL-Scale) to reduce evaluation cost by 75% while maintaining accuracy.

**Strengths:**

1. The benchmark jointly varies graph type, serialization, and prompt scheme—more complete than prior work. This design provides strong coverage for structured reasoning.

2. Includes 241,726 queries across 7 graph generators and 9 prompt types; this ensures statistical reliability and generalizability.

3. Both open-source (Llama‑3, Qwen‑3) and closed-source (GPT‑4o, Claude‑3.5) models are evaluated to ensure representativeness.

4. Error analysis (Sec. 4.1, Result 3; Appendix E.3): Provides qualitative insights into misinterpretations (e.g., misunderstanding diameter definition), strengthening interpretability.

**Weaknesses:**

1. The benchmark provides empirical comparison but lacks analytical justification for observed behaviors—e.g., why serialization types yield specific effects. No direct evidence found in the manuscript.

2. The reuse of m, M, e, E in RL metrics (Sec. 4.4) is not clearly linked back to earlier sections or consistent definitions; may confuse readers.

3. No robustness check under prompt noise: All prompts assume fixed phrasing; real-world variance in natural language not examined.

4. While model output tokens analyzed (Sec. 4.3), inference or hardware cost details (e.g., GPU hours) are missing.

**Questions:**

1. In Sec. 4.4; Table 4, you define Cost = e/E and Rate = m/M but provide no equation for the reward function or RL objective. Could you explicitly describe: the state/action/reward formulation of your RL process.

2. The symbol m is reused for “edge count” in Sec. 2 (Graph Generators) and again for “accuracy” in Sec. 4.4 (Rate = m/M). please clarify notation consistency.

3. Sec. 4.4 presents “RL-Scale” to assess scalability. What feedback signal or evaluation metric was used to balance computational cost and accuracy when adding new serialization factors?

4. The paper systematically varies prompt schemes but all examples appear deterministic (Sec. 2; Appendix A.5). Did you test robustness under paraphrased or noisy prompts to evaluate linguistic invariance?

5. Table 1 omits recent benchmarks in graph reasoning tasks (e.g., GraphWild in GCoder, 2025).

---

> ### Author Response · Authors · 2025-11-20
> **Response from the authors (1/n)**
>
> We thank the reviewers for their valuable feedback and positive comments regarding our work. Here are the one-to-one responses to the reviewer's questions.
>
> ***
>
> # Response to W1 (analytical justification):
>
> We sincerely thank the reviewer for this important comment. We appreciate the opportunity to clarify the analytical justifications in the manuscript, which aim to explain why different serialization choices lead to systematic behavioral differences.
>
> 1. **Mechanistic error analysis. Result 3 and Appendix E.4** provide representative failure cases that reveal the mechanistic pathways through which different serialization formats introduce structural misalignment. These examples serve as analytical evidence explaining how and why specific serialization designs affect the model's internal reasoning process. For example, it brings insights like **Finding 1 (p.7)** to indicate how certain prompt schemes, like Algorithm scheme, can help models perform better.
>
> 2. **Cross-factor analytical study. Figures 8–25 in Appendix E.2 (pp. 43-60)** systematically examine the interaction between serialization format, prompting strategy, task type, difficulty level, and model family. In addition, **Appendix E.1.1 and Tables 15, 18, 21 (pp. 37-41)**, provide detailed comparisons across graph types, identifying structural properties that increase reasoning difficulty or amplify particular capabilities. **To further quantify how graph types interact with these factors**, we add a sensitivity analysis in **Appendix E.3 (pp.61-67)** that measures graph type responses to variations in prompts and formats. The analysis reveals that open-source models exhibit stronger prompt sensitivity while closed-source models show higher format sensitivity, providing analytical justification for why different serialization choices affect models differently: **open-source models rely more on prompts for task comprehension, while closed-source models benefit more from information-rich serialization formats due to stronger intrinsic reasoning capabilities**. These patterns align with **Finding 3 (p. 8)**, which shows that open-source models benefit from multi-shot examples while closed-source models do not. Based on all these results, we also have very specific analyses like **Findings 4 and 5 in Appendix E.6 (p. 101)**, shining more light on how we should understand the effect of different prompt schemes and serialization formats on models of different natures.
>
> 3.  **RL-based actionable framework**. Beyond the analytical insights above, our manuscript provides, in **Appendix D (pp. 33-36)**, an RL-based optimization framework that offers a generalizable and actionable method. While the analytical components explain why different serialization strategies yield specific effects for the evaluated models, the RL framework remains applicable across different model architectures and future LLMs, operationalizing these insights for practical optimization.
>
> In summary, the manuscript combines **mechanistic error analysis**, **systematic cross-factor interaction studies**, and **actionable RL-based optimization** to provide comprehensive analytical justifications for the observed behaviors. We are happy to provide additional details if the reviewer requires further clarification.
>
> ***
>
> # Response to W2 (inconsistent notation):
>
> Thank the reviewer for highlighting the notation inconsistency. We have revised the paper by introducing dedicated RL notations ($k$, $acc_⋆$, $acc_{max}$) and retaining $m$, $M$, $e$, $E$ exclusively for graph-related terms. All notation has now been made globally consistent.

---

> ### Author Response · Authors · 2025-11-20
> **Response from the authors (2/n)**
>
> # Response to W3 and Q4 (robustness check):
>
> We thank the reviewer for raising this important concern about robustness to natural language variation. We address this through a supplementary evaluation on prompt perturbations. **The detailed description and experiment results can be found in Appendix H.3 (pp. 109-114)**. We appreciate the reviewer’s effort in reviewing the paper's content, as it is hard to provide a clear view in the rebuttal boxes, given the multiple figures and tables.
>
> **Design Rationale**: In our main evaluation, we deliberately use deterministic phrasing within each prompt scheme to isolate the effects of our three core dimensions (graph types, serialization formats, prompt schemes) without confounding factors from linguistic variation. This controlled design allows us to systematically attribute performance differences to structural representation choices rather than incidental phrasing variations. However, we agree that robustness to linguistic variation is a practical concern.
>
> **Robustness Study Design:** We conduct a supplementary evaluation on BFS order (one of our tasks which requires a complex structured output) with 4,000 subsampled instances. We generate *noisy prompts* (i.e., semantically equivalent but syntactically diverse variants) through systematic human-authored paraphrasing of natural-language components while preserving character-for-character fidelity of graph data. Our perturbations achieve 17.5-47.9% word-level changes (e.g., formal "Initialize, enqueue, dequeue" → conversational "First, pick your, put it in, take out") across all 9 prompt types and 7 serialization formats. Multi-level verification ensures 100% preservation of the graph structure while 87.9% of prompts are successfully modified. We have very clear examples in **Figure 34 and 35 in Appendix H.3.1 (pp. 110-111)**.
>
> **Key Findings:** We evaluate o4-mini (top closed-source) and Qwen-2.5-72B (top open-source):
> 1. **Main conclusions remain stable (Table 32 and 33, p. 112):** Relative rankings of prompt schemes and serialization formats are preserved between original and perturbed conditions. For example, AL and AS consistently dominate serialization formats across all difficulty levels (AL: 92.26%→93.41% Easy, 83.44%→83.56% Medium, 48.27%→50.33% Hard), confirming that no single configuration works universally but specific formats consistently outperform others.
> 2. **Real perturbation effects confirmed  (Table 32 and 33):** Absolute performance values shift measurably (e.g., CoT: 85.26%→90.98% Easy; confidence intervals widen: ±9.08→±15.83 for 0-Shot Easy), validating that our perturbations introduce meaningful linguistic variation.
> 3. **Differential robustness  (Table 34-37, p. 113):** o4-mini shows high stability (minimal ranking changes, tight confidence intervals), while Qwen-2.5-72B exhibits greater sensitivity (larger performance shifts, wider variance). Crucially, however, **relative rankings** remain consistent for both models.
>
> **Conclusion**: Our robustness analysis validates the reliability of GraphOmni's findings under realistic linguistic perturbation. While absolute values shift, the comparative conclusions about which representations work better hold across both conditions. This confirms that our main findings are not artifacts of specific phrasings. We will include this analysis in the revision and note that our extensible framework readily accommodates prompt perturbation as an additional evaluation dimension for future work.

---

> ### Author Response · Authors · 2025-11-20
> **Response from the authors (3/n)**
>
> # Response to W4 (detailed cost analysis):
>
> We thank the reviewer for this important point. We have added a comprehensive cost analysis in the revised manuscript (**Appendix E.7.6 (p. 104)**, also in Table 1 below). We report per-query costs based on API pricing rather than GPU hours for three reasons: (1) closed-source models (Claude-3.5, GPT-4o, o4-mini, Gemini-2.0) are only accessible via API, making GPU hour measurements infeasible, (2) API pricing reflects the actual deployment cost practitioners face in real-world applications, providing directly actionable insights, and (3) it enables fair comparison across both open-source and closed-source models using a unified metric.
>
> **Our analysis reveals dramatic cost variation across models**. o4-mini achieves the highest accuracy (80.96%) but incurs the highest cost (`$`0.007194 per query, 1369 output tokens on average), while open-source models cost 2-3 orders of magnitude less (`$`0.000049-`$`0.000267) with substantially lower accuracy (29.58%-40.50%). Gemini-2.0 and GPT-4o-mini emerge as cost-efficient alternatives at `$`0.000271 and `$`0.000282 per query, achieving 64.42% and 49.87% accuracy, respectively, a 26x cost reduction compared to o4-mini while retaining 62%-80% of its performance. For the full benchmark (241,726 queries), total costs range from `$`11.85 (Llama-3) to `$`1,739 (o4-mini), a 147x difference with significant implications for large-scale deployments.
>
>
> **Table 1 Per-Query Inference Cost Analysis.**
>
> | Model | Input Cost (\$) | Output Cost (\$) | Total Cost per Query (\$) |
> |-------|----------------|-----------------|-------------------------|
> | **Open-Source Models** | | | |
> | Llama-3.1 (8B) | 0.000019 | 0.000031 | 0.000049 |
> | Mistral (7B) | 0.000187 | 0.000080 | 0.000267 |
> | Phi-4 (14B) | 0.000056 | 0.000084 | 0.000140 |
> | Qwen-2.5 (7B) | 0.000037 | 0.000089 | 0.000126 |
> | **Closed-Source Models** | | | |
> | Claude-3.5 | 0.002799 | 0.002894 | 0.005694 |
> | GPT-4o | 0.002333 | 0.002666 | 0.004998 |
> | GPT-4o-mini | 0.000140 | 0.000142 | 0.000282 |
> | Gemini-2.0 | 0.000093 | 0.000178 | 0.000271 |
> | o4-mini | 0.001026 | 0.006168 | 0.007194 |
>
> ***
>
> # Response to Q1 (clarification on RL design):
>
> Thank the reviewer for the reviewer’s constructive suggestions. We appreciate the request for a clearer description of the RL process. In addition to the detailed formulation already provided in **Appendix D**, we have revised **Section 4.4** in the main text to explicitly describe the **state, action,** and **reward** components of our RL framework (keywords highlighted in red). In our RL formulation, the **state** consists of the task type and difficulty (initial state $s_0$​) together with the previously selected serialization components, while the action at each decision epoch is to choose one specific serialization factor (e.g., prompt, format, or LLM) from the corresponding action space. The **reward** for a complete serialization strategy ($a_1,\ldots,a_T$) is defined as the expected binary correctness of the LLM’s answer, which we approximate in practice by the **average accuracy over N randomly generated graphs** of the same task type.
>
> We have also unified our notation to avoid ambiguity. Specifically, we updated the definitions of Cost and Rate as follows:
>
> > **Cost** measures the proportion of explored combinations:
> $\texttt{Cost} = \frac{k}{K}$,
>  where $k$ is the number of explored combinations and $K$ is the total number of possible combinations.
>
> > **Rate** evaluates the performance of the best combination found by RL:
> $\texttt{Rate} = \frac{acc_*}{acc_{max}}$,
>  where $acc_{*}$ denotes the accuracy achieved by the best RL-found combination, and $acc_{max}$ is the highest accuracy in the benchmark.
>
> These revised definitions have been incorporated into the main text for clarity and consistency.
>
> ***
>
> # Response to Q2 (clarification on notation consistency):
>
> Thank the reviewer for pointing out the symbol reuse. The symbol 𝑚 was previously used for “edge count” in **Section 2** and also (incorrectly) for “accuracy” in **Section 4.4**. We have corrected this inconsistency by introducing the explicit notations $k$, $acc_∗$ and $acc_{max}$  for related quantities, ensuring that $𝑚$ is used exclusively for graph-related terms. We have carefully checked the entire manuscript to confirm global consistency of notation.

---

> ### Author Response · Authors · 2025-11-20
> **Response from the authors (4/4)**
>
> # Response to Q3 (clarification on RL–Scale):
>
> We thank the reviewer for pointing out an ambiguity in our description of the reward design, and we are happy to clarify it here and in **Section 4.4**.
>
> RL-Opt and RL-Scale use the same RL feedback signal. For any given serialization combination, the reward is defined as the LLM’s performance on the target task under that combination, specifically, the average accuracy of the LLMs. In RL-Scale, we evaluate the scalability of the RL-based method as additional serialization factors are introduced and the action space grows combinatorially. For each experiment with a different number of factors, we run an independent RL experiment, and in all settings, the RL method maximizes this same accuracy-based reward. No additional reward engineering is introduced.
>
>
> The balance between computational cost and accuracy is **not** encoded in a single scalar reward. Instead, once RL identifies high-reward combinations for each setting, we **separately** measure the computational **Cost** of these discovered combinations from RL methods. We do not incorporate the **Rate** metric in RL-Scale. As the number of factors increases, the search space grows rapidly. For example, 6 factors yield 75,600 possible combinations. Computing the global maximum accuracy  $acc_{max}$  via exhaustive evaluation is therefore computationally infeasible and also not aligned with the purpose of RL-Scale, which is to assess the **scalability** of the RL method rather than to perform full enumeration.
>
>
> To provide a clearer view of the ‘quality’ of combinations RL method found, we have added **Table 12 2–6 factors: top-3 combinations and corresponding reward from RL-Scale** in **Appendix D**. The “Reward” column corresponds directly to the accuracy metrics discussed above. As shown in the table, when the number of factors increases from 2 to 6, the RL-based method continues to perform well and consistently discovers high-performing combinations. In particular, the best reward found by the RL method improves from 0.30 to 0.63, demonstrating that the approach remains effective. We also recognize that the reviewer’s comment highlights a valid concern regarding the trade-off between reward design complexity and computational cost, especially as the search space grows exponentially. We clarify that the RL method is intended as a generalizable framework for GraphOmni, and integrating explicit cost modeling into the RL objective is a natural and important direction for future work that may further improve search efficiency in large-scale factor spaces.
>
>
> We have revised **Section 4.4** to clarify the reward design and the distinct roles of accuracy, rate, and cost. We thank the reviewer for enabling us to improve the exposition.
>
>
> **Table 2. 2–6 factors: top-3 combinations and corresponding reward from RL-Scale.**
> | Grid Research | Rank | Combination Parameters | Reward |
> |--------------|------|------------------------|--------|
> | **100** | 1 | Edge List, 0-shot, Q:, A:, `\n\t`, no | 0.3000 |
> | **100** | 2 | Edge List, 0-shot, Q:, A:, `\n`, no | 0.2667 |
> | **100** | 3 | Edge List, 0-shot, Q:, A:, , , , no | 0.2000 |
> | **300** | 1 | Edge List, 0-shot, Q:, A:, \|\|, `\t`, no | 0.3667 |
> | **300** | 2 | Edge List, 0-shot, Q:, A:, `--`, `\t`, `\n\t`, no | 0.3333 |
> | **300** | 3 | Edge List, 0-shot, Q:, A:, `--`, `\t`, `-`, no | 0.3333 |
> | **1200** | 1 | Edge List, 0-shot, Q:, A:, `--`, , `\n`, upper | 0.4000 |
> | **1200** | 2 | Edge List, 0-shot, Q:, A:, \|\|, , `:::`, lower | 0.4000 |
> | **1200** | 3 | Edge List, 0-shot, Q:, A:, , `\t`, , upper | 0.3667 |
> | **8400** | 1 | Adjacency Matrix, 0-shot, Q:, A:, `<sep>`, , `\n`, title | 0.5667 |
> | **8400** | 2 | GMoL, 0-shot, Q:, A:, `\n`, `\t`, `:::`, upper | 0.5333 |
> | **8400** | 3 | GMoL, 0-shot, Q:, A:, `; \n`, `\t`, `:::`, upper | 0.5333 |
> | **75600** | 1 | Adjacency Set, Algorithm, Q:, A:, \|\|,  , `:`, lower | 0.6333 |
> | **75600** | 2 | Adjacency Set, Algorithm, Q:, A:, \|\|,  , `:`, no | 0.6333 |
> | **75600** | 3 | Adjacency Matrix, 0-shot, Q:, A:, `\n`,  , `:`, title | 0.6000 |
>
> ***
> # Response to Q5 (more reference needed):
>
> Thank the reviewer for the advice! We are happy to include it in **Table 1** of the revised paper to provide more comprehensive coverage.
>
> ***
>
> **We greatly appreciate the reviewers' time in reading our rebuttal and considering our responses. And we are happy to provide further details or answer any additional questions the reviewers may have.**

---

### Author Response · Authors · 2025-11-21
**A simple clarification on the index used in the author responses**

Dear AC and Reviewers,

To help you better understand our responses, we want to clarify as follows:

1. When we say "Response to W1", we mean "Response to the first point in the Weaknesses section"
2. When we say "Response to Q1", we mean "Response to the first point in the Questions section"

Best,

The authors

---

### Author Response · Authors · 2025-11-25
**General Response to all Reviewers**

Dear Reviewers and AC,


We thank all reviewers for their thoughtful and constructive feedback. We are very grateful that the reviewers recognized the value of GraphOmni, specifically highlighting the **detailed experimental results** (All 4 reviewers), the **comprehensive nature of our evaluation** (Reviewers xfYV, nDLz, 61MW), the **positive contribution of our unified framework** (Reviewers xfYV, nDLz, 61MW) and **insightful and actionable findings** (Reviewer xfYV for mechanistic error analysis, Reviewer nDLz for RL method, Reviewer 61MW for meaningful insights in Section 4.2).


We are committed to addressing the questions posed by the reviewers and have updated the manuscript based on the feedback so far. We summarize the key changes we have made to the paper here:

1. **Robustness Analysis**: We added a supplementary evaluation on prompt perturbations in **Appendix H.3**, confirming that our findings on representation sensitivity hold true even under linguistic variation and noise. (Asked by Reviewer xfYv)
2. **Enhanced Analytical Justification and Cross-Factor Analysis**: To address the request for analytical justifications (Reviewer xfYV) and systematic cross-factor interactions (Reviewer nDLz) , we added a sensitivity-based analysis framework in **Appendix E.3**. These additions quantitatively measure how different graph types respond to variations in prompts and formats, revealing that open-source models are more prompt-sensitive while closed-source models are more format-sensitive. This newly added analysis and its conclusions are consistent with our previous findings, further enriching our overall results and providing deeper insights.
3. **Extended Graph Scale**: We conducted additional experiments on larger graphs (50–100 nodes) in **Appendix H.2**, demonstrating that the fundamental reasoning challenges identified in our main results persist and intensify with scale. (Asked Reviewer 61MW)
4. **Inference Cost Analysis**: We added a comprehensive cost analysis in **Appendix E.7.6**, detailing per-query costs to highlight the efficiency trade-offs between open-source and closed-source models. (Asked by Reviewer xfYv)
5. **Complexity vs. Performance Analysis**: We added a detailed comparison of task time complexity versus LLM performance in **Appendix H.1**, demonstrating that performance is driven not just by algorithmic complexity but also by output structure and reasoning scope. (Asked by Reviewer CVe5)
6. **Clarified RL Framework**: We revised **Section 4.4** to explicitly define the RL state, action, and reward components, unified the notation for cost and rate metrics, and incorporated more detailed descriptions of our experiments. (Asked by Reviewer xfYv)
7. **Updated References**: We updated **Table 1** to include recent benchmarks as suggested. (Asked by Reviewer CVe5)

We believe that this discussion has significantly strengthened the manuscript. We are happy to discuss and clarify any remaining questions the reviewers have.

---

> ### Author Response · Authors · 2025-11-30
> **Summary of Rebuttal to AC**
>
> We want to thank the AC for the extra effort under these special circumstances. Therefore, to assist the AC in navigating the paper, reviews, and rebuttal discussions, we provide a brief recap below:
>
> - **General Overview**: We have included a comprehensive summary of all of our detailed rebuttal responses in the **General Response to all Reviewers** above.
>
> - **Discussion with Reviewer 61MW**: During the rebuttal period, we had a productive discussion with Reviewer 61MW regarding **Graph Perception vs. General Reasoning** and **Scalability/Context Limits**. Notably, the reviewer expressed their inclination to **raise the score**, stating: *"Although I am currently unable to modify my score for technical reasons, I am inclined to **accept** the paper. I believe the work provides valuable observations on graph reasoning with LLMs."*
>
> Best Regards,
>
> The authors

---

### Meta-Review · Area_Chair_omvh · 2026-01-07

**Summary:**

This paper proposes GraphOmni, a large-scale and comprehensive benchmark for evaluating LLMs on graph-theoretic tasks under varying graph types, serialization formats, and prompting schemes, together with an RL-based method to search for effective configurations. The reviewers generally agree that the empirical study is extensive, carefully executed, and clearly presented, and they acknowledge the significant engineering and experimental effort involved. Scores, however, are consistently borderline across reviewers, with no strong consensus toward acceptance.

The discussion and rebuttal were unusually thorough, and the authors addressed many technical questions in detail. Several reviewers (xfYV, nDLz, 61MW) expressed appreciation for the clarity of experiments, additional analyses, and extended discussions during rebuttal, with one reviewer (61MW) indicating an inclination to accept. At the same time, core conceptual concerns raised by CVe5 and echoed implicitly by others remained central throughout the discussion: namely, whether the benchmark leads to genuinely new insights about graph understanding or graph reasoning in LLMs, as opposed to reinforcing already well-known observations from prior LLM and LLM-on-graph literature.

Overall, while I find the study solid and comprehensive, I lean toward rejection. The main reason is that, despite the scale and care of the evaluation, the work does not clearly advance our understanding of graph reasoning in LLMs beyond what is already established. The conclusions largely reiterate known phenomena (prompt sensitivity, benefits of algorithmic prompting, limits of LLMs on explicit graph tasks), and the setting remains constrained to small graphs where classical algorithms trivially dominate. I would not mind if the paper were accepted, but I do not see a strong enough original contribution to warrant acceptance at ICLR this cycle.

**Reviewer Concerns:**

The rebuttal convincingly addressed several technical concerns. Questions about robustness to prompt noise, notation consistency, RL formulation, cost analysis, and cross-factor interactions raised by xfYV and nDLz were handled carefully, with added experiments and clearer explanations. The authors also clarified evaluation procedures and extended analyses to larger graphs, which alleviated some implementation-level doubts. Reviewer 61MW acknowledged these clarifications and explicitly stated a positive inclination toward acceptance after discussion.

However, the core conceptual concerns remain largely unresolved. CVe5’s critique that most findings are incremental, previously known, or not clearly graph-specific, was not fully addressed, despite the length of the rebuttal. The central observations (serialization matters, prompts matter, algorithmic hints help, open-source models are more prompt-sensitive) are well-established in the broader LLM literature and are not unique to graph reasoning. Similarly, concerns raised by 61MW about the distinction between graph understanding and solving explicit graph tasks remain open: strong task performance does not necessarily imply deeper graph understanding, and the benchmark does not clearly disentangle this distinction.

More broadly, several reviewers expressed skepticism about the significance of evaluating LLMs on small (≈30-node) explicit graph problems, a regime where LLMs are known to underperform classical algorithms and where no generalization beyond the benchmark setting is demonstrated. While the authors frame the work as diagnostic, the paper stops short of offering new conceptual insights into how LLMs internalize graph structure or how this differs from general reasoning failures. As a result, the main concerns about novelty, insight, and impact are only partially mitigated.

**Reviewer Scores:**

Reviewer xfYV scored the paper slightly above the acceptance threshold and, based on the discussion, would likely maintain a similar score or increase it marginally. Reviewer nDLz also gave a borderline-positive score and would likely remain in that range after rebuttal. Reviewer 61MW explicitly stated an inclination to accept the paper, although their original score could not be changed, suggesting a potential upward adjustment if possible. Reviewer CVe5, however, remained unconvinced about novelty and insight, and their score would likely stay below or at the borderline even after rebuttal.

Taken together, the scores reflect a consistently borderline paper with no strong push toward acceptance. While some reviewers warmed up after discussion, there is no clear signal that the work meaningfully shifts perspectives in the field. Given the similarity of the main findings to prior work on LLM prompting and representation sensitivity, I do not believe the paper, as it stands, makes a sufficiently original contribution for ICLR. I would encourage the authors, in future work, to more sharply articulate what new understanding of graphs (as opposed to task execution) their benchmarks enable, and to explore settings/generalization beyond graph-theoretic tasks.

---

### Decision · Program_Chairs · 2026-01-26

Accept (Poster)